# THUNDER: Tile-level Histopathology image UNDERstanding benchmark

**Pierre Marza**[1,2◇]**, Leo Fillioux**[1,2*]**, Sofiène Boutaj**[1,2*]**, Kunal Mahatha**[3]**,
Christian Desrosiers**[3]**, Pablo Piantanida**[4]**, Jose Dolz**[3]**, Stergios Christodoulidis**[1,2†]**,
Maria Vakalopoulou**[1,2†]

[1] MICS Laboratory, CentraleSupélec, Université Paris-Saclay
[2] IHU PRISM, National Center for Precision Medicine in Oncology, Gustave Roussy
[3] LIVIA, ILLS, ETS Montreal
[4] ILLS, MILA, Université Paris-Saclay, CNRS, CentraleSupélec

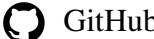 GitHub    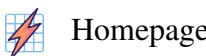 Homepage

## Abstract

Progress in a research field can be hard to assess, in particular when many concurrent methods are proposed in a short period of time. This is the case in digital pathology, where many foundation models have been released recently to serve as feature extractors for tile-level images, being used in a variety of downstream tasks, both for tile- and slide-level problems. Benchmarking available methods then becomes paramount to get a clearer view of the research landscape. In particular, in critical domains such as healthcare, a benchmark should not only focus on evaluating downstream performance, but also provide insights about the main differences between methods, and importantly, further consider uncertainty and robustness to ensure a reliable usage of proposed models. For these reasons, we introduce *THUNDER*, a tile-level benchmark for digital pathology foundation models, allowing for efficient comparison of many models on diverse datasets with a series of downstream tasks, studying their feature spaces and assessing the robustness and uncertainty of predictions informed by their embeddings. *THUNDER* is a fast, easy-to-use, dynamic benchmark that can already support a large variety of state-of-the-art foundation, as well as local user-defined models for direct tile-based comparison. In this paper, we provide a comprehensive comparison of 23 foundation models on 16 different datasets covering diverse tasks, feature analysis, and robustness. The code for *THUNDER* is publicly available at https://github.com/MICS-Lab/thunder.

## 1   Introduction

Histopathology is the gold standard for assessing the structure, cellular phenotypes, and cell-to-cell interactions in tissue samples. It is extensively used in cancer care as it can provide important insights at the level of the tumor microenvironment, allowing for triage, diagnosis, disease sub-typing, or treatment decisions. Digital pathology emerged recently as a research topic aiming to develop automated tools for the processing and analysis of histopathology images that can streamline clinical

---

[*][†]denote equal contribution.
[◇]corresponding author: pierre.marza@centralesupelec.fr

39th Conference on Neural Information Processing Systems (NeurIPS 2025) Track on Datasets and Benchmarks.

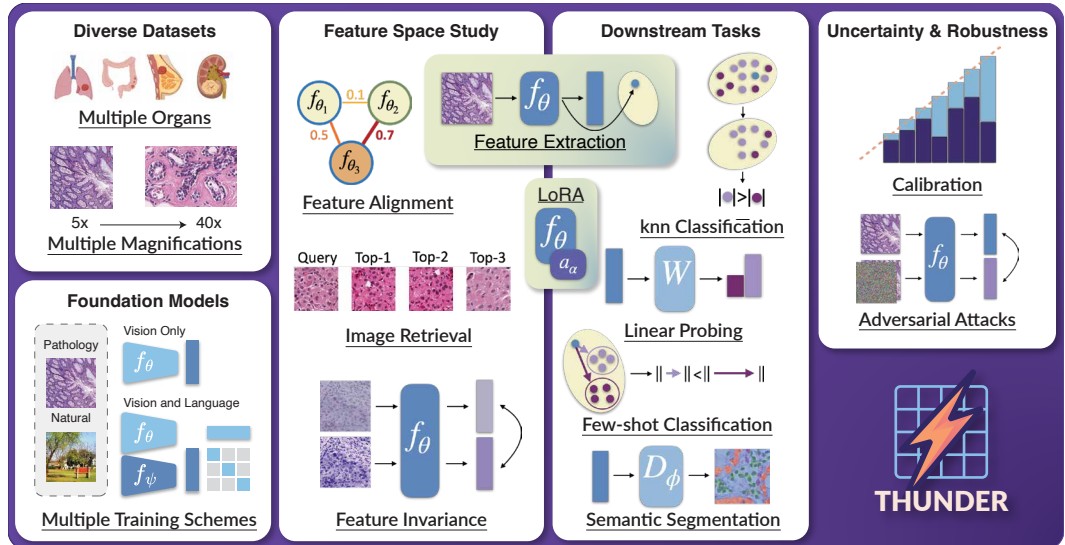

Figure 1: **THUNDER**: We propose a benchmark to compare and study foundation models across three axes: (*i*) downstream task performance, (*ii*) feature space comparisons, and (*iii*) uncertainty and robustness. Our current version integrates 23 foundation models, vision-only, vision-language, trained on pathology or natural images, on 16 datasets covering different magnifications and organs. *THUNDER* also supports the evaluation of new user-defined models for direct comparisons.

practices, making them more robust and efficient, while providing a standardization across various centers and protocols. Large strides have been taken towards this direction lately, especially with the introduction of very large deep learning models trained using self-supervised learning on large curated datasets (i.e., foundation models). Such models trained specifically on domain-specific data stand out thanks to their representative power and versatility [10, 73, 83, 61, 17, 18, 54, 35, 48, 13, 82, 29, 27, 78]. However, their growing number blurs the landscape of current pre-trained vision encoders for digital pathology. Taking also into consideration other general-purpose foundation models [57, 14, 58] that have been trained on even larger and more diverse datasets, assessing their capabilities and understanding better their differences is not a trivial yet crucial step.

There are already a number of published benchmark results on pre-trained models in digital pathology [77, 34, 55, 25, 2, 22, 7, 45, 52, 1, 80, 9]. However, most of them do not come with an open-source implementation, whereas some compare older backbones that are not considered the state-of-the-art today. More importantly, they often focus on reporting downstream performance in specific settings, such as slide-level multiple-instance learning (MIL) training or linear probing, risking to draw conclusions specific to the chosen task. Such evaluation tasks can be time-consuming, and the final performance might be influenced by other factors than the foundation model itself, e.g., the embedding aggregation step in MIL settings. Last, these benchmarks completely disregard the feature space properties of compared models, and often omit a study of uncertainty estimation and robustness, which are, however, crucial, especially for healthcare applications.

Inspired by these observations, we introduce *THUNDER* (Figure 1), a benchmark to compare foundation models on different downstream tasks, but also study their feature spaces and evaluate their robustness and uncertainty when used in challenging settings. We gather 16 diverse recognized datasets spanning different cancer types, magnifications, image and sample sizes, and propose a series of tasks. Importantly, this benchmark is patch/tile-level, meaning that we compare the representations of foundation models for a patch, isolating its representative power from other aggregation processes at the slide level, e.g., MIL, that might blur the possible conclusions to be drawn. Our benchmark currently supports 23 recent state-of-the-art models and shows that we can draw many different conclusions from the diverse evaluation settings considered. We provide this benchmark to the community as a tool to efficiently compare foundation models, making it easy to integrate new ones in an automatic way and compare them.

## 2 Related work

**Foundation models in histopathology** — are presented as general feature extractors, to be leveraged in diverse downstream settings. These models are pretrained using different self-supervised strategies and/or different data modalities. A variety of vision-only [10, 73, 83, 61, 17, 18, 54, 35, 71, 74, 79], as well as vision-language [48, 13, 82, 29, 27, 78, 62] models have been proposed in the last years, each claiming different advantages. Most of them are trained on pathology tiles [10, 73, 83, 61, 17, 18, 54, 35, 48, 82, 29, 27, 78], and even if some are slide-level models [13, 71, 74, 79, 62] they all rely on a patch-level foundation model to extract tile features to be aggregated. Most recent vision encoders are variants of the Vision Transformer (ViT) [14] and are trained in a self-supervised manner, mainly leveraging DINOv2 [57] or iBOT [81] training objectives for vision-only models and CLIP [58]-like loss functions for vision models trained together with a text encoder. One of the main differences between foundation models comes from the training data source, i.e., whether it comes from public [70, 12, 15, 29] or private databases, size, i.e., number of tiles and/or slides, magnification, organs represented. Indeed, models share similar architectures and training objectives, and the main differences rely on how datasets are compiled and pre-processed.

As many foundation models have been released recently, getting a clear understanding of their differences, strengths, and weaknesses thus becomes primordial. This motivates the introduction of a benchmark like *THUNDER*. Importantly, even if it already supports the most recent foundation models, it is not restricted to them, and can be used to evaluate any model, such as new backbones, or lighter CNN-based models [11].

**Benchmarking pathology models** — has already been studied in previous work [77, 34, 55, 25, 2, 22, 32, 7, 45, 52, 1, 80, 9, 50]. Existing benchmarks mainly focus on downstream performance, mostly on slide-level tasks. As the majority of foundation models are trained on patch-level images, a common approach is to train an aggregator, e.g. with a MIL method [30, 49, 63], to provide a prediction from features extracted using pre-trained models for different slide regions. While relevant from a clinical point of view, such a setting adds complexity, both from a computational point of view, but also experimentally, as features from foundation models are not compared directly but through their aggregation from a specific method. Moreover, while predictive performance is important, most benchmarks disregard the uncertainty and robustness of foundation models, which is essential for many medical imaging applications. Finally, very few benchmarks come with an open-source implementation. Exceptions to this are *eva* [22] and *Patho-bench* [80], which both propose public benchmark implementations. However, both put a focus on downstream performance, with *Patho-bench* targeting slide-level only, and *eva* both a patch-level and slide-level tasks. *HEST-Benchmark* [32] and *PathBench* [50] are also to be considered even if they are less directly comparable. We provide a more detailed comparison to open-source benchmarks in appendix (C).

In this study, we propose a benchmark to assess and compare the downstream performance of diverse foundation models, and more than this, also study the differences in their feature spaces and their robustness and uncertainty estimation. By focusing on comparing the performance of foundation models and studying their feature spaces on patch-level datasets, we remove the additional feature aggregation step and thus isolate their own representative power. Finally, we provide an open-source implementation, allowing for efficient comparison of foundation models on tile-level tasks, being complementary to existing slide-level benchmarks.

## 3 Benchmarking foundation models for tile-level digital pathology

*THUNDER* is characterized by the variety of considered datasets and foundation models, the diverse downstream tasks spanning different applicative needs, the study of feature spaces, and of the uncertainty and robustness of pre-trained backbones. By coming with an open-source and easy-to-use implementation, *THUNDER* aims to be the next available tool for a wide benchmark of models on different tasks and analyses.

### 3.1 Models and datasets

*THUNDER* currently supports 23 foundation models. We consider vision encoders from vision-only [10, 73, 83, 61, 17, 18, 54, 35, 71, 74, 79], but also from vision-language [48, 13, 82, 29, 27, 78, 62] models, and study both recent histopathology-specific models as well as backbones pre-trained

Table 1: **Tile-level datasets included in *THUNDER***: Overview of the 16 datasets currently supported, spanning different tasks, numbers of classes and samples, organs, input sizes, magnifications.

| Name | Short name | Labels | Nb. cls. | Organ(s) | Im. size | Magnif. | Nb. im. |
|---|---|---|---|---|---|---|---|
| BACH [4] | bach | Classif. | 4 | Breast | $1,536 \times 2,048$ | 20× | 408 |
| BRACS [6] | bracs | Classif. | 7 | Breast | Variable | 40× | 4,539 |
| BreakHis [66] | break-h | Classif. | 8 | Breast | $700 \times 460$ | 40× | 1,995 |
| Camelyon17 WILDS [38] | wilds | Classif. | 2 | Breast | $96 \times 96$ | 10× | 302,436 |
| Patch Camelyon [72] | pcam | Classif. | 2 | Breast | $96 \times 96$ | 10× | 327,680 |
| CRC-100k [37] | crc | Classif. | 9 | CRC | $224 \times 224$ | 20× | 107,180 |
| MHIST [76] | mhist | Classif. | 2 | CRC | $224 \times 224$ | 5× | 3,152 |
| TCGA CRC-MSI [36] | tcga-crc | Classif. | 2 | CRC | $512 \times 512$ | 20× | 51,918 |
| CCRCC [8] | ccrcc | Classif. | 3 | Renal | $300 \times 300$ | 40× | 52,713 |
| ESCA [69] | esca | Classif. | 11 | Oeso. | $256 \times 256$ | 10× | 367,229 |
| TCGA TILS [33] | tcga-tils | Classif. | 2 | Multi | $100 \times 100$ | 20× | 304,097 |
| TCGA Uniform [41, 42] | tcga-unif | Classif. | 32 | Multi | $256 \times 256$ | 20× | 271,170 |
| Ocelot [60] | ocelot | Segm. | 2 | Multi | $256 \times 256$ | 40× | 10,608 |
| PanNuke [20, 21] | pannuke | Segm. | 6 | Multi | $256 \times 256$ | 40× | 7,901 |
| SegPath Epithelial [39, 43] | segp-ep | Segm. | 2 | Multi | $256 \times 256$ | 40× | 238,581 |
| SegPath Lymphocytes [40, 43] | segp-ly | Segm. | 2 | Multi | $256 \times 256$ | 40× | 110,457 |

on natural images and text. Details about their architecture, number of parameters, training strategy, as well as sources for training data are presented in Table S1 in appendix. Moreover, Table 1 presents the 16 public datasets currently considered in our benchmark [4, 6, 66, 38, 8, 37, 69, 76, 72, 36, 41, 42, 33, 3, 60, 20, 21, 39, 40, 43]. They cover both classification and segmentation with a different number of classes, diverse cancer types, magnifications, as well as image and sample sizes.

## 3.2 Evaluation protocols

**Feature space study** — Understanding the differences between foundation models requires going beyond mere performance evaluation and comparing their representation spaces. We thus consider a series of tasks to assess the alignment of their feature spaces, both original ones and after adaptation, the main patterns they detect relying on image retrieval, and the characteristics in input images they are invariant to. This way, we position each model in the current landscape of models, highlighting their differences and similarities. For all the tasks, we use cosine similarity as the distance to evaluate performance.

**(i) Feature space alignment** is a way to compare the embedding spaces of different foundation models. Following [28], we consider different alignment metrics, and in particular the introduced *Mutual knn*, which computes the size of the intersection of nearest neighbor sets of two foundation models for similar query samples. The larger the intersection, the more aligned the models will be considered, providing a proxy for embedding spaces being similar. **(ii) LoRA adaptation** [26] modulates the embedding space of a pre-trained model. We thus study how the alignment between foundation models evolves when they are adapted. If they tend to align more, provided enough data to perform such adaptation, is the choice of the initial backbone of any importance? On the other hand, if they diverge, could it provide us information about the starting point, i.e., original feature spaces being significantly different? **(iii) Image retrieval** provides a qualitative assessment of differences in model feature spaces. Comparing the top-$k$ closest images to a query in embedding space helps us better understand the information contained in the extracted embeddings, and in particular, the main characteristics extracted for an image, e.g., either style or morphological features. **(iv) Invariance to image transformations** is an important indicator of the information contained in the extracted embeddings. For instance, if the output embedding does not change when altering the contrast or saturation of the input image, then it means that photometric information is not captured by the model. By studying the invariance of foundation models to different image transformations, we can then refine our understanding of the information they store. More than this, we can also evaluate their robustness to certain transforms, providing a proxy to their ability to generalize to specific domain shifts. We thus compute the distance between embedding representations for the original and perturbed images for all models.

Table 2: **Benchmark task runtimes and computational requirements** to evaluate one model (averaged across supported models). $^\dagger$ denotes tasks using pre-computed embeddings. **Emb. comp.** runtimes are computed on the 12 classification datasets.

| Runtime | Emb. comp. | Knn$^\dagger$ | Few-shot$^\dagger$ | Lin. prob. + calib.$^\dagger$ | Segm.$^\dagger$ | Adv. attack |
|---|---|---|---|---|---|---|
| Min. | 00h08 | 00h27 | 00h27 | 00h15 | 05h08 | 00h01 |
| Max. | 02h57 | 01h13 | 11h32 | 18h39 | 12h11 | 01h05 |
| Avg. | 01h14 | 00h37 | 02h12 | 03h21 | 09h10 | 00h37 |
| Cumulative | 14h48 | 07h22 | 26h21 | 40h16 | 36h39 | 07h20 |
| (Nb. datasets) | (12) | (12) | (12) | (12) | (4) | (12) |
| Hardware | $\times$1 V100 | $\times$32 CPUs | $\times$32 CPUs | $\times$1 V100 | $\times$1 V100 | $\times$1 V100 |

**Downstream tasks** — One of the common ways to evaluate the power and capabilities for general performance of foundation models is to challenge them on a variety of tasks and datasets. Such an analysis is usually presented in the original papers proposing foundation models, but since datasets and metrics tend to vary between them, there is a need for a standard benchmark to fairly compare models. In this study, we used different metrics including *accuracy*, *balanced accuracy* and *F1-score* for classification, and *Dice (F1) score*, and *Jaccard index* for segmentation. Specifically, we challenge the models in the following settings.

**(i) knn classification** provides a direct signal of the predictive power of a feature space. For each test sample, we perform a majority voting among the $k$ – the $k$ value being validated on a validation set – training nearest neighbors based on cosine similarity distance measure. **(ii) Linear probing** is another important task to consider. Indeed, it is a standard choice when evaluating pre-trained models as it is parameter-efficient, accommodating black-box adaptation and does not require large computational resources. **(iii) Few-shot classification** is a more challenging setting, as, unlike in *knn classification* and *linear probing*, where we have access to the entire dataset, few-shot learning methods can only use a few support samples per class (1, 2, 4, 8, or 16). We leverage the SimpleShot [75] method to perform few-shot classification from support embeddings extracted with the foundation models. **(iv) Semantic segmentation** evaluates the spatial information contained in the embeddings from pre-trained models. We extract 2D spatial embeddings from the models and train a Segmenter [67] decoder head to perform semantic segmentation by minimizing a Dice loss. The same setting is considered as in *linear probing*: validating hyperparameters on a validation set and testing performance on an independent test set. **(v) LoRA adaptation** is a specific setting we study mainly for classification in this paper, as it is representative of current practices when applying pre-trained models on a downstream task. To this end, we train LoRA adapters [26], as they are lightweight and computationally efficient. In addition to studying its impact on the feature space as presented in the previous sub-section, we also evaluate the performance gains it can bring.

**Uncertainty estimation and robustness** — Lastly, in addition to downstream performance, we are also interested in building robust and reliable predictors based on foundation models. We thus evaluate how well-calibrated linear probes trained on pre-trained features are and how such foundation models are robust to adversarial attacks in image space. We consider standard calibration metrics, i.e., *Expected Calibration Error (ECE)*, *Maximum Calibration Error (MCE)*, *Adaptive Calibration Error (ACE)*, *Threshold Adaptive Calibration Error (TACE)* [24, 56], and assess the robustness to adversarial attacks by measuring the performance drop on the test set between the original and adversarially perturbed images.

**(i) Calibration** is an important property of neural models [24, 56]. In any downstream task, but even more in sensitive contexts such as medical imaging, providing an accurate estimation of the prediction uncertainty is important. We thus compare the calibration of linear classifiers trained on top of embeddings from foundation models, to see whether different feature spaces lead to more or less calibrated classifiers. **(ii) Robustness to adversarial attacks** is a critical consideration before deploying foundation models in high-impact applications [19, 23, 31, 68, 47, 53]. To assess this, we evaluate the robustness of different backbones to additive adversarial noise in input images by applying the Projected Gradient Descent (PGD) attack [51] for different perturbation budget $\epsilon$.

**Main design choices and runtime** — To foster a fair comparison between models and reproducibility of results, we produce a fixed set of data splits for each considered dataset. We follow the standard

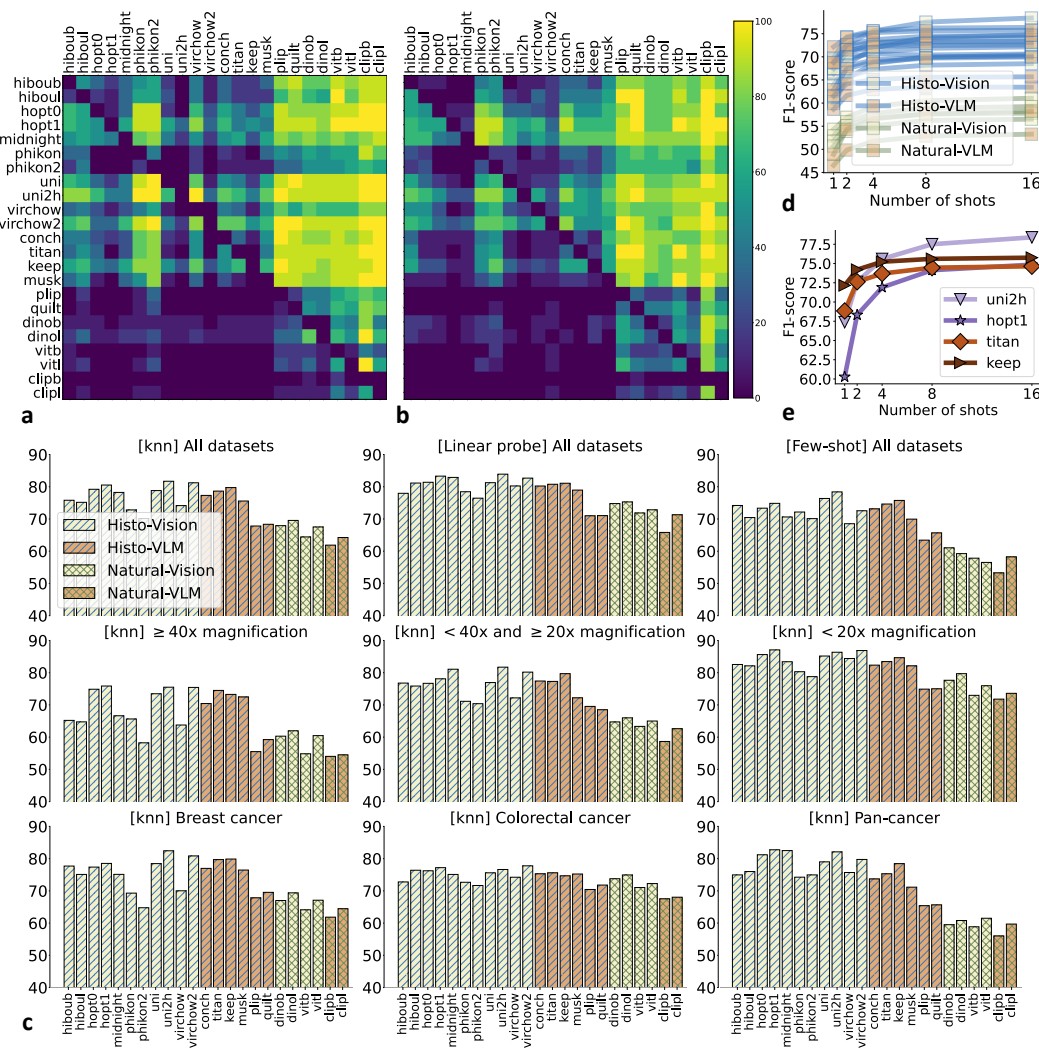

Figure 2: **Classification**: Performance comparison heatmaps for **(a)** knn classification and **(b)** linear probing – **(c)** Distribution of average F1-scores across datasets per model for different tasks and stratified according to magnifications and organs – Few-shot F1-score as a function of shots for **(d)** all models and **(e)** a set of selected models.

train/val/test split when available, and otherwise split the train set into a train and validation sets, and consider publicly available samples outside of the official train set as a test set. The validation sets are used to perform automatic hyperparameter search ($k$ value for knn, learning rate, and weight decay for linear probing and segmentation) to ensure a fair comparison of foundation models as general feature extractors. We also want to emphasize the importance of the computational efficiency of a benchmark and focus on this in our implementation. Table 2 shows the runtime for each downstream and uncertainty/robustness task to evaluate one model. For all tasks different from embedding pre-computing itself, we consider that image embeddings have been extracted a priori as *THUNDER* allows to do it (*emb. pre-comp.* task). The cumulative time is the average total time to run a model across all datasets for a given task. As can be seen, some tasks (*knn*, *few-shot*) can be run on CPU only, and others only require a single V100 GPU for a reasonable amount of time. Note that the cumulative time represents the worst-case scenario where a model is evaluated sequentially on all datasets. However, *THUNDER* allows evaluating a model on different datasets in parallel (separate jobs), reducing the cumulative time to the max time if more resources are available. Additional details about runtimes of feature space study tasks and design choices are provided in appendix (B, E).

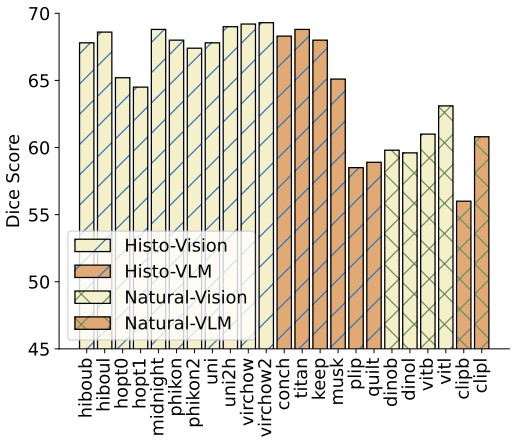

Figure 3: **Segmentation**: Distribution of Dice scores.



Figure 4: **Segmentation**: Predicted masks.

Table 3: **Classification**: Gain in linear probing F1-score from LoRA.

| Dataset | uni | uni2h | virchow | virchow2 | keep | conch |
|---------|-----|-------|---------|----------|------|-------|
| mhist | +2.3 | +4.9 | +2.9 | +3.9 | +7.6 | +5.4 |
| bracs | +4.6 | -1.3 | +2.8 | +1.0 | -1.3 | +2.7 |

## 3.3 Benchmarking at the tile level

Most foundation models for digital pathology are trained at the tile level, and even slide-level encoders leverage a pre-trained patch-level model. To perform predictions at the level of the slide, the latter must be divided into patches to extract patch-specific features, that will then be aggregated. Evaluating them on tiles allows us to isolate the predictive power of vision models independently of aggregation strategies, leading to a more direct evaluation of their representations.

Additionally, working at the tile level allows one to avoid the heavy slide processing which can be compute-demanding. Indeed, as an example, extracting features with *virchow2* [83] at the standard *20X* magnification consumes around 514 V100 GPU hours across the 7 following well-studied datasets: BLCA (437 WSIs, 63h), BRCA (1100, 106h), CAMELYON16 (400, 42h), KIRC (511, 83h), LUAD (456, 69h), LUSC (505, 66h) and UCEC (504, 85h)  a total of around 4000 WSIs. Repeating this for each of the 23 foundation models pushes the bill to more than 10000 GPU hours before any slide-level training is done. By contrast, our benchmark covers all 16 datasets with around 2 million pre-extracted patches; the same 23-model ensemble finishes feature extraction in less than 500 GPU hours, while providing richer supervision (around 2M patch-level labels vs. around 4k slide-level labels). After feature extraction, a Multiple Instance Learning (MIL) aggregator must be trained to aggregate patch-level features to perform a prediction at the level of the slide. Common methods such as Abmil ([30] $\simeq$ 1M parameters) or Transmil ([63] $\simeq$ 3M parameters) require training more parameters than simple linear probes as used in *THUNDER*.

We believe that slide-level benchmarks are important, and rather propose *THUNDER* as a complementary tile-level alternative allowing for faster and more direct evaluation on many different datasets requiring fewer resources. Additionally, using patch-level data enables us to provide a fully reproducible benchmark, which is much more challenging for slide-level tasks due to required pre-processing steps.

## 4 Experiments

We present aggregated performance for the different benchmark tasks, comparing the currently supported 23 recent foundation models to showcase the insights that can be drawn from our benchmark. Importantly, our open-source implementation allows one to benchmark any other pre-trained vision encoder. Detailed results for all datasets and models independently, along with confidence intervals, additional visualizations, and implementation details, are presented in appendix (E, H).

**Classification-related downstream tasks** —  are evaluated in Figure 2. (a) and (b) report the proportion of classification datasets where a model (row) significantly outperforms another (column) on knn classification and linear probing respectively, in terms of per-sample accuracy. We perform a per-dataset Binomial test on per-sample binary accuracies with Benjamini-Hochberg p-value correction [5] for all model pairs. Histopathology models often outperform natural-image models, and

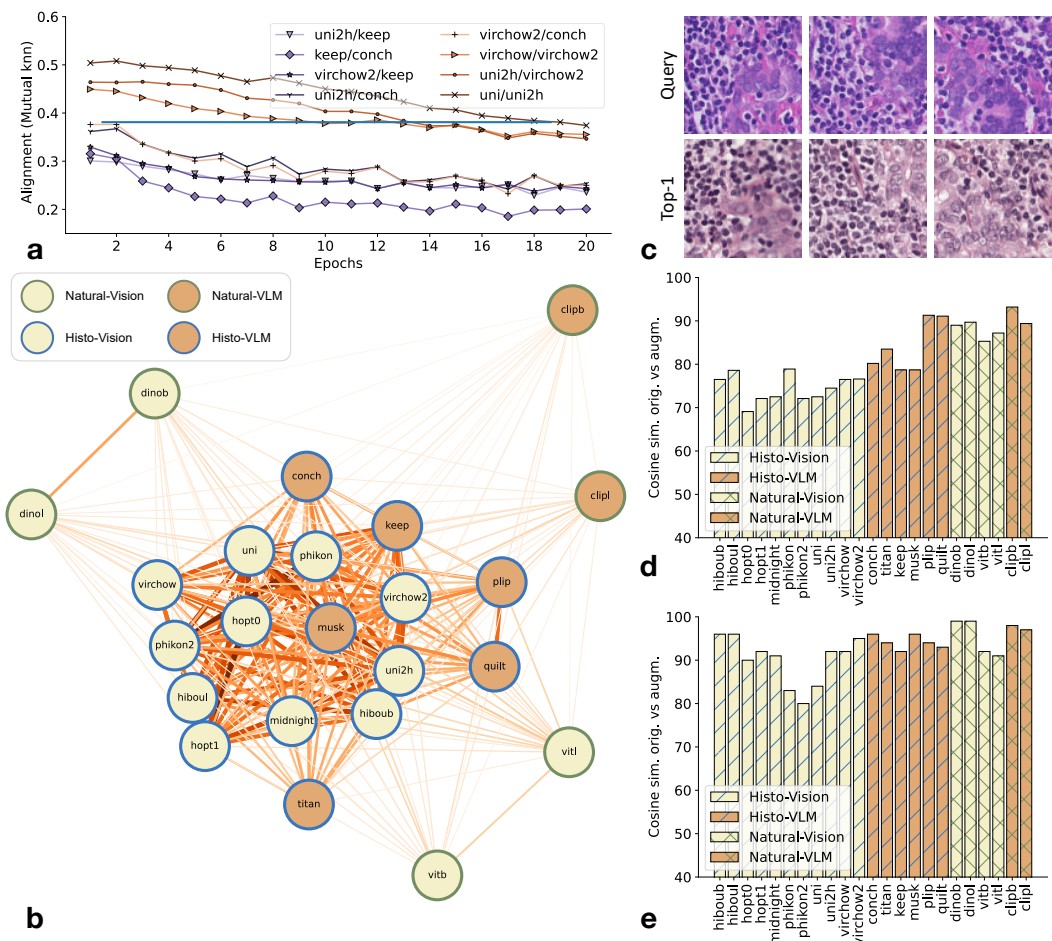

Figure 5: **Feature space study**: **(a)** Evolution of pair-wise alignment between models during LoRA adaptation – **(b)** Average alignment between models across all datasets visualized as a graph – **(c)** Image retrieval samples for *uni2h* on *wilds* – Cosine similarity between embeddings extracted from original and augmented images, **(d)** averaged across all considered augmentations or **(e)** only for the histopathology-specific HED transform.

a few models, e.g. *uni2h*, *virchow2*, *midnight*, *hopt1*, *keep* are superior to many others. Interestingly, gaps between models tend to decrease when transitioning from knn to linear probing. Figure 2(c) presents the average F1-score for the different models on knn classification, linear probing and few-shot classification (16 shots) stratified according to magnification and organs. Performance trends seem to be quite similar between tasks, with vision-language models showing particularly good performance in the few-shot setting. Performance also varies for different magnifications and organs. For instance, the gap between histopathology and natural models widens at $20\times$, which could be explained by the predominance of $20\times$ slides in pre-training datasets. Finally, Figure 2(d) and (e), focus on the few-shot classification. As expected, performance increases with more shots, but more importantly, confirming findings in (c), the strongest vision-language models (*titan* and *keep*) showcase higher performance on low-shot (e.g. 1-shot) settings as well.

**Segmentation downstream task** — Figure 3 presents the average Dice score of Segmenter decoders [67] trained on embeddings extracted from the different foundation models. *virchow2* showcases superior performance, while *plip* and *quilt*, unlike other histopathology VLMs, do not appear to extract relevant spatial information. Figure 4 provides qualitative examples of segmentation predictions from *virchow2* embeddings.

**Feature space study** — results are presented in Figure 5. First, (a) and (b) illustrate feature space alignment. (a) shows the evolution of model pair-wise alignment (*Mutual knn*) during LoRA

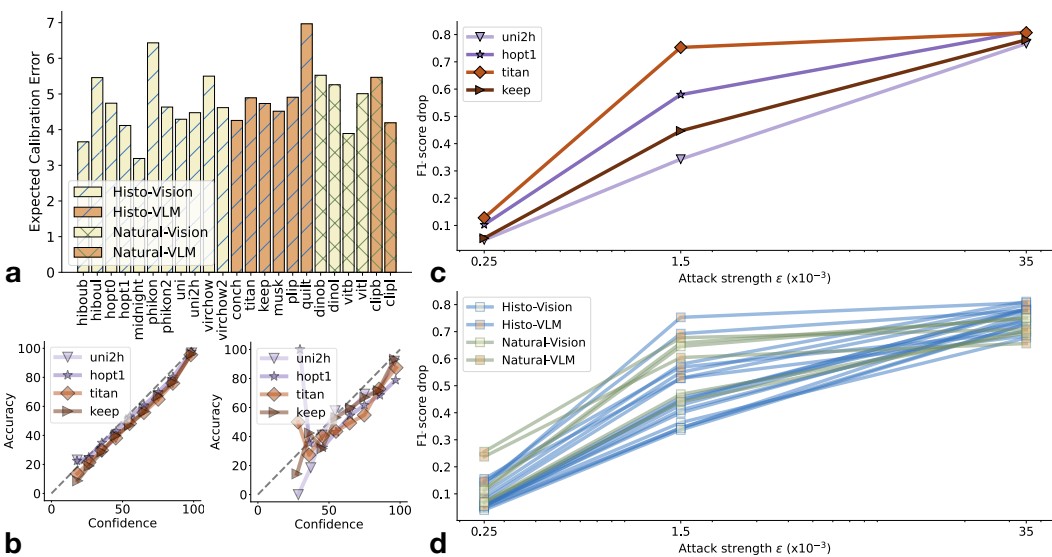

Figure 6: **Uncertainty estimation and robustness**: Distribution of average ECE for **(a)** all models and **(b)** sample calibration curves on 2 datasets (*bracs* and *tcga-unif*) for selected models – **(c)** Drop in F1-score as a function of adversarial attack strength for all models and **(d)** for selected models.

adaptation averaged across the *bracs* and *mhist* datasets. There is a clear overall trend for feature space alignment to decrease while training with adapters, even for methods which are initially well aligned. (b) is a graph visualization of the average alignment (*Mutual knn*) between pairs of models on all 12 classification datasets. Natural-image models seem to be far from pathology models. Among the latter, vision-language and vision-only models tend to have stronger connections between models within respective groups, while this is not true for all of them (e.g. *musk*, *titan*). Figure 5(c) presents 3 queries and top-1 samples when performing image retrieval on the *wilds* dataset with *uni2h* embeddings. The spatial distribution of cells seems to be captured in *uni2h* embedding as queries and top-1 images showcase large similarities. Finally, Figure 5(d) presents the average cosine similarity between embeddings of original and augmented images considering a series of photometric, geometric, morphological transformations (see Table S5 in appendix), while (e) focuses explicitly on the histopathology-specific HED transform [16]. Natural models appear to be more invariant in general, and vision-language pathology models more invariant than vision pathology models. However, the gap is lower when considering the HED transform.

**Uncertainty estimation and robustness** — are illustrated in Figure 6. (a) presents the distribution of average ECE for the different models and (b) specifically visualizes calibration curves for 4 models on the *bracs* (left) and *tcga-unif* (right) datasets. Interestingly, discriminative performance does not seem to correlate with better calibrated estimates for some models, e.g., *uni2h*. From (b) we can also see that some datasets are more challenging from a calibration point of view. It is important to note that calibration is probe-dependent, and the presented differences in performance and ranking are thus conditioned on the chosen classifier, in our case a linear classifier. We indeed present a difference in calibration performance when choosing a linear classifier or an MLP with the same hidden size (256) for all models in appendix (Table S6). However, we believe the linear probe remains a relevant reference point, as it is the only head predicting classes directly from the feature space (no intermediate representations) and requires no hyper-parameter selection (e.g., architecture choices), thereby providing a clearer view of the impact of the embedding space on calibration. Figure 6(c) and (d) focus on the robustness to adversarial attacks for all models and only a set of selected ones respectively. The drop in F1-score increases with the strength of the performed attack. $\epsilon = 35 \cdot 10^{-3}$ leads to a strong drop in performance, while the noisy image is indistinguishable from the original one, showing that foundation models can be strongly influenced by such attacks, which is concerning when considering how sensitive healthcare applications are. With a smaller $\epsilon$ value, we can observe more diverse performance between models, with vision-language being more affected, and pathology vision models performing generally better.

Table 4: **Rank-sum overall performance comparison:** for each task and model, we report the score along with the rank between parentheses – Vision and Vision-Language models.

| Task | Histopathology models | | | | | | | | | | | Natural-image models | | | | | | | | | | | |
|---|---|---|---|---|---|---|---|---|---|---|---|---|---|---|---|---|---|---|---|---|---|---|---|
| | hiboub | hiboul | hopt0 | hopt1 | midnight | phikon | phikon2 | uni | uni2h | virchow | virchow2 | conch | titan | keep | musk | plip | quilt | dinob | dinol | vitb | vitl | clipb | clipl |
| knn ↑ | 75.8 (10) | 75.2 (12) | 79.2 (5) | 80.5 (3) | 78.2 (8) | 72.8 (14) | 70.1 (15) | 78.8 (6) | 81.7 (1) | 74.2 (13) | 81.2 (2) | 77.3 (9) | 78.6 (7) | 79.7 (4) | 75.6 (11) | 67.8 (19) | 68.3 (17) | 67.9 (18) | 69.6 (16) | 64.4 (21) | 67.5 (20) | 61.9 (23) | 64.2 (22) |
| Lin. prob. ↑ | 78.0 (14) | 81.2 (7) | 81.4 (5) | 83.3 (2) | 82.9 (3) | 78.4 (13) | 76.5 (15) | 81.3 (6) | 83.9 (1) | 80.2 (10) | 82.7 (4) | 80.2 (11) | 80.8 (9) | 81.1 (8) | 79.0 (12) | 71.0 (22) | 71.0 (21) | 74.8 (17) | 75.3 (16) | 71.9 (19) | 72.8 (18) | 65.8 (23) | 71.3 (20) |
| Few-shot ↑ | 74.2 (6) | 70.4 (12) | 73.4 (7) | 74.8 (4) | 70.6 (11) | 72.2 (10) | 70.1 (13) | 76.4 (2) | 78.4 (1) | 68.5 (15) | 72.6 (9) | 73.1 (8) | 74.6 (5) | 75.8 (3) | 70.0 (14) | 63.4 (17) | 65.7 (16) | 61.0 (18) | 59.2 (19) | 57.8 (21) | 56.5 (22) | 53.3 (23) | 58.2 (20) |
| Seg. ↑ | 67.8 (10) | 68.6 (6) | 65.2 (13) | 64.5 (15) | 68.8 (4) | 68.0 (9) | 67.4 (12) | 67.8 (11) | 69.0 (3) | 69.2 (2) | 69.3 (1) | 68.3 (7) | 68.8 (5) | 68.0 (8) | 65.1 (14) | 58.5 (22) | 58.9 (21) | 59.8 (19) | 59.6 (20) | 61.0 (17) | 63.1 (16) | 56.0 (23) | 60.8 (18) |
| Calib. ↓ | 3.7 (2) | 5.5 (18) | 4.7 (13) | 4.1 (4) | 3.2 (1) | 6.4 (22) | 4.6 (11) | 4.3 (7) | 4.5 (8) | 5.5 (20) | 4.6 (10) | 4.3 (6) | 4.9 (14) | 4.7 (12) | 4.5 (9) | 4.9 (15) | 7.0 (23) | 5.5 (21) | 5.3 (17) | 3.9 (3) | 5.0 (16) | 5.5 (19) | 4.2 (5) |
| Adv. attack ↓ | 52.8 (14) | 40.0 (5) | 44.2 (9) | 58.0 (17) | 36.3 (4) | 34.4 (3) | 45.6 (11) | 42.8 (7) | 34.3 (2) | 41.0 (6) | 33.6 (1) | 55.0 (15) | 75.3 (23) | 44.7 (10) | 69.3 (22) | 56.9 (16) | 52.7 (13) | 65.8 (20) | 64.5 (19) | 46.8 (12) | 44.1 (8) | 60.4 (18) | 67.8 (21) |
| Rank sum ↓ | 56 (7) | 60 (8) | 52 (6) | 45 (5) | 31 (3) | 71 (11) | 77 (12) | 39 (4) | 16 (1) | 66 (10) | 27 (2) | 56 (7) | 63 (9) | 45 (5) | 82 (13) | 111 (18) | 111 (18) | 113 (19) | 107 (17) | 93 (14) | 100 (15) | 129 (20) | 106 (16) |

**Global ranking of foundation models** — We propose a global ranking of studied foundation models by aggregating the quantitative results from different tasks. To this end, we rank the models for each of them independently, and sum task-specific rankings to obtain a final global ranking. We consider: (*i*) average knn F1-score, (*ii*) average linear probing F1-score, (*iii*) average 16-shot F1-score, (*iv*) average ECE after linear probing, and (*v*) average adversarial attack F1-score drop ($\epsilon = 1.5.10^{-3}$). As shown in Table 4, ranks vary between tasks, with however certain models showing consistent strong performance across them, leading to a top-5 composed of *uni2h*, *virchow2*, *midnight*, *uni*, and *hopt1/keep*. In particular, *uni2h* performs very well on a majority of tasks. Interestingly, no vision-language model is present in the top-4, but the best vision-language model, i.e. *keep*, reaches $5^{th}$ rank (same rank as *hopt1*). We provide a more detailed discussion on quantitative results in appendix (D): we present how our results are aligned with findings from previous studies, how to leverage them to improve models in the long run and look more closely into intra-group discrepancies.

# 5   Conclusion

We present *THUNDER*, an efficient tile-level benchmark to compare foundation models for digital pathology. It currently includes 16 well-known datasets and 23 foundation models. Importantly, it comes with an open-source implementation allowing the evaluation of new foundation models. It also implements tasks for uncertainty estimation and robustness of backbone models, and a way to study their feature spaces to provide more interpretability. Lastly, we present a comprehensive study of the most recent state-of-the-art foundation models for histopathology leveraging all benchmark tasks to draw a clearer picture of their strengths, weaknesses, and differences.

**Limitations** — Currently *THUNDER* only includes H&E stained data, but it could be extended to support other staining protocols (e.g., IHC). Additionally, the datasets considered in this benchmark can introduce biases that are inherent to the gathering protocol. We have included well-studied datasets in the field that have been utilized by many studies dealing with evaluating pathology foundation models as they are the best quality patch-level datasets currently available. While they can still bring an interesting signal about differences between existing foundation models, they have been extensively studied and used which could lead to performance saturation. *THUNDER* is thought of as an evolving benchmark, adapting to the direction the digital pathology community goes toward, and we will keep integrating new relevant datasets when they will be released in the future. Lastly, while the benchmark currently allows gaining insights about the feature spaces of models, we do not study how to combine them to improve performance further.

**Broader impact** — A deep learning benchmark in digital pathology can accelerate research by highlighting state-of-the-art methods and identifying performance gaps, ultimately improving clinical decision-making and patient outcomes. However, introducing such tools into clinical practice requires careful validation, regulatory approval, and consideration of ethical challenges to ensure safety.

## Acknowledgments

This work has been partially supported by *ANR-23-IAHU-0002*, *ANR-21-CE45-0007*, *ANR-23-CE45-0029*, and the *Health Data Hub* (*HDH*) as part of the second edition of the *France-Québec* call for projects *Intelligence Artificielle en santé*. It was performed using computational resources from the *Mésocentre* computing center of *Université Paris-Saclay*, *CentraleSupélec* and *École Normale Supérieure Paris-Saclay* supported by *CNRS* and *Région Île-de-France*, and from *GENCI-IDRIS* (*Grant 2025-AD011016068*).

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

# Appendix

## A   Included foundation models

Following [46], Table S1 presents the different foundation models currently supported by *THUNDER* and studied in the main paper. A detailed comparison including the main architecture used, the number of parameters and the training strategy as well as details about the training data are highlighted for each model. Importantly, our benchmark is not restricted to these models, as any custom model can be evaluated easily.

## B   Additional runtimes

**Runtime of feature space study tasks** — Table S2 presents the runtime of feature space study tasks. As can be seen, such runtimes are fairly low, and thus allow to efficiently and easily study the feature space of a foundation model.

**Per-model embedding pre-computing runtimes** — are provided in Table S3. As can be seen, differences are large between models, mainly depending on their size, but also diverse transformations applied to input images and implementation choices could explain some variations. Most tasks on the benchmark are performed from pre-computed embeddings, and variations between models thus become smaller, which is why we do not provide per-model runtimes for them.

## C   Comparison with existing benchmarks

As presented in our related work, several papers addressed the benchmarking of foundation models for histopathology. However, only a small subset of them comes with an open-source implementation. We can consider the four following open-source benchmarks as comparison points on which our main differences can be summarized as follows: **(i) eva** [22] includes both tile and slide level tasks, and evaluates models on linear probing (classification) and semantic segmentation. The eva benchmark provides less datasets, but also much less tasks and metrics (only focusing on balanced accuracy for linear probing and dice score for segmentation) than we do. **(ii) PathoBench** [80] focuses on slide-level classification and regression tasks (Morphological subtyping, Tumor grading, Molecular subtyping, Mutation prediction, Treatment response and assessment, Survival prediction). PathoBench is a slide-level benchmark, also focusing only on downstream performance, to which we are complementary as we propose a faster and more direct evaluation directly at the level of tiles. **(iii) HEST-Benchmark** [32] targets gene expression regression at the tile level. While being interesting and relevant, this is more specific than the diverse tasks we propose in *THUNDER*. **(iv) PathBench** [50] presents the slide-level performance of foundation models for diverse classification and regression (DFS, DSS, OS prediction) tasks, but does not come with an open-source tool to evaluate a new custom model (only an online open-source leaderboard is provided).

The added value of our benchmark is 3-fold: **(i)** an open-source easy-to-use implementation to seamlessly download datasets, models, generate common train/val/test splits and run any downstream task with automatic hyperparameter search and report performance along with bootstrap confidence intervals on an independent test set, **(ii)** a breadth of tasks going beyond downstream performance only, also providing tools to compare representation spaces of models and study their robustness, **(iii)** a patch-level framework allowing fast evaluation on many diverse datasets decoupling model embeddings from slide-level aggregation techniques. It also enables full reproducibility of the benchmark, which is challenging at the slide level due to required pre-processing steps.

## D   Extended discussion on quantitative results

**Alignment with previous findings** — The conclusions drawn from our benchmark align with previous studies: **(i) Better performance of pathology-specific pretrained models.** First, we show that models pre-trained on pathology images outperform the ones trained on natural images, which had been shown in previous work [9]. **(ii) Strong performance of recent vision-only models.** The high performance of recent vision-only models such as *uni*, *uni2h*, *virchow*, *virchow2*, *hoptimus0*,

Table S1: **Foundation models already included in *THUNDER*** are presented and grouped depending on their pretraining scheme. Training data – TC: Tile-Caption pairs, WR: WSI-Report pairs, TT: Text tokens, C: captions.

| Name | Short name | Vision arch. | Params. | Training method | #Slides | #Tiles | Text | Magn. | Source |
|---|---|---|---|---|---|---|---|---|---|
| *Vision-only, histopathology pretrained* | | | | | | | | | |
| HIBOU-B [54] | hiboub | ViT-B/14 | 86M | DINOv2 | 1.1M | 512M-1.2B | − | 20× | Private |
| HIBOU-L [54] | hiboul | ViT-L/14 | 307M | DINOv2 | 1.1M | 512M-1.2B | − | 20× | Private |
| H-OPTIMUS-0 [61] | hopt0 | ViT-G/14 | 1.1B | DINOv2 | 500K | − | − | − | Private |
| H-OPTIMUS-1 | hopt1 | ViT-G/14 | 1.1B | DINOv2 | 1M | − | − | − | Private |
| MIDNIGHT [35] | midnight | ViT-G/14 | 1.1B | DINOv2 | 12K | − | − | − | TCGA |
| PHIKON [17] | phikon | ViT-B/16 | 86M | iBOT | 6.1K | 43.4M | − | 20× | TCGA |
| PHIKON2 [18] | phikon2 | ViT-L/16 | 307M | DINOv2 | 60K | 456M | − | 20× | TCGA, GTEx, Private |
| UNI [10] | uni | ViT-L/16 | 307M | DINOv2 | 100K | 100M | − | 20× | GTEx, Private |
| UNI2-H [10] | uni2h | ViT-H/14 | 681M | DINOv2 | 350K | 200M | − | 20× | Private |
| VIRCHOW [73] | virchow | ViT-H/14 | 632M | DINOv2 | 1.5M | 2B | − | 20× | Private |
| VIRCHOW2 [83] | virchow2 | ViT-H/14 | 632M | DINOv2 | 3.1M | 2B | − | 5−40× | Private |
| *Vision-language, histopathology pretrained* | | | | | | | | | |
| CONCH [48] | conch | ViT-B/16 | 86M | CoCa, iBOT | 21K | 16M | 1.17M TC | 20× | PMC OA, Private |
| CONCH 1.5 [13] | titan | ViT-L/16 | 307M | CoCa | 336K | − | 423K TC, 183K WR | 20× | GTEx, Private |
| KEEP [82] | keep | ViT-L/16 | 307M | CLIP | − | − | 143K TC | − | Quilt1M, OpenPath |
| MUSK [78] | musk | V-FFN | 202M | CoCa, BEiT-3 | − | 50M | 1B TT, 1M TC | 10−40× | PMC OA, TCGA, Quilt1M, PathCap |
| PLIP [27] | plip | ViT-B/32 | 86M | CLIP | − | − | 208K TC | − | Twitter, PathLAION |
| QUILTNET [29] | quilt | ViT-B/32 | 86M | CLIP | − | 438K | 802K C | 10−40× | Quilt1M |
| *Vision-only, natural image pretrained* | | | | | | | | | |
| DINOv2-B [57] | dinob | ViT-B/14 | 86M | DINOv2 | − | − | − | − | − |
| DINOv2-L [57] | dinol | ViTL/14 | 307M | DINOv2 | − | − | − | − | − |
| ViT-B/16 [14] | vitb | ViT-B/16 | 86M | Imagenet | − | − | − | − | − |
| ViT-L/16 [14] | vitl | ViT-L/16 | 307M | Imagenet | − | − | − | − | − |
| *Vision-language, natural image pretrained* | | | | | | | | | |
| CLIP-B/32 [58] | clipb | ViT-B/32 | 86M | CLIP | − | − | − | − | − |
| CLIP-L/14 [58] | clipl | ViT-L/14 | 307M | CLIP | − | − | − | − | − |

Table S2: **Runtime of feature space study tasks**. † denotes tasks using pre-computed embeddings.

| Runtime | Feature space alignment† | Image retrieval† | Transformation invariance |
|---|---|---|---|
| Min. | 00h01 | 00h07 | 00h07 |
| Max. | 00h10 | 00h57 | 00h27 |
| Avg. | 00h07 | 00h20 | 00h11 |
| Cumulative | 01h20 | 04h00 | 02h18 |
| (Nb. datasets) | (12) | (12) | (11) |
| Hardware | ×32 CPUs | ×32 CPUs | ×1 V100 |

*hoptimus1* models trained with a DiNOv2 training objective was showcased in previous experimental studies [22, 32, 50] both at tile and slide levels. We also confirm this, in particular showing that newer versions of these models (e.g. *uni2h* or *virchow2*) outperform previous versions. **(iii) Competitive performance of VLM models.** VLMs such as *conch* and *titan* (*conch1.5*) were also highlighted in previous work [55], which is also the case in our work, along with the newer *keep* model that performs well on many different tasks.

In addition to confirming findings in previous work, we also go one step further by considering very recent models (e.g. the competitive *midnight* and *keep*) that were not included in previous benchmarks, and also new tasks (feature space alignment, calibration, robustness).

**Explaining differences in performance and how to improve models** — We provide additional insights into differences between foundation models that are currently included in the benchmark: **Impact of SSL methods.** Among vision-only models, it appears that the only model trained with iBot (*phikon*) performs worse than all others trained with DINOv2. This might indicate the superiority of DINOv2 as SSL pretraining strategy, which could be confirmed by its large adoption across most foundation models. However, it should be noted that this could also be partly explained by differences in training data. **Impact of datasets.** Indeed, generally, models trained from large and diverse datasets such as *uni2h* or *virchow2* have higher performance. Interestingly, on the other hand, *midnight* appears as an exception because it reaches strong performance while being trained on the smaller TCGA dataset only. Going forward into analyzing the impact of different characteristics of pre-training data on final performance is quite difficult since most models (e.g. *uni2h* or *virchow2*) are partly trained on private data. **Impact of number of parameters of the models.** The number of model parameters can also play a role in final performance: the trend is a bit clearer within VLMs

Table S3: **Embedding pre-computing runtimes** on the 12 classification datasets.

| Model | Avg. | Cumulative |
|---|---|---|
| dinob | 00h23 | 04h32 |
| vitb | 00h24 | 04h49 |
| quiltnet | 00h25 | 04h57 |
| phikon | 00h25 | 04h58 |
| plip | 00h25 | 05h00 |
| clipb | 00h27 | 05h25 |
| hiboub | 00h27 | 05h27 |
| vitl | 00h36 | 07h10 |
| phikon2 | 00h37 | 07h19 |
| uni | 00h37 | 07h20 |
| keep | 00h40 | 07h54 |
| dinol | 00h44 | 08h45 |
| hiboul | 00h47 | 09h20 |
| clipl | 00h47 | 09h28 |
| conch | 00h47 | 09h29 |
| virchow | 01h22 | 16h25 |
| virchow2 | 01h24 | 16h44 |
| uni2h | 01h25 | 17h00 |
| hopt1 | 02h05 | 24h58 |
| titan | 02h12 | 26h22 |
| midnight | 02h15 | 26h56 |
| hopt0 | 03h05 | 37h01 |
| musk | 06h45 | 81h04 |

where models with more parameters, i.e. *keep* and *titan* (*conch1.5*), seem to outperform others on downstream task performance (*knn*, *linear probing*, *few-shot*).

These insights can serve as pointers for the development of future foundation models, and we hope *THUNDER* can be used as a tool to strengthen such conclusions provided more models and datasets in the future. Moreover, an interesting finding is the potential gain a simple LoRA adaptation can bring to tasks with small datasets, even for strong foundation models, as shown in Table 3. Efficient adaptation of foundation models thus appears as a relevant direction. Improving foundation models also requires a better understanding of their inner mechanisms. We believe in the power of alignment metrics such as *Mutual knn* and hope to provide a common framework for researchers to extend our study. Indeed, the alignment graphs we could build along with the evolution of alignment during LoRA adaptation are quite intriguing and deserve more studies in future work.

# E   Implementation details

**knn classification** — Distance is measured using cosine similarity and the best $k$ value is validated among $\{1, 3, 5, 10, 20, 30, 40, 50\}$ on a validation set.

**Linear probing** — hyperparameters are summarized in Table S4.

**Few-shot classification** — leverages the SimpleShot [75] method. More specifically, we consider access to a support (train) set composed, for each class $c \in \mathcal{C}$ where $\mathcal{C}$ is the set of all classes, of $N_c$ samples. The goal is then to predict the class of each sample within a query (test) set. We first center all support and query embeddings by subtracting the support set embedding mean. Then, a prototype embedding is computed for each class by taking the mean of class-specific centered embeddings. Finally, for a given query embedding, the predicted class is the one associated with the closest centroid. In our experiments, we consider the following numbers of shots $N_s$: $\{1, 2, 4, 8, 16\}$.

**Semantic segmentation** — hyperparameters are summarized in Table S4. We perform different numbers of epochs depending on the size of the datasets: 200 epochs for *ocelot* and *pannuke*, respectively 9 and 21 for *segp-ep* and *segp-ly* as those two datasets are much larger. We leverage the Segmenter [67] decoder as our segmentation probe. It is a Transformer-based decoder fed with token embeddings from the vision encoder and one learned token for each class. The final segmentation is performed by computing a scalar product between spatial token representations and token embeddings.

Table S4: **Training-based downstream task hyperparameters**

| Hyperparameter | Linear probing | Linear probing + LoRA | Segmentation |
|---|---|---|---|
| Optimizer | Adam | Adam | Adam |
| Loss | Cross Entropy | Cross Entropy | Dice |
| Batch size | 64 | 2 | 32 |
| Epochs | 200 | 20 | $\leq 200$ |
| Searched learning rates | $\{10^{-3}, 10^{-4}, 10^{-5}\}$ | $\{10^{-3}, 10^{-4}, 10^{-5}\}$ | $\{10^{-3}, 10^{-4}, 10^{-5}\}$ |
| Searched weight decays | $\{0, 10^{-3}, 10^{-4}\}$ | $\{0, 10^{-3}, 10^{-4}\}$ | $\{0, 10^{-3}, 10^{-4}\}$ |
| Probe | Linear | Linear | Segmenter [67] |

This is followed by a bilinear interpolation to reach the required segmentation mask size. In terms of hyperparameters, we use 2 Transformer layers, with 8 attention heads, and an internal representation size of 768. The reported test performance is averaged across test patches, with a reduced weight for the ones containing only *background* pixels.

**Feature space alignment** — *THUNDER* supports different feature alignment metrics ([65, 59, 44], and additional knn-based metrics introduced by [28]). The main one we focus on in this paper is *Mutual knn* [28], which measures the average size of the intersection of nearest-neighbors sets for different query samples between two foundation models. Following notations from [28], let us consider two models $f$ and $g$. Provided an input $x_i$, $\phi_i = f(x_i)$ and $\psi_i = g(x_i)$ are the respected extracted embeddings. $\Phi$ and $\Psi$ are the sets of all embeddings of a set of samples $\{x_i\}_{i=[1,\cdots,N]}$, and $d_{\mathrm{knn}}$ is the function returning the set of indices of $k$-nearest neighbors of an embedding as follows,

$$d_{\mathrm{knn}}\left(\phi_i, \Phi \backslash \phi_i\right) = \mathcal{S}\left(\phi_i\right),$$

$$d_{\mathrm{knn}}\left(\psi_i, \Psi \backslash \psi_i\right) = \mathcal{S}\left(\psi_i\right).$$

The *Mutual knn* function $m_{\mathrm{knn}}$ is then defined as,

$$m_{\mathrm{knn}}\left(\phi_i, \psi_i\right) = \frac{1}{k}\left|\mathcal{S}\left(\phi_i\right) \cap \mathcal{S}\left(\psi_i\right)\right|.$$

In our experiments, we pick $k = 10$.

**Image retrieval** — We sort all training samples based on their cosine similarity with query test samples. We compute top-1, top-3, top-5, top-10 classification metrics (F1-score, balanced accuracy), but more importantly, provide qualitative visualizations of top-10 retrieved samples to compare foundation model feature spaces.

**Invariance to image transformations** — We consider a set of images $\{x_i\}_{i=[1,\cdots,N]}$. For each image $x_i$, we sample a stochastic transformation $\tau$ described in Table S5, compute the embeddings $z_i = f_\theta(x_i)$ and $z_i^\tau = f_\theta(\tau(x_i))$, and measure their agreement with cosine similarity. In our experiments, we pick $N = 1000$ as the number of images sampled from each dataset.

**LoRA adaptation** — LoRA [26] is a parameter-efficient finetuning (PEFT) method, which consists in freezing the whole pretrained Transformer model and introducing small trainable modules. Specifically, the LoRA adapters are introduced in parallel to the query and value branches of the multi-head attention of each Transformer encoder layer. With a feature dimension $d \in \mathbb{N}_*^+$ and a rank $r < d \in \mathbb{N}_*^+$, the only trainable modules are $\mathbf{A} \in \mathbb{R}^{r \times d}$ and $\mathbf{B} \in \mathbb{R}^{d \times r}$. The forward pass through the linear layer for computing the query and value components is transformed from $\mathbf{h} = \mathbf{W}_0\mathbf{x}$ to $\mathbf{h} = \mathbf{W}_0\mathbf{x} + \frac{\alpha}{r}\mathbf{W}_{\mathrm{LoRA}}\mathbf{x}$, for a scaling hyperparameter $\alpha \in \mathbb{R}$ with,

$$\Delta\mathbf{W}_{\mathrm{LoRA}}\mathbf{x} = \mathbf{B}\mathbf{A}\mathbf{x}.$$

We use $\alpha = 16$ and $r = 16$, other hyperparameters are summarized in Table S4.

**Calibration** — metrics are computed on classifiers trained during linear probing experiments. For all metrics, predictions are divided into $\mathcal{B}$ bins based on their confidence. Let us denote $y_i$, $\hat{y}_i$ and $p_i$

Table S5: **Stochastic image transformations** used to compute embedding transformation invariance.

| Transformation | Description | Sampling range |
|---|---|---|
| Crop | Crop randomly from 4 corners or center | Crop side size : $\min(H, W)/2$ |
| Elastic | Elastic deformation | Amplitude $\alpha = 250$, smoothing $\sigma = 6$ |
| Dilation | Morphological dilation | Square kernel $k \in \{3, 5\}$ |
| Erosion | Morphological erosion | Square kernel $k \in \{3, 5\}$ |
| Opening | Erosion then dilation | Same $k$ as above |
| Closing | Dilation then erosion | Same $k$ as above |
| Blur | Gaussian blur | Fixed $15 \times 15$ kernel |
| Jitter | Brightness/contrast/saturation/hue jitter | $b, c, s \sim U[0.5, 1.5]$, $h \sim U[-0.35, 0.35]$ |
| Translate | Random affine shift/scale/shear | $\|\Delta x\| \leq W/5$; $\|\Delta y\| \leq H/5$; scale $\in [0.8, 1.2]$; shear $\in [-1, 1]$ |
| Cutout | Random square mask of zeros | Square side size : $u \sim U[0.1, 0.5] \min(H, W)$ |
| HED | Histology colour perturbation in HED space [16] | - |
| RandStain | Unified stain normalization and augmentation that normalizes and perturbs stain appearance [64] | - |
| Flip | Horizontal *or* vertical flip | Probability 0.5 each |
| Rotate | Rigid rotation | Angle $\in \{90°, 180°, 270°\}$ |
| Gamma | Power-law intensity transform | $\gamma \sim U[0.5, 1.5]$ |

as the respective class prediction, class ground-truth and confidence for a sample $x_i$. For each bin $B_b$, we can then compute the average confidence and accuracy of samples within it as,

$$\text{Acc}(B_b) = \frac{1}{|B_b|} \sum_{i \in B_b} \mathbb{1}(\hat{y}_i = y_i)$$

$$\text{Conf}(B_b) = \frac{1}{|B_b|} \sum_{i \in B_b} p_i$$

Calibration metrics are then defined as follows,

- **ECE :** It measures the average discrepancy between confidence and accuracy. Each bins contribution is weighted by the number of samples in that bin.

$$\text{ECE} = \sum_{b=1}^{\mathcal{B}} \frac{|B_b|}{N} |\text{Acc}(B_b) - \text{Conf}(B_b)|$$

- **MCE :** It captures the worst-case mis-calibration. Unlike ECE, which takes an average, MCE focuses on the largest deviation between accuracy and confidence across all bins.

$$\text{MCE} = \max_b |\text{Acc}(B_b) - \text{Conf}(B_b)|$$

- **SCE :** It is similar to Expected Calibration Error (ECE) but treats all bins equally. This makes SCE more robust to class imbalance and ensures a fair contribution from all bins.

$$\text{SCE} = \frac{1}{\mathcal{B}} \sum_{b=1}^{\mathcal{B}} |\text{Acc}(B_b) - \text{Conf}(B_b)|$$

- **ACE :** It is an extension of ECE that uses adaptive binning. Instead of using fixed confidence intervals, ACE ensures that each bin has an equal number of samples.

$$\text{ACE} = \frac{1}{\mathcal{B}} \sum_{b=1}^{\mathcal{B}} |\text{Acc}(B_b) - \text{Conf}(B_b)|$$

- **TACE :** It is a modification of ACE that focuses only on high-confidence predictions. Predictions with confidence scores below a certain threshold are ignored. We thus only consider the $\mathcal{B}^*$ bins associated with highest confidence. This is useful in high-stakes applications where only confident predictions are acted upon, e.g., medical imaging.

$$\text{TACE} = \frac{1}{\mathcal{B}^*} \sum_{b \in \mathcal{B}^*} |\text{Acc}(B_b) - \text{Conf}(B_b)|$$

Table S6: **Impact of the architecture of the classifier on calibration performance** (ECE): linear classifier vs MLP – Vision and Vision-Language models.

| Decoder | Histopathology models | | | | | | | | | | | | | | | | | Natural-image models | | | | | |
|---|---|---|---|---|---|---|---|---|---|---|---|---|---|---|---|---|---|---|---|---|---|---|---|
| | hiboub | hiboul | hopt0 | hopt1 | midnight | phikon | phikon2 | uni | uni2h | virchow | virchow2 | conch | titan | keep | musk | plip | quilt | dinob | dinol | vitb | vitl | clipb | clipl |
| Linear | 3.7 (2) | 5.5 (18) | 4.7 (13) | 4.1 (4) | 3.2 (1) | 6.4 (22) | 4.6 (11) | 4.3 (7) | 4.5 (8) | 5.5 (20) | 4.6 (10) | 4.3 (6) | 4.9 (14) | 4.7 (12) | 4.5 (9) | 4.9 (15) | 7.0 (23) | 5.5 (21) | 5.3 (17) | 3.9 (3) | 5.0 (16) | 5.5 (19) | 4.2 (5) |
| MLP | 6.0 (4) | 5.6 (3) | 6.8 (9) | 6.8 (10) | 6.2 (5) | 7.4 (16) | 9.0 (17) | 7.0 (11) | 6.5 (7) | 7.3 (15) | 6.3 (6) | 4.5 (1) | 7.3 (14) | 5.3 (2) | 6.7 (8) | 7.2 (12) | 9.5 (19) | 9.2 (18) | 7.2 (13) | 11.1 (23) | 10.3 (22) | 10.2 (21) | 9.8 (20) |

We also generate reliability diagrams that visually represent how predicted probabilities align with actual correctness by plotting bin average accuracies as a function of bin average confidences. A perfectly calibrated model follows the $y=x$ line. If the curve is below the diagonal, the model is overconfident. Conversely, if the curve is above the diagonal, the model is under-confident.

**Robustness to adversarial attacks** — To assess the vulnerability of each frozen backbone $f_\theta(\cdot)$ and its linear probe $W$ to additive adversarial noise, we employ a **Projected Gradient Descent (PGD)** attack constrained in $\ell_\infty$ norm [51, 53, 19, 23]. Let $\mathbf{x}$ be a normalized image and $y \in \{1, \ldots, K\}$ its label. Image classification is performed as follows,

$$c(\mathbf{x}) = W\big(f_\theta(\mathbf{x})\big) \in \mathbb{R}^K.$$

PGD iteratively constructs a perturbation $\boldsymbol{\delta}_t$ that *maximizes* the cross-entropy loss $\mathcal{L}\big(c(\mathbf{x} + \boldsymbol{\delta}_t), y\big)$ while remaining inside the $\ell_\infty$ ball $\|\boldsymbol{\delta}\|_\infty \leq \varepsilon$:

$$\boldsymbol{\delta}_{t+1} = \Pi_{\|\boldsymbol{\delta}\|_\infty \leq \varepsilon}\Big(\boldsymbol{\delta}_t + \alpha \, \text{sign}\big(\nabla_{\mathbf{x}}\mathcal{L}(c(\mathbf{x} + \boldsymbol{\delta}_t), y)\big)\Big)$$

where $\alpha$ is the step size and $\Pi_{\|\boldsymbol{\delta}\|_\infty \leq \varepsilon}$ projects back onto the intersection of the $\ell_\infty$ ball.

We perform `num_steps` $= 5$ gradient steps and keep the network in `eval()` mode so that only input gradients are computed. We use three perturbation budgets $\varepsilon \in \{0.25 \times 10^{-3}, \ 1.5 \times 10^{-3}, \ 35 \times 10^{-3}\}$ and record the average **F1-score drop**:

$$\Delta\text{F1}(\varepsilon) = \text{F1}_{\text{clean}} - \text{F1}_{\text{adv}}(\varepsilon)$$

Here, $\text{F1}_{\text{clean}}$ denotes the F1-score obtained on clean (i.e., unperturbed) test samples, while $\text{F1}_{\text{adv}}(\varepsilon)$ denotes the F1-score computed on the same samples after being perturbed by the PGD attack with budget $\varepsilon$.

To ensure a fair comparison and efficient evaluation, both scores are computed over the same set of up to (depending on each dataset test set size) 10,000 randomly selected test samples.

## F  Impact of the choice of probe on calibration performance

Table S6 shows the difference in calibration of two different classification heads, i.e. a linear classifier which is our default choice in this paper and an MLP with a single hidden layer with a fixed 256-dim, trained on top of embeddings from the different considered foundation models. As can be seen, calibration is probe-dependent, rankings vary when changing the nature of the trained classifier. However, we believe the linear classifier is the best default choice as it has no intermediate representation and thus less expressive power, which makes it a good candidate to assess the impact of extracted embeddings themselves on calibration.

## G  Different pre-processing for the *bracs* dataset

Images from the *bracs* dataset vary in size and are on average quite large compared with other datasets. We thus provide an alternative pre-processing method for this specific dataset: instead of extracting embeddings from the full image as input, we divide it into $512 \times 512$ patches (as the magnification is $40\times$), extract one embedding per patch and aggregate them with a mean pooling operation to obtain a final embedding. Table S7 presents the difference in *knn* performance when using the full image as input vs dividing into patches with mean pooling. We provide the latter as an alternative variant in the benchmark but keep the standard approach, i.e. using the full image, as done for other datasets, in all our experiments.

Table S7: **Comparison between image pre-processing options** for the *bracs* dataset on the *knn* task (F1-score).

| Pre-proc. | hiboub | hiboul | hopt0 | hopt1 | midnight | phikon | phikon2 | uni | uni2h | virchow | virchow2 | conch | titan | keep | musk | plip | quiltnet | dinob | dinol | vitb | vitl | clipb | clipl |
|---|---|---|---|---|---|---|---|---|---|---|---|---|---|---|---|---|---|---|---|---|---|---|---|
| Full img. | 56.9 | 56.2 | 52.2 | 55.0 | 50.2 | 50.0 | 45.9 | 55.6 | 56.1 | 51.3 | 54.9 | 56.9 | 59.4 | 53.0 | 57.8 | 48.2 | 50.7 | 43.7 | 46.8 | 45.4 | 46.9 | 42.5 | 46.6 |
| Patches + pool. | 54.1 | 50.1 | 54.2 | 55.0 | 45.6 | 43.1 | 43.8 | 53.2 | 52.4 | 49.7 | 52.1 | 52.1 | 55.0 | 53.3 | 48.9 | 43.6 | 44.1 | 36.9 | 39.2 | 44.6 | 42.7 | 35.3 | 41.5 |

Table S8: **Additional experimental results**: Detailed results per task, model, dataset and overall are presented in the following material.

| Task | Aggregated | Per-dataset |
|---|---|---|
| knn | Fig. S4(a) and S5(a) , Tab. S11 and S12 | Tab. S36 and S37 |
| Linear probing | Fig. S4(b) and S5(b), Tab. S13 and S14 | Tab. S38 and S39 |
| Few-shot classification | Fig. S6, Tab. S15-S24 | Tab. S40-S49 |
| Segmentation | — | Tab. S50, S51 |
| Calibration | Tab. S25-S29 | Fig. S7, Tab. S52-S56 |
| Robustness to adversarial attacks | Tab. S30-S35 | Tab. S57- S62 |
| Image retrieval | — | Fig. S9-S19 |
| Feature space alignment | — | Fig. S20-S31 |
| Transformation invariance | Tab. S9 | Tab. S10 |
| LoRA adaptation | — | Fig. S8 |

# H   Additional experimental results

We provide additional results to complement those presented in the main paper. Table S8 refers to the different tables and figures that can be found below. Importantly, all tables present average performance (for different metrics specified in caption), and Tables  S36-S62 also report $95\%$ bootstrap confidence intervals (metric score [95% CI]) computed using the *percentile* method with 3000 resamples.

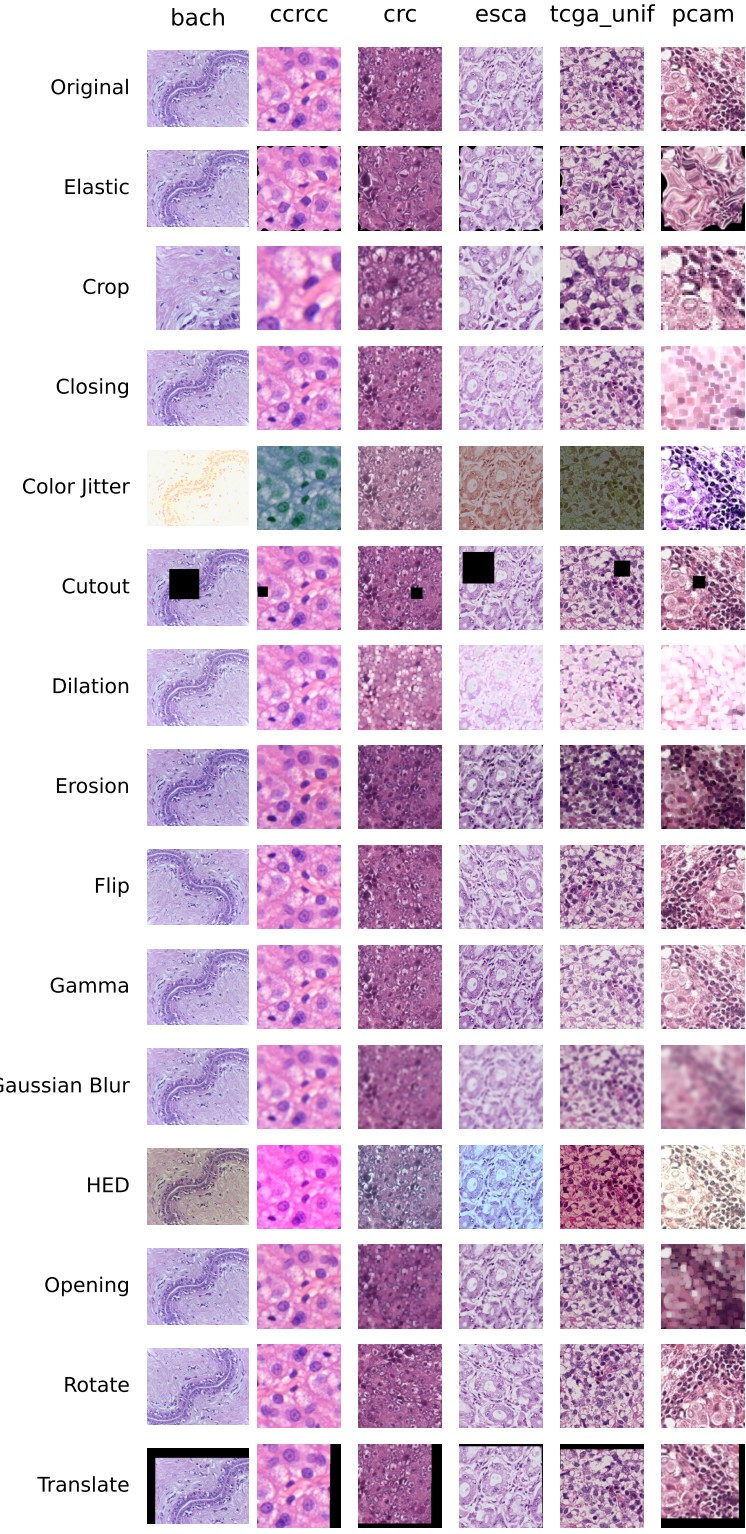

Figure S1: **Transformation invariance**: Visualization of transformations across datasets. One representative patch per dataset (*bach*, *ccrcc*, *crc*, *esca*, *tcga-unif*, and *pcam*) is shown under various transformations.

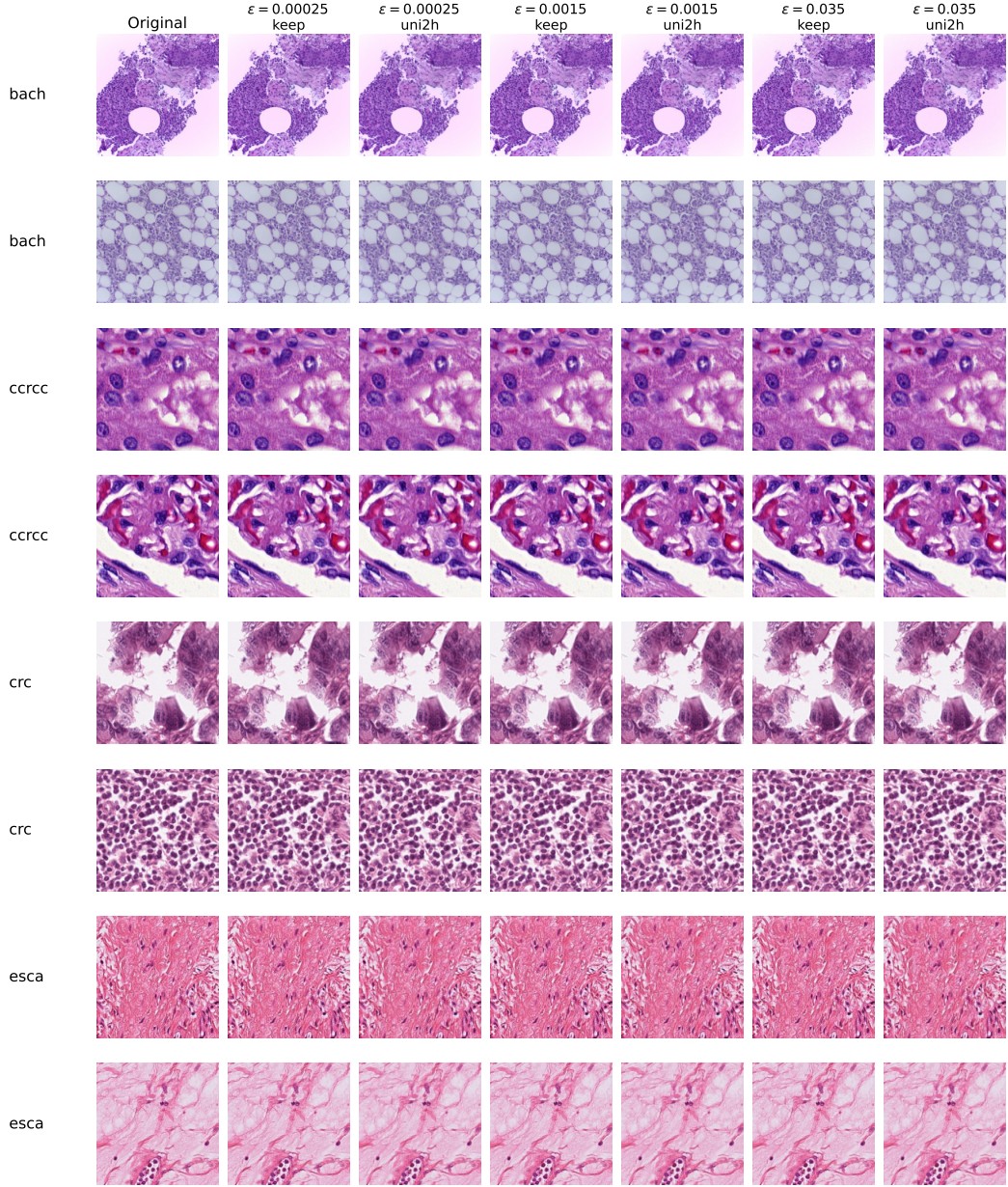

Figure S2: **Robustness to adversarial attacks**: Visualization of adversarial samples for the *bach*, *ccrcc*, *crc*, and *esca* datasets. Three different perturbation budgets ($\epsilon = 0.00025$, $0.0015$, and $0.035$) and two foundation models (*keep* and *uni2h*) are considered. Despite the increasing perturbation magnitude, no perceptually distinguishable differences are observable between the original and adversarial samples under visual inspection.

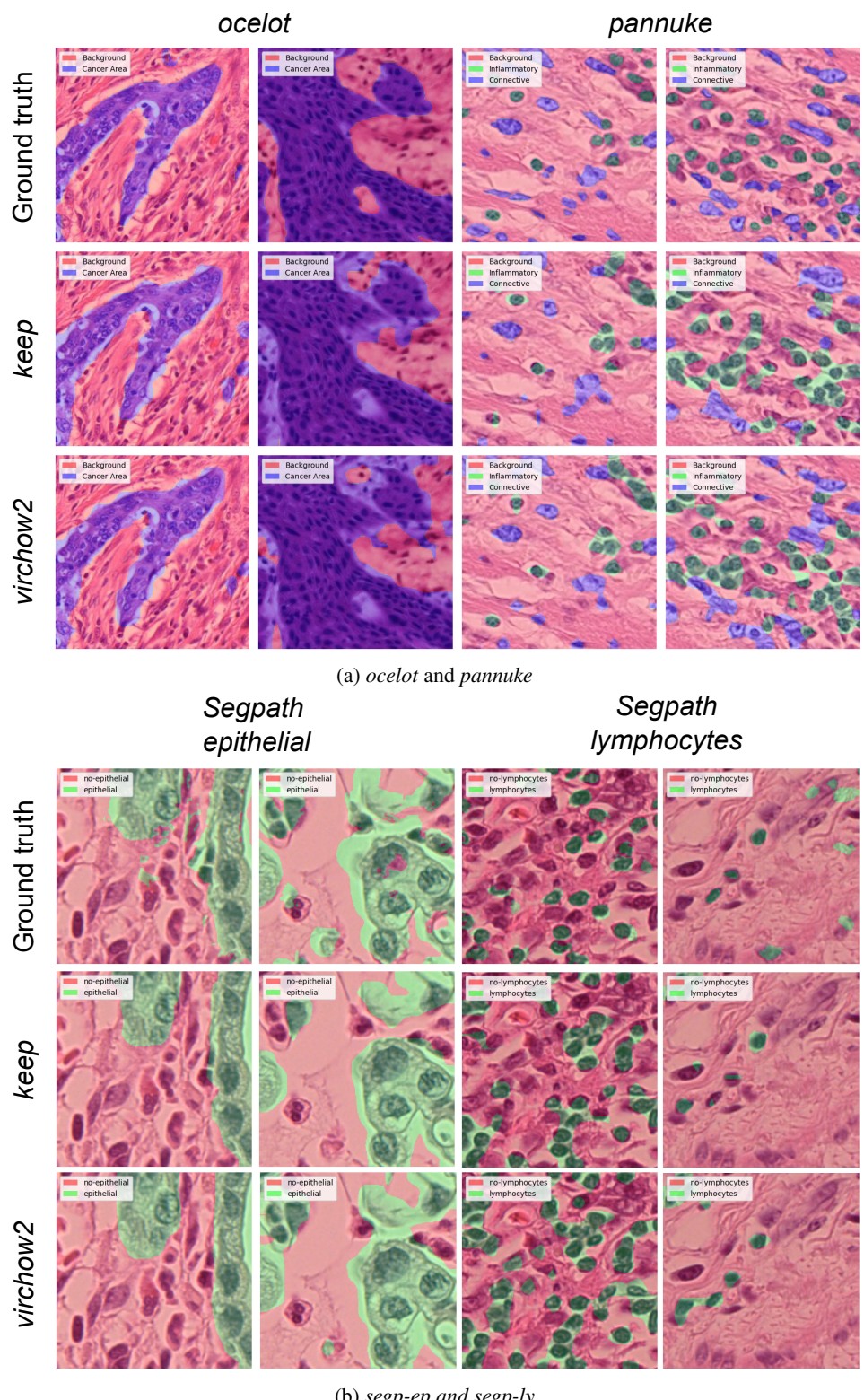

(a) *ocelot* and *pannuke*

(b) *segp-ep and segp-ly*

Figure S3: **Segmentation**: Visualization of segmentation samples for the *ocelot*, *pannuke*, *segp-ep*, and *segp-ly* datasets. Two foundation models (*keep* and *virchow2*) are considered.

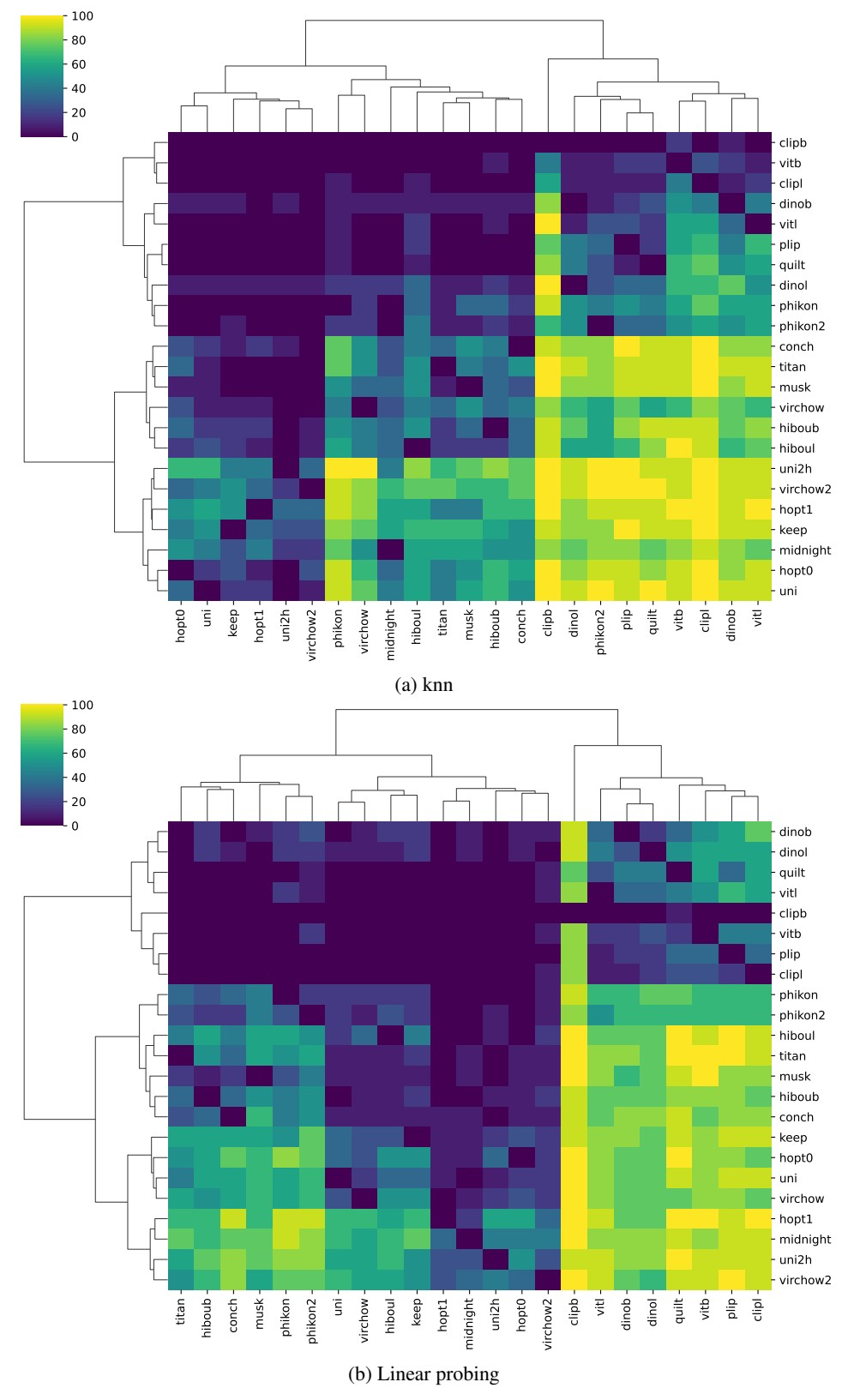

(a) knn

(b) Linear probing

Figure S4: **Performance comparison hierarchical clustering**: Hierarchical clustering of performance comparison heatmaps (Figure 2(a) and (b)) for *knn* and *linear probing* tasks.

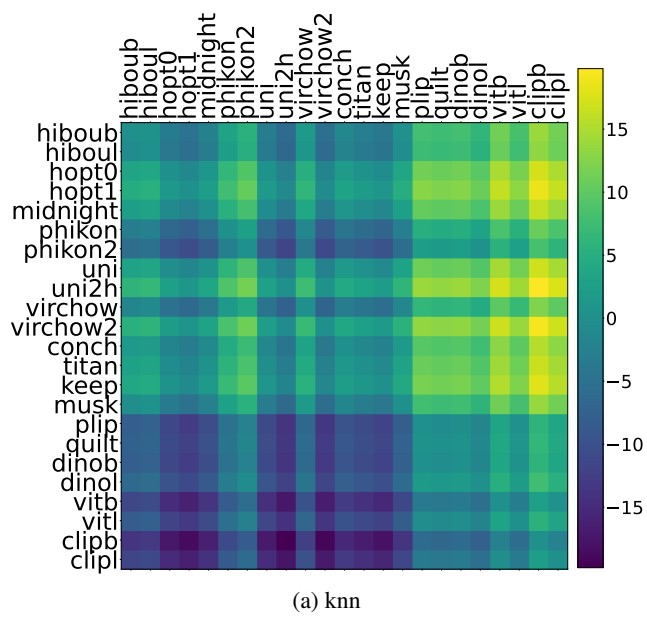

(a) knn

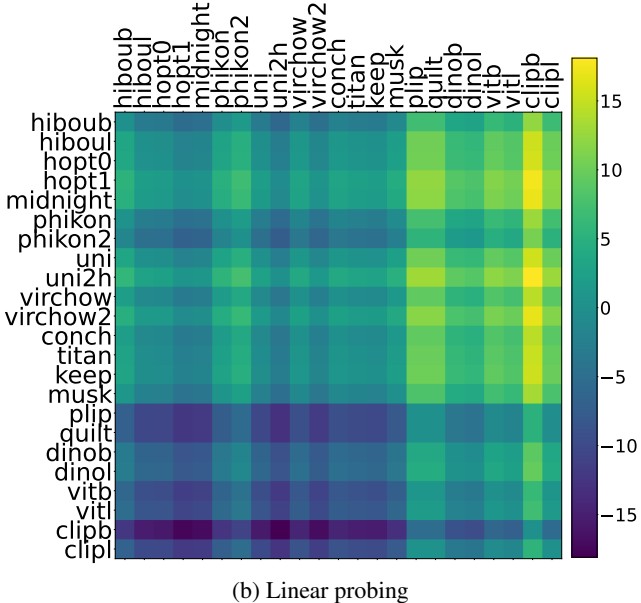

(b) Linear probing

Figure S5: **Gain heatmaps**: Heatmaps showing the difference between the average F1-score of row ($s_r$) and column ($s_c$) models ($s_r - s_c$) for the *knn* and *linear probing* tasks.

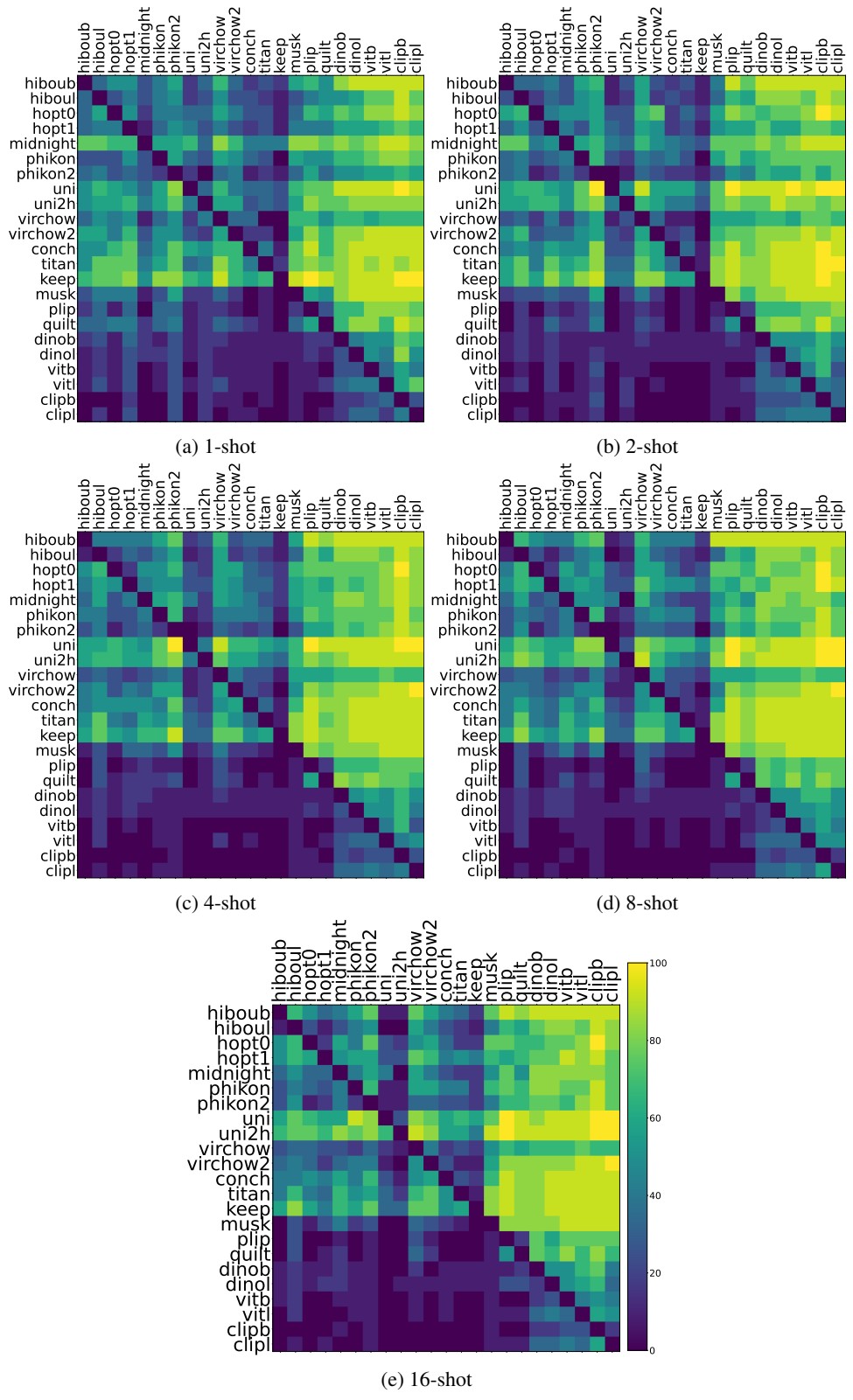

Figure S6: **Few-shot performance comparison heatmaps**: Heatmaps for different numbers of shots (1, 2, 4, 8, 16). Each cell shows the proportion of classification datasets where the row model is significantly better than the column model. This is assessed by performing a Binomial test on per-sample binary accuracies, followed by a Benjamini-Hochberg p-value correction for each pair of models.

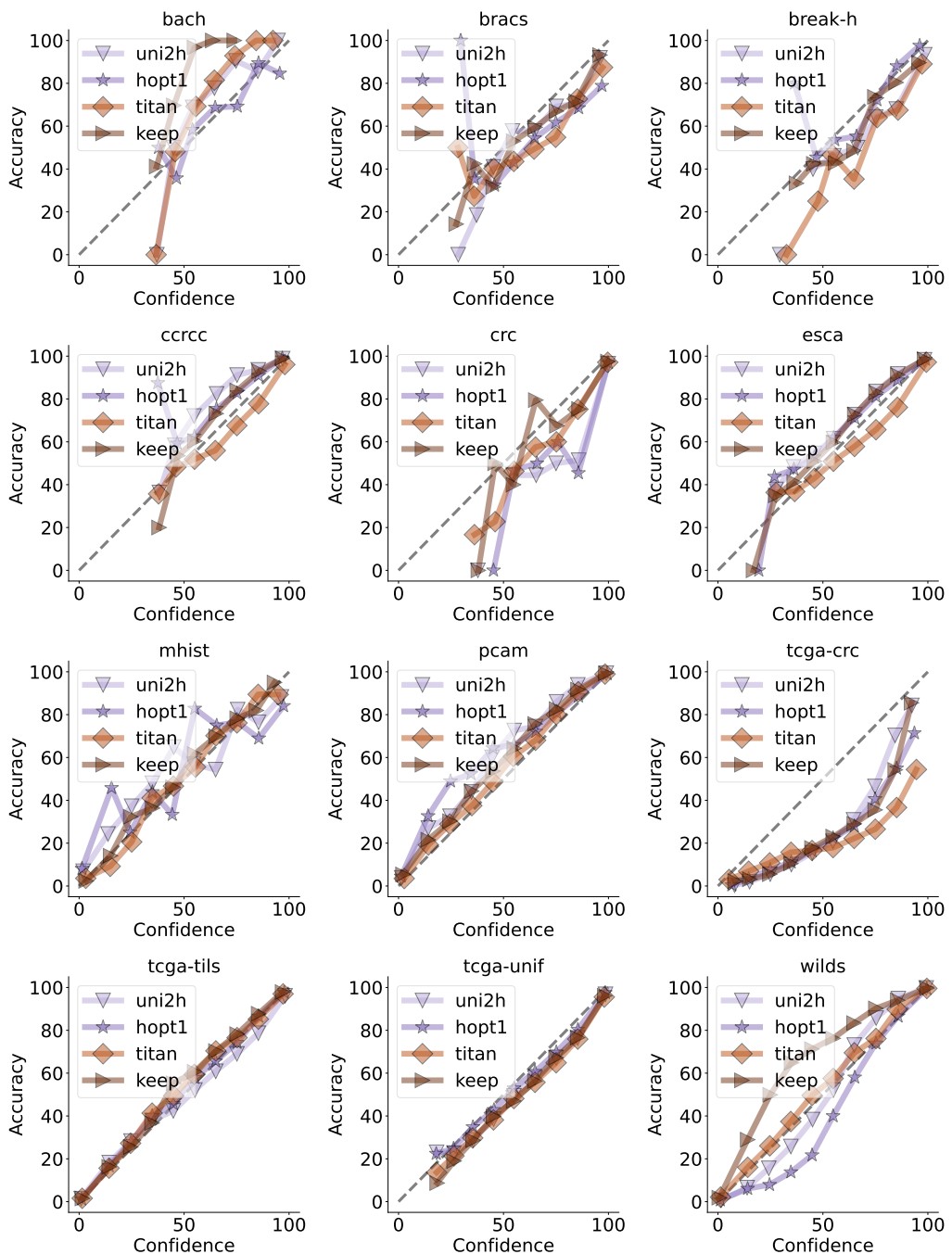

Figure S7: **Calibration**: Calibration curves for 4 selected models (*uni2h*, *hopt1*, *titan*, *keep*) on all datasets.

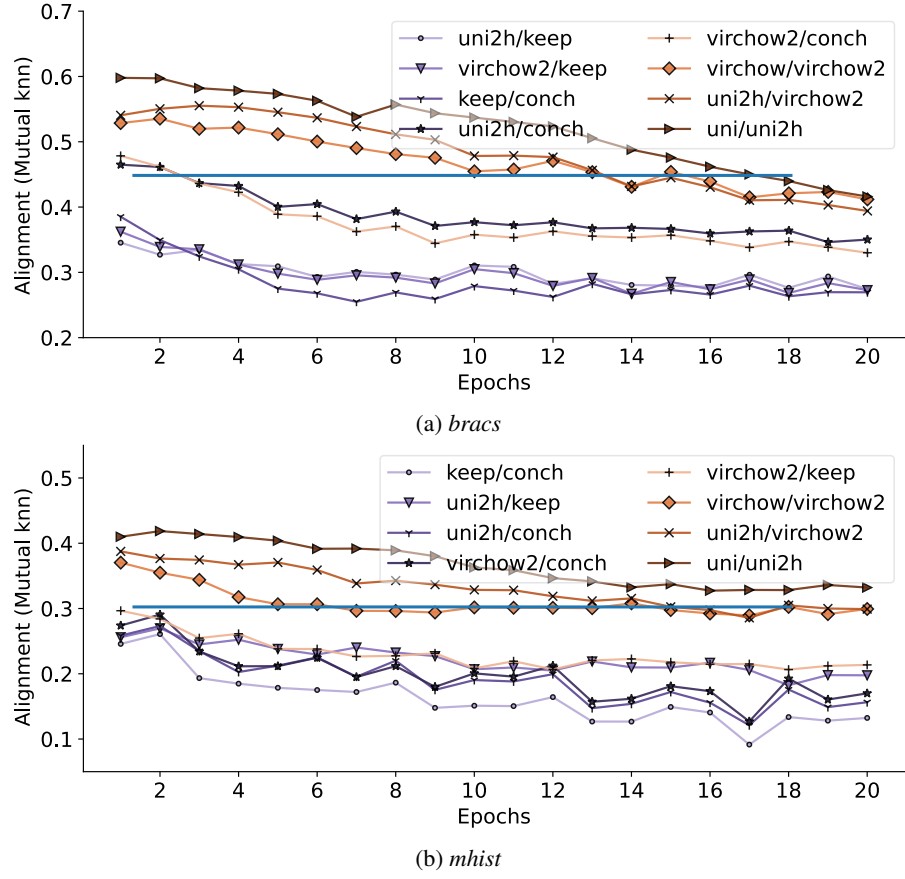

(a) *bracs*

(b) *mhist*

Figure S8: **Pair-wise alignment evolution**: Evolution of pair-wise alignment (Mutual knn) between models during LoRA adaptation for the *bracs* and *mhist* datasets.

Table S9: **Transformation invariance**: Cosine similarity between original embeddings and embeddings of the transformed image, averaged across classification datasets (with per-model and per-transform means)

| Model | Crop | Elastic | Dilation | Erosion | Opening | Closing | Gaussian | Jitter | Translate | Cutout | HED | RandStain | Flip | Rotate | Gamma | Mean |
|---|---|---|---|---|---|---|---|---|---|---|---|---|---|---|---|---|
| hopt0 | 0.53 | 0.38 | 0.41 | 0.37 | 0.50 | 0.43 | 0.52 | 0.83 | 0.84 | 0.89 | 0.90 | 0.87 | 0.97 | 0.96 | 0.97 | 0.69 |
| hopt1 | 0.56 | 0.40 | 0.45 | 0.39 | 0.53 | 0.46 | 0.63 | 0.84 | 0.88 | 0.95 | 0.92 | 0.89 | 0.98 | 0.97 | 0.97 | 0.72 |
| phikon2 | 0.61 | 0.49 | 0.53 | 0.52 | 0.61 | 0.57 | 0.60 | 0.76 | 0.82 | 0.94 | 0.80 | 0.76 | 0.95 | 0.90 | 0.95 | 0.72 |
| midnight | 0.55 | 0.31 | 0.44 | 0.47 | 0.61 | 0.53 | 0.63 | 0.86 | 0.82 | 0.94 | 0.91 | 0.87 | 0.99 | 0.98 | 0.97 | 0.73 |
| uni | 0.53 | 0.41 | 0.53 | 0.52 | 0.64 | 0.59 | 0.65 | 0.78 | 0.81 | 0.92 | 0.84 | 0.81 | 0.95 | 0.93 | 0.97 | 0.73 |
| uni2h | 0.61 | 0.40 | 0.53 | 0.48 | 0.65 | 0.57 | 0.69 | 0.87 | 0.84 | 0.93 | 0.92 | 0.90 | 0.92 | 0.89 | 0.98 | 0.75 |
| virchow | 0.73 | 0.46 | 0.55 | 0.57 | 0.65 | 0.62 | 0.65 | 0.84 | 0.79 | 0.88 | 0.92 | 0.90 | 0.98 | 0.96 | 0.97 | 0.76 |
| virchow2 | 0.61 | 0.45 | 0.54 | 0.48 | 0.63 | 0.56 | 0.63 | 0.91 | 0.86 | 0.97 | 0.95 | 0.93 | 0.99 | 0.99 | 0.99 | 0.77 |
| hiboub | 0.55 | 0.54 | 0.60 | 0.59 | 0.68 | 0.65 | 0.69 | 0.79 | 0.69 | 0.84 | 0.96 | 0.96 | 0.98 | 0.98 | 0.98 | 0.77 |
| musk | 0.82 | 0.45 | 0.54 | 0.55 | 0.60 | 0.60 | 0.75 | 0.85 | 0.87 | 0.93 | 0.96 | 0.94 | 0.99 | 0.98 | 0.98 | 0.79 |
| hiboul | 0.58 | 0.61 | 0.58 | 0.56 | 0.67 | 0.63 | 0.72 | 0.78 | 0.84 | 0.92 | 0.96 | 0.97 | 1.00 | 0.99 | 0.98 | 0.79 |
| keep | 0.68 | 0.52 | 0.58 | 0.59 | 0.72 | 0.66 | 0.73 | 0.77 | 0.86 | 0.96 | 0.92 | 0.89 | 0.98 | 0.96 | 0.98 | 0.79 |
| phikon | 0.66 | 0.60 | 0.65 | 0.66 | 0.74 | 0.71 | 0.70 | 0.77 | 0.88 | 0.96 | 0.83 | 0.80 | 0.97 | 0.94 | 0.96 | 0.79 |
| conch | 0.75 | 0.58 | 0.65 | 0.61 | 0.73 | 0.68 | 0.65 | 0.79 | 0.86 | 0.87 | 0.96 | 0.94 | 0.99 | 0.98 | 0.99 | 0.80 |
| titan | 0.79 | 0.65 | 0.73 | 0.74 | 0.79 | 0.78 | 0.71 | 0.77 | 0.88 | 0.89 | 0.94 | 0.92 | 0.97 | 0.97 | 0.99 | 0.84 |
| vitb | 0.82 | 0.66 | 0.77 | 0.79 | 0.83 | 0.82 | 0.83 | 0.68 | 0.90 | 0.94 | 0.92 | 0.90 | 0.98 | 0.96 | 0.99 | 0.85 |
| vitl | 0.85 | 0.72 | 0.82 | 0.84 | 0.86 | 0.87 | 0.83 | 0.71 | 0.91 | 0.92 | 0.91 | 0.90 | 0.98 | 0.97 | 0.99 | 0.87 |
| dinob | 0.88 | 0.53 | 0.75 | 0.84 | 0.80 | 0.85 | 0.86 | 0.96 | 0.96 | 0.98 | 0.99 | 0.98 | 0.99 | 0.98 | 1.00 | 0.89 |
| clipl | 0.89 | 0.71 | 0.81 | 0.86 | 0.85 | 0.86 | 0.83 | 0.90 | 0.91 | 0.91 | 0.97 | 0.95 | 0.99 | 0.98 | 0.99 | 0.89 |
| dinol | 0.92 | 0.49 | 0.77 | 0.85 | 0.83 | 0.86 | 0.88 | 0.96 | 0.97 | 0.98 | 0.99 | 0.98 | 0.99 | 0.99 | 1.00 | 0.90 |
| quilt | 0.88 | 0.86 | 0.87 | 0.87 | 0.89 | 0.89 | 0.86 | 0.83 | 0.94 | 0.99 | 0.93 | 0.91 | 0.99 | 0.98 | 0.98 | 0.91 |
| plip | 0.89 | 0.85 | 0.88 | 0.88 | 0.89 | 0.90 | 0.85 | 0.83 | 0.94 | 0.97 | 0.94 | 0.92 | 0.99 | 0.98 | 0.99 | 0.91 |
| clipb | 0.92 | 0.86 | 0.88 | 0.91 | 0.88 | 0.92 | 0.84 | 0.92 | 0.96 | 0.97 | 0.98 | 0.98 | 0.99 | 0.98 | 0.99 | 0.93 |
| Mean | 0.72 | 0.56 | 0.65 | 0.65 | 0.72 | 0.70 | 0.73 | 0.83 | 0.87 | 0.93 | 0.93 | 0.91 | 0.98 | 0.96 | 0.98 | 0.81 |

Table S10: **Transformation invariance**: Per-dataset cosine similarity between original embeddings and embeddings of the transformed image, averaged across transforms.

| Model | bach | ccrcc | crc | esca | pcam | tcga-crc | tcga-unif | tcga-tils | wilds | mhist | break-his | Mean |
|---|---|---|---|---|---|---|---|---|---|---|---|---|
| hopt0 | 0.90 | 0.70 | 0.64 | 0.67 | 0.62 | 0.80 | 0.68 | 0.54 | 0.57 | 0.74 | 0.76 | 0.69 |
| hopt1 | 0.90 | 0.73 | 0.68 | 0.72 | 0.62 | 0.81 | 0.71 | 0.58 | 0.62 | 0.78 | 0.78 | 0.72 |
| phikon2 | 0.90 | 0.73 | 0.69 | 0.73 | 0.62 | 0.82 | 0.74 | 0.54 | 0.61 | 0.74 | 0.79 | 0.72 |
| midnight | 0.92 | 0.80 | 0.70 | 0.78 | 0.59 | 0.83 | 0.69 | 0.59 | 0.61 | 0.72 | 0.77 | 0.73 |
| uni | 0.90 | 0.75 | 0.70 | 0.73 | 0.57 | 0.83 | 0.74 | 0.61 | 0.61 | 0.73 | 0.81 | 0.73 |
| uni2h | 0.94 | 0.78 | 0.67 | 0.75 | 0.60 | 0.85 | 0.75 | 0.67 | 0.61 | 0.77 | 0.81 | 0.75 |
| virchow | 0.88 | 0.85 | 0.74 | 0.79 | 0.65 | 0.79 | 0.79 | 0.68 | 0.67 | 0.77 | 0.79 | 0.76 |
| virchow2 | 0.95 | 0.78 | 0.74 | 0.76 | 0.63 | 0.88 | 0.78 | 0.67 | 0.69 | 0.71 | 0.84 | 0.77 |
| hiboub | 0.93 | 0.84 | 0.76 | 0.75 | 0.58 | 0.89 | 0.78 | 0.70 | 0.66 | 0.72 | 0.82 | 0.77 |
| musk | 0.94 | 0.80 | 0.79 | 0.75 | 0.74 | 0.80 | 0.75 | 0.77 | 0.76 | NaN | NaN | 0.79 |
| hiboul | 0.96 | 0.86 | 0.78 | 0.77 | 0.58 | 0.91 | 0.80 | 0.70 | 0.66 | 0.78 | 0.85 | 0.79 |
| keep | 0.96 | 0.80 | 0.77 | 0.80 | 0.67 | 0.89 | 0.79 | 0.72 | 0.71 | 0.72 | 0.83 | 0.79 |
| phikon | 0.92 | 0.80 | 0.76 | 0.82 | 0.71 | 0.87 | 0.81 | 0.63 | 0.73 | 0.78 | 0.85 | 0.79 |
| conch | 0.93 | 0.84 | 0.79 | 0.79 | 0.74 | 0.83 | 0.80 | 0.82 | 0.77 | 0.71 | 0.80 | 0.80 |
| titan | 0.94 | 0.86 | 0.83 | 0.82 | 0.79 | 0.86 | 0.83 | 0.85 | 0.83 | 0.78 | 0.79 | 0.84 |
| vitb | 0.93 | 0.89 | 0.82 | 0.87 | 0.76 | 0.93 | 0.89 | 0.80 | 0.80 | 0.82 | 0.89 | 0.85 |
| vitl | 0.93 | 0.91 | 0.85 | 0.89 | 0.79 | 0.93 | 0.90 | 0.83 | 0.84 | 0.84 | 0.89 | 0.87 |
| dinob | 0.98 | 0.92 | 0.89 | 0.91 | 0.78 | 0.96 | 0.92 | 0.81 | 0.80 | 0.88 | 0.93 | 0.89 |
| clipl | 0.98 | 0.92 | 0.90 | 0.87 | 0.80 | 0.93 | 0.88 | 0.87 | 0.86 | 0.87 | 0.94 | 0.89 |
| dinol | 0.99 | 0.93 | 0.89 | 0.92 | 0.79 | 0.97 | 0.92 | 0.82 | 0.80 | 0.91 | 0.95 | 0.90 |
| quilt | 0.97 | 0.95 | 0.94 | 0.85 | 0.87 | 0.91 | 0.87 | 0.94 | 0.91 | 0.87 | 0.95 | 0.91 |
| plip | 0.96 | 0.93 | 0.92 | 0.88 | 0.88 | 0.92 | 0.90 | 0.92 | 0.90 | 0.88 | 0.94 | 0.91 |
| clipb | 0.98 | 0.95 | 0.94 | 0.93 | 0.86 | 0.96 | 0.94 | 0.91 | 0.90 | 0.89 | 0.97 | 0.93 |
| Mean | 0.94 | 0.84 | 0.79 | 0.81 | 0.71 | 0.88 | 0.81 | 0.74 | 0.74 | 0.79 | 0.85 | 0.81 |

Table S11: Aggregated quantitative performance (Balanced accuracy) on knn classification.

| Model | all | nb classes | | magnif. | | | cancer type | | |
|---|---|---|---|---|---|---|---|---|---|
| | | 2 | >2 | ≥40× ≥20× | <40× | <20× | breast | crc | multi |
| hiboub | 76.7 | 80.7 | 73.9 | 66.4 | 77.5 | 83.5 | 78.3 | 74.2 | 73.0 |
| hiboul | 76.2 | 81.1 | 72.7 | 66.6 | 76.7 | 82.8 | 75.8 | 78.0 | 73.8 |
| hopt0 | 80.1 | 83.4 | 77.7 | 75.8 | 77.7 | 86.2 | 78.1 | 78.8 | 79.7 |
| hopt1 | 81.2 | **84.1** | 79.1 | 76.0 | 79.5 | 87.2 | 79.1 | **79.2** | 81.3 |
| midnight | 78.6 | 80.7 | 77.1 | 67.0 | 81.8 | 83.4 | 75.8 | 76.1 | **81.7** |
| phikon | 73.5 | 77.7 | 70.4 | 67.2 | 71.3 | 80.8 | 69.8 | 74.0 | 72.3 |
| phikon2 | 70.8 | 75.8 | 67.3 | 59.3 | 70.9 | 79.3 | 65.4 | 73.0 | 73.3 |
| uni | 79.6 | 82.4 | 77.6 | 74.7 | 77.6 | 85.7 | 79.3 | 77.1 | 77.2 |
| uni2h | **82.2** | 83.6 | **81.3** | 75.3 | **82.8** | 86.8 | **82.6** | 78.0 | 81.0 |
| virchow | 74.7 | 81.9 | 69.5 | 64.3 | 72.8 | 84.8 | 70.0 | 76.2 | 74.3 |
| virchow2 | 81.7 | 83.2 | 80.6 | **76.0** | 80.7 | **87.3** | 81.5 | 79.1 | 78.2 |
| conch | 78.0 | 79.9 | 76.6 | 71.3 | 78.4 | 82.6 | 78.4 | 76.5 | 72.0 |
| titan | 79.0 | 81.0 | 77.7 | 75.0 | 78.1 | 83.3 | 80.3 | 77.0 | 73.6 |
| keep | 80.6 | 82.0 | 79.5 | 74.8 | 80.3 | 85.2 | 80.9 | 75.7 | 77.6 |
| musk | 76.3 | 80.5 | 73.3 | 73.8 | 73.0 | 82.3 | 77.7 | 76.4 | 69.2 |
| plip | 68.3 | 76.8 | 62.2 | 57.1 | 69.7 | 75.0 | 68.1 | 72.0 | 63.5 |
| quilt | 69.2 | 77.7 | 63.1 | 60.6 | 69.2 | 75.7 | 70.4 | 73.4 | 63.8 |
| dinob | 68.8 | 77.7 | 62.5 | 61.6 | 66.6 | 77.0 | 68.0 | 75.5 | 58.6 |
| dinol | 70.2 | 79.2 | 63.7 | 63.3 | 67.1 | 79.2 | 70.1 | 76.7 | 59.2 |
| vitb | 65.1 | 74.6 | 58.3 | 56.5 | 64.0 | 72.9 | 65.0 | 72.1 | 57.1 |
| vitl | 68.3 | 76.8 | 62.2 | 61.9 | 66.4 | 75.5 | 68.4 | 74.3 | 59.8 |
| clipb | 62.7 | 73.5 | 55.1 | 55.8 | 59.6 | 71.9 | 62.7 | 69.5 | 53.7 |
| clipl | 64.8 | 74.3 | 58.0 | 55.7 | 63.4 | 73.5 | 65.4 | 69.3 | 57.4 |

Table S12: Aggregated quantitative performance (F1-score) on knn classification.

| Model | all | nb classes | | magnif. | | | cancer type | | |
|---|---|---|---|---|---|---|---|---|---|
| | | 2 | >2 | ≥40× ≥20× | <40× | <20× | breast | crc | multi |
| hiboub | 75.8 | 80.5 | 72.5 | 65.2 | 76.8 | 82.5 | 77.7 | 72.8 | 75.0 |
| hiboul | 75.2 | 80.8 | 71.1 | 64.7 | 75.9 | 82.1 | 75.1 | 76.4 | 76.0 |
| hopt0 | 79.2 | 82.4 | 76.9 | 74.9 | 76.7 | 85.6 | 77.4 | 76.2 | 81.2 |
| hopt1 | 80.5 | **83.2** | 78.6 | **75.9** | 78.1 | **87.1** | 78.5 | 77.2 | **82.8** |
| midnight | 78.2 | 80.3 | 76.7 | 66.6 | 81.1 | 83.4 | 75.1 | 75.1 | 82.5 |
| phikon | 72.8 | 77.3 | 69.6 | 65.6 | 71.1 | 80.3 | 69.3 | 72.7 | 74.2 |
| phikon2 | 70.1 | 75.4 | 66.4 | 58.2 | 70.4 | 78.8 | 64.8 | 71.7 | 74.9 |
| uni | 78.8 | 81.9 | 76.6 | 73.5 | 76.9 | 85.2 | 78.5 | 75.6 | 79.0 |
| uni2h | **81.7** | 83.1 | **80.7** | 75.5 | **81.7** | 86.3 | **82.4** | 76.6 | 82.1 |
| virchow | 74.2 | 81.3 | 69.1 | 63.8 | 72.2 | 84.4 | 70.0 | 74.2 | 75.7 |
| virchow2 | 81.2 | 83.0 | 79.9 | 75.4 | 80.2 | 86.9 | 80.8 | **77.8** | 79.8 |
| conch | 77.3 | 79.8 | 75.5 | 70.4 | 77.4 | 82.3 | 77.0 | 75.3 | 73.7 |
| titan | 78.6 | 80.7 | 77.2 | 74.5 | 77.3 | 83.5 | 79.7 | 75.6 | 75.3 |
| keep | 79.7 | 81.6 | 78.4 | 73.3 | 79.7 | 84.7 | 79.9 | 74.7 | 78.5 |
| musk | 75.6 | 80.3 | 72.2 | 72.5 | 72.2 | 82.1 | 76.5 | 75.2 | 71.2 |
| plip | 67.8 | 76.4 | 61.7 | 55.5 | 69.5 | 74.9 | 67.8 | 70.4 | 65.4 |
| quilt | 68.3 | 77.3 | 62.0 | 59.2 | 68.5 | 75.0 | 69.5 | 71.8 | 65.7 |
| dinob | 67.9 | 76.9 | 61.5 | 60.3 | 64.7 | 77.7 | 67.0 | 73.8 | 59.5 |
| dinol | 69.5 | 78.6 | 63.1 | 62.0 | 66.0 | 79.7 | 69.4 | 74.9 | 60.8 |
| vitb | 64.4 | 74.4 | 57.3 | 54.8 | 63.3 | 73.0 | 64.1 | 71.0 | 58.9 |
| vitl | 67.5 | 76.0 | 61.4 | 60.5 | 65.0 | 76.0 | 67.1 | 72.3 | 61.5 |
| clipb | 61.9 | 73.3 | 53.7 | 54.0 | 58.7 | 71.8 | 61.8 | 67.5 | 56.0 |
| clipl | 64.2 | 74.1 | 57.2 | 54.5 | 62.6 | 73.6 | 64.5 | 68.0 | 59.7 |

Table S13: Aggregated quantitative performance (Balanced accuracy) on linear probing.

| Model | all | nb classes | | magnif. | | | cancer type | | |
|---|---|---|---|---|---|---|---|---|---|
| | | 2 | >2 | ≥40× | <40× ≥20× | <20× | breast | crc | multi |
| hiboub | 78.5 | 85.1 | 73.8 | 68.1 | 78.6 | 86.2 | 75.8 | 80.4 | 78.5 |
| hiboul | 81.9 | 87.4 | 77.9 | 73.1 | 82.2 | 88.0 | 80.2 | 83.4 | 81.4 |
| hopt0 | 81.5 | **87.6** | 77.1 | 71.5 | 81.1 | 89.5 | 77.2 | 83.5 | 83.8 |
| hopt1 | 83.8 | 87.5 | 81.2 | 77.3 | 83.0 | 89.8 | 81.5 | 83.7 | 84.8 |
| midnight | 83.1 | 86.2 | 80.8 | 70.0 | **85.9** | 89.3 | 80.1 | 81.7 | **87.1** |
| phikon | 79.3 | 85.0 | 75.3 | 69.8 | 79.2 | 86.7 | 75.7 | 79.9 | 82.2 |
| phikon2 | 77.5 | 84.1 | 72.8 | 64.5 | 78.3 | 86.4 | 73.4 | 78.7 | 83.2 |
| uni | 81.6 | 86.6 | 78.0 | 73.4 | 80.3 | 89.3 | 78.9 | 82.4 | 81.2 |
| uni2h | **84.5** | 87.0 | **82.6** | **77.5** | 84.6 | 89.5 | **84.5** | 82.1 | 83.6 |
| virchow | 80.6 | 86.9 | 76.0 | 70.9 | 78.9 | **89.8** | 75.8 | 82.5 | 81.4 |
| virchow2 | 83.0 | 86.4 | 80.5 | 75.9 | 82.5 | 88.9 | 79.0 | **84.0** | 83.5 |
| conch | 80.6 | 84.4 | 77.9 | 71.9 | 80.5 | 87.3 | 80.2 | 81.1 | 76.1 |
| titan | 81.9 | 85.3 | 79.4 | 75.4 | 80.8 | 88.0 | 81.8 | 81.6 | 78.7 |
| keep | 81.7 | 85.3 | 79.2 | 73.2 | 81.7 | 88.1 | 79.7 | 81.0 | 79.8 |
| musk | 79.2 | 84.0 | 75.9 | 74.1 | 77.0 | 85.9 | 79.5 | 79.4 | 75.8 |
| plip | 71.4 | 80.9 | 64.6 | 59.2 | 71.0 | 81.0 | 68.6 | 76.6 | 69.9 |
| quilt | 71.2 | 81.1 | 64.2 | 59.9 | 71.3 | 79.7 | 71.6 | 76.8 | 62.0 |
| dinob | 75.3 | 81.8 | 70.6 | 69.7 | 72.1 | 83.4 | 74.7 | 79.6 | 68.5 |
| dinol | 75.8 | 82.1 | 71.4 | 69.7 | 73.6 | 83.2 | 75.0 | 80.4 | 69.3 |
| vitb | 72.7 | 80.4 | 67.3 | 66.8 | 70.5 | 80.0 | 71.3 | 77.9 | 67.8 |
| vitl | 73.4 | 81.2 | 67.8 | 66.3 | 70.8 | 81.8 | 71.6 | 78.3 | 70.0 |
| clipb | 66.8 | 77.1 | 59.4 | 54.9 | 66.0 | 76.8 | 62.3 | 75.4 | 64.2 |
| clipl | 72.0 | 80.1 | 66.2 | 62.5 | 69.9 | 81.6 | 70.7 | 75.7 | 68.0 |

Table S14: Aggregated quantitative performance (F1-score) on linear probing.

| Model | all | nb classes | | magnif. | | | cancer type | | |
|---|---|---|---|---|---|---|---|---|---|
| | | 2 | >2 | ≥40× | <40× ≥20× | <20× | breast | crc | multi |
| hiboub | 78.0 | 84.6 | 73.3 | 68.5 | 77.1 | 86.1 | 75.5 | 78.9 | 79.9 |
| hiboul | 81.2 | 86.2 | 77.5 | 73.5 | 80.6 | 87.6 | 80.0 | 80.8 | 82.8 |
| hopt0 | 81.4 | **86.6** | 77.7 | 72.6 | 80.2 | 89.5 | 77.7 | 81.2 | 85.2 |
| hopt1 | 83.3 | 86.5 | 81.0 | 77.3 | 81.4 | **90.1** | 80.9 | 81.5 | 86.0 |
| midnight | 82.9 | 85.7 | 80.9 | 70.4 | **85.1** | 89.5 | 80.0 | 80.5 | **88.1** |
| phikon | 78.5 | 84.0 | 74.5 | 68.9 | 78.2 | 85.9 | 75.0 | 77.8 | 83.3 |
| phikon2 | 76.5 | 83.6 | 71.4 | 62.6 | 77.3 | 85.8 | 72.5 | 77.4 | 84.3 |
| uni | 81.3 | 85.9 | 78.0 | 73.6 | 79.5 | 89.2 | 78.8 | 80.5 | 82.7 |
| uni2h | **83.9** | 86.0 | **82.4** | **77.8** | 82.9 | 89.7 | **84.5** | 79.8 | 84.6 |
| virchow | 80.2 | 85.9 | 76.2 | 71.4 | 78.2 | 89.4 | 76.0 | 80.5 | 82.8 |
| virchow2 | 82.7 | 85.4 | 80.8 | 76.2 | 81.7 | 88.8 | 78.8 | **82.1** | 84.7 |
| conch | 80.2 | 84.0 | 77.5 | 72.3 | 79.7 | 86.9 | 80.2 | 79.2 | 78.0 |
| titan | 80.8 | 84.3 | 78.3 | 75.0 | 78.4 | 88.0 | 80.5 | 79.1 | 79.7 |
| keep | 81.1 | 84.3 | 78.8 | 73.8 | 79.8 | 88.2 | 79.0 | 78.8 | 81.3 |
| musk | 79.0 | 83.6 | 75.7 | 75.4 | 75.9 | 85.5 | 79.2 | 78.1 | 77.7 |
| plip | 71.0 | 79.7 | 64.8 | 60.2 | 70.0 | 80.3 | 68.9 | 74.2 | 71.0 |
| quilt | 71.0 | 80.4 | 64.3 | 61.3 | 70.3 | 79.3 | 71.8 | 75.2 | 70.5 |
| dinob | 74.8 | 80.8 | 70.5 | 70.1 | 70.7 | 83.3 | 74.4 | 77.2 | 70.2 |
| dinol | 75.3 | 81.0 | 71.2 | 70.2 | 72.2 | 83.0 | 75.1 | 77.4 | 71.1 |
| vitb | 71.9 | 79.7 | 66.3 | 66.7 | 69.2 | 79.0 | 70.8 | 75.8 | 69.6 |
| vitl | 72.8 | 80.5 | 67.4 | 66.1 | 70.0 | 81.5 | 71.3 | 76.5 | 71.7 |
| clipb | 65.8 | 75.3 | 59.1 | 55.4 | 63.8 | 76.2 | 61.9 | 71.3 | 65.8 |
| clipl | 71.3 | 79.1 | 65.8 | 62.3 | 68.6 | 81.5 | 70.6 | 73.3 | 69.5 |

Table S15: Aggregated quantitative performance (Balanced accuracy) on 1-shot classification.

| Model | all | nb classes | | magnif. | | | cancer type | | |
|---|---|---|---|---|---|---|---|---|---|
| | | 2 | >2 | ≥40× | <40× ≥20× | <20× | breast | crc | multi |
| hiboub | 73.6 | 73.4 | 73.7 | 69.8 | 72.9 | 77.3 | 74.7 | 71.3 | 67.8 |
| hiboul | 71.0 | 72.2 | 70.1 | 68.6 | 70.8 | 73.1 | 70.5 | 66.3 | 71.2 |
| hopt0 | 72.4 | 71.4 | 73.2 | 69.5 | 74.5 | 72.1 | 65.1 | 71.6 | **77.0** |
| hopt1 | 69.5 | 61.7 | 75.0 | 69.0 | 71.9 | 66.8 | 62.3 | 72.8 | 64.8 |
| midnight | 72.5 | 73.4 | 71.8 | 59.4 | 75.5 | 78.6 | 67.2 | 72.7 | 72.0 |
| phikon | 73.4 | 75.9 | 71.6 | 69.1 | 71.6 | 78.8 | 71.2 | 72.7 | 68.1 |
| phikon2 | 69.3 | 67.6 | 70.4 | 66.6 | 64.2 | 77.5 | 68.0 | 67.8 | 58.0 |
| uni | 76.2 | 76.1 | 76.3 | **73.6** | 74.3 | 80.4 | 76.6 | 72.7 | 71.2 |
| uni2h | 75.3 | 71.3 | **78.1** | 71.1 | 73.6 | 80.5 | 76.3 | **73.9** | 62.0 |
| virchow | 68.1 | 69.9 | 66.8 | 62.5 | 63.5 | 78.0 | 63.5 | 71.6 | 56.7 |
| virchow2 | 72.8 | 72.2 | 73.3 | 65.2 | 75.2 | 75.5 | 71.1 | 68.5 | 73.0 |
| conch | 74.5 | 74.7 | 74.4 | 68.1 | 73.8 | 80.2 | 75.6 | 70.5 | 66.0 |
| titan | 75.0 | 74.7 | 75.2 | 70.8 | 72.2 | 81.7 | **77.2** | 72.2 | 62.5 |
| keep | **77.0** | **78.2** | 76.1 | 69.8 | **76.6** | **83.0** | 76.4 | 72.4 | 72.4 |
| musk | 70.0 | 68.9 | 70.8 | 68.0 | 69.1 | 72.8 | 69.3 | 70.1 | 60.6 |
| plip | 65.0 | 68.2 | 62.7 | 62.5 | 63.7 | 68.5 | 62.9 | 67.7 | 57.9 |
| quilt | 67.7 | 72.2 | 64.5 | 66.4 | 64.8 | 72.4 | 67.4 | 70.6 | 58.7 |
| dinob | 61.9 | 63.8 | 60.6 | 62.2 | 56.9 | 67.9 | 64.3 | 67.8 | 44.0 |
| dinol | 60.7 | 63.4 | 58.8 | 58.9 | 57.2 | 66.6 | 60.1 | 68.9 | 44.8 |
| vitb | 59.2 | 64.5 | 55.4 | 55.0 | 56.8 | 65.3 | 59.6 | 64.0 | 48.0 |
| vitl | 59.0 | 63.3 | 55.9 | 55.0 | 55.5 | 66.3 | 57.5 | 65.9 | 43.8 |
| clipb | 54.7 | 59.7 | 51.1 | 51.6 | 54.0 | 57.8 | 51.9 | 61.3 | 47.9 |
| clipl | 57.6 | 60.9 | 55.3 | 52.6 | 55.5 | 64.0 | 56.2 | 63.5 | 45.3 |

Table S16: Aggregated quantitative performance (F1-score) on 1-shot classification.

| Model | all | nb classes | | magnif. | | | cancer type | | |
|---|---|---|---|---|---|---|---|---|---|
| | | 2 | >2 | ≥40× | <40× ≥20× | <20× | breast | crc | multi |
| hiboub | 66.3 | 65.9 | 66.6 | 65.8 | 65.0 | 68.5 | 71.4 | 64.5 | 54.9 |
| hiboul | 64.0 | 63.9 | 64.0 | 65.7 | 63.1 | 63.8 | 66.6 | 58.8 | 60.1 |
| hopt0 | 66.8 | 65.2 | 67.9 | 65.3 | 69.7 | 64.2 | 61.8 | 66.4 | **66.4** |
| hopt1 | 60.3 | 48.1 | 68.9 | 64.0 | 61.8 | 55.6 | 55.1 | 64.9 | 45.9 |
| midnight | 69.2 | 71.1 | 67.9 | 57.8 | **72.1** | 74.2 | 65.3 | **71.5** | 65.2 |
| phikon | 66.3 | 67.3 | 65.7 | 66.0 | 63.4 | 70.2 | 68.9 | 64.0 | 55.3 |
| phikon2 | 59.9 | 54.4 | 63.8 | 63.3 | 50.8 | 68.7 | 64.8 | 57.5 | 35.8 |
| uni | 69.8 | 69.1 | 70.2 | **71.0** | 67.0 | 72.4 | **74.2** | 66.9 | 57.7 |
| uni2h | 67.4 | 60.2 | **72.5** | 68.6 | 62.5 | 72.5 | 73.1 | 66.2 | 41.7 |
| virchow | 61.7 | 62.0 | 61.4 | 58.7 | 55.6 | 71.4 | 60.7 | 67.2 | 39.8 |
| virchow2 | 67.5 | 66.4 | 68.2 | 63.6 | 70.0 | 67.2 | 68.8 | 63.1 | 62.3 |
| conch | 68.7 | 69.3 | 68.3 | 65.2 | 68.0 | 72.2 | 72.7 | 65.9 | 54.1 |
| titan | 68.8 | 67.9 | 69.5 | 67.7 | 64.6 | 74.9 | 74.0 | 67.7 | 47.8 |
| keep | **72.2** | **73.9** | 70.9 | 66.4 | 71.8 | **76.9** | 73.9 | 69.4 | 61.9 |
| musk | 62.9 | 61.5 | 63.9 | 64.0 | 61.3 | 64.0 | 65.9 | 63.8 | 46.2 |
| plip | 58.7 | 61.0 | 57.0 | 58.4 | 57.3 | 60.6 | 60.1 | 61.4 | 45.5 |
| quilt | 61.5 | 66.1 | 58.2 | 63.3 | 57.9 | 64.6 | 64.8 | 65.0 | 46.7 |
| dinob | 54.3 | 54.2 | 54.5 | 59.5 | 46.7 | 60.1 | 57.0 | 63.5 | 26.2 |
| dinol | 52.9 | 53.9 | 52.2 | 55.0 | 47.8 | 57.8 | 54.5 | 65.0 | 27.7 |
| vitb | 51.8 | 56.8 | 48.2 | 49.1 | 48.9 | 57.4 | 54.3 | 58.1 | 35.9 |
| vitl | 50.7 | 53.9 | 48.4 | 49.1 | 44.5 | 59.6 | 51.7 | 58.8 | 27.2 |
| clipb | 46.5 | 48.8 | 44.9 | 48.5 | 45.6 | 46.2 | 45.3 | 54.0 | 34.3 |
| clipl | 48.9 | 49.7 | 48.3 | 48.1 | 44.7 | 54.8 | 51.3 | 55.0 | 28.2 |

Table S17: Aggregated quantitative performance (Balanced accuracy) on 2-shot classification.

| Model | all | nb classes | | magnif. | | | cancer type | | |
|---|---|---|---|---|---|---|---|---|---|
| | | 2 | >2 | ≥40×
≥20× | <40× | <20× | breast | crc | multi |
| hiboub | 76.6 | 78.8 | 75.1 | 71.8 | 74.9 | 82.4 | 77.6 | 73.8 | 71.7 |
| hiboul | 73.2 | 74.5 | 72.2 | 70.1 | 73.3 | 75.4 | 72.0 | 70.5 | 72.6 |
| hopt0 | 76.3 | 81.0 | 73.0 | 68.7 | 75.7 | 82.9 | 71.3 | 75.2 | **80.2** |
| hopt1 | 74.6 | 72.6 | 76.1 | 69.5 | 76.1 | 76.7 | 69.2 | 75.9 | 73.3 |
| midnight | 72.7 | 74.8 | 71.1 | 56.7 | 76.1 | 80.3 | 68.6 | 72.3 | 73.4 |
| phikon | 74.9 | 79.7 | 71.4 | 68.9 | 73.3 | 81.3 | 70.9 | 73.8 | 74.4 |
| phikon2 | 72.1 | 73.5 | 71.2 | 65.4 | 70.2 | 79.6 | 69.2 | 69.8 | 68.2 |
| uni | **79.3** | **82.2** | 77.3 | **75.0** | 77.0 | **85.5** | 78.5 | 75.7 | 77.5 |
| uni2h | 78.7 | 77.9 | **79.3** | 71.9 | 77.6 | 85.2 | **79.5** | **76.9** | 69.5 |
| virchow | 71.0 | 77.4 | 66.5 | 61.1 | 67.1 | 83.4 | 66.2 | 73.1 | 64.8 |
| virchow2 | 75.0 | 77.7 | 73.2 | 64.5 | 77.1 | 80.4 | 73.4 | 72.5 | 75.7 |
| conch | 76.2 | 78.1 | 74.9 | 68.6 | 75.8 | 82.5 | 76.5 | 74.0 | 68.9 |
| titan | 77.5 | 79.4 | 76.2 | 71.2 | 76.4 | 83.6 | 78.5 | 75.0 | 70.1 |
| keep | 78.1 | 79.9 | 76.8 | 70.6 | **78.0** | 83.8 | 77.5 | 72.9 | 75.2 |
| musk | 73.3 | 76.4 | 71.1 | 69.3 | 71.0 | 79.2 | 73.2 | 72.0 | 66.6 |
| plip | 66.8 | 73.8 | 61.8 | 61.4 | 64.0 | 74.2 | 64.9 | 70.0 | 60.0 |
| quilt | 69.5 | 75.6 | 65.1 | 67.4 | 65.8 | 75.6 | 69.5 | 72.1 | 61.0 |
| dinob | 63.4 | 67.2 | 60.6 | 62.2 | 58.7 | 70.1 | 62.3 | 69.4 | 48.6 |
| dinol | 62.7 | 67.3 | 59.3 | 58.9 | 58.8 | 70.3 | 61.9 | 70.2 | 47.7 |
| vitb | 61.2 | 68.5 | 55.9 | 55.2 | 58.4 | 69.1 | 62.1 | 65.8 | 50.7 |
| vitl | 60.5 | 67.2 | 55.7 | 54.4 | 57.5 | 68.7 | 58.2 | 67.8 | 48.7 |
| clipb | 56.5 | 63.2 | 51.8 | 53.3 | 55.2 | 60.7 | 54.0 | 63.5 | 50.5 |
| clipl | 61.1 | 69.2 | 55.3 | 52.8 | 57.8 | 71.5 | 60.2 | 66.6 | 50.7 |

Table S18: Aggregated quantitative performance (F1-score) on 2-shot classification.

| Model | all | nb classes | | magnif. | | | cancer type | | |
|---|---|---|---|---|---|---|---|---|---|
| | | 2 | >2 | ≥40×
≥20× | <40× | <20× | breast | crc | multi |
| hiboub | 71.0 | 73.7 | 69.1 | 68.8 | 68.8 | 75.5 | 75.0 | 69.0 | 61.6 |
| hiboul | 67.9 | 68.9 | 67.2 | 67.9 | 67.5 | 68.5 | 69.0 | 65.7 | 63.5 |
| hopt0 | 72.1 | 77.3 | 68.4 | 65.7 | 71.7 | 77.5 | 69.0 | 70.8 | **73.5** |
| hopt1 | 68.3 | 64.9 | 70.8 | 64.9 | 68.7 | 70.4 | 65.0 | 70.3 | 60.1 |
| midnight | 70.0 | 73.0 | 68.0 | 55.8 | 73.0 | 77.0 | 66.9 | **71.3** | 67.9 |
| phikon | 69.3 | 74.3 | 65.7 | 65.3 | 68.0 | 73.9 | 68.4 | 67.6 | 66.1 |
| phikon2 | 65.3 | 65.4 | 65.2 | 62.4 | 62.0 | 71.5 | 66.4 | 62.9 | 53.8 |
| uni | **74.4** | **77.7** | 72.0 | **72.6** | 71.4 | 79.3 | 76.3 | 71.1 | 68.5 |
| uni2h | 72.6 | 69.8 | **74.6** | 69.8 | 68.8 | **79.5** | **77.0** | 70.5 | 55.0 |
| virchow | 65.7 | 71.6 | 61.4 | 57.6 | 60.6 | 78.1 | 63.6 | 68.8 | 53.0 |
| virchow2 | 70.8 | 73.5 | 68.9 | 63.2 | 72.6 | 74.4 | 71.6 | 68.0 | 67.8 |
| conch | 71.5 | 74.2 | 69.6 | 66.6 | 70.7 | 76.2 | 74.2 | 70.3 | 59.5 |
| titan | 72.6 | 74.6 | 71.2 | 68.9 | 70.4 | 78.2 | 76.0 | 71.2 | 59.5 |
| keep | 74.2 | 77.0 | 72.2 | 67.8 | **74.2** | 79.0 | 75.5 | 70.6 | 67.5 |
| musk | 67.5 | 71.0 | 64.9 | 66.2 | 64.7 | 71.8 | 70.6 | 66.2 | 56.0 |
| plip | 61.7 | 69.0 | 56.4 | 57.9 | 59.1 | 67.7 | 62.7 | 65.4 | 50.5 |
| quilt | 64.1 | 70.9 | 59.3 | 64.6 | 59.8 | 69.2 | 67.2 | 67.1 | 51.5 |
| dinob | 56.9 | 60.3 | 54.5 | 59.3 | 50.5 | 63.2 | 58.4 | 65.7 | 34.2 |
| dinol | 55.9 | 60.1 | 53.0 | 55.4 | 50.8 | 62.8 | 57.2 | 67.1 | 33.2 |
| vitb | 55.1 | 62.9 | 49.6 | 50.2 | 52.1 | 62.6 | 58.6 | 61.1 | 40.1 |
| vitl | 53.9 | 60.8 | 49.0 | 49.1 | 49.3 | 63.2 | 53.2 | 62.9 | 36.0 |
| clipb | 49.8 | 55.4 | 45.9 | 50.7 | 48.6 | 50.8 | 48.6 | 58.3 | 39.2 |
| clipl | 54.3 | 62.1 | 48.8 | 48.5 | 49.4 | 64.9 | 56.8 | 59.9 | 37.8 |

Table S19: Aggregated quantitative performance (Balanced accuracy) on 4-shot classification.

| Model | all | nb classes | | magnif. | | | cancer type | | |
|---|---|---|---|---|---|---|---|---|---|
| | | 2 | >2 | ≥40×
≥20× | <40× | <20× | breast | crc | multi |
| hiboub | 77.7 | 81.2 | 75.3 | 71.7 | 75.8 | 84.7 | 78.7 | 74.9 | 73.3 |
| hiboul | 73.8 | 75.8 | 72.3 | 69.3 | 74.1 | 76.7 | 72.3 | 71.3 | 73.8 |
| hopt0 | 76.8 | 81.8 | 73.2 | 68.0 | 76.2 | 84.0 | 72.0 | 76.1 | **80.8** |
| hopt1 | 76.7 | 77.6 | 76.2 | 70.2 | 78.3 | 79.7 | 70.8 | 77.4 | 80.4 |
| midnight | 72.9 | 75.1 | 71.4 | 56.3 | 76.9 | 80.3 | 69.5 | 72.3 | 73.9 |
| phikon | 75.6 | 81.0 | 71.7 | 70.1 | 73.4 | 82.5 | 71.0 | 75.2 | 76.0 |
| phikon2 | 74.0 | 77.3 | 71.6 | 66.1 | 73.7 | 80.2 | 69.3 | 71.3 | 76.7 |
| uni | 79.8 | **83.3** | 77.2 | **74.3** | 77.7 | **86.5** | 78.2 | 77.3 | 78.8 |
| uni2h | **80.5** | 81.7 | **79.7** | 71.8 | **81.1** | 86.4 | **79.8** | **78.0** | 77.7 |
| virchow | 71.9 | 78.8 | 67.0 | 61.3 | 68.7 | 84.0 | 66.5 | 73.1 | 68.5 |
| virchow2 | 75.9 | 79.4 | 73.5 | 64.3 | 77.5 | 82.8 | 74.4 | 75.1 | 75.4 |
| conch | 76.5 | 78.8 | 74.8 | 68.6 | 76.2 | 82.6 | 76.2 | 74.7 | 70.0 |
| titan | 78.0 | 80.1 | 76.5 | 71.2 | 77.4 | 83.7 | 78.7 | 75.4 | 72.2 |
| keep | 78.8 | 80.2 | 77.8 | 71.8 | 78.8 | 84.0 | 78.5 | 73.7 | 75.9 |
| musk | 74.1 | 78.2 | 71.2 | 70.0 | 71.1 | 81.0 | 73.9 | 73.3 | 67.9 |
| plip | 67.6 | 75.5 | 62.0 | 62.0 | 64.0 | 76.3 | 65.9 | 71.6 | 60.1 |
| quilt | 69.8 | 76.9 | 64.7 | 67.1 | 65.3 | 77.4 | 69.5 | 73.1 | 61.1 |
| dinob | 64.8 | 70.9 | 60.5 | 61.5 | 59.3 | 73.5 | 64.1 | 70.5 | 51.6 |
| dinol | 63.7 | 70.1 | 59.1 | 58.7 | 58.9 | 73.3 | 63.5 | 70.7 | 49.1 |
| vitb | 62.2 | 70.8 | 56.0 | 54.9 | 59.1 | 71.5 | 63.2 | 67.3 | 52.1 |
| vitl | 60.9 | 68.7 | 55.4 | 53.3 | 58.4 | 69.8 | 57.8 | 68.7 | 51.1 |
| clipb | 57.7 | 66.3 | 51.5 | 52.8 | 55.7 | 63.8 | 55.1 | 65.7 | 51.1 |
| clipl | 62.6 | 72.6 | 55.5 | 53.3 | 58.2 | 75.2 | 62.6 | 67.8 | 52.1 |

Table S20: Aggregated quantitative performance (F1-score) on 4-shot classification.

| Model | all | nb classes | | magnif. | | | cancer type | | |
|---|---|---|---|---|---|---|---|---|---|
| | | 2 | >2 | ≥40×
≥20× | <40× | <20× | breast | crc | multi |
| hiboub | 72.9 | 76.9 | 70.0 | 69.7 | 70.2 | 78.5 | 76.5 | 70.3 | 65.2 |
| hiboul | 69.4 | 71.2 | 68.1 | 68.1 | 69.0 | 70.8 | 70.2 | 66.4 | 66.8 |
| hopt0 | 73.2 | 78.7 | 69.3 | 66.1 | 72.6 | 79.2 | 70.2 | 71.7 | **76.0** |
| hopt1 | 71.9 | 72.1 | 71.8 | 67.2 | 72.6 | 74.6 | 67.8 | 72.4 | 71.3 |
| midnight | 70.6 | 73.2 | 68.8 | 56.2 | 73.8 | 77.5 | 68.4 | 71.1 | 68.7 |
| phikon | 71.0 | 76.7 | 66.9 | 67.7 | 69.0 | 75.9 | 69.4 | 69.3 | 69.8 |
| phikon2 | 68.5 | 71.8 | 66.2 | 63.7 | 67.8 | 73.0 | 67.1 | 65.4 | 67.3 |
| uni | **75.6** | **79.8** | 72.6 | **72.8** | 73.0 | 81.1 | 76.6 | **72.5** | 72.5 |
| uni2h | 75.6 | 75.7 | **75.5** | 69.9 | 74.2 | **81.5** | **77.8** | 72.4 | 67.3 |
| virchow | 67.3 | 74.0 | 62.5 | 58.4 | 63.1 | 79.3 | 64.4 | 68.9 | 58.8 |
| virchow2 | 72.2 | 75.7 | 69.7 | 63.6 | 72.9 | 77.8 | 73.0 | 71.3 | 68.0 |
| conch | 72.1 | 75.1 | 70.0 | 66.9 | 71.4 | 77.0 | 74.4 | 70.7 | 61.8 |
| titan | 73.7 | 76.0 | 72.0 | 69.4 | 72.0 | 79.1 | 76.6 | 71.9 | 63.5 |
| keep | 75.2 | 77.5 | 73.6 | 69.5 | **75.1** | 79.7 | 76.9 | 70.9 | 69.3 |
| musk | 68.9 | 73.5 | 65.6 | 67.6 | 65.4 | 74.2 | 71.8 | 67.7 | 59.0 |
| plip | 62.9 | 71.0 | 57.2 | 59.5 | 58.9 | 70.4 | 64.2 | 67.0 | 51.5 |
| quilt | 65.1 | 72.7 | 59.7 | 65.5 | 59.7 | 71.5 | 67.9 | 68.2 | 52.8 |
| dinob | 59.2 | 65.3 | 54.9 | 59.2 | 52.4 | 67.8 | 61.2 | 66.9 | 39.1 |
| dinol | 57.8 | 63.9 | 53.3 | 55.7 | 51.5 | 67.1 | 59.9 | 67.8 | 36.0 |
| vitb | 56.7 | 66.1 | 50.1 | 50.4 | 53.1 | 66.0 | 60.2 | 63.0 | 43.0 |
| vitl | 55.1 | 63.2 | 49.3 | 49.1 | 50.7 | 65.1 | 53.6 | 64.1 | 40.0 |
| clipb | 51.8 | 59.7 | 46.1 | 50.6 | 49.0 | 56.1 | 51.3 | 60.5 | 40.4 |
| clipl | 56.6 | 66.5 | 49.5 | 49.3 | 50.8 | 69.2 | 59.9 | 61.8 | 40.8 |

Table S21: Aggregated quantitative performance (Balanced accuracy) on 8-shot classification.

| Model | all | nb classes | | magnif. | | | cancer type | | |
|---|---|---|---|---|---|---|---|---|---|
| | | 2 | >2 | ≥40× | <40× ≥20× | <20× | breast | crc | multi |
| hiboub | 78.3 | 81.9 | 75.7 | 72.4 | 76.2 | 85.2 | 79.4 | 75.8 | 73.8 |
| hiboul | 73.8 | 76.1 | 72.1 | 68.6 | 74.3 | 76.9 | 72.3 | 71.8 | 74.0 |
| hopt0 | 76.5 | 81.7 | 72.8 | 67.2 | 76.2 | 83.9 | 71.4 | 77.0 | 80.7 |
| hopt1 | 78.1 | 80.5 | 76.4 | 70.2 | 79.5 | 82.4 | 72.7 | 78.6 | **82.3** |
| midnight | 72.6 | 75.3 | 70.6 | 54.5 | 77.4 | 80.1 | 68.9 | 72.7 | 74.2 |
| phikon | 76.1 | 81.5 | 72.2 | 71.0 | 73.4 | 83.2 | 71.7 | 76.0 | 75.8 |
| phikon2 | 74.7 | 78.2 | 72.1 | 67.1 | 74.2 | 80.9 | 70.1 | 72.0 | 77.9 |
| uni | 79.8 | **83.7** | 77.0 | **74.4** | 77.4 | 86.9 | 78.1 | 78.3 | 78.8 |
| uni2h | 81.5 | 83.3 | **80.3** | 72.3 | **82.6** | **87.1** | **80.6** | **79.1** | 80.2 |
| virchow | 72.3 | 79.5 | 67.2 | 61.8 | 69.5 | 83.8 | 66.8 | 73.2 | 70.1 |
| virchow2 | 75.9 | 79.5 | 73.4 | 63.8 | 77.6 | 82.9 | 74.7 | 75.7 | 74.8 |
| conch | 76.8 | 79.0 | 75.2 | 69.4 | 76.7 | 82.6 | 77.0 | 74.9 | 70.3 |
| titan | 78.2 | 80.3 | 76.7 | 71.7 | 77.9 | 83.3 | 78.8 | 75.3 | 73.5 |
| keep | 78.8 | 80.3 | 77.8 | 71.8 | 78.9 | 84.1 | 78.4 | 73.9 | 76.1 |
| musk | 74.4 | 78.6 | 71.5 | 70.0 | 71.6 | 81.3 | 74.1 | 73.9 | 68.6 |
| plip | 67.7 | 75.9 | 61.9 | 61.3 | 64.3 | 76.8 | 66.1 | 71.9 | 60.4 |
| quilt | 70.0 | 77.2 | 64.8 | 66.9 | 65.6 | 77.8 | 69.7 | 73.1 | 61.4 |
| dinob | 65.6 | 72.6 | 60.7 | 60.4 | 61.7 | 74.4 | 64.9 | 71.5 | 53.6 |
| dinol | 64.1 | 71.2 | 59.1 | 58.8 | 59.1 | 74.4 | 64.0 | 70.9 | 50.1 |
| vitb | 62.7 | 71.7 | 56.3 | 55.4 | 59.3 | 72.4 | 63.5 | 68.4 | 52.9 |
| vitl | 61.5 | 69.9 | 55.5 | 53.5 | 59.3 | 70.3 | 58.2 | 69.5 | 53.1 |
| clipb | 58.1 | 67.8 | 51.2 | 52.6 | 55.6 | 65.4 | 56.0 | 66.0 | 51.6 |
| clipl | 63.1 | 73.5 | 55.7 | 53.4 | 59.0 | 75.6 | 63.3 | 68.1 | 53.6 |

Table S22: Aggregated quantitative performance (F1-score) on 8-shot classification.

| Model | all | nb classes | | magnif. | | | cancer type | | |
|---|---|---|---|---|---|---|---|---|---|
| | | 2 | >2 | ≥40× | <40× ≥20× | <20× | breast | crc | multi |
| hiboub | 73.9 | 78.1 | 70.9 | 70.7 | 71.3 | 79.6 | 77.4 | 71.2 | 67.2 |
| hiboul | 69.9 | 71.9 | 68.5 | 68.0 | 69.7 | 71.5 | 70.7 | 66.9 | 67.8 |
| hopt0 | 73.3 | 79.1 | 69.1 | 65.7 | 72.8 | 79.5 | 69.7 | 72.8 | **77.0** |
| hopt1 | 74.1 | 76.5 | 72.4 | 67.6 | 74.9 | 78.0 | 70.2 | 74.2 | 75.9 |
| midnight | 70.4 | 73.3 | 68.3 | 54.7 | 74.1 | 77.5 | 68.0 | 71.3 | 69.2 |
| phikon | 71.8 | 77.8 | 67.6 | 68.5 | 69.6 | 77.1 | 70.2 | 70.5 | 70.8 |
| phikon2 | 70.0 | 74.1 | 67.0 | 64.8 | 69.6 | 74.4 | 67.9 | 67.0 | 71.2 |
| uni | 76.2 | **80.8** | 73.0 | **73.1** | 73.4 | 82.1 | 76.7 | 73.9 | 74.0 |
| uni2h | 77.5 | 78.7 | **76.7** | 71.1 | **77.0** | **82.9** | **79.0** | **74.3** | 72.7 |
| virchow | 68.2 | 75.4 | 63.1 | 59.3 | 64.4 | 79.7 | 64.9 | 69.5 | 61.9 |
| virchow2 | 72.4 | 76.1 | 69.8 | 63.3 | 73.1 | 78.4 | 73.3 | 72.1 | 68.1 |
| conch | 72.9 | 75.7 | 71.0 | 68.3 | 72.0 | 77.5 | 75.5 | 71.1 | 63.1 |
| titan | 74.5 | 76.9 | 72.7 | 70.3 | 73.1 | 79.3 | 77.1 | 72.2 | 66.3 |
| keep | 75.6 | 77.9 | 74.0 | 70.1 | 75.4 | 80.0 | 77.1 | 71.3 | 70.4 |
| musk | 69.7 | 74.4 | 66.3 | 68.1 | 66.5 | 75.0 | 72.4 | 68.4 | 61.2 |
| plip | 63.5 | 71.8 | 57.5 | 59.2 | 59.9 | 71.2 | 64.7 | 67.4 | 53.0 |
| quilt | 65.3 | 73.3 | 60.1 | 65.4 | 60.4 | 72.2 | 68.3 | 68.3 | 54.2 |
| dinob | 60.5 | 67.7 | 55.3 | 58.4 | 54.8 | 69.2 | 62.2 | 67.6 | 43.2 |
| dinol | 58.9 | 65.8 | 54.0 | 56.4 | 52.5 | 68.7 | 61.1 | 67.9 | 38.6 |
| vitb | 57.6 | 67.7 | 50.5 | 51.1 | 53.7 | 67.4 | 60.7 | 64.5 | 44.8 |
| vitl | 56.2 | 65.4 | 49.7 | 49.3 | 52.6 | 66.0 | 54.2 | 65.2 | 44.1 |
| clipb | 52.9 | 62.2 | 46.2 | 51.0 | 49.4 | 58.7 | 53.1 | 60.9 | 42.2 |
| clipl | 57.6 | 68.2 | 50.1 | 49.7 | 52.6 | 69.9 | 60.8 | 62.3 | 44.2 |

Table S23: Aggregated quantitative performance (Balanced accuracy) on 16-shot classification.

| Model | all | nb classes | | magnif. | | | cancer type | | |
|---|---|---|---|---|---|---|---|---|---|
| | | 2 | >2 | ≥40× | <40× ≥20× | <20× | breast | crc | multi |
| hiboub | 78.2 | 81.9 | 75.5 | 71.6 | 76.5 | 85.3 | 79.1 | 76.0 | 74.2 |
| hiboul | 74.1 | 76.4 | 72.4 | 68.4 | 74.8 | 77.3 | 72.7 | 71.8 | 74.7 |
| hopt0 | 76.4 | 81.5 | 72.8 | 67.1 | 75.9 | 84.0 | 71.3 | 77.1 | 80.7 |
| hopt1 | 78.4 | 81.3 | 76.3 | 70.2 | 79.3 | 83.3 | 73.5 | 77.9 | **83.0** |
| midnight | 72.6 | 75.2 | 70.8 | 54.5 | 77.5 | 80.0 | 69.2 | 72.2 | 74.5 |
| phikon | 76.1 | 81.3 | 72.4 | 71.4 | 73.1 | 83.4 | 72.1 | 75.7 | 75.9 |
| phikon2 | 74.4 | 78.1 | 71.8 | 67.1 | 73.7 | 80.8 | 69.5 | 72.0 | 77.9 |
| uni | 79.7 | 83.7 | 76.9 | **74.0** | 77.3 | 87.0 | 77.8 | 78.5 | 78.9 |
| uni2h | 81.9 | **83.7** | **80.5** | 72.6 | **83.1** | **87.3** | **80.8** | **79.2** | 81.5 |
| virchow | 72.4 | 79.6 | 67.3 | 61.8 | 69.7 | 83.8 | 66.8 | 73.0 | 71.0 |
| virchow2 | 75.8 | 79.5 | 73.3 | 63.6 | 77.7 | 82.7 | 74.3 | 75.8 | 75.3 |
| conch | 76.8 | 79.3 | 75.1 | 68.9 | 76.8 | 82.8 | 76.7 | 75.3 | 70.7 |
| titan | 78.2 | 80.2 | 76.8 | 71.6 | 78.1 | 83.3 | 78.7 | 75.4 | 74.0 |
| keep | 78.8 | 80.1 | 77.9 | 71.7 | 79.0 | 83.9 | 78.8 | 73.8 | 76.3 |
| musk | 74.4 | 78.4 | 71.6 | 70.3 | 71.5 | 81.2 | 74.2 | 73.7 | 68.7 |
| plip | 67.6 | 75.8 | 61.7 | 61.2 | 64.1 | 76.7 | 65.8 | 71.9 | 60.4 |
| quilt | 69.8 | 77.1 | 64.7 | 66.9 | 65.3 | 77.6 | 69.3 | 73.4 | 61.2 |
| dinob | 65.8 | 72.8 | 60.8 | 60.6 | 62.0 | 74.4 | 64.9 | 71.6 | 54.5 |
| dinol | 64.1 | 71.7 | 58.7 | 57.9 | 59.3 | 74.7 | 63.9 | 71.0 | 50.5 |
| vitb | 62.5 | 71.8 | 56.0 | 55.3 | 59.0 | 72.4 | 62.9 | 68.8 | 53.0 |
| vitl | 61.4 | 70.2 | 55.1 | 53.1 | 59.3 | 70.3 | 57.4 | 70.0 | 53.8 |
| clipb | 58.2 | 68.0 | 51.3 | 52.4 | 55.8 | 65.6 | 56.0 | 66.2 | 52.0 |
| clipl | 63.5 | 73.8 | 56.2 | 54.5 | 59.2 | 75.8 | 63.8 | 68.4 | 54.1 |

Table S24: Aggregated quantitative performance (F1-score) on 16-shot classification.

| Model | all | nb classes | | magnif. | | | cancer type | | |
|---|---|---|---|---|---|---|---|---|---|
| | | 2 | >2 | ≥40× | <40× ≥20× | <20× | breast | crc | multi |
| hiboub | 74.2 | 78.4 | 71.2 | 70.4 | 71.8 | 80.0 | 77.5 | 71.3 | 68.4 |
| hiboul | 70.4 | 72.3 | 69.1 | 68.0 | 70.6 | 72.1 | 71.4 | 67.0 | 69.3 |
| hopt0 | 73.4 | 79.0 | 69.3 | 65.9 | 72.7 | 79.8 | 69.7 | 72.7 | 77.7 |
| hopt1 | 74.8 | 78.0 | 72.6 | 68.1 | 75.2 | 79.4 | 71.3 | 73.6 | **78.4** |
| midnight | 70.6 | 73.3 | 68.7 | 55.1 | 74.4 | 77.5 | 68.6 | 70.7 | 69.8 |
| phikon | 72.2 | 77.9 | 68.1 | 69.2 | 69.7 | 77.4 | 70.7 | 70.4 | 71.5 |
| phikon2 | 70.1 | 74.6 | 66.9 | 64.7 | 69.8 | 74.5 | 67.4 | 67.1 | 72.9 |
| uni | 76.4 | **80.9** | 73.1 | **73.1** | 73.5 | 82.4 | 76.6 | 74.0 | 74.9 |
| uni2h | 78.4 | 79.9 | **77.3** | 71.7 | **78.4** | **83.4** | **79.6** | **74.6** | 75.8 |
| virchow | 68.5 | 75.8 | 63.3 | 59.6 | 64.9 | 79.7 | 65.0 | 69.2 | 63.9 |
| virchow2 | 72.5 | 76.3 | 69.9 | 63.3 | 73.5 | 78.3 | 73.1 | 72.1 | 69.5 |
| conch | 73.1 | 76.0 | 71.1 | 68.3 | 72.2 | 77.9 | 75.5 | 71.4 | 63.9 |
| titan | 74.6 | 77.0 | 73.0 | 70.4 | 73.4 | 79.3 | 77.1 | 72.1 | 67.6 |
| keep | 75.8 | 77.9 | 74.3 | 70.4 | 75.6 | 80.0 | 77.2 | 71.2 | 71.2 |
| musk | 70.0 | 74.4 | 66.8 | 68.7 | 66.7 | 75.0 | 72.7 | 68.2 | 62.2 |
| plip | 63.4 | 71.8 | 57.5 | 59.4 | 59.7 | 71.1 | 64.5 | 67.4 | 53.6 |
| quilt | 65.7 | 73.3 | 60.2 | 65.9 | 60.3 | 72.3 | 68.1 | 68.6 | 54.7 |
| dinob | 61.0 | 68.3 | 55.8 | 59.0 | 55.7 | 69.2 | 62.6 | 67.7 | 45.4 |
| dinol | 59.2 | 66.6 | 54.0 | 55.9 | 53.2 | 69.3 | 61.4 | 67.9 | 40.2 |
| vitb | 57.8 | 68.0 | 50.6 | 51.3 | 54.0 | 67.5 | 60.6 | 65.0 | 45.6 |
| vitl | 56.5 | 66.0 | 49.7 | 49.2 | 53.3 | 66.1 | 53.8 | 65.9 | 45.8 |
| clipb | 53.3 | 62.7 | 46.6 | 51.2 | 49.9 | 59.0 | 53.4 | 61.0 | 43.5 |
| clipl | 58.2 | 68.8 | 50.7 | 50.7 | 53.2 | 70.2 | 61.5 | 62.6 | 45.9 |

Table S25: Aggregated quantitative performance (ECE) on linear probing.

| Model | all | nb classes | | magnif. | | | cancer type | | |
|---|---|---|---|---|---|---|---|---|---|
| | | 2 | >2 | ≥40× | <40× ≥20× | <20× | breast | crc | multi |
| hiboub | 3.7 | 1.7 | 5.1 | **3.7** | 4.8 | 2.2 | 5.3 | 2.6 | 2.2 |
| hiboul | 5.5 | 2.3 | 7.7 | 12.0 | 3.7 | 2.7 | 7.1 | 4.1 | 1.4 |
| hopt0 | 4.7 | 2.0 | 6.7 | 7.9 | 4.8 | 2.3 | 7.1 | 2.9 | 1.8 |
| hopt1 | 4.1 | 3.0 | 4.9 | 7.8 | **2.4** | 3.4 | 5.5 | 4.9 | 1.1 |
| midnight | **3.2** | 1.2 | **4.6** | 6.0 | 3.4 | **0.9** | 4.3 | 2.3 | 2.9 |
| phikon | 6.4 | 2.5 | 9.3 | 12.1 | 5.6 | 3.1 | 10.3 | 4.3 | 2.5 |
| phikon2 | 4.6 | 1.3 | 7.0 | 12.3 | 2.6 | 1.4 | 7.1 | 2.8 | **0.9** |
| uni | 4.3 | 1.8 | 6.1 | 9.5 | 2.9 | 2.1 | 6.8 | 2.8 | 1.8 |
| uni2h | 4.5 | 3.0 | 5.5 | 7.1 | 4.4 | 2.6 | 4.8 | 5.3 | 2.4 |
| virchow | 5.5 | 1.8 | 8.1 | 8.9 | 6.7 | 1.5 | 9.4 | 3.3 | 3.3 |
| virchow2 | 4.6 | 3.5 | 5.4 | 5.4 | 5.3 | 3.1 | 6.6 | 3.4 | 3.8 |
| conch | 4.3 | 2.0 | 5.9 | 8.3 | 3.7 | 1.9 | 6.3 | 3.6 | 1.7 |
| titan | 4.9 | 1.7 | 7.2 | 8.6 | 5.7 | 1.1 | 7.3 | 3.5 | 2.8 |
| keep | 4.7 | **1.1** | 7.3 | 5.8 | 7.0 | 1.1 | 8.3 | **1.9** | 2.3 |
| musk | 4.5 | 3.1 | 5.5 | 5.0 | 4.6 | 4.0 | 6.3 | 4.5 | 1.5 |
| plip | 4.9 | 1.9 | 7.1 | 8.3 | 4.6 | 2.8 | 6.3 | 3.5 | 3.2 |
| quilt | 7.0 | 2.1 | 10.4 | 13.2 | 6.5 | 2.9 | 7.4 | 3.6 | 4.1 |
| dinob | 5.5 | 1.6 | 8.3 | 11.9 | 4.9 | 1.6 | 9.6 | 1.9 | 1.6 |
| dinol | 5.3 | 3.3 | 6.7 | 7.0 | 4.5 | 4.9 | 5.3 | 4.4 | 5.6 |
| vitb | 3.9 | 2.0 | 5.2 | 6.2 | 2.8 | 3.5 | 4.6 | 2.8 | 2.9 |
| vitl | 5.0 | 2.0 | 7.2 | 9.3 | 4.7 | 2.2 | 8.1 | 2.4 | 1.3 |
| clipb | 5.5 | 4.0 | 6.5 | 5.9 | 5.7 | 4.8 | 7.4 | 4.9 | 1.4 |
| clipl | 4.2 | 1.7 | 6.0 | 7.1 | 4.3 | 1.9 | **3.7** | 4.0 | 3.4 |

Table S26: Aggregated quantitative performance (MCE) on linear probing.

| Model | all | nb classes | | magnif. | | | cancer type | | |
|---|---|---|---|---|---|---|---|---|---|
| | | 2 | >2 | ≥40× | <40× ≥20× | <20× | breast | crc | multi |
| hiboub | 9.8 | 2.8 | 14.8 | **6.0** | 13.7 | 7.8 | 8.2 | 14.1 | 4.7 |
| hiboul | 12.9 | 4.1 | 19.2 | 26.2 | 8.7 | 8.2 | 15.0 | 11.2 | 3.6 |
| hopt0 | 11.9 | 5.7 | 16.3 | 14.3 | 14.2 | 7.2 | 12.2 | 15.1 | 6.6 |
| hopt1 | 17.7 | 9.8 | 23.4 | 37.1 | 7.5 | 15.9 | 18.8 | 13.9 | 3.7 |
| midnight | 12.1 | **2.4** | 19.0 | 12.7 | 19.6 | **2.4** | 15.0 | 14.7 | 10.8 |
| phikon | 15.1 | 4.6 | 22.6 | 22.8 | 17.7 | 6.1 | 21.3 | 15.0 | 5.9 |
| phikon2 | 11.6 | 2.6 | 18.0 | 24.9 | 7.5 | 6.8 | 14.7 | 9.0 | **2.2** |
| uni | 10.4 | 3.6 | 15.3 | 19.6 | 8.4 | 6.0 | 10.8 | 8.7 | 4.7 |
| uni2h | 15.8 | 6.1 | 22.8 | 21.7 | 18.9 | 7.7 | 19.2 | 18.0 | 5.9 |
| virchow | 23.0 | 3.3 | 37.2 | 22.6 | 23.4 | 22.9 | 25.3 | 15.2 | 6.6 |
| virchow2 | 17.7 | 7.8 | 24.7 | 28.6 | 15.5 | 12.3 | 18.2 | 14.8 | 10.2 |
| conch | 11.3 | 4.6 | 16.1 | 14.2 | 14.7 | 4.9 | 12.4 | 16.3 | 4.3 |
| titan | 12.1 | **2.4** | 19.0 | 19.5 | 15.0 | 2.8 | 17.8 | 9.4 | 5.5 |
| keep | 12.2 | 3.0 | 18.9 | 11.3 | 17.6 | 6.3 | 11.1 | 14.6 | 6.2 |
| musk | 9.6 | 5.6 | **12.4** | 9.2 | 11.5 | 7.4 | 14.2 | 7.8 | 3.3 |
| plip | 12.5 | 4.2 | 18.4 | 16.7 | 14.1 | 7.4 | 14.2 | 8.9 | 5.9 |
| quilt | 17.8 | 3.5 | 28.0 | 18.9 | 13.4 | 22.4 | 10.8 | 11.1 | 7.4 |
| dinob | 16.8 | 3.6 | 26.2 | 34.6 | 13.7 | 7.2 | 26.4 | 10.5 | 2.6 |
| dinol | 16.5 | 5.5 | 24.3 | 30.3 | 13.6 | 9.6 | 18.7 | 18.0 | 9.2 |
| vitb | **9.1** | 4.2 | 12.6 | 18.2 | **6.3** | 5.9 | 12.2 | **7.4** | 5.5 |
| vitl | 12.0 | 3.3 | 18.3 | 16.3 | 13.0 | 7.7 | 16.2 | 8.2 | 2.5 |
| clipb | 13.8 | 10.2 | 16.4 | 10.3 | 13.5 | 16.8 | 10.5 | 16.2 | 2.3 |
| clipl | 9.7 | 3.5 | 14.1 | 13.3 | 11.9 | 4.3 | **7.5** | 13.5 | 5.8 |

Table S27: Aggregated quantitative performance (ACE) on linear probing.

| Model | all | nb classes | | magnif. | | | cancer type | | |
|---|---|---|---|---|---|---|---|---|---|
| | | 2 | >2 | ≥40× | <40× ≥20× | <20× | breast | crc | multi |
| hiboub | 3.8 | 1.7 | 5.3 | **4.4** | 4.8 | 2.2 | 5.7 | 2.7 | 2.2 |
| hiboul | 5.3 | 2.3 | 7.5 | 11.7 | 3.6 | 2.7 | 6.8 | 4.1 | 1.4 |
| hopt0 | 4.6 | 1.8 | 6.7 | 7.8 | 4.8 | 2.1 | 7.1 | 2.6 | 1.7 |
| hopt1 | 3.9 | 3.0 | **4.7** | 7.1 | **2.4** | 3.5 | 5.1 | 4.7 | 1.1 |
| midnight | **3.4** | 1.3 | 4.8 | 6.5 | 3.5 | **0.9** | 4.6 | 2.5 | 2.7 |
| phikon | 6.2 | 2.6 | 8.9 | 12.2 | 5.1 | 3.2 | 9.8 | 4.3 | 2.5 |
| phikon2 | 4.6 | **1.2** | 7.1 | 12.4 | 2.5 | 1.4 | 7.1 | 2.6 | **0.8** |
| uni | 4.3 | 1.6 | 6.3 | 9.7 | 2.9 | 2.1 | 6.9 | 2.6 | 1.8 |
| uni2h | 4.6 | 3.1 | 5.7 | 7.3 | 4.4 | 2.8 | 4.9 | 5.3 | 2.4 |
| virchow | 5.3 | 1.8 | 7.7 | 8.5 | 6.3 | 1.6 | 8.8 | 3.3 | 3.3 |
| virchow2 | 5.0 | 3.7 | 5.8 | 5.5 | 5.8 | 3.4 | 7.1 | 3.6 | 3.9 |
| conch | 4.0 | 1.9 | 5.4 | 7.9 | 3.4 | 1.8 | 5.7 | 3.4 | 1.7 |
| titan | 5.0 | 2.1 | 7.0 | 8.7 | 5.5 | 1.5 | 7.1 | 4.1 | 2.8 |
| keep | 4.8 | 1.3 | 7.4 | 5.6 | 7.1 | 1.4 | 8.3 | **2.1** | 2.3 |
| musk | 4.6 | 3.0 | 5.7 | 5.4 | 4.7 | 3.8 | 6.5 | 4.3 | 1.5 |
| plip | 5.0 | 1.8 | 7.3 | 8.6 | 4.7 | 2.6 | 6.5 | 3.4 | 3.2 |
| quilt | 7.1 | 2.1 | 10.6 | 13.2 | 6.7 | 3.0 | 7.6 | 3.7 | 4.0 |
| dinob | 5.5 | 1.9 | 8.0 | 11.8 | 4.5 | 1.9 | 9.2 | 2.4 | 1.6 |
| dinol | 5.1 | 3.2 | 6.5 | 6.9 | 4.3 | 4.8 | 5.0 | 4.4 | 5.5 |
| vitb | 3.9 | 2.2 | 5.1 | 5.9 | 2.8 | 3.8 | 4.5 | 3.1 | 2.9 |
| vitl | 5.0 | 2.1 | 7.1 | 9.2 | 4.7 | 2.3 | 8.1 | 2.5 | 1.3 |
| clipb | 5.6 | 4.1 | 6.7 | 6.4 | 5.7 | 5.0 | 7.7 | 5.1 | 1.3 |
| clipl | 4.4 | 1.7 | 6.2 | 7.5 | 4.4 | 1.9 | **4.1** | 4.0 | 3.5 |

Table S28: Aggregated quantitative performance (TACE) on linear probing.

| Model | all | nb classes | | magnif. | | | cancer type | | |
|---|---|---|---|---|---|---|---|---|---|
| | | 2 | >2 | ≥40× | <40× ≥20× | <20× | breast | crc | multi |
| hiboub | 3.8 | 1.7 | 5.3 | **4.4** | 4.8 | 2.2 | 5.7 | 2.7 | 2.2 |
| hiboul | 5.3 | 2.3 | 7.5 | 11.7 | 3.6 | 2.7 | 6.8 | 4.1 | 1.4 |
| hopt0 | 4.6 | 1.8 | 6.7 | 7.8 | 4.8 | 2.1 | 7.1 | 2.6 | 1.7 |
| hopt1 | 3.9 | 3.0 | **4.7** | 7.1 | **2.4** | 3.5 | 5.1 | 4.7 | 1.1 |
| midnight | **3.4** | 1.3 | 4.8 | 6.5 | 3.5 | **0.9** | 4.6 | 2.5 | 2.7 |
| phikon | 6.2 | 2.6 | 8.9 | 12.2 | 5.1 | 3.2 | 9.8 | 4.3 | 2.5 |
| phikon2 | 4.6 | **1.2** | 7.1 | 12.4 | 2.5 | 1.4 | 7.1 | 2.6 | **0.8** |
| uni | 4.3 | 1.6 | 6.3 | 9.7 | 2.9 | 2.1 | 6.9 | 2.6 | 1.8 |
| uni2h | 4.6 | 3.1 | 5.7 | 7.3 | 4.4 | 2.8 | 4.9 | 5.3 | 2.4 |
| virchow | 5.3 | 1.8 | 7.7 | 8.5 | 6.3 | 1.6 | 8.8 | 3.3 | 3.3 |
| virchow2 | 5.0 | 3.7 | 5.8 | 5.5 | 5.8 | 3.4 | 7.1 | 3.6 | 3.9 |
| conch | 4.0 | 1.9 | 5.4 | 7.9 | 3.4 | 1.8 | 5.7 | 3.4 | 1.7 |
| titan | 5.0 | 2.1 | 7.0 | 8.7 | 5.5 | 1.5 | 7.1 | 4.1 | 2.8 |
| keep | 4.8 | 1.3 | 7.4 | 5.6 | 7.1 | 1.4 | 8.3 | **2.1** | 2.3 |
| musk | 4.6 | 3.0 | 5.7 | 5.4 | 4.7 | 3.8 | 6.5 | 4.3 | 1.5 |
| plip | 5.0 | 1.8 | 7.3 | 8.6 | 4.7 | 2.6 | 6.5 | 3.4 | 3.2 |
| quilt | 7.1 | 2.1 | 10.6 | 13.2 | 6.7 | 3.0 | 7.6 | 3.7 | 4.0 |
| dinob | 5.5 | 1.9 | 8.0 | 11.8 | 4.5 | 1.9 | 9.2 | 2.4 | 1.6 |
| dinol | 5.1 | 3.2 | 6.5 | 6.9 | 4.3 | 4.8 | 5.0 | 4.4 | 5.5 |
| vitb | 3.9 | 2.2 | 5.1 | 5.9 | 2.8 | 3.8 | 4.5 | 3.1 | 2.9 |
| vitl | 5.0 | 2.1 | 7.1 | 9.2 | 4.7 | 2.3 | 8.1 | 2.5 | 1.3 |
| clipb | 5.6 | 4.1 | 6.7 | 6.4 | 5.7 | 5.0 | 7.7 | 5.1 | 1.3 |
| clipl | 4.4 | 1.7 | 6.2 | 7.5 | 4.4 | 1.9 | **4.1** | 4.0 | 3.5 |

Table S29: Aggregated quantitative performance (SCE) on linear probing.

| Model | all | nb classes | | magnif. | | | cancer type | | |
|---|---|---|---|---|---|---|---|---|---|
| | | 2 | >2 | ≥40× | <40× ≥20× | <20× | breast | crc | multi |
| hiboub | 4.0 | 1.2 | 6.0 | **2.7** | 6.0 | 2.4 | 4.1 | 4.8 | 2.5 |
| hiboul | 5.6 | 1.8 | 8.4 | 11.2 | 4.2 | 3.2 | 6.8 | 4.7 | 2.0 |
| hopt0 | 5.4 | 2.2 | 7.7 | 7.1 | 6.5 | 2.7 | 5.8 | 6.1 | 3.5 |
| hopt1 | 6.0 | 3.0 | 8.1 | 11.3 | 3.1 | 5.6 | 6.2 | 4.7 | 1.8 |
| midnight | 4.6 | 1.1 | 7.1 | 5.5 | 7.0 | **0.9** | 5.1 | 4.8 | 6.2 |
| phikon | 6.9 | 1.6 | 10.7 | 10.5 | 8.4 | 2.4 | 9.3 | 7.6 | 2.8 |
| phikon2 | 4.4 | **0.9** | 6.9 | 10.2 | 2.6 | 2.2 | 6.2 | **2.4** | **1.1** |
| uni | 4.6 | 1.3 | 7.0 | 9.8 | 2.9 | 3.0 | 5.6 | 2.8 | 2.2 |
| uni2h | 6.4 | 2.3 | 9.3 | 8.6 | 7.5 | 3.3 | 6.9 | 7.5 | 3.2 |
| virchow | 8.3 | 1.2 | 13.4 | 9.8 | 9.7 | 5.4 | 10.6 | 5.6 | 4.3 |
| virchow2 | 6.6 | 2.9 | 9.2 | 9.0 | 6.4 | 4.9 | 7.1 | 3.9 | 6.2 |
| conch | 4.6 | 1.6 | 6.7 | 6.8 | 5.3 | 2.0 | 5.2 | 5.7 | 2.4 |
| titan | 5.4 | 1.0 | 8.5 | 8.7 | 6.8 | 1.1 | 7.7 | 4.4 | 3.2 |
| keep | 5.1 | 0.9 | 8.1 | 5.7 | 6.9 | 2.4 | 5.2 | 4.4 | 3.9 |
| musk | 4.1 | 2.0 | 5.6 | 4.2 | 5.1 | 2.9 | 5.9 | 3.4 | 1.9 |
| plip | 5.2 | 1.3 | 8.0 | 8.0 | 5.4 | 2.9 | 6.1 | 3.1 | 3.6 |
| quilt | 7.4 | 1.3 | 11.7 | 9.4 | 6.6 | 6.9 | 5.4 | 4.2 | 4.4 |
| dinob | 6.5 | 1.2 | 10.2 | 14.2 | 5.3 | 2.0 | 10.6 | 3.3 | 1.6 |
| dinol | 6.5 | 2.0 | 9.6 | 10.9 | 5.6 | 4.2 | 6.5 | 5.7 | 5.7 |
| vitb | **3.7** | 1.4 | **5.3** | 6.9 | **2.5** | 2.7 | 4.5 | 2.5 | 2.9 |
| vitl | 5.2 | 1.3 | 8.1 | 8.3 | 5.4 | 2.7 | 7.1 | 3.0 | 1.6 |
| clipb | 5.6 | 3.2 | 7.3 | 5.3 | 5.6 | 5.8 | 5.4 | 5.3 | 1.5 |
| clipl | 4.4 | 1.2 | 6.7 | 6.8 | 5.1 | 1.7 | **3.3** | 5.2 | 3.5 |

Table S30: Aggregated quantitative performance (Drop in Balanced accuracy) on adversarial attack ($\epsilon = 0.25 \cdot 10^{-3}$).

| Model | all | nb classes | | magnif. | | | cancer type | | |
|---|---|---|---|---|---|---|---|---|---|
| | | 2 | >2 | 0.25 | 0.5 | 1.0 | breast | crc | multi |
| hiboub | 7.1 | 7.2 | 7.1 | 6.7 | 8.4 | 5.9 | 5.8 | 9.2 | 9.6 |
| hiboul | 5.5 | 5.6 | 5.5 | 6.3 | 5.1 | 5.5 | 5.3 | 7.0 | 6.2 |
| hopt0 | 5.4 | 6.2 | 4.7 | 5.3 | 5.0 | 5.8 | 5.6 | 5.7 | 6.2 |
| hopt1 | 9.8 | 12.9 | 7.6 | 9.2 | 7.6 | 13.0 | 11.1 | 13.2 | 8.2 |
| midnight | 5.7 | 5.4 | 6.0 | 8.5 | 3.9 | 5.9 | 6.4 | 7.4 | **3.9** |
| phikon | 5.0 | 5.4 | 4.7 | 4.8 | 5.4 | 4.7 | 4.8 | 6.6 | 5.2 |
| phikon2 | 6.2 | 7.0 | 5.6 | 6.0 | 5.7 | 7.0 | 5.5 | 9.2 | 5.7 |
| uni | 5.2 | 5.9 | 4.7 | 5.0 | 5.5 | 4.9 | 4.6 | 6.5 | 6.7 |
| uni2h | 4.3 | 5.1 | 3.8 | 4.7 | **3.9** | 4.5 | **3.2** | 7.0 | 5.1 |
| virchow | 5.3 | 5.4 | 5.2 | 5.7 | 5.4 | 4.8 | 4.8 | 7.4 | 5.8 |
| virchow2 | **3.9** | **4.3** | **3.7** | **3.1** | 4.3 | **4.0** | 4.0 | **3.7** | 5.5 |
| conch | 8.5 | 9.4 | 7.8 | 7.3 | 8.2 | 9.7 | 8.1 | 10.0 | 9.6 |
| titan | 12.4 | 14.2 | 11.1 | 11.0 | 12.3 | 13.6 | 9.0 | 16.9 | 16.6 |
| keep | 5.2 | 6.4 | 4.4 | 4.6 | 5.5 | 5.3 | 4.0 | 7.3 | 7.7 |
| musk | 13.9 | 15.9 | 12.5 | 11.4 | 14.7 | 14.8 | 11.5 | 17.7 | 20.1 |
| plip | 15.1 | 19.2 | 12.2 | 9.9 | 15.6 | 18.5 | 13.3 | 20.3 | 17.3 |
| quilt | 13.4 | 15.4 | 12.0 | 10.3 | 13.1 | 16.2 | 11.6 | 15.9 | 18.4 |
| dinob | 11.8 | 12.1 | 11.5 | 12.3 | 12.7 | 10.2 | 11.5 | 12.4 | 13.4 |
| dinol | 11.0 | 12.0 | 10.3 | 8.5 | 13.1 | 10.2 | 9.9 | 14.1 | 12.7 |
| vitb | 6.9 | 6.6 | 7.1 | 6.8 | 7.3 | 6.5 | 6.0 | 8.4 | 7.7 |
| vitl | 6.6 | 5.9 | 7.1 | 7.8 | 6.6 | 5.9 | 6.3 | 8.1 | 7.0 |
| clipb | 22.8 | 27.8 | 19.2 | 15.4 | 24.6 | 26.0 | 18.6 | 26.9 | 28.2 |
| clipl | 25.1 | 26.3 | 24.2 | 21.4 | 27.6 | 24.8 | 24.5 | 27.9 | 27.8 |

Table S31: Aggregated quantitative performance (Drop in F1-score) on adversarial attack ($\epsilon = 0.25 \cdot 10^{-3}$).

| Model | all | nb classes | | magnif. | | | cancer type | | |
|---|---|---|---|---|---|---|---|---|---|
| | | 2 | >2 | 0.25 | 0.5 | 1.0 | breast | crc | multi |
| hiboub | 7.7 | 7.7 | 7.7 | 7.3 | 8.8 | 6.8 | 6.6 | 9.3 | 10.2 |
| hiboul | 5.8 | 6.1 | 5.6 | 6.3 | 5.3 | 6.2 | 5.7 | 7.2 | 6.3 |
| hopt0 | 5.9 | 7.2 | 5.0 | 5.3 | 5.2 | 7.2 | 6.5 | 5.8 | 6.3 |
| hopt1 | 10.3 | 13.4 | 8.0 | 9.5 | 7.7 | 14.0 | 11.9 | 13.2 | 8.3 |
| midnight | 5.8 | 5.2 | 6.2 | 8.9 | **3.9** | 6.0 | 6.1 | 7.5 | **4.0** |
| phikon | 5.1 | 5.7 | 4.7 | 4.7 | 5.4 | 5.1 | 5.0 | 6.5 | 5.4 |
| phikon2 | 6.4 | 7.2 | 5.8 | 6.1 | 5.8 | 7.2 | 5.5 | 9.2 | 5.9 |
| uni | 5.4 | 6.3 | 4.8 | 4.8 | 5.7 | 5.5 | 4.9 | 6.5 | 6.9 |
| uni2h | 4.7 | 5.5 | 4.1 | 4.9 | 4.2 | 5.1 | **3.5** | 7.3 | 5.3 |
| virchow | 5.3 | 5.5 | 5.2 | 5.6 | 5.4 | 5.1 | 4.8 | 7.1 | 6.1 |
| virchow2 | **4.0** | **4.6** | **3.6** | **3.1** | 4.4 | **4.2** | 4.1 | **3.8** | 5.7 |
| conch | 8.8 | 9.9 | 8.0 | 7.5 | 8.4 | 10.2 | 8.5 | 10.0 | 9.9 |
| titan | 12.8 | 14.4 | 11.7 | 11.7 | 12.4 | 14.1 | 9.6 | 16.6 | 17.3 |
| keep | 5.3 | 6.2 | 4.6 | 4.5 | 5.9 | 5.2 | 4.0 | 7.3 | 8.1 |
| musk | 14.1 | 16.0 | 12.8 | 11.5 | 15.0 | 14.9 | 11.2 | 18.2 | 20.9 |
| plip | 15.5 | 18.8 | 13.2 | 10.9 | 15.7 | 18.8 | 13.9 | 19.3 | 18.8 |
| quilt | 14.5 | 16.4 | 13.2 | 11.3 | 14.0 | 17.6 | 12.7 | 15.8 | 20.6 |
| dinob | 11.8 | 11.9 | 11.7 | 12.2 | 12.8 | 10.2 | 11.2 | 11.9 | 14.5 |
| dinol | 11.2 | 12.2 | 10.5 | 8.4 | 13.3 | 10.8 | 10.2 | 13.6 | 14.0 |
| vitb | 7.0 | 6.8 | 7.2 | 6.9 | 7.3 | 6.8 | 6.2 | 8.3 | 8.2 |
| vitl | 6.8 | 6.0 | 7.4 | 7.7 | 6.9 | 6.1 | 6.5 | 8.0 | 7.5 |
| clipb | 23.7 | 29.0 | 19.9 | 16.4 | 25.5 | 27.1 | 19.8 | 26.7 | 31.1 |
| clipl | 25.6 | 26.3 | 25.1 | 22.7 | 27.6 | 25.3 | 25.0 | 26.8 | 29.9 |

Table S32: Aggregated quantitative performance (Drop in Balanced accuracy) on adversarial attack ($\epsilon = 1.5 \cdot 10^{-3}$).

| Model | all | nb classes | | magnif. | | | cancer type | | |
|---|---|---|---|---|---|---|---|---|---|
| | | 2 | >2 | 0.25 | 0.5 | 1.0 | breast | crc | multi |
| hiboub | 53.0 | 62.5 | 46.3 | 40.8 | 56.4 | 58.1 | 54.3 | 58.0 | 57.9 |
| hiboul | 39.9 | 41.1 | 39.1 | 39.1 | 39.9 | 40.5 | 40.2 | 45.8 | 42.4 |
| hopt0 | 44.4 | 55.7 | 36.3 | 34.4 | 41.8 | 55.1 | 48.4 | 45.4 | 46.6 |
| hopt1 | 58.4 | 71.2 | 49.2 | 51.8 | 52.4 | 70.8 | 69.4 | 59.0 | 54.9 |
| midnight | 35.7 | **37.2** | 34.5 | 37.9 | **29.4** | 41.9 | 40.5 | 39.1 | **26.4** |
| phikon | 34.9 | 41.1 | 30.6 | 31.4 | 35.8 | 36.6 | 35.1 | 40.6 | 39.3 |
| phikon2 | 46.0 | 54.9 | 39.6 | 34.9 | 43.8 | 57.0 | 49.5 | 51.8 | 42.2 |
| uni | 42.9 | 56.9 | 32.9 | 31.3 | 42.7 | 51.8 | 45.5 | 45.4 | 51.1 |
| uni2h | 33.7 | 39.7 | 29.4 | 30.7 | 33.7 | **35.9** | **32.3** | 42.0 | 39.1 |
| virchow | 41.5 | 49.5 | 35.8 | 36.4 | 39.1 | 48.2 | 45.6 | 45.5 | 42.3 |
| virchow2 | **33.2** | 44.1 | **25.3** | **26.1** | 31.5 | 40.5 | 36.4 | **32.4** | 42.0 |
| conch | 55.9 | 60.7 | 52.5 | 53.8 | 53.7 | 60.1 | 52.4 | 62.6 | 57.2 |
| titan | 77.0 | 82.9 | 72.8 | 71.5 | 74.6 | 84.0 | 76.7 | 79.1 | 74.5 |
| keep | 44.3 | 53.2 | 37.9 | 38.7 | 44.4 | 48.2 | 46.0 | 46.6 | 52.0 |
| musk | 70.2 | 81.7 | 62.0 | 60.7 | 69.2 | 78.6 | 68.3 | 76.1 | 74.9 |
| plip | 56.6 | 66.2 | 49.7 | 40.7 | 60.6 | 63.5 | 50.7 | 68.3 | 61.2 |
| quilt | 51.9 | 58.2 | 47.4 | 40.9 | 55.4 | 55.9 | 51.2 | 56.9 | 58.8 |
| dinob | 66.7 | 74.6 | 61.1 | 63.1 | 65.5 | 71.0 | 67.2 | 70.2 | 64.5 |
| dinol | 65.6 | 74.2 | 59.5 | 59.2 | 65.9 | 70.0 | 65.3 | 70.4 | 64.1 |
| vitb | 48.0 | 53.4 | 44.2 | 43.9 | 45.8 | 54.0 | 49.6 | 52.2 | 43.5 |
| vitl | 44.1 | 47.6 | 41.7 | 39.3 | 42.8 | 49.5 | 48.0 | 45.6 | 40.3 |
| clipb | 61.8 | 72.9 | 53.8 | 48.4 | 61.9 | 71.6 | 58.4 | 67.6 | 61.6 |
| clipl | 68.9 | 77.6 | 62.7 | 59.2 | 66.2 | 79.6 | 67.6 | 72.1 | 65.8 |

Table S33: Aggregated quantitative performance (Drop in F1-score) on adversarial attack ($\epsilon = 1.5 \cdot 10^{-3}$).

| Model | all | nb classes | | magnif. | | | cancer type | | |
|---|---|---|---|---|---|---|---|---|---|
| | | 2 | >2 | 0.25 | 0.5 | 1.0 | breast | crc | multi |
| hiboub | 52.8 | 60.4 | 47.3 | 42.7 | 54.5 | 58.2 | 54.4 | 55.3 | 58.7 |
| hiboul | 40.0 | 40.8 | 39.4 | 39.1 | 39.9 | 40.8 | 39.8 | 44.4 | 44.3 |
| hopt0 | 44.2 | 55.0 | 36.4 | 35.0 | 41.1 | 54.9 | 48.7 | 43.1 | 47.6 |
| hopt1 | 58.0 | 70.5 | 49.0 | 51.6 | 51.0 | 71.4 | 69.5 | 56.4 | 55.6 |
| midnight | 36.4 | **37.4** | 35.6 | 38.7 | **29.9** | 42.6 | 40.6 | 39.0 | **28.1** |
| phikon | 34.4 | 40.0 | 30.5 | 30.5 | 35.4 | **36.1** | 34.5 | 38.3 | 41.4 |
| phikon2 | 45.6 | 54.0 | 39.7 | 35.9 | 43.1 | 56.2 | 48.8 | 49.6 | 44.0 |
| uni | 42.8 | 56.0 | 33.4 | 31.6 | 41.7 | 52.5 | 45.7 | 43.5 | 51.5 |
| uni2h | 34.3 | 40.0 | 30.3 | 31.1 | 34.4 | 36.7 | **32.7** | 41.5 | 40.7 |
| virchow | 41.1 | 48.3 | 35.9 | 36.9 | 38.6 | 47.2 | 45.0 | 43.4 | 44.0 |
| virchow2 | **33.6** | 45.0 | **25.6** | **25.6** | 31.6 | 42.3 | 37.5 | **31.0** | 43.2 |
| conch | 55.0 | 58.7 | 52.3 | 54.2 | 52.4 | 58.8 | 52.2 | 59.5 | 57.8 |
| titan | 75.3 | 80.6 | 71.5 | 70.9 | 72.1 | 82.6 | 74.9 | 76.5 | 74.2 |
| keep | 44.7 | 52.9 | 38.8 | 38.0 | 45.0 | 49.3 | 47.0 | 45.4 | 53.1 |
| musk | 69.3 | 80.1 | 61.6 | 61.5 | 67.9 | 76.8 | 67.0 | 74.1 | 76.2 |
| plip | 56.9 | 65.0 | 51.0 | 43.3 | 58.9 | 64.5 | 52.8 | 64.9 | 61.0 |
| quilt | 52.7 | 57.2 | 49.5 | 44.7 | 55.4 | 55.5 | 52.2 | 55.0 | 60.9 |
| dinob | 65.8 | 72.9 | 60.7 | 63.3 | 63.1 | 71.2 | 66.4 | 66.7 | 64.5 |
| dinol | 64.5 | 72.2 | 59.1 | 60.0 | 63.2 | 69.6 | 65.5 | 66.1 | 64.1 |
| vitb | 46.8 | 52.1 | 43.1 | 43.9 | 43.9 | 52.8 | 48.2 | 49.7 | 45.2 |
| vitl | 44.0 | 47.1 | 41.9 | 39.7 | 42.2 | 49.7 | 47.5 | 44.9 | 42.4 |
| clipb | 60.4 | 70.3 | 53.4 | 49.0 | 59.9 | 69.6 | 57.3 | 63.2 | 63.5 |
| clipl | 67.8 | 75.7 | 62.1 | 59.2 | 63.9 | 78.9 | 67.1 | 68.7 | 66.4 |

Table S34: Aggregated quantitative performance (Drop in Balanced accuracy) on adversarial attack ($\epsilon = 35 \cdot 10^{-3}$).

| Model | all | nb classes | | magnif. | | | cancer type | | |
|---|---|---|---|---|---|---|---|---|---|
| | | 2 | >2 | 0.25 | 0.5 | 1.0 | breast | crc | multi |
| hiboub | 78.8 | 86.1 | 73.6 | 67.6 | 78.5 | 87.5 | 76.9 | 79.9 | 78.6 |
| hiboul | 76.1 | 83.3 | 70.8 | 68.2 | 76.1 | 81.9 | 77.4 | 77.7 | 76.1 |
| hopt0 | 79.0 | 87.5 | 72.9 | 68.4 | 78.2 | 87.8 | 77.5 | 80.1 | 80.9 |
| hopt1 | 81.9 | 88.4 | 77.3 | 75.7 | 81.1 | 87.6 | 82.7 | 81.5 | 83.5 |
| midnight | 67.6 | **72.8** | 63.9 | 56.8 | **64.6** | 79.4 | 72.5 | **69.8** | **59.5** |
| phikon | 73.6 | 81.9 | 67.7 | 63.0 | 75.4 | 79.3 | 71.5 | 74.7 | 80.7 |
| phikon2 | 75.5 | 83.7 | 69.7 | 62.1 | 76.3 | 84.7 | 73.1 | 76.8 | 81.3 |
| uni | 78.4 | 87.4 | 72.0 | 69.2 | 77.5 | 86.5 | 78.5 | 78.3 | 80.2 |
| uni2h | 77.5 | 86.6 | 71.0 | 68.0 | 78.7 | 83.1 | 81.6 | 76.9 | 81.2 |
| virchow | 74.6 | 83.4 | 68.3 | 65.4 | 73.4 | 83.0 | 73.9 | 76.6 | 76.7 |
| virchow2 | 73.8 | 83.7 | 66.8 | 65.4 | 76.1 | 77.3 | 75.8 | 75.1 | 79.5 |
| conch | 80.6 | 84.6 | 77.7 | 71.8 | 79.9 | 88.0 | 80.8 | 80.6 | 75.1 |
| titan | 82.0 | 86.0 | 79.2 | 75.4 | 80.3 | 89.2 | 82.7 | 81.4 | 77.7 |
| keep | 78.9 | 85.1 | 74.5 | 69.8 | 79.7 | 84.8 | 79.0 | 79.7 | 78.9 |
| musk | 78.7 | 83.6 | 75.2 | 73.9 | 76.3 | 85.2 | 77.4 | 79.7 | 75.8 |
| plip | 70.7 | 80.9 | 63.5 | 57.4 | 70.3 | 81.2 | 67.9 | 76.0 | 68.9 |
| quilt | 68.9 | 78.1 | 62.3 | 57.3 | 69.8 | **76.5** | 68.5 | 73.9 | 68.5 |
| dinob | 75.6 | 82.6 | 70.5 | 69.7 | 71.7 | 84.8 | 75.9 | 79.1 | 67.8 |
| dinol | 75.8 | 82.4 | 71.1 | 69.7 | 73.2 | 83.7 | 75.6 | 79.8 | 68.8 |
| vitb | 71.0 | 78.7 | 65.6 | 65.2 | 68.7 | 78.4 | 69.5 | 75.7 | 66.8 |
| vitl | 71.6 | 79.9 | 65.7 | 63.9 | 68.6 | 81.2 | 70.9 | 74.4 | 68.3 |
| clipb | **67.0** | 78.1 | **59.1** | **54.8** | 65.6 | 78.1 | **63.7** | 74.8 | 63.4 |
| clipl | 71.8 | 79.6 | 66.2 | 62.5 | 69.7 | 81.2 | 70.3 | 75.3 | 67.8 |

Table S35: Aggregated quantitative performance (Drop in F1-score) on adversarial attack ($\epsilon = 35 \cdot 10^{-3}$).

| Model | all | nb classes | | magnif. | | | cancer type | | |
|---|---|---|---|---|---|---|---|---|---|
| | | 2 | >2 | 0.25 | 0.5 | 1.0 | breast | crc | multi |
| hiboub | 78.2 | 85.5 | 73.1 | 68.1 | 77.0 | 87.4 | 76.6 | 78.3 | 80.2 |
| hiboul | 75.0 | 81.6 | 70.2 | 69.3 | 73.7 | 80.8 | 77.0 | 74.6 | 76.2 |
| hopt0 | 78.4 | 85.9 | 73.1 | 69.4 | 76.7 | 87.3 | 77.8 | 77.6 | 81.2 |
| hopt1 | 81.1 | 87.2 | 76.8 | 75.6 | 79.3 | 87.6 | 82.1 | 79.3 | 84.2 |
| midnight | 67.7 | **72.3** | 64.4 | 57.8 | 64.3 | 79.4 | 72.4 | **68.7** | **60.6** |
| phikon | 72.9 | 80.5 | 67.4 | 62.6 | 74.3 | 78.8 | 70.6 | 72.7 | 81.2 |
| phikon2 | 74.1 | 82.5 | 68.1 | 60.3 | 75.1 | 83.2 | 71.5 | 75.3 | 82.1 |
| uni | 77.8 | 86.5 | 71.6 | 69.2 | 76.2 | 86.2 | 78.3 | 75.8 | 81.6 |
| uni2h | 76.7 | 85.2 | 70.6 | 68.1 | 77.0 | 82.8 | 81.0 | 75.0 | 81.3 |
| virchow | 73.8 | 81.6 | 68.2 | 65.5 | 72.1 | 82.1 | 73.7 | 74.7 | 76.3 |
| virchow2 | 72.8 | 82.0 | 66.2 | 64.8 | 74.8 | 76.2 | 75.0 | 72.7 | 79.6 |
| conch | 79.8 | 83.6 | 77.1 | 72.2 | 78.9 | 86.7 | 80.3 | 78.6 | 77.0 |
| titan | 80.7 | 84.5 | 78.0 | 75.0 | 77.9 | 88.5 | 81.1 | 78.9 | 78.6 |
| keep | 78.0 | 83.6 | 74.1 | 70.2 | 77.9 | 84.2 | 78.2 | 77.1 | 80.1 |
| musk | 77.9 | 82.1 | 74.8 | 75.1 | 75.0 | 83.4 | 77.1 | 77.9 | 77.6 |
| plip | 69.9 | 78.7 | 63.7 | 58.5 | 69.2 | 79.4 | 67.5 | 73.3 | 70.0 |
| quilt | 68.4 | 77.0 | 62.3 | 58.8 | 68.3 | **75.7** | 68.5 | 71.3 | 69.7 |
| dinob | 74.9 | 81.3 | 70.2 | 70.1 | 70.0 | 84.5 | 75.4 | 76.4 | 69.3 |
| dinol | 75.1 | 81.2 | 70.8 | 70.1 | 71.5 | 83.5 | 75.8 | 76.3 | 70.4 |
| vitb | 69.6 | 77.1 | 64.2 | 64.9 | 66.8 | 76.5 | 67.9 | 72.7 | 68.5 |
| vitl | 70.4 | 77.8 | 65.0 | 63.5 | 67.1 | 79.6 | 69.5 | 71.4 | 69.7 |
| clipb | 65.6 | 75.3 | **58.7** | **55.3** | **63.3** | 76.3 | **62.5** | 70.4 | 65.0 |
| clipl | 70.8 | 78.1 | 65.6 | 62.2 | 68.1 | 80.7 | 70.0 | 72.5 | 69.2 |

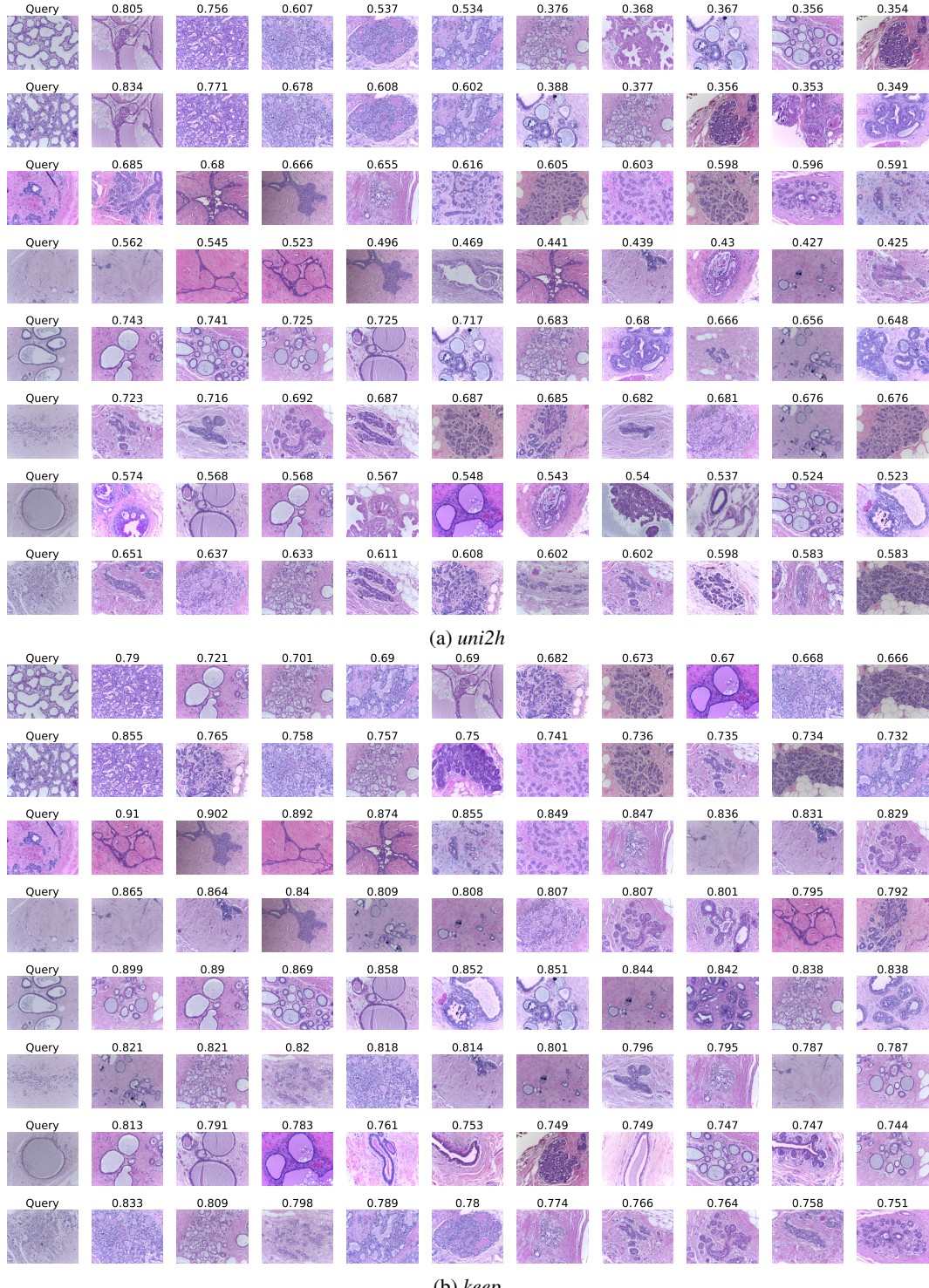

Figure S9: **Image retrieval**: Qualitative samples (query + top-10 with cosine similarity) on *bach*.

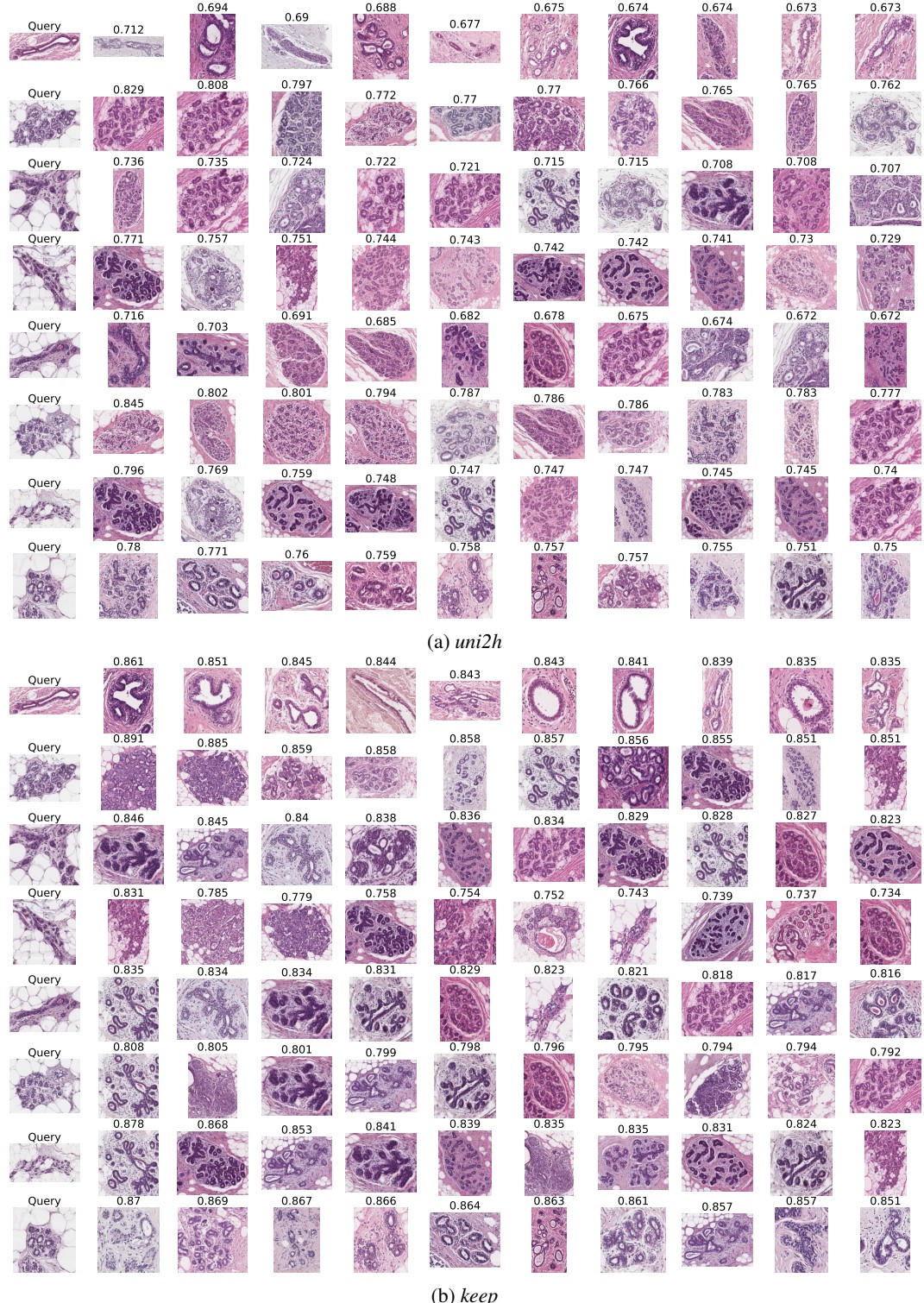

(a) *uni2h*

(b) *keep*

Figure S10: **Image retrieval**: Qualitative samples (query + top-10 with cosine similarity) on *bracs*.

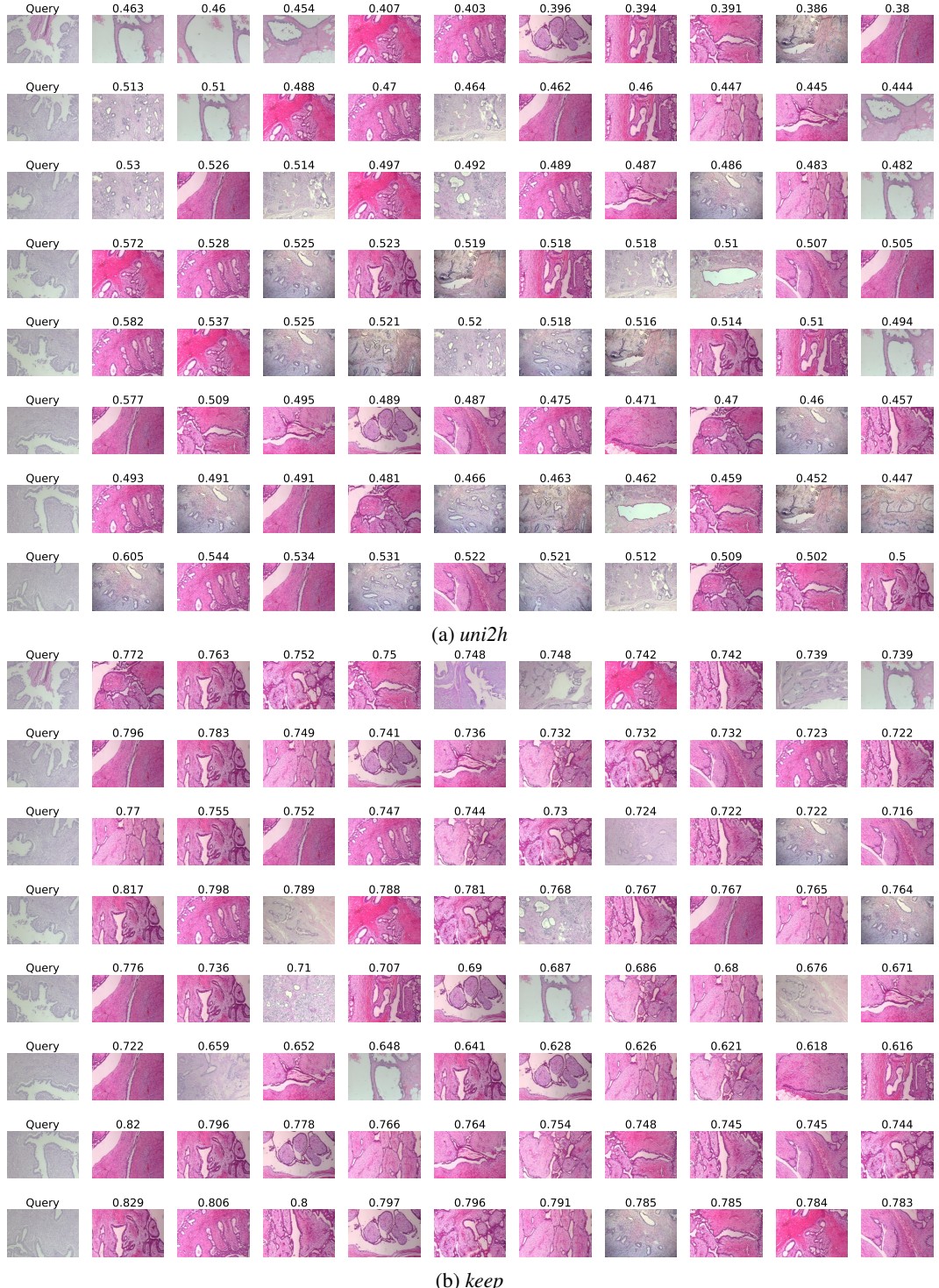

(a) *uni2h*

(b) *keep*

Figure S11: **Image retrieval**: Qualitative samples (query + top-10 with cosine similarity) on *break-h*.

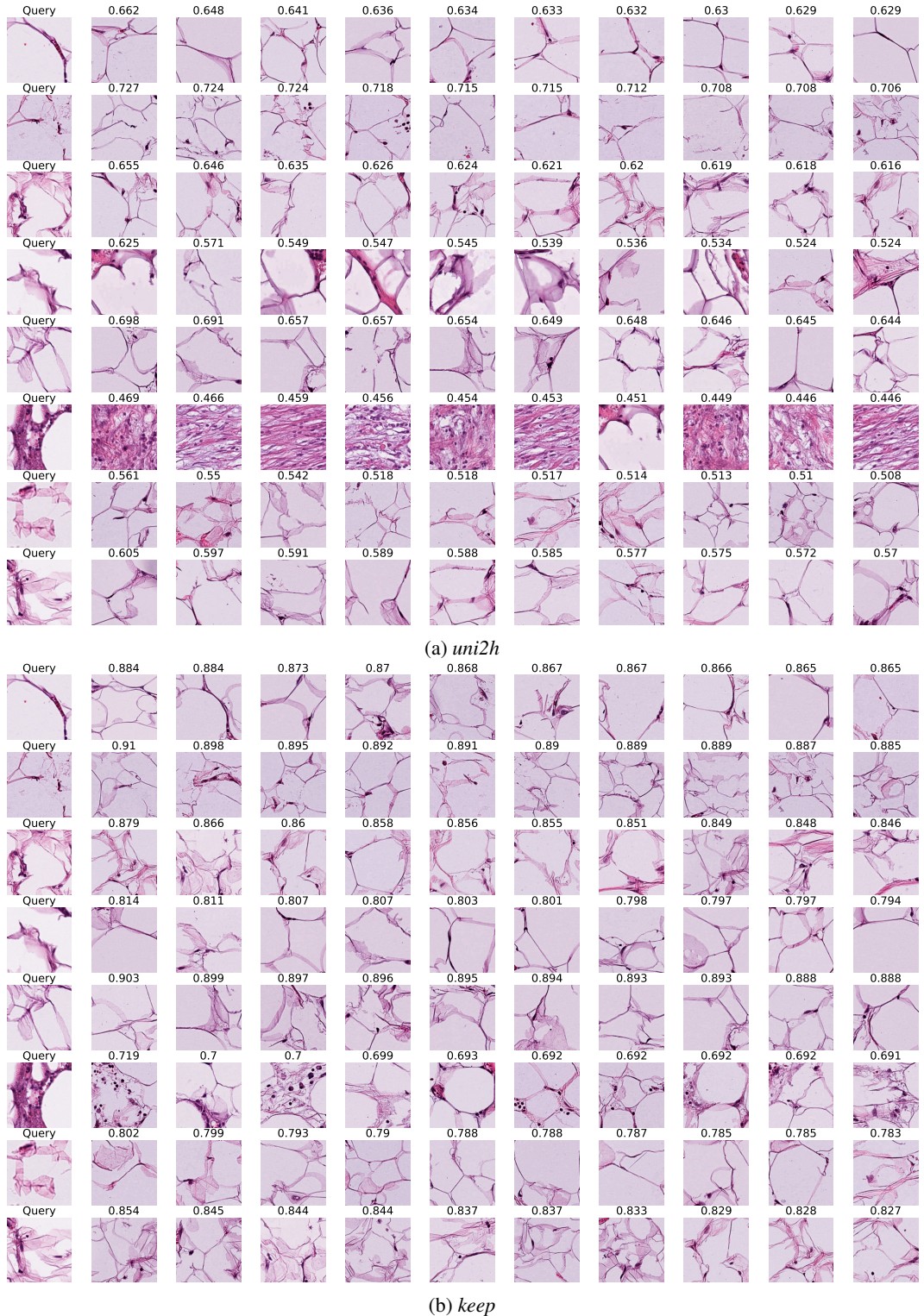

(a) *uni2h*

(b) *keep*

Figure S12: **Image retrieval**: Qualitative samples (query + top-10 with cosine similarity) on *crc*.

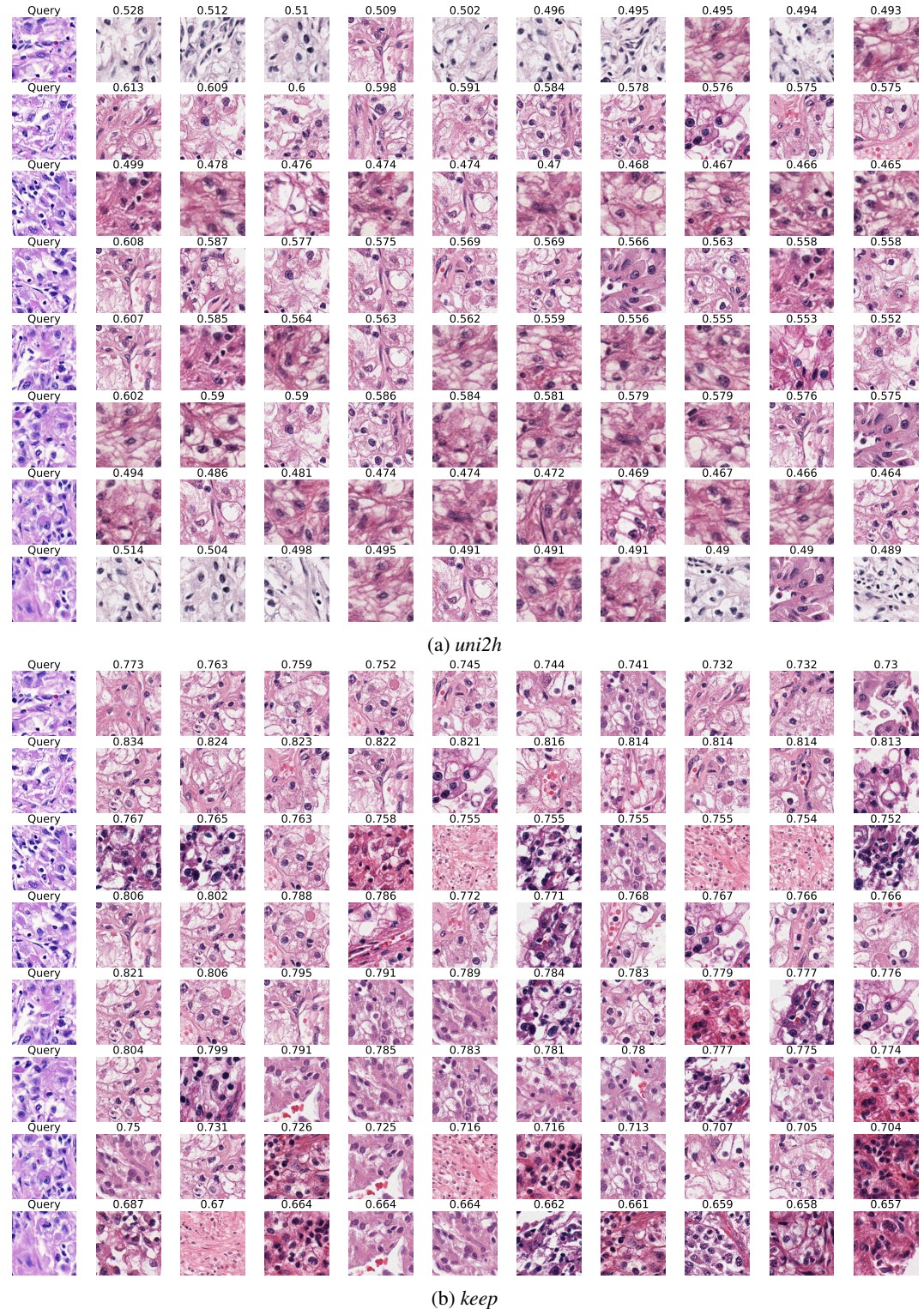

(a) *uni2h*

(b) *keep*

Figure S13: **Image retrieval**: Qualitative samples (query + top-10 with cosine similarity) on *ccrcc*.

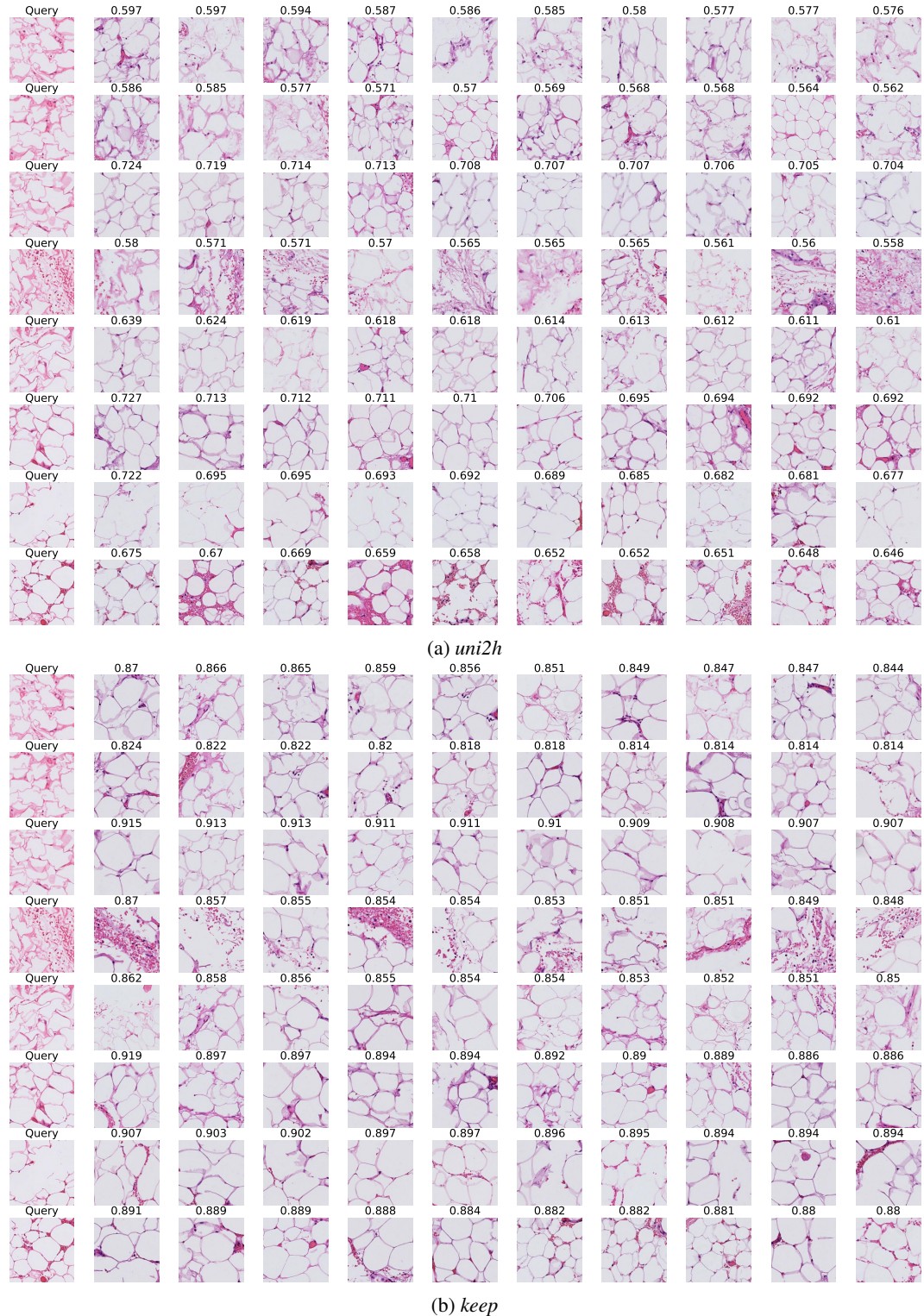

(a) *uni2h*

(b) *keep*

Figure S14: **Image retrieval**: Qualitative samples (query + top-10 with cosine similarity) on *esca*.

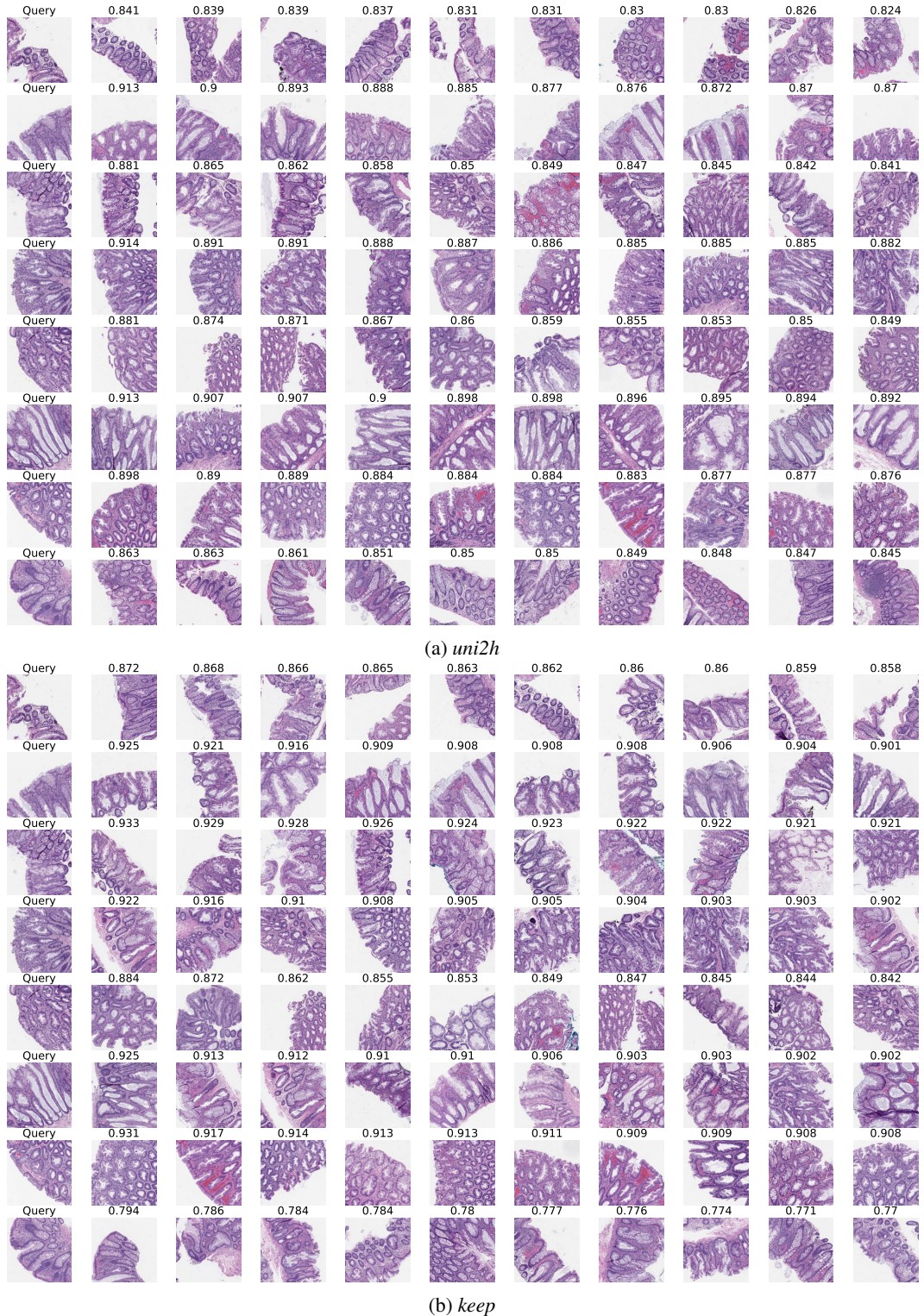

(a) *uni2h*

(b) *keep*

Figure S15: **Image retrieval**: Qualitative samples (query + top-10 with cosine similarity) on *mhist*.

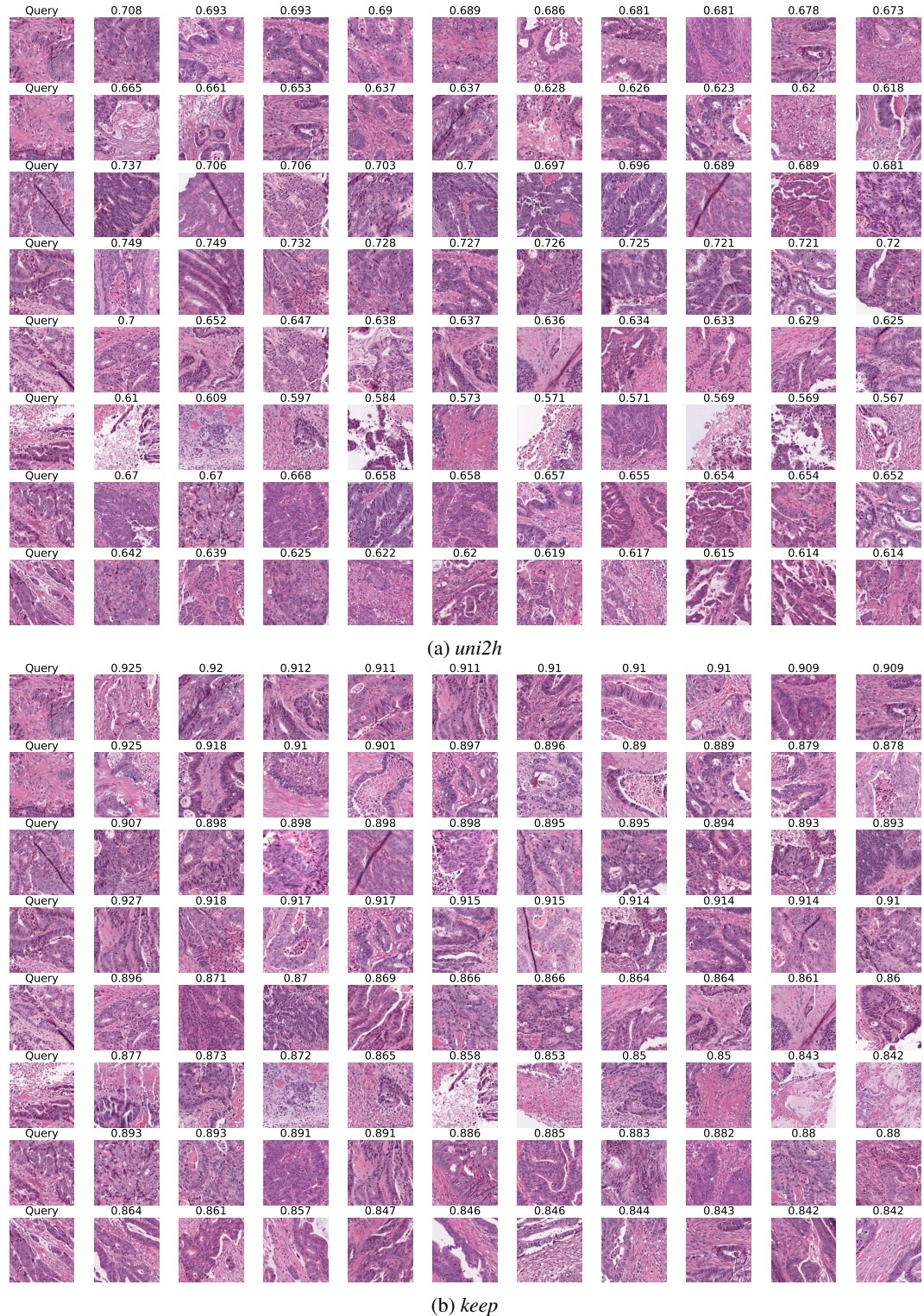

Figure S16: **Image retrieval**: Qualitative samples (query + top-10 with cosine similarity) on *tcga-crc*.

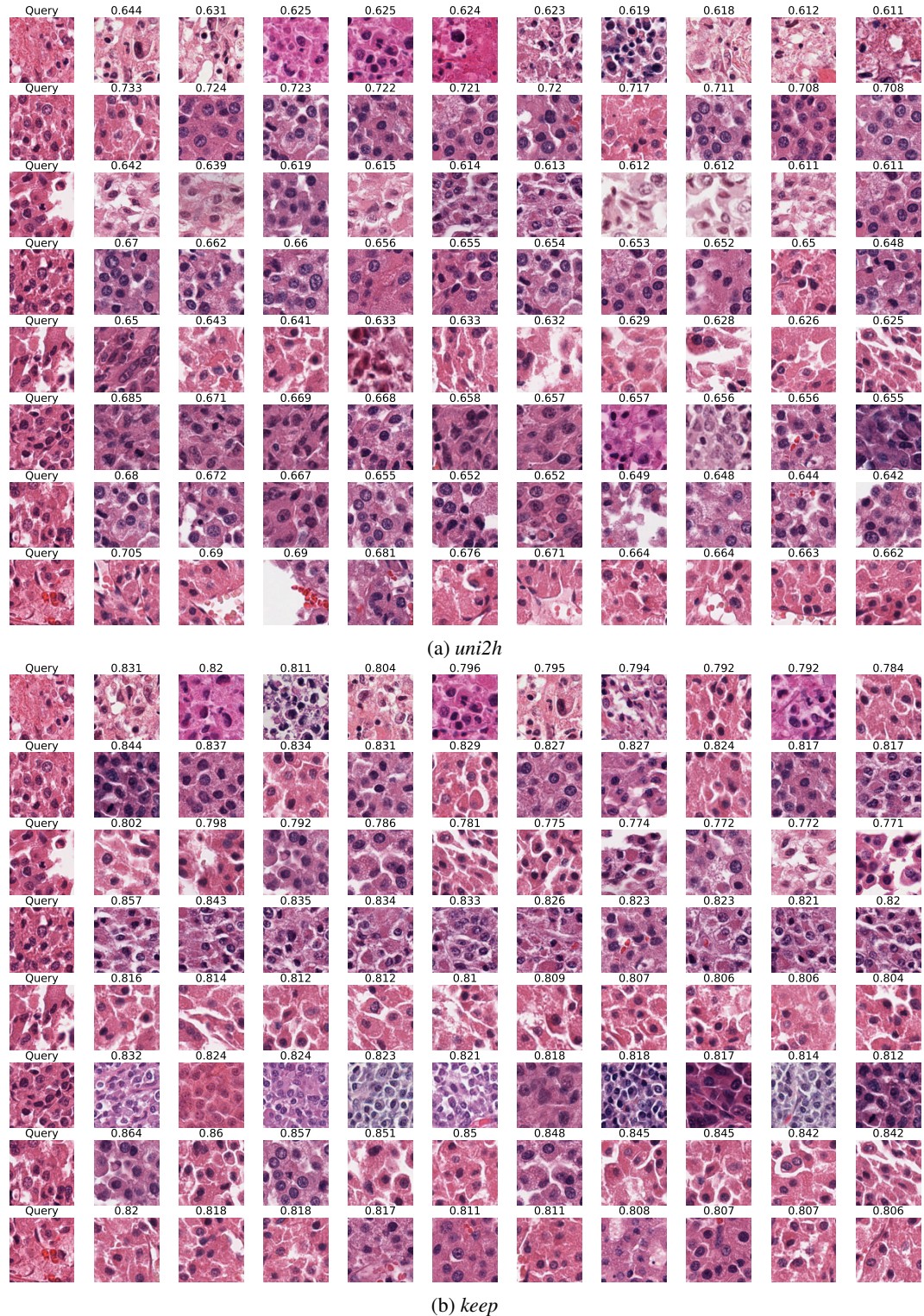

(a) *uni2h*

(b) *keep*

Figure S17: **Image retrieval**: Qualitative samples (query + top-10 with cosine similarity) on *tcga-tils*.

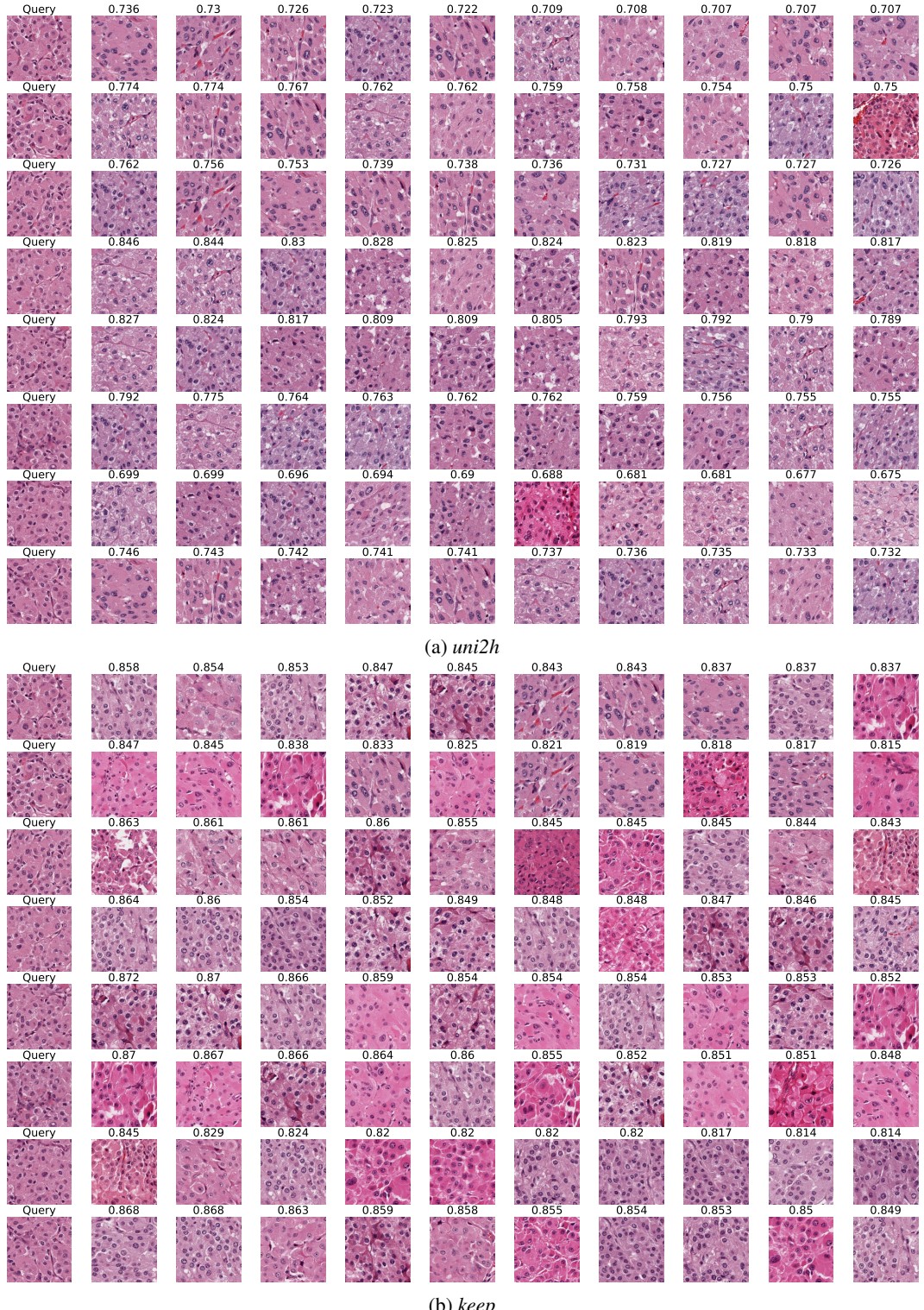

(a) *uni2h*

(b) *keep*

Figure S18: **Image retrieval**: Qualitative samples (query + top-10 with cosine similarity) on *tcga-unif*.

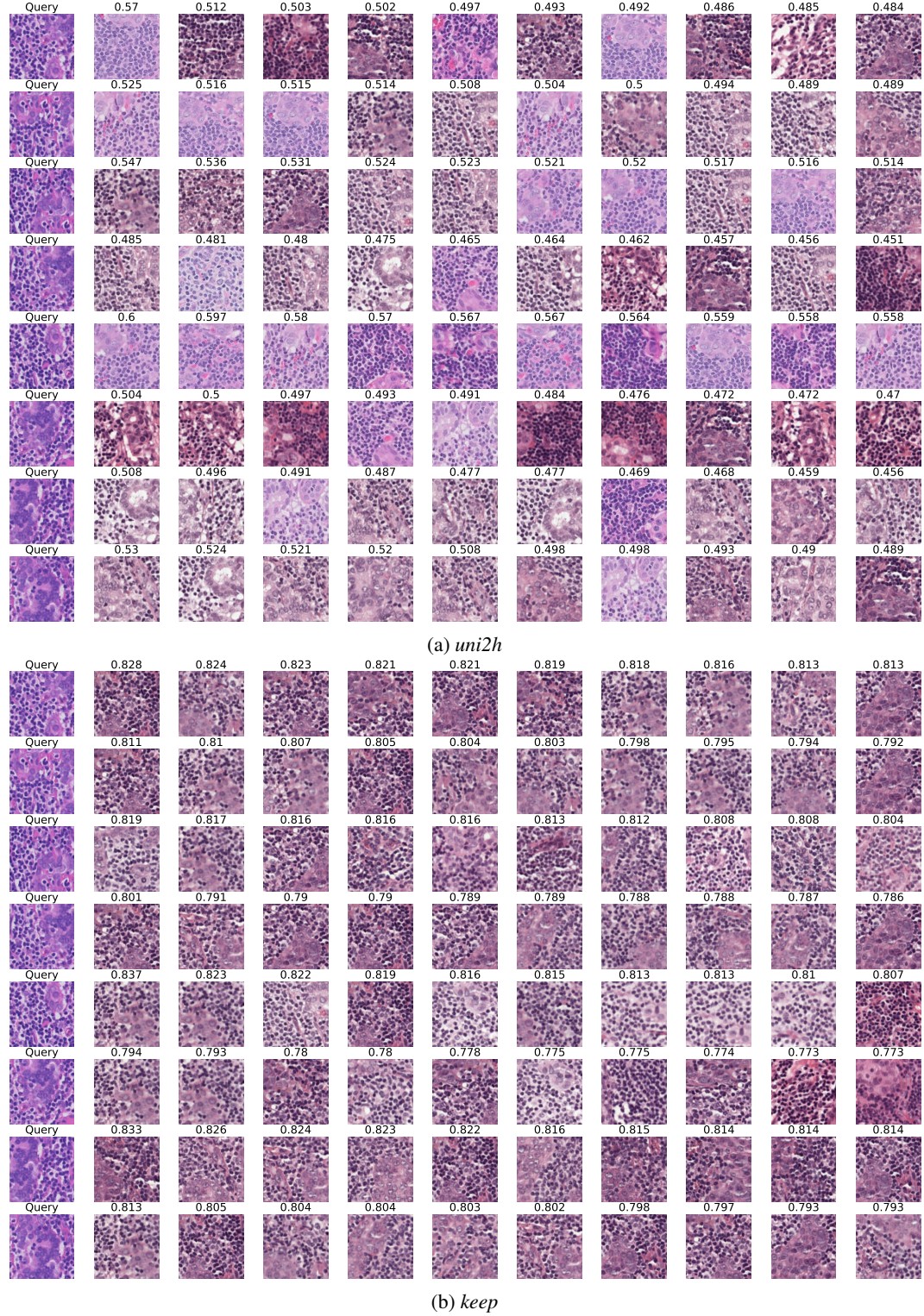

(a) *uni2h*

(b) *keep*

Figure S19: **Image retrieval**: Qualitative samples (query + top-10 with cosine similarity) on *wilds*.

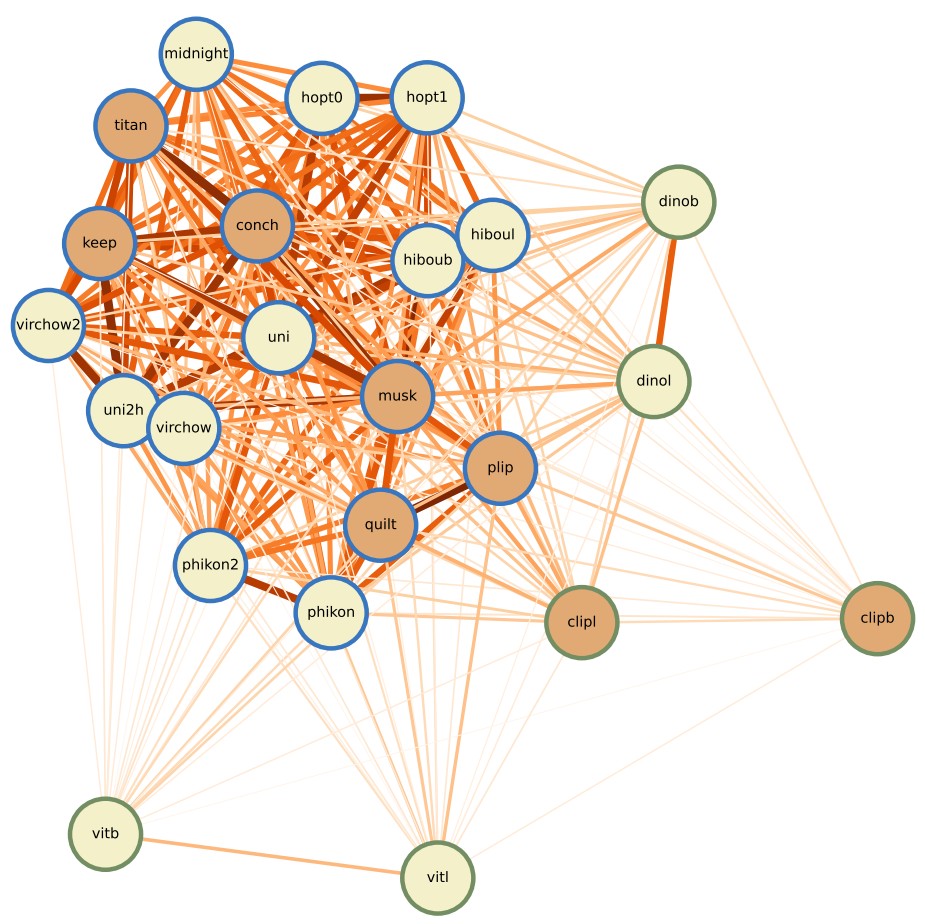

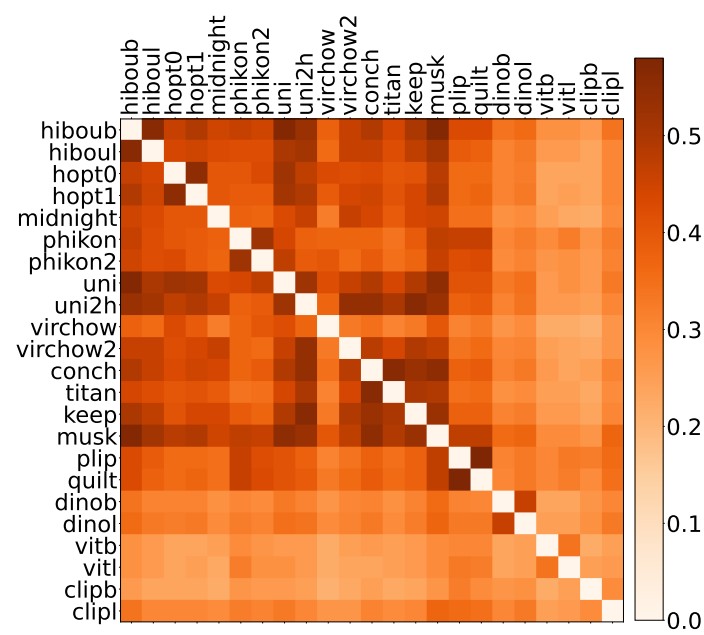

Figure S20: **Alignment scoring** (*Mutual knn*) on *bach*.

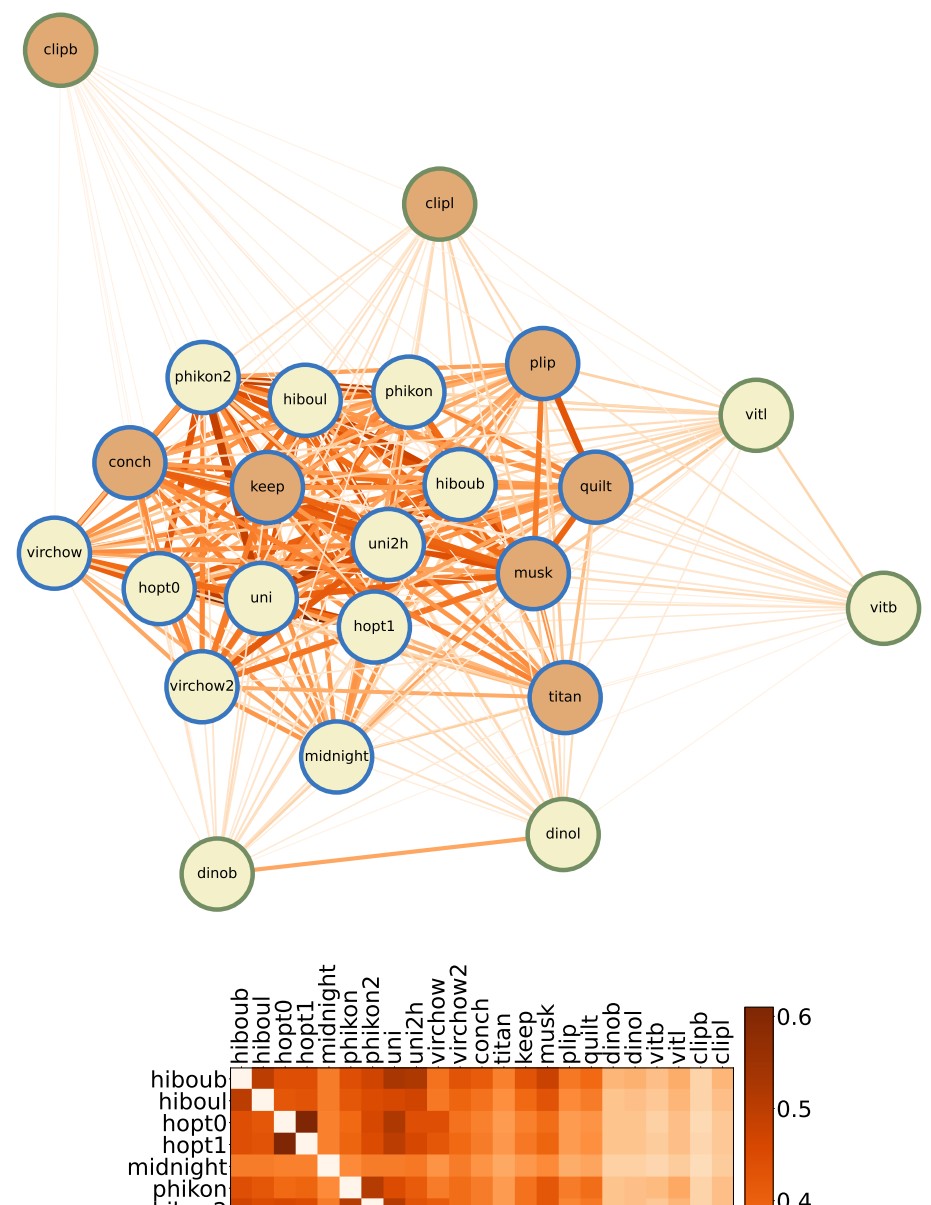

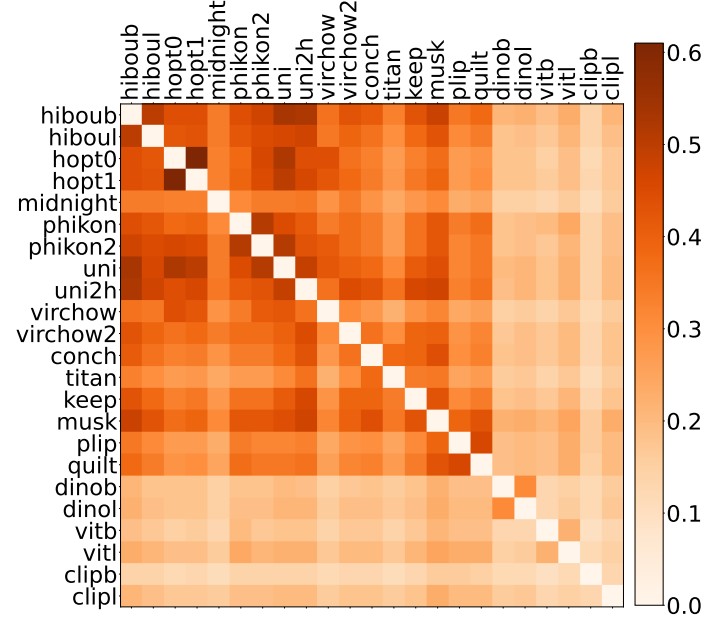

Figure S21: **Alignment scoring** (*Mutual knn*) on *bracs*.

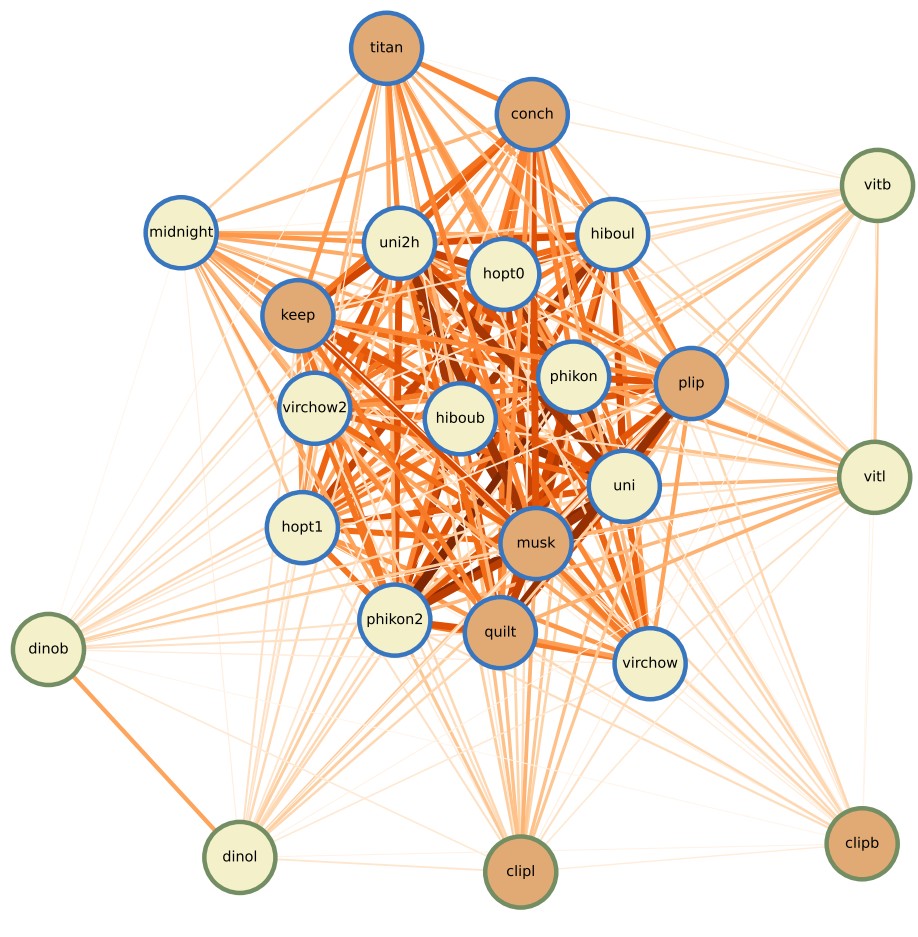

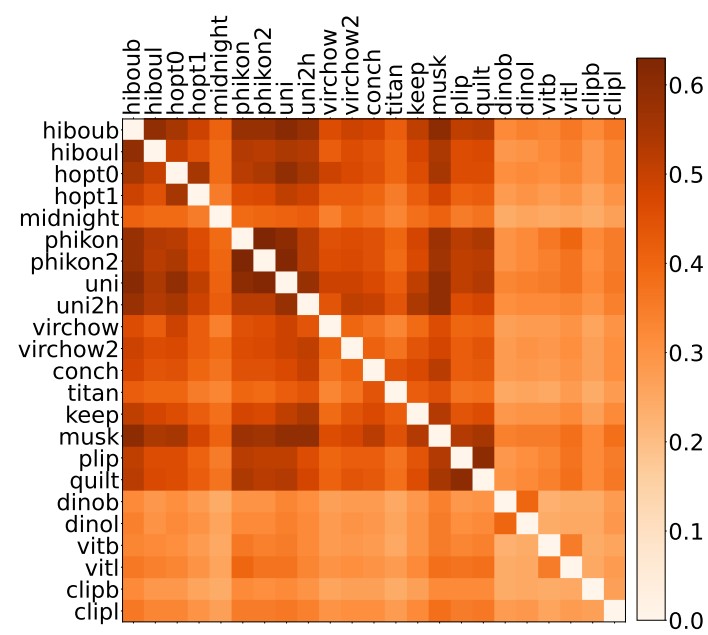

Figure S22: **Alignment scoring** (*Mutual knn*) on *break his*.

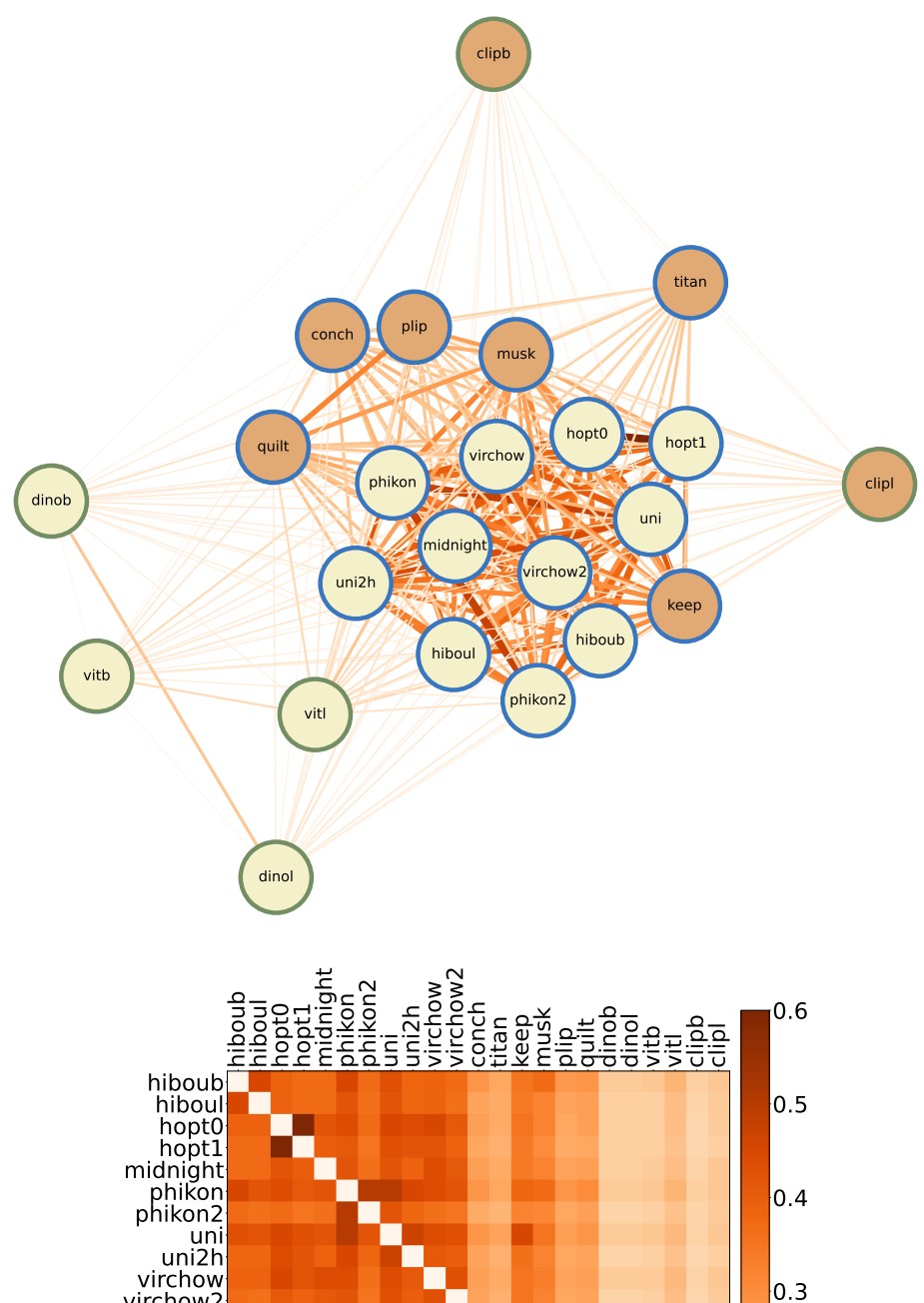

Figure S23: **Alignment scoring** (*Mutual knn*) on *ccrcc*.

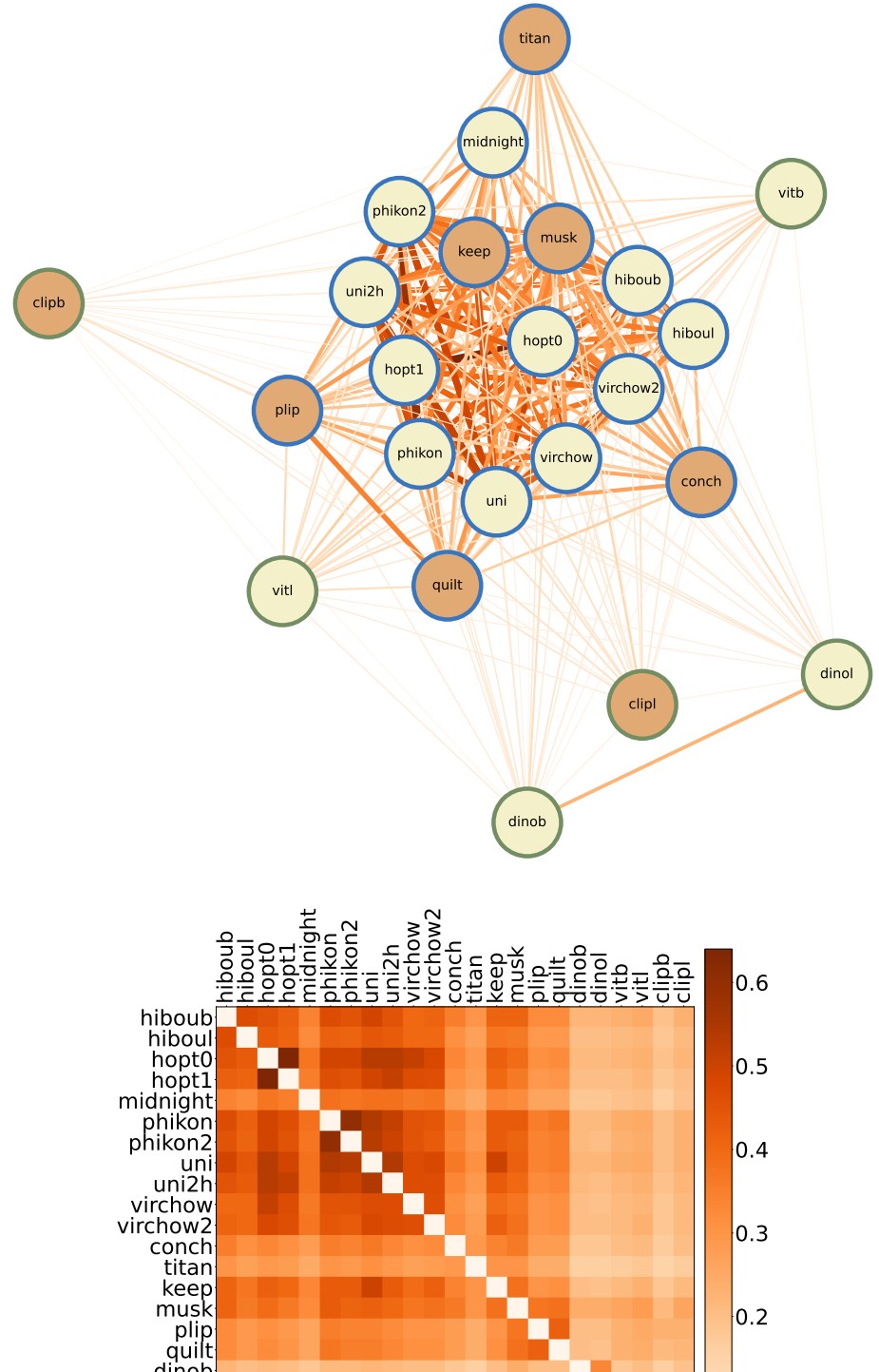

Figure S24: **Alignment scoring** (*Mutual knn*) on *crc*.

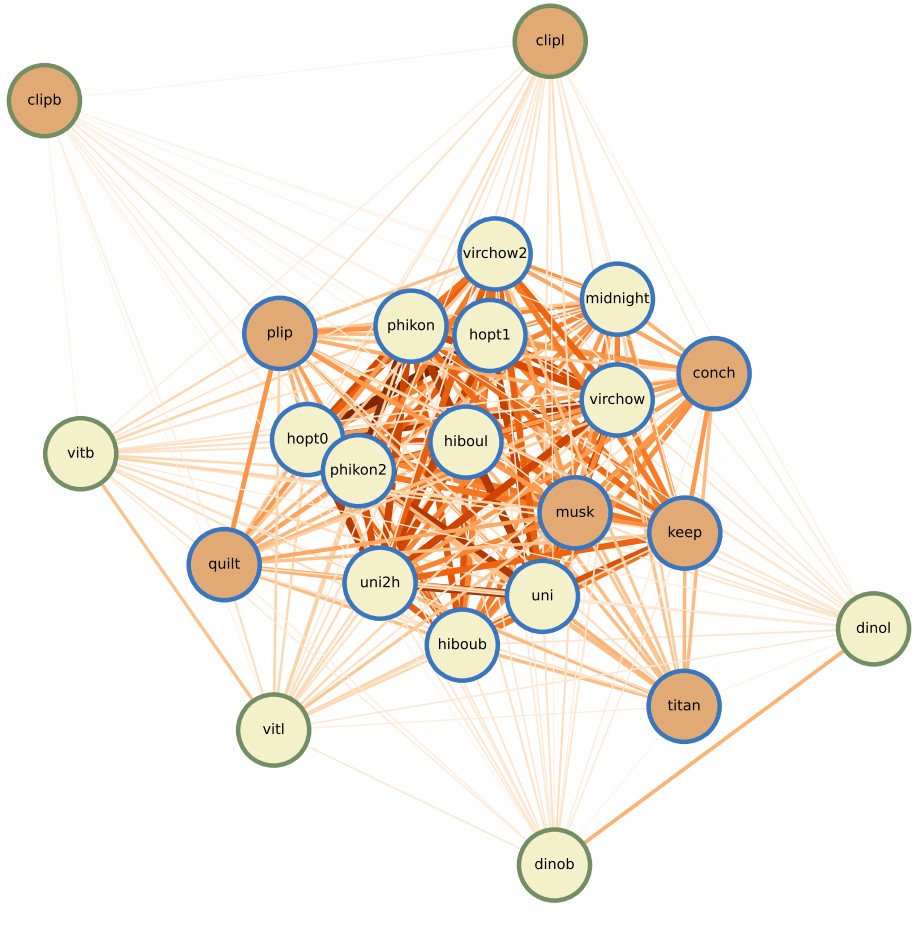

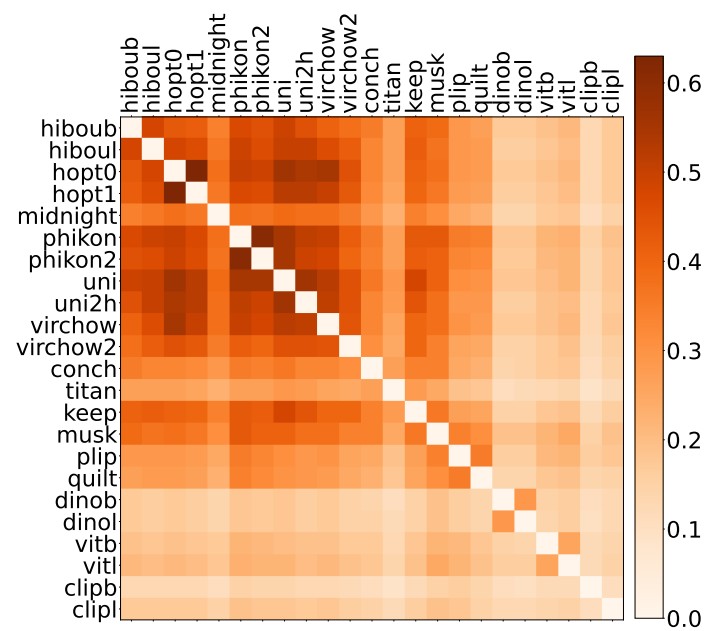

Figure S25: **Alignment scoring** (*Mutual knn*) on *esca*.

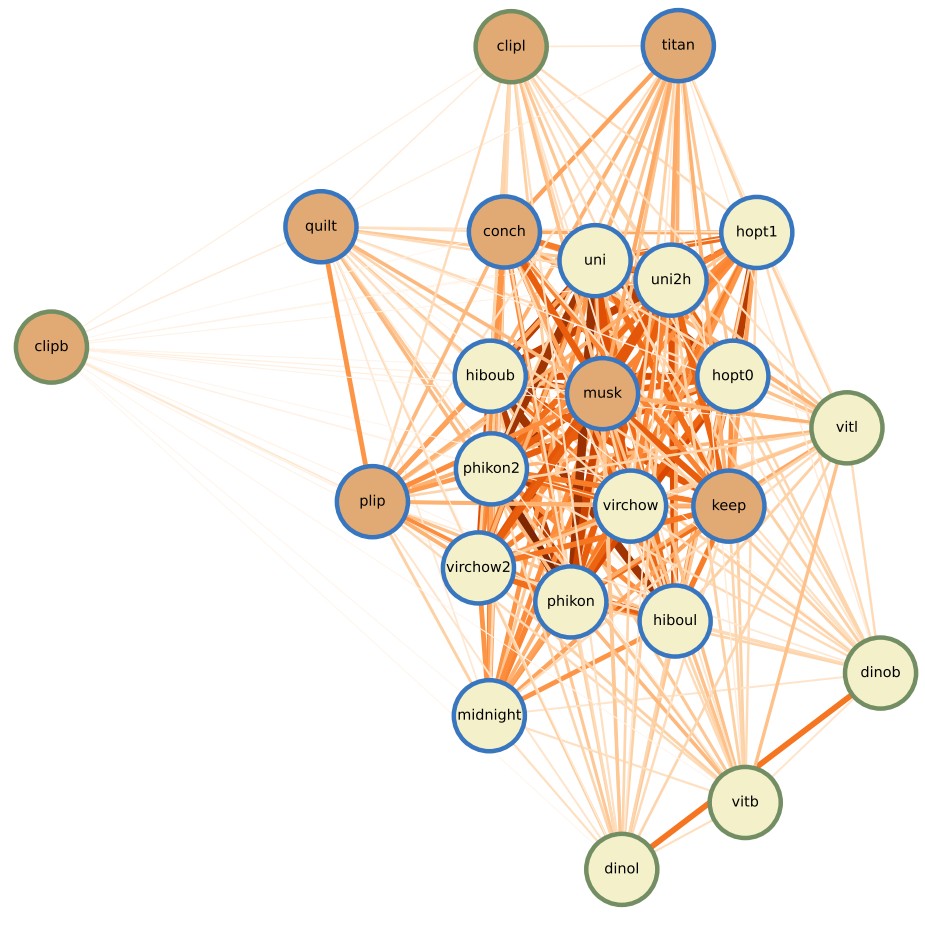

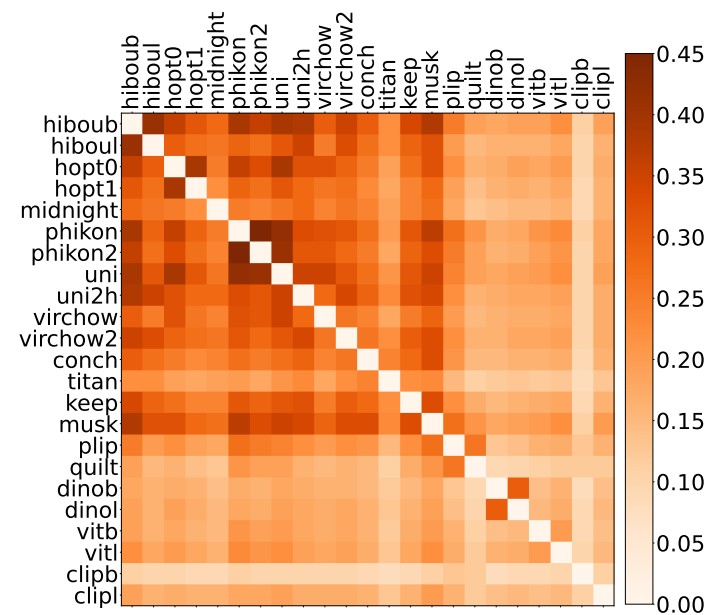

Figure S26: **Alignment scoring** (*Mutual knn*) on *mhist*.

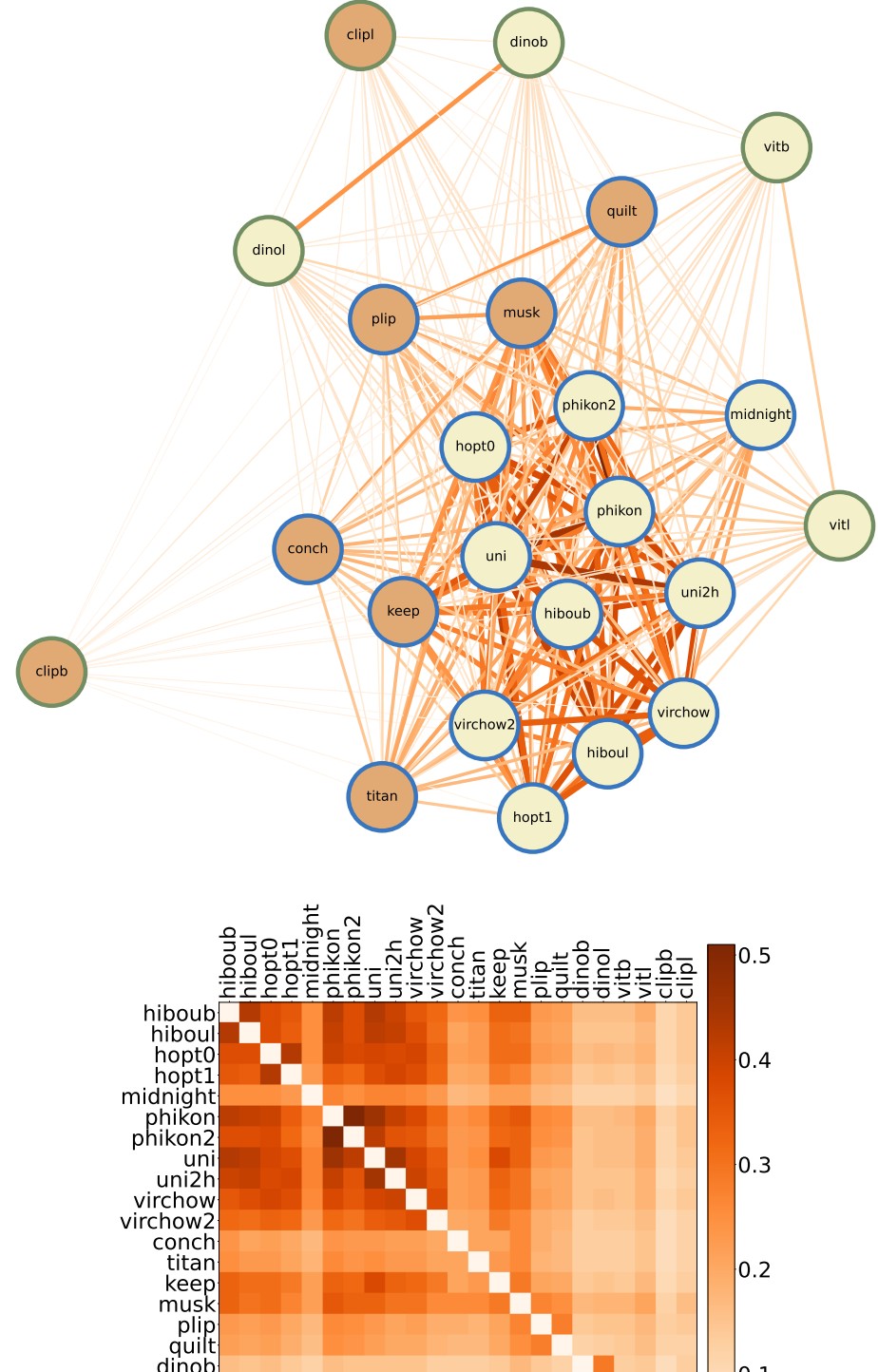

Figure S27: **Alignment scoring** (*Mutual knn*) on *patch camelyon*.

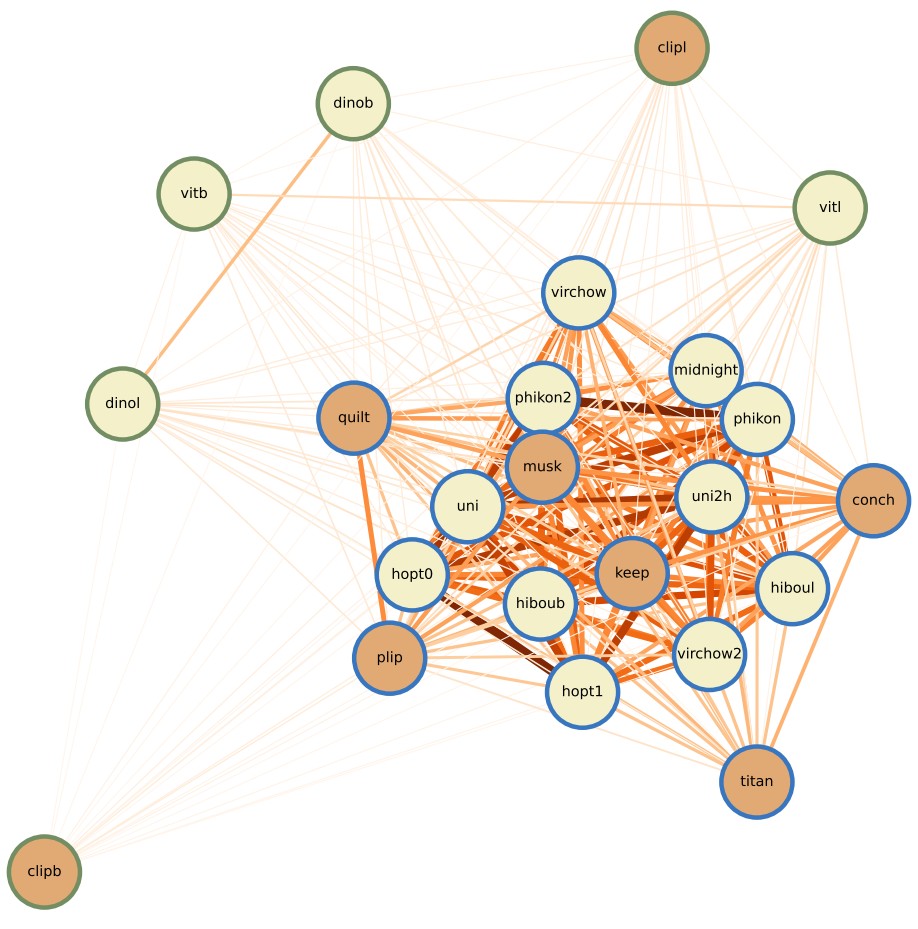

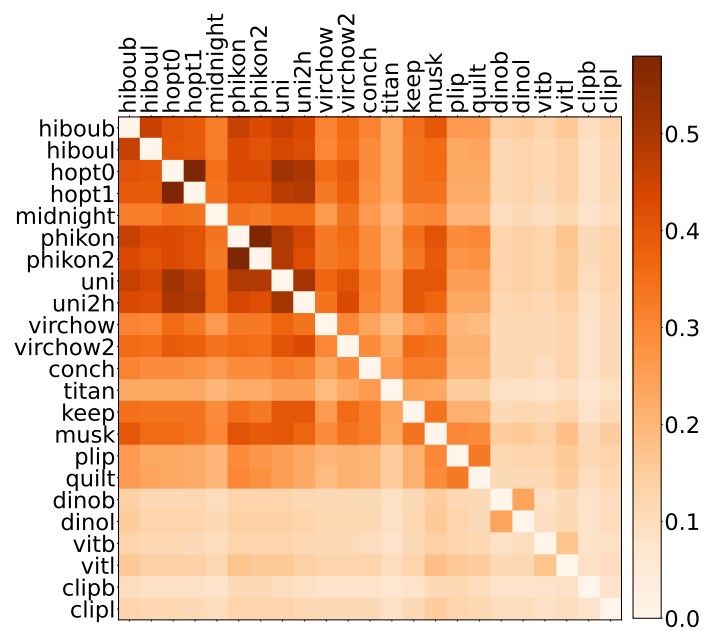

Figure S28: **Alignment scoring** (*Mutual knn*) on *tcga crc msi*.

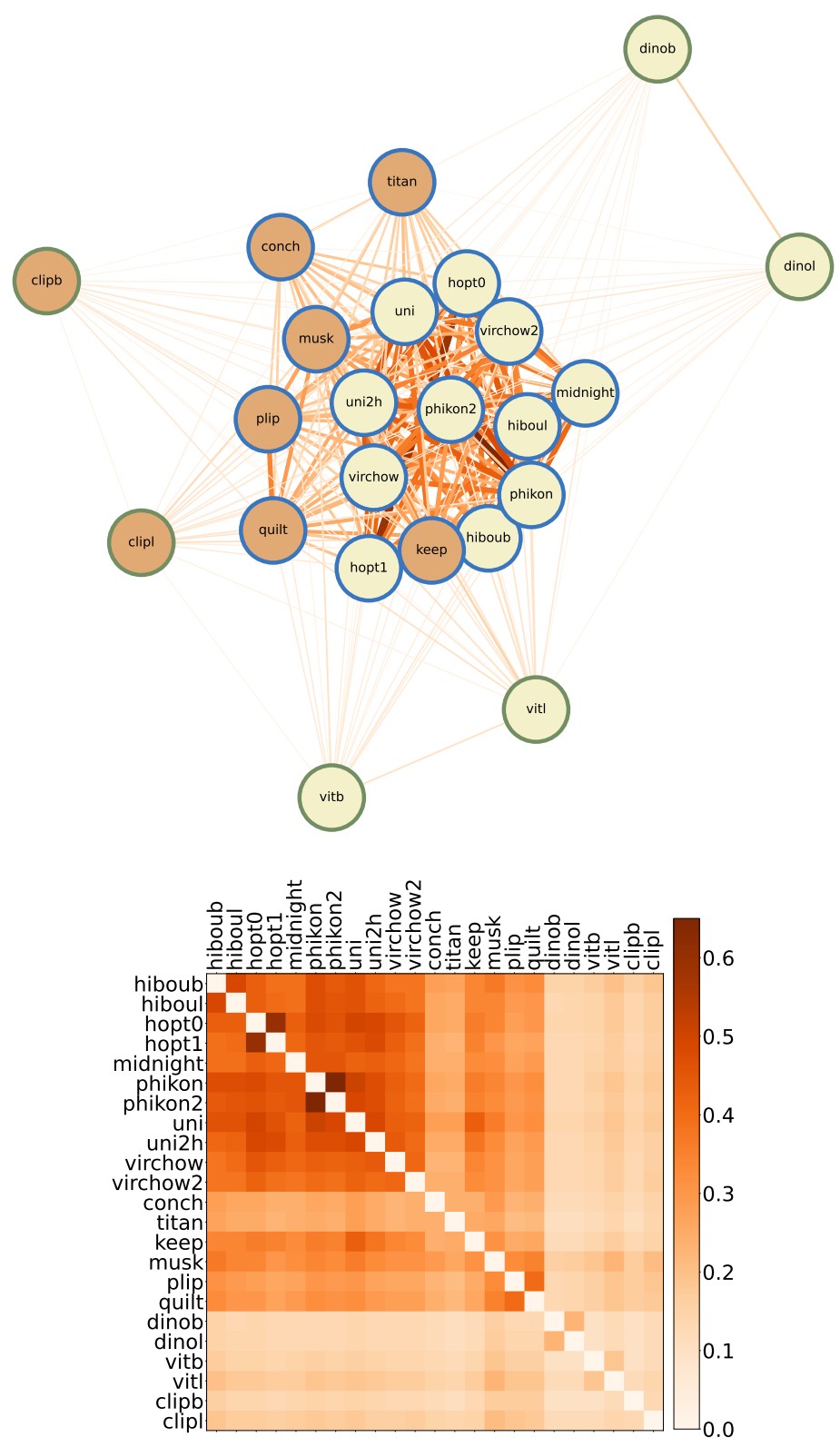

Figure S29: **Alignment scoring** (*Mutual knn*) on *tcga tils*.

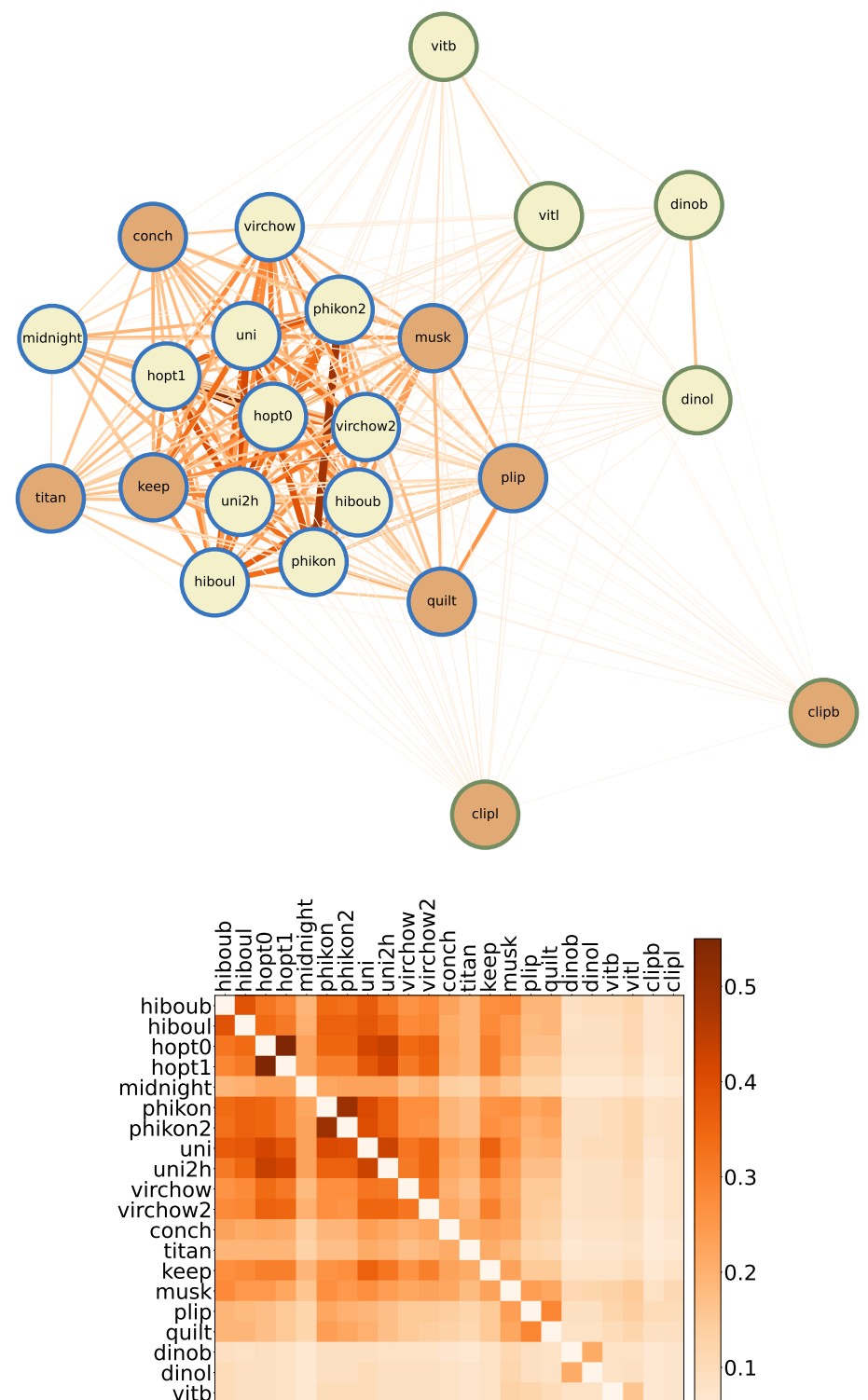

Figure S30: **Alignment scoring** (*Mutual knn*) on *tcga uniform*.

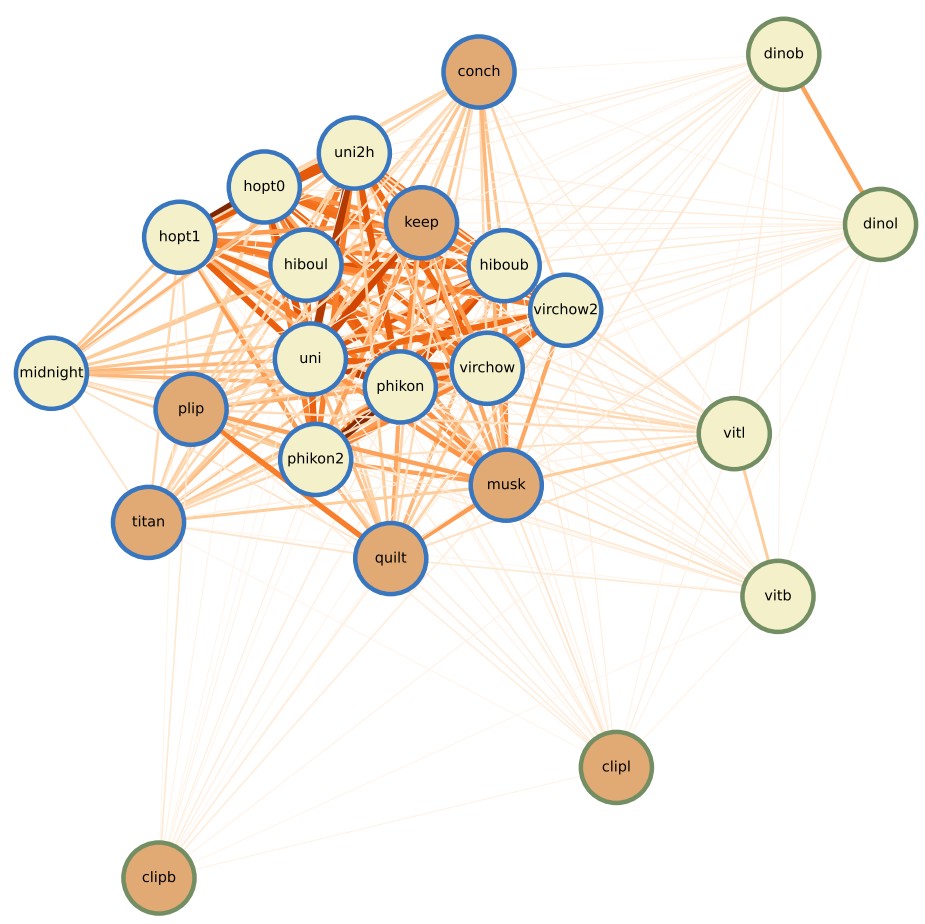

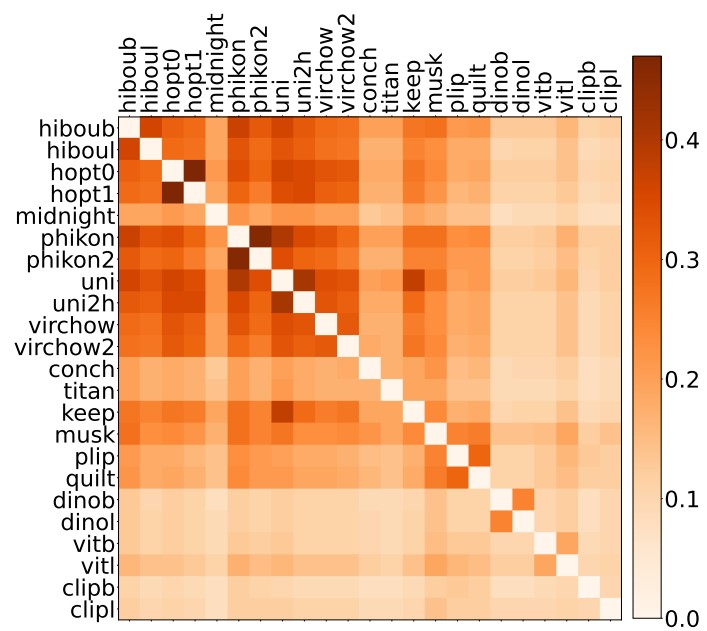

Figure S31: **Alignment scoring** (*Mutual knn*) on *wilds*.

Table S36: Quantitative performance (Balanced accuracy) on knn classification.

| Model | bach | bracs | break-h | ccrcc | crc | esca | mhist | pcam | tcga-crc | tcga-tils | tcga-unif | wilds |
|---|---|---|---|---|---|---|---|---|---|---|---|---|
| | 82.9 | 58.8 | 60.4 | 80.0 | 93.4 | 80.4 | 63.8 | 91.9 | 65.5 | 84.8 | 61.1 | 97.7 |
| hiboub | [77.1, | [55.1, | [55.1, | [79.1, | [92.7, | [79.9, | [60.7, | [91.6, | [64.8, | [84.4, | [60.6, | [97.6, |
| | 88.3] | 62.3] | 65.6] | 80.8] | 94.0] | 80.9] | 66.9] | 92.1] | 66.2] | 85.3] | 61.6] | 97.8] |
| | 76.4 | 57.6 | 65.7 | 76.5 | 91.6 | 77.1 | 74.9 | 91.1 | 67.6 | 83.4 | 64.3 | 88.3 |
| hiboul | [69.7, | [53.7, | [60.1, | [75.6, | [90.9, | [76.6, | [72.1, | [90.8, | [66.9, | [82.9, | [63.8, | [88.1, |
| | 83.1] | 61.2] | 71.3] | 77.4] | 92.3] | 77.6] | 77.7] | 91.4] | 68.3] | 83.8] | 64.8] | 88.5] |
| | 67.0 | 54.4 | 82.0 | 91.1 | 93.3 | 83.6 | 74.1 | 89.3 | 69.0 | 86.8 | 72.6 | 97.9 |
| hopt0 | [59.3, | [51.0, | [77.1, | [90.5, | [92.7, | [83.1, | [71.3, | [89.0, | [68.3, | [86.4, | [72.1, | [97.8, |
| | 74.6] | 57.9] | 86.8] | 91.8] | 94.0] | 84.0] | 77.0] | 89.6] | 69.7] | 87.2] | 73.0] | 98.0] |
| | 71.7 | 57.2 | 77.7 | 93.0 | 92.6 | 85.2 | 74.7 | 90.7 | 70.4 | 86.1 | 76.6 | 98.4 |
| hopt1 | [64.5, | [53.8, | [72.0, | [92.4, | [91.9, | [84.8, | [71.7, | [90.4, | [69.7, | [85.6, | [76.1, | [98.3, |
| | 78.8] | 60.8] | 83.1] | 93.6] | 93.3] | 85.6] | 77.6] | 91.0] | 71.1] | 86.5] | 77.1] | 98.5] |
| | 86.3 | 52.9 | 56.7 | 91.3 | 94.3 | 81.4 | 69.2 | 88.1 | 64.9 | 86.4 | 76.9 | 95.0 |
| midnight | [80.7, | [49.4, | [50.5, | [90.6, | [93.7, | [81.0, | [66.1, | [87.8, | [64.2, | [86.0, | [76.4, | [94.8, |
| | 91.5] | 56.2] | 62.9] | 92.0] | 94.9] | 81.8] | 72.2] | 88.4] | 65.6] | 86.8] | 77.4] | 95.1] |
| | 57.2 | 50.6 | 64.5 | 86.4 | 93.3 | 79.3 | 67.1 | 87.5 | 61.6 | 82.8 | 61.8 | 89.3 |
| phikon | [48.5, | [46.7, | [59.1, | [85.7, | [92.7, | [78.9, | [64.2, | [87.2, | [60.8, | [82.3, | [61.3, | [89.1, |
| | 65.8] | 54.3] | 69.8] | 87.2] | 94.0] | 79.8] | 70.1] | 87.9] | 62.3] | 83.3] | 62.4] | 89.5] |
| | 54.8 | 49.1 | 49.7 | 79.2 | 92.3 | 77.6 | 66.1 | 82.6 | 60.6 | 78.5 | 68.2 | 91.0 |
| phikon2 | [45.8, | [45.6, | [44.1, | [78.2, | [91.6, | [77.1, | [63.0, | [82.2, | [59.8, | [78.0, | [67.7, | [90.9, |
| | 63.7] | 52.6] | 55.7] | 80.1] | 92.9] | 78.0] | 69.1] | 83.0] | 61.3] | 79.0] | 68.7] | 91.2] |
| | 72.9 | 57.3 | 78.6 | 88.2 | 93.8 | 84.3 | 71.0 | 89.8 | 66.6 | 86.6 | 67.9 | 97.8 |
| uni | [65.5, | [53.6, | [73.4, | [87.4, | [93.1, | [83.9, | [68.1, | [89.5, | [65.9, | [86.1, | [67.4, | [97.7, |
| | 79.8] | 60.8] | 83.6] | 88.8] | 94.4] | 84.7] | 73.8] | 90.1] | 67.3] | 87.0] | 68.4] | 97.9] |
| | 87.7 | 57.1 | 76.7 | 92.2 | 95.3 | 85.5 | 70.1 | 93.4 | 68.7 | 87.6 | 74.5 | 98.0 |
| uni2h | [83.0, | [53.7, | [71.1, | [91.6, | [94.8, | [85.1, | [67.2, | [93.1, | [68.0, | [87.2, | [74.0, | [98.0, |
| | 92.4] | 60.6] | 82.1] | 92.8] | 95.9] | 85.9] | 72.9] | 93.7] | 69.4] | 88.0] | 75.0] | 98.1] |
| | 55.7 | 52.7 | 54.1 | 86.2 | 93.6 | 82.7 | 68.9 | 90.0 | 66.0 | 87.1 | 61.5 | 97.5 |
| virchow | [46.7, | [48.9, | [48.9, | [85.4, | [93.0, | [82.3, | [65.9, | [89.6, | [65.2, | [86.7, | [61.0, | [97.4, |
| | 64.4] | 56.3] | 59.3] | 87.0] | 94.2] | 83.1] | 72.0] | 90.3] | 66.7] | 87.6] | 62.0] | 97.6] |
| | 83.6 | 57.6 | 78.9 | 91.6 | 94.7 | 87.9 | 73.9 | 89.8 | 68.6 | 86.3 | 70.2 | 97.6 |
| virchow2 | [77.1, | [54.1, | [74.4, | [90.9, | [94.1, | [87.5, | [71.0, | [89.5, | [67.9, | [85.9, | [69.7, | [97.5, |
| | 89.2] | 60.8] | 83.5] | 92.3] | 95.3] | 88.4] | 76.7] | 90.1] | 69.3] | 86.8] | 70.7] | 97.7] |
| | 86.9 | 59.7 | 64.7 | 89.5 | 94.4 | 80.9 | 68.6 | 85.7 | 66.5 | 86.3 | 60.3 | 95.1 |
| conch | [82.6, | [56.3, | [59.6, | [88.7, | [93.7, | [80.4, | [65.6, | [85.4, | [65.8, | [83.4, | [59.8, | [94.9, |
| | 90.9] | 62.8] | 69.5] | 90.3] | 94.9] | 81.3] | 71.6] | 86.1] | 67.2] | 84.3] | 60.8] | 95.2] |
| | 82.8 | 60.2 | 76.8 | 88.0 | 93.3 | 80.8 | 70.7 | 87.7 | 67.1 | 85.4 | 61.7 | 94.1 |
| titan | [76.7, | [56.4, | [71.4, | [87.3, | [92.7, | [80.3, | [67.7, | [86.4, | [66.4, | [84.9, | [61.1, | [93.9, |
| | 88.5] | 63.6] | 81.6] | 88.8] | 93.9] | 81.3] | 73.4] | 88.1] | 67.8] | 85.8] | 62.2] | 94.2] |
| | 86.5 | 57.2 | 73.7 | 93.6 | 92.2 | 86.2 | 67.5 | 90.2 | 67.4 | 88.0 | 67.2 | 97.0 |
| keep | [80.6, | [54.1, | [68.0, | [93.0, | [91.4, | [85.8, | [64.6, | [89.9, | [66.8, | [87.5, | [66.7, | [96.9, |
| | 92.1] | 60.3] | 79.2] | 94.1] | 92.9] | 86.7] | 70.5] | 90.5] | 68.1] | 88.3] | 67.7] | 97.1] |
| | 68.4 | 59.1 | 80.1 | 82.3 | 93.1 | 77.4 | 70.8 | 85.6 | 65.2 | 85.5 | 53.0 | 95.5 |
| musk | [61.4, | [55.6, | [75.0, | [81.4, | [92.4, | [76.9, | [67.8, | [85.2, | [64.5, | [85.1, | [52.5, | [95.3, |
| | 75.0] | 62.5] | 85.0] | 83.2] | 93.7] | 77.9] | 73.6] | 85.9] | 66.0] | 85.9] | 53.5] | 95.6] |
| | 67.3 | 49.7 | 45.5 | 76.0 | 92.7 | 59.7 | 62.1 | 84.0 | 61.3 | 82.4 | 44.6 | 94.0 |
| plip | [58.8, | [46.2, | [41.0, | [75.0, | [92.0, | [59.3, | [59.0, | [83.6, | [60.5, | [82.0, | [44.2, | [93.8, |
| | 75.8] | 53.2] | 50.0] | 77.0] | 93.3] | 60.2] | 65.2] | 84.4] | 62.0] | 82.8] | 45.1] | 94.2] |
| | 64.8 | 53.4 | 55.2 | 73.3 | 92.6 | 57.5 | 66.6 | 84.2 | 61.1 | 82.4 | 45.1 | 94.3 |
| quilt | [56.7, | [50.1, | [50.3, | [72.3, | [92.0, | [57.1, | [63.6, | [83.8, | [60.4, | [82.0, | [44.5, | [94.1, |
| | 72.4] | 56.9] | 60.3] | 74.4] | 93.3] | 57.9] | 69.8] | 84.5] | 61.7] | 82.9] | 45.6] | 94.4] |
| | 64.8 | 46.8 | 60.7 | 77.2 | 87.6 | 65.0 | 75.2 | 80.7 | 63.6 | 81.8 | 35.4 | 87.1 |
| dinob | [57.1, | [43.3, | [55.0, | [76.3, | [86.8, | [64.5, | [72.5, | [80.3, | [62.9, | [81.4, | [34.9, | [86.8, |
| | 72.2] | 50.2] | 66.8] | 78.1] | 88.4] | 65.5] | 78.0] | 81.1] | 64.3] | 82.3] | 35.8] | 87.3] |
| | 67.4 | 50.4 | 61.7 | 77.7 | 87.0 | 65.1 | 80.4 | 80.9 | 62.7 | 81.9 | 36.4 | 90.3 |
| dinol | [59.5, | [47.0, | [56.6, | [76.7, | [86.2, | [64.6, | [77.7, | [80.5, | [62.0, | [81.4, | [35.9, | [90.1, |
| | 75.0] | 53.6] | 67.0] | 78.6] | 87.8] | 65.7] | 83.2] | 81.3] | 63.5] | 82.3] | 36.8] | 90.5] |
| | 57.6 | 49.0 | 53.3 | 67.1 | 88.6 | 58.5 | 68.2 | 78.5 | 59.5 | 80.2 | 34.0 | 86.6 |
| vitb | [49.0, | [45.5, | [47.9, | [65.9, | [87.9, | [58.0, | [65.2, | [78.1, | [58.8, | [79.8, | [33.6, | [86.4, |
| | 65.8] | 52.4] | 58.7] | 68.2] | 89.5] | 59.0] | 71.1] | 78.9] | 60.3] | 80.7] | 34.5] | 86.9] |
| | 59.4 | 49.3 | 64.1 | 72.2 | 90.0 | 62.8 | 69.7 | 79.1 | 63.2 | 81.8 | 37.7 | 90.2 |
| vitl | [51.6, | [45.8, | [58.0, | [71.1, | [89.2, | [62.4, | [66.6, | [78.7, | [62.5, | [81.4, | [37.2, | [90.0, |
| | 67.1] | 52.9] | 70.1] | 73.3] | 90.8] | 63.3] | 72.6] | 79.6] | 63.9] | 82.3] | 38.2] | 90.4] |
| | 48.5 | 45.7 | 52.6 | 69.2 | 83.6 | 54.0 | 66.6 | 78.6 | 58.4 | 75.5 | 31.9 | 88.3 |
| clipb | [40.1, | [42.2, | [46.7, | [68.1, | [82.7, | [53.6, | [63.6, | [78.2, | [57.7, | [75.1, | [31.4, | [88.1, |
| | 56.6] | 49.1] | 58.8] | 70.2] | 84.4] | 54.5] | 69.6] | 79.0] | 59.2] | 76.0] | 32.3] | 88.6] |
| | 56.2 | 49.1 | 50.2 | 67.9 | 86.6 | 60.0 | 62.2 | 81.1 | 59.2 | 78.6 | 36.2 | 90.6 |
| clipl | [48.1, | [45.5, | [44.9, | [66.7, | [85.8, | [59.5, | [59.2, | [80.7, | [58.5, | [78.1, | [35.8, | [90.4, |
| | 64.0] | 52.6] | 55.9] | 69.0] | 87.6] | 60.5] | 65.4] | 81.5] | 59.9] | 79.0] | 36.7] | 90.8] |

Table S37: Quantitative performance (F1-score) on knn classification.

| Model | bach | bracs | break-h | ccrcc | crc | esca | mhist | pcam | tcga-crc | tcga-tils | tcga-unif | wilds |
|---|---|---|---|---|---|---|---|---|---|---|---|---|
| | 79.5 | 56.9 | 62.6 | 76.1 | 92.9 | 76.9 | 63.8 | 91.8 | 61.6 | 87.5 | 62.4 | 97.7 |
| hiboub | [72.1, | [52.8, | [56.1, | [75.0, | [92.2, | [76.6, | [60.7, | [91.5, | [61.0, | [87.1, | [61.8, | [97.6, |
| | 86.0] | 60.6] | 68.7] | 77.2] | 93.5] | 77.3] | 66.9] | 92.1] | 62.2] | 87.9] | 62.9] | 97.8] |
| | 73.6 | 56.2 | 66.5 | 71.5 | 91.0 | 73.8 | 75.5 | 91.0 | 62.7 | 86.6 | 65.4 | 88.2 |
| hiboul | [66.1, | [52.2, | [61.0, | [70.4, | [90.3, | [73.4, | [72.7, | [90.7, | [62.1, | [86.3, | [64.9, | [88.0, |
| | 80.6] | 59.8] | 71.7] | 72.6] | 91.7] | 74.2] | 78.3] | 91.3] | 63.3] | 87.0] | 66.0] | 88.4] |
| | 65.6 | 52.2 | 81.9 | 90.5 | 93.1 | 82.1 | 73.2 | 89.3 | 62.4 | 89.2 | 73.2 | 97.9 |
| hopt0 | [57.0, | [48.4, | [77.1, | [89.8, | [92.4, | [81.7, | [70.3, | [88.9, | [61.8, | [88.9, | [72.7, | [97.8, |
| | 73.6] | 55.9] | 86.3] | 91.2] | 93.8] | 82.4] | 76.1] | 89.6] | 63.0] | 89.6] | 73.7] | 98.0] |
| | 68.3 | 55.0 | 80.3 | 92.3 | 92.6 | 84.4 | 74.9 | 90.6 | 64.1 | 88.1 | 77.4 | 98.4 |
| hopt1 | [59.7, | [51.1, | [75.0, | [91.6, | [91.9, | [84.0, | [71.9, | [90.3, | [63.5, | [87.7, | [76.9, | [98.3, |
| | 75.8] | 58.7] | 84.9] | 93.0] | 93.3] | 84.8] | 77.7] | 90.9] | 64.7] | 88.5] | 77.8] | 98.5] |
| | 84.3 | 50.2 | 58.1 | 91.6 | 94.2 | 81.4 | 69.2 | 88.0 | 61.9 | 87.6 | 77.4 | 95.0 |
| midnight | [77.8, | [46.5, | [51.6, | [90.9, | [93.5, | [81.0, | [66.1, | [87.7, | [61.3, | [87.3, | [77.0, | [94.8, |
| | 89.9] | 53.8] | 64.0] | 92.2] | 94.8] | 81.8] | 72.1] | 88.4] | 62.6] | 88.0] | 77.9] | 95.1] |
| | 56.7 | 50.0 | 63.2 | 83.7 | 93.1 | 77.0 | 67.6 | 87.4 | 57.3 | 85.1 | 63.4 | 89.2 |
| phikon | [47.6, | [46.0, | [57.2, | [82.7, | [92.5, | [76.6, | [64.5, | [87.0, | [56.7, | [84.7, | [62.9, | [88.9, |
| | 64.5] | 53.7] | 68.5] | 84.6] | 93.8] | 77.3] | 70.7] | 87.7] | 57.9] | 85.5] | 63.9] | 89.4] |
| | 53.1 | 45.9 | 51.6 | 77.2 | 92.1 | 75.8 | 66.1 | 82.2 | 56.8 | 80.8 | 69.1 | 91.0 |
| phikon2 | [44.2, | [42.4, | [45.5, | [76.1, | [91.4, | [75.5, | [63.0, | [81.8, | [56.2, | [80.4, | [68.6, | [90.8, |
| | 61.5] | 49.3] | 57.7] | 78.2] | 92.8] | 76.2] | 69.1] | 82.6] | 57.4] | 81.3] | 69.6] | 91.2] |
| | 71.0 | 55.6 | 78.2 | 86.6 | 93.8 | 82.1 | 71.1 | 89.7 | 61.9 | 88.9 | 69.1 | 97.8 |
| uni | [62.1, | [51.6, | [72.9, | [85.7, | [93.1, | [81.8, | [68.1, | [89.4, | [61.3, | [88.6, | [68.6, | [97.7, |
| | 78.4] | 59.3] | 83.0] | 87.4] | 94.4] | 82.5] | 73.9] | 90.0] | 62.5] | 89.3] | 69.6] | 97.9] |
| | 85.3 | 56.1 | 79.4 | 91.0 | 95.0 | 83.2 | 70.8 | 93.4 | 64.1 | 89.1 | 75.1 | 98.0 |
| uni2h | [79.4, | [52.5, | [74.1, | [90.2, | [94.4, | [82.8, | [67.8, | [93.1, | [63.5, | [88.8, | [74.6, | [98.0, |
| | 90.8] | 59.6] | 83.9] | 91.7] | 95.6] | 83.6] | 73.7] | 93.6] | 64.7] | 89.5] | 75.6] | 98.1] |
| | 55.2 | 51.3 | 56.2 | 83.8 | 92.5 | 81.8 | 68.4 | 89.9 | 61.8 | 88.7 | 62.7 | 97.5 |
| virchow | [46.0, | [47.2, | [49.7, | [82.8, | [91.9, | [81.4, | [65.4, | [89.6, | [61.2, | [88.4, | [62.1, | [97.4, |
| | 63.3] | 54.9] | 62.2] | 84.7] | 93.2] | 82.1] | 71.4] | 90.2] | 62.4] | 89.1] | 63.2] | 97.6] |
| | 81.6 | 54.9 | 80.3 | 91.1 | 94.3 | 86.6 | 73.5 | 89.8 | 65.5 | 88.7 | 70.8 | 97.6 |
| virchow2 | [74.5, | [51.0, | [75.4, | [90.4, | [93.7, | [86.2, | [70.7, | [89.4, | [64.9, | [88.3, | [70.3, | [97.5, |
| | 87.8] | 58.5] | 84.8] | 91.8] | 94.9] | 87.0] | 76.1] | 90.1] | 66.2] | 89.0] | 71.3] | 97.7] |
| | 82.4 | 56.9 | 64.9 | 89.5 | 93.7 | 80.1 | 68.6 | 85.6 | 63.6 | 86.3 | 61.1 | 95.1 |
| conch | [76.1, | [52.9, | [59.4, | [88.7, | [93.1, | [79.7, | [65.6, | [85.2, | [62.9, | [85.9, | [60.6, | [94.9, |
| | 88.1] | 60.5] | 69.9] | 90.3] | 94.3] | 80.5] | 71.7] | 86.0] | 64.2] | 86.7] | 61.6] | 95.2] |
| | 80.5 | 59.4 | 76.9 | 87.2 | 92.9 | 80.7 | 71.4 | 87.6 | 62.5 | 87.8 | 62.8 | 94.1 |
| titan | [73.5, | [55.4, | [71.4, | [86.4, | [92.3, | [80.2, | [68.4, | [87.3, | [61.9, | [87.4, | [62.3, | [93.9, |
| | 86.6] | 62.9] | 81.4] | 88.0] | 93.6] | 81.1] | 74.2] | 88.0] | 63.1] | 88.1] | 63.4] | 94.2] |
| | 85.5 | 53.0 | 73.8 | 93.0 | 92.5 | 83.6 | 67.9 | 90.1 | 63.6 | 89.5 | 67.4 | 97.0 |
| keep | [79.3, | [49.4, | [68.1, | [92.4, | [91.8, | [83.3, | [64.9, | [89.8, | [63.0, | [89.2, | [66.9, | [96.8, |
| | 91.2] | 56.6] | 78.7] | 93.6] | 93.2] | 83.9] | 70.9] | 90.4] | 64.2] | 89.9] | 67.9] | 97.1] |
| | 64.3 | 57.8 | 79.4 | 80.2 | 92.8 | 76.4 | 71.3 | 85.4 | 61.6 | 87.9 | 54.4 | 95.4 |
| musk | [55.6, | [53.9, | [74.3, | [79.2, | [92.1, | [76.0, | [68.3, | [85.0, | [61.0, | [87.5, | [53.9, | [95.3, |
| | 72.0] | 61.3] | 83.9] | 81.2] | 93.5] | 76.8] | 74.2] | 85.8] | 62.2] | 88.2] | 54.9] | 95.6] |
| | 67.8 | 48.2 | 45.3 | 73.0 | 92.1 | 59.5 | 62.2 | | 57.0 | 84.9 | 45.9 | 94.0 |
| plip | [59.4, | [44.5, | [40.2, | [71.8, | [91.4, | [59.0, | [59.0, | [83.5, | [56.4, | [84.5, | [45.4, | [93.8, |
| | 75.5] | 51.6] | 49.8] | 74.1] | 92.8] | 60.1] | 65.2] | 84.3] | 57.6] | 85.3] | 46.4] | 94.2] |
| | 62.2 | 50.7 | 56.5 | 70.5 | 92.1 | 55.3 | 66.4 | 84.0 | 56.9 | 84.8 | 46.5 | 94.3 |
| quilt | [53.2, | [46.9, | [50.7, | [69.3, | [91.4, | [54.9, | [63.4, | [83.6, | [56.4, | [84.4, | [45.9, | [94.1, |
| | 69.9] | 54.4] | 62.3] | 71.7] | 92.8] | 55.8] | 69.5] | 84.4] | 57.5] | 85.2] | 47.0] | 94.4] |
| | 59.7 | 43.7 | 64.1 | 73.1 | 86.7 | 66.7 | 76.4 | 80.6 | 58.2 | 82.5 | 36.5 | 86.9 |
| dinob | [51.4, | [39.9, | [58.1, | [72.0, | [85.9, | [66.1, | [73.5, | [80.2, | [57.6, | [82.1, | [36.0, | [86.7, |
| | 67.2] | 47.2] | 69.7] | 74.2] | 87.6] | 67.2] | 79.2] | 81.1] | 58.7] | 82.9] | 37.1] | 87.2] |
| | 64.0 | 46.8 | 65.0 | 74.1 | 86.3 | 67.2 | 80.5 | 80.8 | 58.0 | 83.6 | 38.0 | 90.3 |
| dinol | [55.7, | [43.0, | [59.0, | [72.9, | [85.4, | [66.6, | [77.8, | [80.4, | [57.4, | [83.2, | [37.5, | [90.1, |
| | 71.6] | 50.2] | 70.7] | 75.2] | 87.1] | 67.8] | 83.1] | 81.2] | 58.5] | 84.0] | 38.6] | 90.5] |
| | 54.7 | 45.4 | 55.7 | 63.4 | 87.8 | 58.3 | 68.8 | 78.3 | 56.5 | 82.0 | 35.7 | 86.5 |
| vitb | [46.0, | [41.7, | [49.5, | [62.2, | [87.0, | [57.8, | [65.7, | [77.9, | [55.9, | [81.6, | [35.2, | [86.3, |
| | 62.3] | 49.0] | 61.5] | 64.6] | 88.7] | 58.8] | 71.7] | 78.8] | 57.1] | 82.4] | 36.2] | 86.8] |
| | 55.4 | 46.9 | 64.2 | 70.3 | 89.2 | 64.6 | 70.2 | 78.8 | 57.4 | 83.6 | 39.4 | 90.2 |
| vitl | [46.8, | [42.9, | [58.2, | [69.1, | [88.4, | [64.0, | [67.1, | [78.4, | [56.8, | [83.2, | [38.8, | [90.0, |
| | 63.3] | 50.5] | 69.7] | 71.5] | 90.0] | 65.2] | 73.2] | 79.3] | 58.0] | 84.0] | 39.9] | 90.4] |
| | 45.3 | 42.5 | 54.6 | 65.0 | 81.3 | 53.8 | 66.6 | 78.5 | 54.7 | 78.4 | 33.6 | 88.3 |
| clipb | [37.0, | [38.9, | [48.2, | [63.8, | [80.4, | [53.3, | [63.7, | [78.0, | [54.1, | [77.9, | [33.0, | [88.1, |
| | 52.9] | 46.0] | 60.6] | 66.2] | 82.2] | 54.4] | 69.6] | 78.9] | 55.3] | 78.9] | 34.1] | 88.6] |
| | 52.0 | 46.6 | 52.1 | 64.8 | 86.3 | 60.4 | 64.2 | 81.0 | 55.4 | 81.3 | 38.1 | 90.6 |
| clipl | [43.4, | [42.7, | [46.1, | [63.6, | [85.4, | [59.8, | [59.3, | [80.6, | [54.8, | [80.9, | [37.6, | [90.4, |
| | 59.9] | 50.2] | 58.2] | 66.0] | 87.2] | 60.9] | 65.7] | 81.5] | 55.9] | 81.8] | 38.7] | 90.7] |

Table S38: Quantitative performance (Balanced accuracy) on linear probing.

| Model | bach | bracs | break-h | ccrcc | crc | esca | mhist | pcam | tcga-crc | tcga-tils | tcga-unif | wilds |
|---|---|---|---|---|---|---|---|---|---|---|---|---|
| hiboub | 70.7 [64.2, 77.2] | 62.0 [58.2, 65.8] | 55.8 [49.9, 61.5] | 86.6 [85.8, 87.4] | 94.5 [93.9, 95.1] | 78.5 [77.9, 79.0] | 75.8 [73.0, 78.6] | 93.4 [93.1, 93.6] | 71.0 [70.3, 71.7] | 88.3 [87.9, 88.7] | 68.7 [68.1, 69.2] | 97.1 [97.0, 97.2] |
| hiboul | 79.3 [72.9, 85.6] | 63.0 [59.2, 66.8] | 67.2 [61.8, 72.7] | 89.2 [88.4, 89.9] | 93.8 [93.1, 94.4] | 79.3 [78.8, 79.9] | 81.2 [78.6, 83.7] | 93.3 [93.1, 93.6] | 75.1 [74.5, 75.8] | 89.3 [88.9, 89.7] | 73.6 [73.1, 74.1] | 98.1 [98.0, 98.2] |
| hopt0 | 71.0 [62.9, 78.6] | 60.6 [56.7, 64.4] | 62.8 [56.7, 68.6] | 91.0 [90.3, 91.7] | 93.5 [92.8, 94.1] | 82.5 [82.0, 83.0] | 83.7 [81.2, 85.9] | 93.0 [92.7, 93.3] | 73.3 [72.6, 74.0] | 89.5 [89.2, 89.9] | 78.1 [77.6, 78.5] | 98.7 [98.7, 98.8] |
| hopt1 | 76.1 [69.5, 82.5] | 65.2 [61.7, 68.8] | 74.6 [68.9, 80.1] | 92.1 [91.5, 92.8] | 94.7 [94.1, 95.3] | 86.0 [85.5, 86.5] | 81.5 [79.0, 84.1] | 93.3 [93.1, 93.6] | 74.9 [74.2, 75.5] | 89.7 [89.3, 90.1] | 79.8 [79.4, 80.3] | 98.2 [98.1, 98.3] |
| midnight | 90.1 [85.4, 94.5] | 64.1 [60.3, 67.8] | 54.3 [48.8, 59.8] | 91.6 [90.9, 92.2] | 95.9 [95.4, 96.4] | 85.6 [85.1, 86.1] | 79.6 [76.9, 82.1] | 93.6 [93.3, 93.8] | 69.5 [68.7, 70.1] | 89.9 [89.5, 90.3] | 84.2 [83.8, 84.7] | 98.3 [98.2, 98.3] |
| phikon | 70.2 [62.3, 77.8] | 57.5 [53.8, 61.3] | 62.2 [56.6, 68.0] | 89.7 [88.9, 90.4] | 92.9 [92.2, 93.5] | 79.6 [79.1, 80.1] | 78.3 [75.7, 80.9] | 91.0 [90.7, 91.3] | 68.6 [67.8, 69.3] | 89.4 [89.0, 89.8] | 74.9 [74.4, 75.5] | 97.8 [97.7, 97.9] |
| phikon2 | 68.1 [60.5, 75.9] | 59.1 [55.5, 62.6] | 53.1 [47.2, 59.1] | 81.4 [80.7, 82.1] | 92.2 [91.5, 92.8] | 79.5 [79.0, 80.1] | 79.4 [76.7, 82.1] | 90.8 [90.5, 91.1] | 64.6 [63.9, 65.3] | 90.1 [89.7, 90.4] | 76.3 [75.8, 76.7] | 95.8 [95.7, 95.9] |
| uni | 74.7 [67.2, 81.7] | 59.5 [55.7, 63.4] | 69.1 [63.4, 75.0] | 91.7 [91.1, 92.4] | 93.6 [93.0, 94.3] | 83.7 [83.2, 84.3] | 82.5 [79.9, 84.9] | 93.8 [93.5, 94.1] | 71.0 [70.3, 71.7] | 88.6 [88.2, 89.0] | 73.7 [73.2, 74.2] | 97.2 [97.1, 97.3] |
| uni2h | 87.4 [82.3, 92.3] | 65.9 [62.2, 69.3] | 75.4 [69.8, 80.8] | 91.3 [90.7, 91.9] | 94.5 [93.9, 95.0] | 86.3 [85.9, 86.8] | 77.9 [75.1, 80.7] | 95.0 [94.7, 95.2] | 73.9 [73.2, 74.5] | 89.4 [89.0, 89.8] | 77.7 [77.2, 78.1] | 98.9 [98.8, 99.0] |
| virchow | 67.0 [58.5, 74.8] | 60.1 [56.2, 64.0] | 60.6 [54.8, 66.4] | 92.1 [91.5, 92.8] | 93.5 [92.8, 94.3] | 85.4 [84.9, 85.8] | 82.7 [80.2, 85.1] | 93.6 [93.4, 93.9] | 71.2 [70.6, 71.9] | 89.3 [89.0, 89.7] | 73.6 [73.1, 74.1] | 97.6 [97.5, 97.7] |
| virchow2 | 78.4 [71.6, 85.1] | 63.7 [60.2, 67.2] | 70.2 [65.2, 75.6] | 93.7 [93.2, 94.3] | 92.7 [92.0, 93.5] | 88.0 [87.5, 88.5] | 84.8 [82.3, 87.0] | 92.8 [92.5, 93.1] | 74.4 [73.7, 75.0] | 89.9 [89.5, 90.2] | 77.1 [76.6, 77.6] | 90.1 [89.9, 90.3] |
| conch | 87.3 [82.1, 92.3] | 60.1 [56.3, 63.9] | 65.9 [60.2, 71.5] | 89.6 [88.9, 90.3] | 94.1 [93.4, 94.7] | 81.4 [80.9, 81.9] | 80.0 [77.2, 82.5] | 90.6 [90.3, 90.9] | 69.1 [68.4, 69.8] | 85.2 [84.7, 85.6] | 66.9 [66.4, 67.4] | 97.2 [97.1, 97.3] |
| titan | 83.2 [79.2, 87.4] | 63.2 [59.6, 66.8] | 74.6 [69.2, 80.0] | 88.3 [87.6, 89.1] | 94.3 [93.7, 94.9] | 82.8 [82.2, 83.4] | 81.2 [78.5, 83.7] | 91.4 [91.1, 91.7] | 69.2 [68.5, 69.9] | 88.2 [87.8, 88.6] | 69.1 [68.6, 69.6] | 96.7 [96.6, 96.8] |
| keep | 82.0 [76.5, 87.1] | 63.1 [59.4, 66.7] | 63.2 [57.3, 68.8] | 93.4 [92.9, 94.1] | 95.0 [94.4, 95.5] | 86.1 [85.7, 86.5] | 76.2 [73.3, 78.9] | 92.7 [92.4, 92.9] | 71.9 [71.2, 72.5] | 88.1 [87.8, 88.6] | 71.4 [70.9, 72.0] | 97.4 [97.3, 97.5] |
| musk | 74.3 [68.0, 80.5] | 64.2 [60.5, 67.8] | 73.5 [68.5, 78.5] | 84.7 [83.8, 85.6] | 91.0 [90.2, 91.8] | 79.0 [78.4, 79.5] | 79.2 [76.6, 81.9] | 89.4 [89.0, 89.7] | 68.1 [67.4, 68.8] | 87.1 [86.6, 87.5] | 64.5 [63.9, 65.1] | 96.0 [95.9, 96.2] |
| plip | 64.3 [56.3, 72.1] | 57.3 [53.5, 60.9] | 44.2 [39.2, 49.3] | 76.2 [75.1, 77.3] | 88.3 [87.5, 89.1] | 68.2 [67.7, 68.7] | 79.0 [76.3, 81.6] | 87.9 [87.5, 88.2] | 62.4 [61.7, 63.1] | 86.3 [85.9, 86.7] | 53.5 [52.9, 54.0] | 89.1 [88.9, 89.3] |
| quilt | 60.9 [53.1, 68.7] | 56.9 [53.3, 60.7] | 58.1 [52.2, 63.9] | 64.6 [63.5, 65.8] | 92.4 [91.7, 93.2] | 63.2 [62.8, 63.6] | 73.5 [70.6, 76.3] | 87.4 [87.0, 87.8] | 64.4 [63.7, 65.1] | 85.3 [84.9, 85.7] | 53.3 [52.7, 53.8] | 94.7 [94.5, 94.8] |
| dinob | 66.9 [59.1, 74.4] | 53.3 [49.4, 57.0] | 76.4 [70.8, 81.9] | 79.5 [78.5, 80.6] | 90.3 [89.6, 91.1] | 74.7 [74.1, 75.2] | 81.9 [79.3, 84.3] | 86.3 [85.9, 86.7] | 66.5 [65.8, 67.2] | 83.5 [83.1, 83.9] | 53.4 [52.8, 53.9] | 90.7 [90.5, 90.9] |
| dinol | 71.1 [63.3, 78.7] | 51.7 [47.9, 55.4] | 74.9 [69.5, 80.4] | 82.6 [81.7, 83.5] | 91.3 [90.5, 92.0] | 72.4 [71.9, 72.9] | 82.8 [80.2, 85.3] | 87.2 [86.8, 87.5] | 67.1 [66.4, 67.9] | 83.1 [82.6, 83.5] | 55.5 [55.0, 56.1] | 90.3 [90.1, 90.5] |
| vitb | 60.3 [51.8, 68.3] | 56.9 [53.1, 60.7] | 63.4 [57.9, 69.3] | 80.1 [79.1, 81.1] | 91.1 [90.4, 91.9] | 66.7 [66.2, 67.2] | 77.2 [74.4, 79.9] | 84.7 [84.3, 85.1] | 65.5 [64.8, 66.2] | 83.1 [82.6, 83.5] | 52.4 [51.9, 53.0] | 91.3 [91.1, 91.5] |
| vitl | 58.4 [50.0, 66.6] | 55.8 [52.0, 59.4] | 65.5 [59.6, 71.1] | 77.5 [76.5, 78.6] | 92.9 [92.3, 93.6] | 69.9 [69.4, 70.4] | 79.1 [76.4, 81.8] | 85.8 [85.4, 86.2] | 62.9 [62.1, 63.6] | 85.6 [85.1, 86.0] | 54.4 [53.9, 54.9] | 92.6 [92.5, 92.8] |
| clipb | 53.0 [44.6, 61.2] | 52.7 [48.9, 56.6] | 39.1 [35.1, 43.3] | 72.8 [71.7, 74.0] | 88.4 [87.6, 89.1] | 62.9 [62.5, 63.4] | 77.8 [75.0, 80.4] | 82.8 [82.4, 83.2] | 60.0 [59.3, 60.6] | 81.3 [80.8, 81.7] | 47.1 [46.6, 47.6] | 83.8 [83.5, 84.0] |
| clipl | 66.0 [58.3, 73.7] | 57.2 [53.3, 60.9] | 51.3 [45.1, 57.2] | 79.1 [78.1, 80.0] | 88.5 [87.8, 89.4] | 68.1 [67.6, 68.7] | 79.5 [76.7, 82.1] | 85.6 [85.2, 85.9] | 59.1 [58.4, 59.9] | 83.1 [82.7, 83.6] | 53.0 [52.5, 53.6] | 93.2 [93.1, 93.4] |

Table S39: Quantitative performance (F1-score) on linear probing.

| Model | bach | bracs | break-h | ccrcc | crc | esca | mhist | pcam | tcga-crc | tcga-tils | tcga-unif | wilds |
|---|---|---|---|---|---|---|---|---|---|---|---|---|
| hiboub | 65.6 [57.3, 73.3] | 61.7 [57.8, 65.5] | 59.6 [53.0, 65.5] | 84.3 [83.4, 85.2] | 93.9 [93.3, 94.5] | 77.5 [77.0, 78.1] | 76.5 [73.7, 79.2] | 93.4 [93.1, 93.6] | 66.2 [65.6, 66.8] | 89.6 [89.3, 89.9] | 70.2 [69.6, 70.7] | 97.1 [97.0, 97.2] |
| hiboul | 76.5 [69.1, 82.9] | 62.7 [58.8, 66.5] | 69.4 [63.9, 74.5] | 88.4 [87.6, 89.2] | 93.0 [92.3, 93.7] | 77.5 [77.0, 77.9] | 81.3 [78.8, 83.8] | 93.3 [93.0, 93.6] | 68.1 [67.5, 68.7] | 90.3 [90.0, 90.6] | 75.2 [74.7, 75.6] | 98.1 [98.0, 98.2] |
| hopt0 | 69.7 [61.2, 77.1] | 61.1 [57.3, 64.7] | 66.0 [59.5, 71.8] | 90.8 [90.1, 91.5] | 93.5 [92.9, 94.2] | 83.5 [83.1, 84.0] | 82.8 [80.3, 85.1] | 93.0 [92.7, 93.3] | 67.3 [66.7, 67.9] | 91.0 [90.7, 91.3] | 79.3 [78.8, 79.7] | 98.7 [98.7, 98.8] |
| hopt1 | 72.5 [64.4, 79.7] | 64.3 [60.6, 67.9] | 76.0 [70.4, 80.9] | 91.7 [91.0, 92.4] | 94.6 [93.9, 95.2] | 87.1 [86.7, 87.5] | 82.0 [79.5, 84.5] | 93.3 [93.1, 93.6] | 67.9 [67.3, 68.5] | 91.0 [90.7, 91.3] | 81.0 [80.5, 81.4] | 98.2 [98.1, 98.3] |
| midnight | 87.9 [82.0, 93.2] | 63.8 [59.9, 67.4] | 56.7 [49.8, 62.7] | 90.8 [90.1, 91.5] | 95.6 [95.0, 96.1] | 86.2 [85.7, 86.6] | 80.2 [77.5, 82.6] | 93.5 [93.3, 93.8] | 65.6 [65.0, 66.2] | 91.0 [90.7, 91.3] | 85.2 [84.8, 85.6] | 98.3 [98.2, 98.3] |
| phikon | 69.3 [61.2, 76.4] | 56.9 [52.9, 60.6] | 60.0 [53.8, 65.5] | 89.8 [89.1, 90.6] | 92.1 [91.4, 92.7] | 76.7 [76.3, 77.1] | 78.1 [75.4, 80.7] | 90.9 [90.6, 91.2] | 63.1 [62.5, 63.7] | 90.1 [89.8, 90.4] | 76.6 [76.1, 77.1] | 97.8 [97.7, 97.9] |
| phikon2 | 64.7 [56.3, 72.4] | 58.2 [54.3, 61.7] | 53.0 [46.7, 59.1] | 76.7 [75.7, 77.8] | 92.0 [91.3, 92.7] | 77.3 [76.9, 77.7] | 79.2 [76.6, 81.8] | 90.8 [90.5, 91.1] | 61.1 [60.5, 61.7] | 91.0 [90.7, 91.3] | 77.7 [77.3, 78.2] | 95.8 [95.7, 95.9] |
| uni | 73.0 [65.1, 79.7] | 59.4 [55.5, 63.1] | 70.8 [65.3, 75.9] | 90.7 [89.9, 91.4] | 93.1 [92.5, 93.8] | 83.4 [83.0, 83.8] | 82.4 [79.8, 84.7] | 93.8 [93.5, 94.0] | 65.9 [65.3, 66.5] | 90.1 [89.7, 90.4] | 75.3 [74.8, 75.8] | 97.2 [97.1, 97.3] |
| uni2h | 84.7 [78.1, 90.6] | 65.4 [61.5, 68.8] | 78.4 [73.0, 83.2] | 89.5 [88.7, 90.3] | 93.9 [93.2, 94.5] | 86.4 [86.0, 86.8] | 78.6 [75.7, 81.3] | 95.0 [94.7, 95.2] | 66.9 [66.2, 67.4] | 90.4 [90.0, 90.7] | 78.8 [78.4, 79.3] | 98.9 [98.8, 99.0] |
| virchow | 66.2 [57.4, 73.9] | 59.8 [55.7, 63.6] | 62.7 [56.6, 68.4] | 91.7 [91.0, 92.3] | 93.7 [93.0, 94.4] | 84.3 [83.9, 84.7] | 82.3 [79.8, 84.6] | 93.6 [93.3, 93.9] | 65.5 [64.9, 66.1] | 90.7 [90.4, 91.0] | 74.8 [74.3, 75.3] | 97.6 [97.5, 97.7] |
| virchow2 | 76.5 [69.2, 83.0] | 62.4 [58.6, 66.0] | 72.5 [66.8, 77.6] | 93.6 [93.0, 94.2] | 93.0 [92.3, 93.7] | 88.9 [88.5, 89.2] | 83.6 [81.2, 86.0] | 92.8 [92.5, 93.0] | 69.6 [69.0, 70.2] | 90.9 [90.6, 91.2] | 78.4 [77.9, 78.8] | 90.0 [89.8, 90.2] |
| conch | 84.4 [78.2, 90.2] | 60.0 [56.0, 63.7] | 68.8 [62.9, 74.1] | 88.1 [87.3, 88.9] | 93.4 [92.8, 94.0] | 80.1 [79.6, 80.5] | 79.7 [76.9, 82.2] | 90.6 [90.3, 90.9] | 64.4 [63.8, 65.0] | 88.1 [87.8, 88.5] | 68.0 [67.5, 68.5] | 97.2 [97.1, 97.3] |
| titan | 76.8 [70.1, 83.2] | 62.4 [58.5, 66.0] | 75.3 [70.4, 79.9] | 87.4 [86.5, 88.2] | 93.8 [93.2, 94.4] | 82.7 [82.2, 83.1] | 81.2 [78.5, 83.7] | 91.4 [ , 91.7] | 62.3 [61.7, 62.9] | 89.9 [89.6, 90.2] | 69.4 [ , 69.9] | 96.7 [96.6, 96.8] |
| keep | 77.0 [69.8, 83.5] | 62.2 [58.4, 65.9] | 65.9 [60.4, 70.8] | 93.2 [92.5, 93.8] | 94.7 [94.1, 95.3] | 85.8 [85.5, 86.2] | 76.8 [73.8, 79.4] | 92.6 [92.4, 92.9] | 64.9 [64.3, 65.5] | 89.9 [89.6, 90.3] | 72.7 [72.1, 73.2] | 97.4 [97.3, 97.5] |
| musk | 69.9 [62.0, 77.5] | 63.3 [59.3, 66.9] | 77.6 [71.9, 82.6] | 85.3 [84.4, 86.1] | 91.1 [90.4, 91.9] | 76.4 [76.0, 76.8] | 80.0 [77.5, 82.6] | 89.4 [89.0, 89.7] | 63.2 [62.6, 63.8] | 89.4 [89.0, 89.7] | 66.0 [65.4, 66.5] | 96.0 [95.9, 96.2] |
| plip | 63.9 [55.5, 71.4] | 56.5 [52.6, 60.2] | 47.2 [40.4, 53.6] | 76.9 [75.8, 78.0] | 88.2 [87.3, 89.0] | 65.7 [65.1, 66.2] | 78.5 [75.8, 81.0] | 87.9 [87.5, 88.2] | 56.0 [55.4, 56.5] | 87.2 [86.8, 87.5] | 54.9 [54.3, 55.4] | 89.0 [88.8, 89.2] |
| quilt | 58.4 [50.2, 66.2] | 55.9 [52.0, 59.8] | 62.4 [55.9, 68.0] | 65.5 [64.2, 66.9] | 92.5 [91.7, 93.1] | 61.5 [60.9, 62.0] | 73.5 [70.6, 76.3] | 87.4 [87.0, 87.7] | 59.6 [59.0, 60.2] | 87.0 [86.6, 87.3] | 54.1 [53.6, 54.7] | 94.7 [94.5, 94.8] |
| dinob | 64.4 [55.7, 72.4] | 52.9 [49.1, 56.4] | 77.8 [72.3, 82.8] | 79.7 [78.7, 80.8] | 90.3 [89.5, 91.0] | 73.8 [73.3, 74.3] | 82.6 [80.1, 85.0] | 86.3 [85.9, 86.7] | 58.6 [58.0, 59.2] | 85.9 [85.6, 86.3] | 54.4 [53.8, 54.9] | 90.7 [90.5, 90.9] |
| dinol | 68.9 [60.4, 76.6] | 50.6 [46.6, 54.2] | 78.5 [72.9, 83.6] | 81.4 [80.4, 82.4] | 90.7 [89.9, 91.4] | 72.1 [71.6, 72.6] | 82.3 [79.6, 84.7] | 87.2 [86.8, 87.5] | 59.3 [58.7, 59.9] | 85.9 [85.5, 86.3] | 56.2 [55.6, 56.7] | 90.3 [90.1, 90.5] |
| vitb | 57.0 [48.2, 64.8] | 56.6 [52.6, 60.2] | 64.5 [58.7, 70.1] | 79.1 [78.0, 80.1] | 90.6 [89.8, 91.3] | 62.7 [62.3, 63.2] | 77.6 [74.8, 80.2] | 84.6 [84.2, 85.0] | 59.1 [58.5, 59.7] | 85.8 [85.4, 86.2] | 53.4 [52.8, 53.9] | 91.3 [91.1, 91.4] |
| vitl | 56.9 [47.7, 64.7] | 54.8 [50.8, 58.5] | 66.3 [60.2, 71.8] | 77.1 [76.1, 78.2] | 92.8 [92.2, 93.5] | 67.9 [67.4, 68.4] | 79.8 [77.1, 82.4] | 85.7 [85.3, 86.1] | 56.8 [56.2, 57.4] | 87.5 [87.2, 87.9] | 55.9 [55.3, 56.4] | 92.6 [92.4, 92.8] |
| clipb | 51.1 [42.6, 58.7] | 52.4 [48.6, 55.9] | 40.0 [33.8, 45.6] | 73.7 [72.4, 74.8] | 87.3 [86.4, 88.1] | 61.0 [60.5, 61.6] | 77.5 [74.7, 79.9] | 82.7 [82.3, 83.1] | 49.0 [48.5, 49.5] | 83.8 [83.4, 84.2] | 47.9 [47.4, 48.5] | 83.5 [83.2, 83.7] |
| clipl | 63.5 [55.2, 71.2] | 56.4 [52.5, 60.1] | 54.5 [47.5, 60.6] | 75.9 [74.8, 76.9] | 88.3 [87.5, 89.2] | 67.6 [67.1, 68.2] | 79.6 [76.9, 82.1] | 85.6 [85.2, 85.9] | 52.0 [51.4, 52.6] | 85.0 [84.6, 85.4] | 54.1 [53.5, 54.6] | 93.2 [93.1, 93.4] |

Table S40: Quantitative performance (Balanced accuracy) on 1-shot classification.

| Model | bach | bracs | break-h | ccrcc | crc | esca | mhist | pcam | tcga-crc | tcga-tils | tcga-unif | wilds |
|---|---|---|---|---|---|---|---|---|---|---|---|---|
| | 74.0 | 54.1 | 77.7 | 77.7 | 92.0 | 82.2 | 59.2 | 88.4 | 62.8 | 77.2 | 58.5 | 79.5 |
| hiboub | [68.6, | [51.1, | [72.7, | [76.7, | [91.2, | [81.9, | [57.8, | [88.1, | [62.1, | [76.9, | [57.9, | [79.3, |
| | 79.4] | 56.9] | 82.5] | 78.6] | 92.7] | 82.5] | 60.9] | 88.8] | 63.5] | 77.4] | 59.0] | 79.8] |
| | 71.4 | 51.6 | 73.4 | 80.7 | 74.6 | 78.0 | 58.4 | 91.7 | 65.8 | 81.1 | 61.2 | 64.2 |
| hiboul | [66.0, | [48.5, | [68.2, | [79.9, | [73.6, | [77.6, | [57.0, | [91.4, | [65.1, | [80.7, | [60.7, | [63.9, |
| | 76.7] | 54.8] | 78.5] | 81.6] | 75.5] | 78.3] | 60.0] | 92.0] | 66.4] | 81.4] | 61.7] | 64.4] |
| | 62.6 | 45.5 | 70.6 | 92.4 | 86.8 | 82.5 | 59.0 | 71.8 | 69.1 | 82.1 | 71.9 | 74.9 |
| hopt0 | [54.6, | [42.4, | [64.8, | [91.7, | [85.9, | [82.2, | [57.4, | [71.4, | [68.4, | [81.8, | [71.5, | [74.6, |
| | 70.5] | 48.5] | 76.2] | 93.0] | 87.7] | 82.8] | 60.6] | 72.3] | 69.8] | 82.4] | 72.4] | 75.1] |
| | 67.3 | 48.2 | 69.5 | 89.4 | 90.9 | 85.1 | 55.6 | 72.6 | 71.9 | 54.6 | 74.9 | 53.9 |
| hopt1 | [60.7, | [45.0, | [64.6, | [88.7, | [90.1, | [84.8, | [54.4, | [72.2, | [71.2, | [54.5, | [74.5, | [53.8, |
| | 73.6] | 51.4] | 74.4] | 90.2] | 91.7] | 85.4] | 56.9] | 73.0] | 72.5] | 54.7] | 75.3] | 54.1] |
| | 82.7 | 47.0 | 42.0 | 89.1 | 94.2 | 82.6 | 67.4 | 78.1 | 56.4 | 79.0 | 65.1 | 86.2 |
| midnight | [77.4, | [43.9, | [37.1, | [88.2, | [93.6, | [82.2, | [64.7, | [77.7, | [55.8, | [78.6, | [64.7, | [86.0, |
| | 87.8] | 50.1] | 46.8] | 89.9] | 94.9] | 83.0] | 70.1] | 78.5] | 57.0] | 79.4] | 65.6] | 86.4] |
| | 63.2 | 44.1 | 71.4 | 91.7 | 91.6 | 78.8 | 59.5 | 84.2 | 67.0 | 76.1 | 60.1 | 92.9 |
| phikon | [55.3, | [40.9, | [66.4, | [91.0, | [90.9, | [78.5, | [57.9, | [83.8, | [66.4, | [75.8, | [59.5, | [92.7, |
| | 70.5] | 47.3] | 76.4] | 92.3] | 92.4] | 79.1] | 61.1] | 84.6] | 67.6] | 76.3] | 60.6] | 93.1] |
| | 60.5 | 44.3 | 65.3 | 90.1 | 85.5 | 81.4 | 58.6 | 80.2 | 59.2 | 50.0 | 66.0 | 89.9 |
| phikon2 | [53.0, | [41.1, | [59.6, | [89.3, | [84.7, | [81.1, | [57.0, | [79.8, | [58.6, | [50.0, | [65.5, | [89.7, |
| | 67.7] | 47.4] | 71.0] | 90.8] | 86.3] | 81.7] | 60.2] | 80.6] | 59.8] | 50.0] | 66.4] | 90.1] |
| | 73.0 | 52.5 | 81.0 | 87.3 | 89.6 | 83.3 | 61.9 | 88.0 | 66.7 | 75.3 | 67.1 | 88.4 |
| uni | [66.6, | [49.3, | [76.1, | [86.6, | [88.7, | [83.0, | [60.3, | [87.6, | [65.9, | [75.1, | [66.6, | [88.2, |
| | 79.2] | 55.7] | 85.8] | 88.0] | 90.4] | 83.6] | 63.7] | 88.3] | 67.4] | 75.6] | 67.6] | 88.6] |
| | 79.0 | 54.6 | 69.3 | 89.5 | 95.2 | 86.5 | 56.7 | 82.4 | 69.9 | 51.4 | 72.7 | 96.3 |
| uni2h | [74.5, | [52.0, | [63.6, | [88.7, | [94.6, | [86.1, | [55.3, | [82.0, | [69.2, | [51.3, | [72.3, | [96.2, |
| | 83.4] | 57.1] | 74.9] | 90.3] | 95.8] | 86.8] | 58.1] | 82.7] | 70.6] | 51.5] | 73.2] | 96.4] |
| | 51.5 | 42.1 | 58.6 | 86.8 | 89.0 | 85.0 | 62.1 | 86.4 | 63.7 | 58.6 | 54.7 | 78.7 |
| virchow | [44.0, | [38.8, | [52.6, | [86.0, | [88.2, | [84.6, | [60.3, | [86.0, | [63.0, | [58.4, | [54.1, | [78.5, |
| | 59.1] | 45.3] | 64.5] | 87.5] | 89.8] | 85.3] | 64.1] | 86.7] | 64.4] | 58.8] | 55.2] | 79.0] |
| | 80.5 | 53.0 | 58.6 | 84.1 | 89.1 | 82.8 | 55.9 | 74.6 | 60.6 | 81.3 | 64.7 | 88.6 |
| virchow2 | [75.1, | [50.1, | [52.7, | [83.1, | [88.3, | [82.4, | [54.7, | [74.1, | [60.0, | [80.9, | [64.2, | [88.4, |
| | 85.5] | 55.9] | 64.4] | 85.0] | 90.0] | 83.2] | 57.3] | 75.1] | 61.3] | 81.6] | 65.2] | 88.8] |
| | 84.0 | 52.4 | 63.5 | 88.4 | 92.4 | 84.2 | 58.6 | 83.2 | 60.6 | 76.2 | 55.7 | 95.0 |
| conch | [79.5, | [50.2, | [58.5, | [87.6, | [91.7, | [83.8, | [57.1, | [82.8, | [59.9, | [75.9, | [55.2, | [94.9, |
| | 88.5] | 54.6] | 68.6] | 89.3] | 93.2] | 84.5] | 60.2] | 83.6] | 61.2] | 76.6] | 56.2] | 95.2] |
| | 80.3 | 55.8 | 68.7 | 87.8 | 91.0 | 84.5 | 60.8 | 87.3 | 64.8 | 66.3 | 58.6 | 94.1 |
| titan | [76.1, | [53.2, | [62.9, | [87.0, | [90.2, | [84.1, | [59.1, | [87.0, | [64.1, | [66.1, | [58.2, | [94.0, |
| | 84.6] | 58.3] | 74.3] | 88.6] | 91.7] | 84.9] | 62.6] | 87.7] | 65.5] | 66.6] | 59.1] | 94.3] |
| | 84.3 | 53.0 | 62.6 | 93.7 | 90.8 | 86.3 | 63.5 | 86.2 | 62.9 | 82.6 | 62.2 | 95.8 |
| keep | [78.4, | [50.9, | [57.3, | [93.1, | [90.0, | [85.9, | [61.2, | [85.8, | [62.2, | [82.3, | [61.8, | [95.7, |
| | 89.8] | 55.1] | 67.6] | 94.2] | 91.6] | 86.7] | 65.7] | 86.5] | 63.6] | 82.9] | 62.7] | 96.0] |
| | 72.0 | 50.6 | 71.6 | 81.7 | 90.1 | 80.7 | 58.1 | 74.1 | 62.0 | 72.0 | 49.2 | 78.1 |
| musk | [65.6, | [48.2, | [66.8, | [80.7, | [89.3, | [80.3, | [56.7, | [73.7, | [61.3, | [71.7, | [48.6, | [77.8, |
| | 78.0] | 52.9] | 76.2] | 82.6] | 90.8] | 81.1] | 59.6] | 74.6] | 62.8] | 72.3] | 49.7] | 78.3] |
| | 59.7 | 49.0 | 59.0 | 79.6 | 84.9 | 67.0 | 60.3 | 73.9 | 58.0 | 75.8 | 40.0 | 72.9 |
| plip | [51.5, | [46.0, | [53.6, | [78.7, | [83.9, | [66.5, | [58.6, | [73.4, | [57.2, | [75.5, | [39.5, | [72.6, |
| | 67.6] | 51.9] | 64.3] | 80.6] | 85.8] | 67.5] | 62.1] | 74.3] | 58.7] | 76.2] | 40.5] | 73.1] |
| | 56.6 | 49.8 | 70.4 | 79.1 | 88.9 | 67.5 | 62.0 | 80.4 | 61.0 | 78.0 | 39.4 | 79.6 |
| quilt | [49.3, | [46.8, | [65.5, | [78.2, | [88.1, | [67.1, | [60.3, | [80.0, | [60.2, | [77.7, | [38.9, | [79.3, |
| | 63.5] | 52.6] | 75.4] | 80.1] | 89.6] | 67.9] | 63.8] | 80.8] | 61.7] | 78.3] | 40.0] | 79.8] |
| | 63.2 | 41.2 | 68.7 | 76.7 | 76.5 | 65.8 | 72.8 | 65.0 | 56.9 | 56.3 | 31.8 | 68.0 |
| dinob | [55.4, | [38.1, | [62.9, | [75.6, | [75.5, | [65.3, | [70.6, | [64.6, | [56.2, | [56.2, | [31.3, | [67.7, |
| | 70.5] | 44.3] | 74.3] | 77.8] | 77.5] | 66.3] | 75.0] | 65.4] | 57.6] | 56.5] | 32.3] | 68.2] |
| | 63.2 | 43.6 | 66.9 | 66.1 | 73.8 | 66.1 | 73.4 | 60.8 | 59.4 | 57.4 | 32.2 | 66.0 |
| dinol | [55.5, | [40.5, | [61.4, | [65.0, | [72.8, | [65.7, | [70.9, | [60.5, | [58.6, | [57.2, | [31.6, | [65.7, |
| | 70.7] | 46.6] | 72.2] | 67.3] | 74.8] | 66.6] | 75.8] | 61.1] | 60.1] | 57.5] | 32.7] | 66.2] |
| | 59.1 | 45.1 | 59.5 | 60.4 | 73.1 | 63.8 | 63.4 | 68.7 | 55.6 | 69.4 | 26.6 | 65.4 |
| vitb | [52.4, | [42.2, | [54.3, | [59.3, | [72.1, | [63.2, | [61.5, | [68.3, | [54.8, | [69.0, | [26.0, | [65.1, |
| | 65.6] | 48.0] | 64.8] | 61.6] | 74.0] | 64.3] | 65.3] | 69.2] | 56.3] | 69.7] | 27.1] | 65.6] |
| | 56.5 | 47.0 | 47.9 | 70.1 | 76.3 | 64.8 | 64.3 | 68.2 | 57.0 | 59.0 | 28.5 | 67.9 |
| vitl | [50.1, | [44.1, | [42.3, | [69.0, | [75.3, | [64.2, | [62.4, | [67.7, | [56.3, | [58.8, | [28.0, | [67.6, |
| | 62.6] | 49.8] | 53.2] | 71.3] | 77.3] | 65.3] | 66.3] | 68.7] | 57.7] | 59.2] | 29.0] | 68.1] |
| | 50.4 | 39.5 | 52.9 | 62.5 | 68.7 | 54.1 | 60.1 | 63.9 | 55.2 | 66.5 | 29.3 | 53.0 |
| clipb | [42.7, | [36.8, | [47.4, | [61.4, | [67.8, | [53.6, | [58.4, | [63.5, | [54.5, | [66.2, | [28.7, | [52.9, |
| | 58.0] | 42.3] | 58.7] | 63.6] | 69.7] | 54.5] | 61.8] | 64.3] | 56.0] | 66.8] | 29.8] | 53.1] |
| | 57.1 | 43.4 | 50.1 | 64.4 | 75.1 | 65.4 | 60.5 | 63.9 | 54.8 | 58.8 | 31.8 | 66.3 |
| clipl | [50.3, | [40.4, | [44.6, | [63.3, | [74.1, | [64.9, | [58.8, | [63.5, | [54.1, | [58.6, | [31.3, | [66.0, |
| | 63.6] | 46.4] | 55.8] | 65.5] | 76.1] | 65.9] | 62.2] | 64.3] | 55.5] | 59.0] | 32.4] | 66.5] |

Table S41: Quantitative performance (F1-score) on 1-shot classification.

| Model | bach | bracs | break-h | ccrcc | crc | esca | mhist | pcam | tcga-crc | tcga-tils | tcga-unif | wilds |
|---|---|---|---|---|---|---|---|---|---|---|---|---|
| hiboub | 66.7 [58.9, 73.8] | 47.6 [43.9, 51.0] | 75.6 [70.4, 79.9] | 74.2 [73.1, 75.3] | 92.0 [91.2, 92.7] | 61.6 [61.3, 61.9] | 45.0 [41.9, 48.2] | 88.4 [88.0, 88.7] | 56.5 [55.9, 57.0] | 61.0 [60.6, 61.4] | 48.8 [48.4, 49.3] | 78.8 [78.6, 79.1] |
| hiboul | 62.5 [54.9, 69.6] | 47.0 [43.4, 50.6] | 72.6 [67.8, 77.1] | 77.5 [76.5, 78.6] | 76.5 [75.6, 77.5] | 60.9 [60.5, 61.2] | 43.6 [40.5, 46.7] | 91.7 [91.4, 92.0] | 56.2 [55.7, 56.8] | 69.0 [68.6, 69.4] | 51.2 [50.8, 51.7] | 59.0 [58.7, 59.3] |
| hopt0 | 61.3 [52.8, 69.2] | 39.0 [35.4, 42.7] | 64.9 [59.1, 69.9] | 92.1 [91.4, 92.8] | 87.5 [86.5, 88.4] | 68.0 [67.7, 68.2] | 45.1 [42.0, 48.2] | 70.5 [70.0, 71.0] | 66.7 [66.0, 67.3] | 70.3 [69.8, 70.7] | 62.5 [62.1, 63.0] | 73.3 [73.0, 73.6] |
| hopt1 | 61.0 [52.2, 68.5] | 43.8 [40.1, 47.4] | 58.1 [52.7, 63.0] | 90.0 [89.3, 90.8] | 91.5 [90.7, 92.3] | 71.7 [71.4, 72.0] | 38.4 [35.7, 41.4] | 70.9 [70.4, 71.4] | 64.7 [64.1, 65.3] | 25.2 [24.9, 25.5] | 66.5 [66.1, 67.0] | 41.5 [41.2, 41.8] |
| midnight | 78.5 [71.1, 85.0] | 41.7 [38.4, 45.0] | 42.0 [37.2, 46.3] | 89.8 [89.0, 90.6] | 94.3 [93.7, 95.0] | 69.9 [69.6, 70.3] | 62.9 [60.0, 65.9] | 78.1 [77.6, 78.5] | 57.2 [56.5, 57.9] | 71.1 [70.7, 71.5] | 59.2 [58.7, 59.7] | 86.1 [85.9, 86.4] |
| phikon | 60.0 [51.8, 67.1] | 39.1 [35.4, 42.8] | 68.6 [63.2, 73.3] | 90.4 [89.7, 91.2] | 92.1 [91.3, 92.8] | 58.7 [58.4, 59.0] | 45.6 [42.6, 48.8] | 83.9 [83.5, 84.3] | 54.3 [53.8, 54.9] | 59.9 [59.5, 60.3] | 50.8 [50.4, 51.2] | 92.8 [92.7, 93.0] |
| phikon2 | 54.1 [45.4, 62.0] | 38.5 [34.9, 41.9] | 61.5 [55.9, 66.8] | 89.8 [89.0, 90.5] | 86.0 [85.1, 86.9] | 60.8 [60.5, 61.1] | 44.3 [41.2, 47.5] | 80.0 [79.6, 80.5] | 42.3 [41.7, 42.8] | 15.7 [15.5, 16.0] | 55.9 [55.4, 56.3] | 89.8 [89.6, 90.0] |
| uni | 68.3 [60.5, 75.1] | 47.6 [43.8, 51.2] | 78.6 [73.8, 82.9] | 86.7 [85.8, 87.4] | 89.8 [89.0, 90.6] | 63.7 [63.4, 64.0] | 49.5 [46.3, 52.6] | 88.0 [87.6, 88.3] | 61.4 [60.9, 62.1] | 58.4 [58.0, 58.8] | 57.0 [56.6, 57.5] | 88.3 [88.1, 88.5] |
| uni2h | 71.3 [64.2, 77.9] | 47.1 [43.6, 50.6] | 68.6 [63.3, 73.5] | 90.2 [89.5, 90.9] | 94.9 [94.2, 95.5] | 70.8 [70.5, 71.2] | 40.9 [37.9, 43.8] | 82.1 [81.7, 82.5] | 62.9 [62.3, 63.5] | 18.7 [18.5, 19.0] | 64.7 [64.3, 65.1] | 96.3 [96.2, 96.4] |
| virchow | 47.5 [39.5, 55.1] | 36.8 [33.1, 40.4] | 55.0 [49.6, 60.1] | 84.4 [83.5, 85.3] | 89.2 [88.3, 90.1] | 70.6 [70.3, 70.9] | 50.7 [47.5, 53.9] | 86.4 [86.0, 86.7] | 61.8 [61.1, 62.4] | 33.1 [32.7, 33.5] | 46.4 [45.9, 46.9] | 78.0 [77.7, 78.2] |
| virchow2 | 75.0 [67.4, 81.6] | 46.8 [43.4, 50.2] | 59.0 [53.3, 64.1] | 84.9 [84.0, 85.8] | 89.6 [88.8, 90.3] | 66.8 [66.5, 67.2] | 39.1 [36.1, 42.0] | 74.6 [74.1, 75.1] | 60.7 [60.1, 61.4] | 69.3 [68.9, 69.7] | 55.3 [54.9, 55.8] | 88.5 [88.3, 88.7] |
| conch | 78.5 [71.5, 84.8] | 42.6 [39.5, 45.5] | 64.4 [57.9, 69.6] | 88.9 [88.1, 89.8] | 91.9 [91.2, 92.6] | 66.4 [66.1, 66.7] | 44.4 [41.5, 47.5] | 83.2 [82.8, 83.6] | 61.3 [60.6, 62.0] | 62.7 [62.3, 63.2] | 45.5 [45.0, 45.9] | 95.0 [94.9, 95.2] |
| titan | 72.7 [65.7, 79.4] | 48.5 [44.8, 52.0] | 67.4 [61.3, 72.6] | 87.2 [86.3, 88.0] | 90.7 [90.0, 91.5] | 70.2 [69.9, 70.4] | 48.2 [45.1, 51.3] | 87.3 [87.0, 87.7] | 64.3 [63.7, 65.0] | 45.6 [45.2, 46.0] | 49.9 [49.4, 50.3] | 94.1 [94.0, 94.3] |
| keep | 81.2 [74.1, 87.3] | 42.5 [39.7, 45.2] | 63.7 [57.4, 69.0] | 93.1 [92.5, 93.7] | 91.2 [90.3, 91.9] | 71.3 [70.9, 71.6] | 54.5 [51.3, 57.7] | 86.1 [85.8, 86.5] | 62.6 [61.9, 63.2] | 70.6 [70.2, 71.0] | 53.3 [52.8, 53.7] | 95.8 [95.7, 96.0] |
| musk | 65.9 [57.9, 73.3] | 42.4 [39.3, 45.4] | 69.8 [64.8, 74.3] | 79.9 [78.9, 80.9] | 88.9 [88.0, 89.7] | 61.7 [61.4, 62.0] | 43.0 [40.0, 46.3] | 73.6 [73.1, 74.1] | 59.5 [58.9, 60.1] | 53.9 [53.5, 54.3] | 38.5 [38.1, 38.9] | 77.6 [77.3, 77.9] |
| plip | 58.3 [49.3, 66.2] | 43.0 [39.4, 46.2] | 54.7 [49.4, 59.7] | 77.5 [76.5, 78.6] | 85.1 [84.0, 86.0] | 50.6 [50.4, 50.9] | 47.2 [44.0, 50.4] | 73.1 [72.6, 73.6] | 52.0 [51.5, 52.6] | 61.4 [61.0, 61.8] | 29.6 [29.2, 29.9] | 71.5 [71.2, 71.8] |
| quilt | 51.1 [42.3, 58.8] | 44.0 [40.3, 47.6] | 69.9 [64.8, 74.8] | 76.1 [75.0, 77.2] | 88.0 [87.1, 88.9] | 49.3 [49.0, 49.5] | 50.0 [46.9, 53.1] | 80.2 [79.7, 80.6] | 57.0 [56.4, 57.5] | 64.1 [63.7, 64.5] | 29.3 [28.9, 29.7] | 79.0 [78.4, 79.3] |
| dinob | 57.5 [49.0, 65.1] | 35.1 [31.6, 38.4] | 66.2 [60.7, 71.1] | 77.2 [76.1, 78.3] | 74.7 [73.6, 75.8] | 46.6 [46.3, 46.8] | 67.2 [64.3, 70.1] | 60.8 [60.3, 61.3] | 48.7 [48.2, 49.3] | 28.6 [28.3, 29.0] | 23.9 [23.5, 24.2] | 65.6 [65.2, 65.9] |
| dinol | 57.5 [49.2, 65.3] | 37.1 [33.7, 40.1] | 62.1 [56.6, 67.0] | 65.7 [64.4, 67.0] | 72.2 [71.0, 73.3] | 46.1 [45.9, 46.3] | 68.9 [65.9, 71.7] | 53.9 [53.3, 54.4] | 53.9 [53.4, 54.5] | 30.8 [30.4, 31.2] | 24.6 [24.2, 24.9] | 62.1 [61.8, 62.5] |
| vitb | 51.1 [42.9, 58.8] | 37.7 [34.4, 40.9] | 53.1 [47.3, 58.6] | 56.5 [55.3, 57.7] | 71.8 [70.7, 72.9] | 47.3 [47.0, 47.5] | 52.5 [49.5, 55.6] | 67.5 [67.0, 68.0] | 49.9 [49.4, 50.5] | 51.9 [51.5, 52.4] | 19.8 [19.5, 20.1] | 62.2 [61.9, 62.5] |
| vitl | 45.7 [38.7, 52.7] | 39.8 [36.3, 43.2] | 38.8 [33.6, 43.7] | 68.7 [67.5, 69.9] | 75.1 [74.0, 76.2] | 50.5 [50.2, 50.7] | 54.0 [50.8, 57.2] | 68.1 [67.6, 68.6] | 47.4 [46.9, 48.0] | 34.1 [33.7, 34.5] | 20.4 [20.1, 20.8] | 65.9 [65.6, 66.2] |
| clipb | 44.5 [36.4, 52.0] | 31.5 [28.3, 34.5] | 50.6 [45.4, 55.9] | 63.4 [62.0, 64.7] | 64.7 [63.6, 65.8] | 38.0 [37.8, 38.2] | 46.8 [43.6, 50.0] | 60.0 [59.4, 60.5] | 50.4 [49.9, 51.0] | 47.0 [46.6, 47.4] | 21.6 [21.2, 21.9] | 39.9 [39.6, 40.1] |
| clipl | 49.3 [41.0, 57.2] | 36.9 [33.4, 40.4] | 47.9 [42.5, 53.2] | 59.5 [58.2, 60.7] | 72.3 [71.2, 73.4] | 49.3 [49.0, 49.5] | 47.5 [44.4, 50.6] | 59.6 [59.0, 60.1] | 45.3 [44.8, 45.8] | 33.4 [33.0, 33.8] | 23.1 [22.8, 23.5] | 62.7 [62.4, 63.1] |

Table S42: Quantitative performance (Balanced accuracy) on 2-shot classification.

| Model | bach | bracs | break-h | ccrcc | crc | esca | mhist | pcam | tcga-crc | tcga-tils | tcga-unif | wilds |
|---|---|---|---|---|---|---|---|---|---|---|---|---|
| hiboub | 75.5 | 55.1 | 76.9 | 83.4 | 91.8 | 83.6 | 65.8 | 92.0 | 63.7 | 84.0 | 59.3 | 88.4 |
| | [69.4, | [52.2, | [71.7, | [82.5, | [91.0, | [83.2, | [63.8, | [91.6, | [62.9, | [83.7, | [58.8, | [88.2, |
| | 81.7] | 58.1] | 81.7] | 84.2] | 92.6] | 83.9] | 67.9] | 92.2] | 64.4] | 84.3] | 59.8] | 88.6] |
| hiboul | 74.7 | 52.7 | 74.0 | 83.6 | 79.9 | 78.0 | 65.2 | 91.4 | 66.5 | 82.6 | 62.6 | 67.0 |
| | [69.0, | [49.4, | [69.1, | [82.7, | [79.0, | [77.7, | [63.2, | [91.1, | [65.7, | [82.2, | [62.1, | [66.8, |
| | 80.4] | 55.8] | 78.9] | 84.5] | 80.8] | 78.4] | 67.2] | 91.7] | 67.2] | 83.0] | 63.1] | 67.2] |
| hopt0 | 59.6 | 44.9 | 71.8 | 89.4 | 88.2 | 84.3 | 67.2 | 84.2 | 70.2 | 87.6 | 72.8 | 95.8 |
| | [51.9, | [41.9, | [66.1, | [88.6, | [87.4, | [83.9, | [65.2, | [83.8, | [69.5, | [87.2, | [72.4, | [95.6, |
| | 67.8] | 47.9] | 77.4] | 90.2] | 89.1] | 84.6] | 69.3] | 84.6] | 70.9] | 87.9] | 73.3] | 95.9] |
| hopt1 | 70.6 | 46.6 | 73.0 | 88.8 | 91.3 | 86.3 | 64.6 | 84.3 | 71.8 | 70.8 | 75.8 | 71.4 |
| | [63.2, | [43.4, | [68.0, | [88.0, | [90.5, | [86.0, | [62.8, | [83.9, | [71.1, | [70.6, | [75.4, | [71.2, |
| | 77.4] | 49.7] | 77.7] | 89.6] | 92.1] | 86.6] | 66.5] | 84.7] | 72.4] | 71.1] | 76.2] | 71.6] |
| midnight | 83.4 | 47.9 | 40.2 | 82.1 | 94.7 | 83.2 | 66.3 | 82.4 | 55.8 | 80.3 | 66.5 | 89.3 |
| | [78.0, | [44.8, | [35.1, | [81.1, | [94.1, | [82.8, | [63.2, | [82.0, | [55.1, | [79.9, | [66.0, | [89.1, |
| | 88.7] | 51.0] | 45.3] | 83.1] | 95.3] | 83.6] | 69.3] | 82.8] | 56.4] | 80.8] | 67.0] | 89.5] |
| phikon | 59.6 | 43.2 | 70.3 | 93.2 | 91.8 | 80.8 | 63.3 | 86.9 | 66.2 | 87.6 | 61.2 | 94.4 |
| | [51.1, | [40.3, | [65.1, | [92.6, | [91.1, | [80.5, | [61.5, | [86.5, | [65.5, | [87.3, | [60.7, | [94.2, |
| | 67.4] | 46.3] | 75.2] | 93.8] | 92.5] | 81.1] | 65.2] | 87.2] | 66.9] | 87.9] | 61.8] | 94.5] |
| phikon2 | 64.8 | 43.9 | 61.3 | 91.0 | 87.6 | 82.8 | 59.6 | 83.6 | 62.2 | 69.5 | 66.9 | 92.5 |
| | [57.1, | [40.7, | [55.5, | [90.3, | [86.8, | [82.5, | [58.0, | [83.2, | [61.5, | [69.2, | [66.4, | [92.3, |
| | 72.3] | 47.1] | 67.1] | 91.7] | 88.3] | 83.2] | 61.3] | 84.0] | 62.9] | 69.7] | 67.4] | 92.7] |
| uni | 71.6 | 52.5 | 80.5 | 92.0 | 91.4 | 85.0 | 68.9 | 91.4 | 66.8 | 87.1 | 67.9 | 96.6 |
| | [64.4, | [49.3, | [75.6, | [91.3, | [90.6, | [84.6, | [66.8, | [91.1, | [66.0, | [86.8, | [67.4, | [96.5, |
| | 78.4] | 55.8] | 85.3] | 92.7] | 92.2] | 85.3] | 71.1] | 91.7] | 67.5] | 87.4] | 68.3] | 96.7] |
| uni2h | 83.6 | 55.5 | 69.9 | 90.2 | 95.4 | 87.2 | 65.4 | 92.4 | 69.9 | 65.7 | 73.2 | 95.9 |
| | [78.8, | [52.8, | [64.2, | [89.5, | [94.8, | [86.9, | [63.4, | [92.1, | [69.2, | [65.5, | [72.7, | [95.8, |
| | 88.3] | 58.2] | 75.4] | 91.0] | 95.9] | 87.5] | 67.4] | 92.7] | 70.5] | 65.9] | 73.6] | 96.1] |
| virchow | 52.3 | 41.6 | 54.1 | 87.5 | 89.5 | 85.0 | 65.8 | 88.2 | 64.0 | 74.4 | 55.3 | 94.7 |
| | [44.6, | [38.4, | [48.0, | [86.8, | [88.7, | [84.6, | [63.6, | [87.8, | [63.3, | [74.1, | [54.8, | [94.5, |
| | 59.6] | 44.8] | 60.3] | 88.3] | 90.3] | 85.4] | 68.0] | 88.5] | 64.7] | 74.7] | 55.8] | 94.8] |
| virchow2 | 81.2 | 54.9 | 56.5 | 82.0 | 89.9 | 82.7 | 64.5 | 79.1 | 63.0 | 86.4 | 65.0 | 95.4 |
| | [75.0, | [52.1, | [50.9, | [81.0, | [89.2, | [82.3, | [62.7, | [78.6, | [62.3, | [86.1, | [64.6, | [95.3, |
| | 86.7] | 57.8] | 62.2] | 83.0] | 90.8] | 83.1] | 66.5] | 79.5] | 63.7] | 86.8] | 65.5] | 95.6] |
| conch | 84.7 | 54.6 | 63.4 | 87.9 | 92.2 | 84.8 | 65.2 | 84.1 | 64.5 | 80.9 | 56.9 | 95.7 |
| | [80.0, | [52.0, | [58.3, | [87.0, | [91.4, | [84.4, | [63.0, | [83.7, | [63.8, | [80.5, | [56.4, | [95.5, |
| | 89.2] | 57.1] | 68.6] | 88.8] | 93.0] | 85.2] | 67.4] | 84.5] | 65.2] | 81.3] | 57.4] | 95.8] |
| titan | 84.0 | 57.8 | 68.5 | 87.4 | 91.1 | 84.9 | 67.1 | 87.3 | 66.9 | 80.7 | 59.5 | 95.0 |
| | [79.5, | [55.0, | [62.8, | [86.5, | [90.3, | [84.5, | [64.9, | [86.9, | [66.2, | [80.4, | [59.0, | [94.8, |
| | 88.5] | 60.5] | 74.0] | 88.2] | 91.8] | 85.3] | 69.3] | 87.6] | 67.6] | 81.0] | 59.9] | 95.1] |
| keep | 85.0 | 54.8 | 63.4 | 93.5 | 91.0 | 86.9 | 64.1 | 88.3 | 63.7 | 87.2 | 63.1 | 96.1 |
| | [78.6, | [52.5, | [58.3, | [92.9, | [90.2, | [86.5, | [61.3, | [88.0, | [63.0, | [86.8, | [62.6, | [96.0, |
| | 90.7] | 57.1] | 68.8] | 94.1] | 91.8] | 87.3] | 66.8] | 88.7] | 64.4] | 87.5] | 63.5] | 96.2] |
| musk | 67.8 | 52.5 | 72.8 | 82.6 | 90.2 | 81.7 | 62.1 | 84.5 | 63.7 | 83.2 | 50.0 | 88.5 |
| | [60.7, | [49.9, | [68.3, | [81.6, | [89.4, | [81.2, | [60.4, | [84.1, | [63.0, | [82.9, | [49.5, | [88.3, |
| | 74.5] | 55.1] | 77.3] | 83.4] | 90.9] | 82.1] | 63.9] | 84.9] | 64.4] | 83.5] | 50.5] | 88.7] |
| plip | 55.3 | 49.2 | 55.5 | 79.4 | 85.1 | 67.4 | 65.2 | 82.4 | 59.8 | 79.5 | 40.5 | 81.9 |
| | [47.4, | [46.2, | [50.8, | [78.4, | [84.2, | [66.9, | [63.2, | [81.9, | [59.0, | [79.1, | [40.0, | [81.6, |
| | 63.0] | 52.1] | 60.4] | 80.4] | 86.0] | 67.9] | 67.3] | 82.8] | 60.5] | 79.9] | 41.1] | 82.1] |
| quilt | 56.1 | 51.2 | 70.9 | 80.1 | 89.2 | 67.9 | 65.3 | 83.5 | 61.7 | 81.9 | 40.0 | 85.6 |
| | [48.5, | [48.4, | [65.6, | [79.1, | [88.5, | [67.4, | [63.2, | [83.1, | [61.0, | [81.5, | [39.4, | [85.4, |
| | 63.3] | 53.9] | 76.2] | 81.0] | 90.0] | 68.3] | 67.5] | 83.9] | 62.4] | 82.3] | 40.5] | 85.8] |
| dinob | 61.1 | 41.5 | 68.0 | 77.2 | 78.0 | 66.4 | 73.1 | 65.8 | 57.0 | 65.2 | 32.0 | 75.0 |
| | [53.2, | [38.5, | [62.2, | [76.1, | [77.0, | [65.9, | [70.7, | [65.4, | [56.2, | [65.0, | [31.4, | [74.7, |
| | 68.9] | 44.7] | 73.4] | 78.3] | 78.9] | 66.9] | 75.6] | 66.2] | 57.7] | 65.5] | 32.5] | 75.3] |
| dinol | 63.5 | 44.6 | 63.2 | 68.9 | 74.9 | 67.7 | 75.3 | 62.2 | 60.3 | 62.8 | 32.6 | 75.9 |
| | [55.7, | [41.6, | [57.5, | [67.8, | [73.9, | [67.2, | [72.6, | [61.8, | [59.6, | [62.4, | [32.1, | [75.7, |
| | 71.0] | 47.7] | 68.6] | 70.1] | 75.9] | 68.2] | 77.7] | 62.5] | 61.0] | 63.0] | 33.2] | 76.2] |
| vitb | 59.5 | 46.0 | 58.9 | 60.7 | 74.8 | 63.8 | 66.3 | 72.4 | 56.2 | 73.8 | 27.6 | 73.9 |
| | [51.9, | [42.9, | [53.2, | [59.6, | [73.9, | [63.3, | [64.2, | [71.9, | [55.5, | [73.4, | [27.1, | [73.6, |
| | 66.9] | 49.0] | 64.6] | 61.9] | 75.8] | 64.4] | 68.3] | 72.8] | 56.9] | 74.2] | 28.1] | 74.2] |
| vitl | 55.0 | 46.8 | 47.5 | 69.0 | 77.7 | 64.8 | 68.0 | 69.9 | 57.7 | 68.4 | 28.9 | 72.0 |
| | [47.7, | [43.9, | [41.8, | [67.8, | [76.7, | [64.2, | [65.6, | [69.4, | [57.0, | [68.0, | [28.4, | [71.7, |
| | 61.7] | 49.7] | 52.9] | 70.2] | 78.6] | 65.3] | 70.2] | 70.4] | 58.4] | 68.7] | 29.4] | 72.2] |
| clipb | 48.7 | 40.6 | 55.6 | 63.7 | 71.3 | 53.1 | 64.2 | 69.5 | 55.1 | 71.4 | 29.5 | 55.8 |
| | [41.3, | [37.8, | [49.8, | [62.5, | [70.3, | [52.6, | [62.3, | [69.1, | [54.4, | [71.1, | [29.0, | [55.7, |
| | 56.0] | 43.5] | 61.1] | 64.8] | 72.2] | 53.6] | 66.1] | 69.9] | 55.8] | 71.8] | 30.1] | 56.0] |
| clipl | 52.9 | 43.5 | 49.5 | 65.4 | 77.6 | 65.8 | 65.2 | 73.8 | 56.9 | 69.0 | 32.4 | 81.2 |
| | [45.6, | [40.6, | [44.4, | [64.3, | [76.6, | [65.3, | [63.4, | [73.4, | [56.2, | [68.7, | [31.8, | [81.0, |
| | 60.3] | 46.5] | 55.3] | 66.5] | 78.6] | 66.2] | 67.2] | 74.2] | 57.6] | 69.3] | 32.9] | 81.5] |

Table S43: Quantitative performance (F1-score) on 2-shot classification.

| Model | bach | bracs | break-h | ccrcc | crc | esca | mhist | pcam | tcga-crc | tcga-tils | tcga-unif | wilds |
|---|---|---|---|---|---|---|---|---|---|---|---|---|
| hiboub | 69.7 [61.6, 77.1] | 49.4 [45.6, 53.1] | 75.8 [71.0, 79.9] | 81.3 [80.3, 82.3] | 91.9 [91.1, 92.6] | 65.4 [65.1, 65.8] | 56.2 [53.1, 59.3] | 92.0 [91.6, 92.2] | 59.0 [58.4, 59.6] | 73.1 [72.6, 73.5] | 50.1 [49.6, 50.5] | 88.3 [88.1, 88.5] |
| hiboul | 68.3 [60.5, 75.6] | 48.3 [44.6, 51.8] | 73.8 [69.6, 77.6] | 81.6 [80.6, 82.6] | 81.1 [80.2, 81.9] | 64.3 [64.0, 64.7] | 55.1 [51.9, 58.2] | 91.4 [91.1, 91.7] | 61.0 [60.5, 61.6] | 73.9 [73.6, 74.4] | 53.0 [52.6, 53.5] | 63.2 [62.9, 63.5] |
| hopt0 | 57.8 [49.1, 65.8] | 39.1 [35.4, 42.7] | 68.1 [62.4, 73.0] | 90.0 [89.2, 90.7] | 88.9 [88.0, 89.8] | 71.1 [70.8, 71.3] | 58.7 [55.7, 61.9] | 84.2 [83.8, 84.6] | 64.7 [64.1, 65.3] | 83.2 [82.8, 83.6] | 63.9 [63.5, 64.4] | 95.8 [95.6, 95.9] |
| hopt1 | 66.9 [58.3, 74.4] | 41.7 [38.2, 45.1] | 63.4 [58.0, 68.3] | 89.5 [88.8, 90.2] | 92.0 [91.2, 92.7] | 74.1 [73.8, 74.4] | 54.2 [51.0, 57.3] | 84.2 [83.8, 84.6] | 64.7 [64.1, 65.3] | 52.2 [51.8, 52.6] | 67.9 [67.5, 68.4] | 69.0 [68.7, 69.3] |
| midnight | 79.6 [72.2, 86.2] | 42.7 [39.4, 46.0] | 40.7 [35.8, 45.5] | 83.9 [82.9, 84.9] | 94.7 [94.0, 95.3] | 72.5 [72.2, 72.9] | 64.2 [61.1, 67.2] | 82.2 [81.8, 82.6] | 55.0 [54.3, 55.6] | 74.2 [73.7, 74.6] | 61.6 [61.1, 62.1] | 89.2 [89.0, 89.4] |
| phikon | 57.5 [49.1, 65.0] | 37.6 [34.2, 41.2] | 65.7 [59.9, 70.8] | 92.5 [91.9, 93.2] | 92.1 [91.3, 92.8] | 62.3 [61.9, 62.6] | 52.1 [48.8, 55.4] | 86.8 [86.4, 87.1] | 58.5 [57.9, 59.0] | 79.9 [79.5, 80.3] | 52.2 [51.8, 52.7] | 94.4 [94.2, 94.5] |
| phikon2 | 59.5 [51.1, 67.1] | 38.2 [34.7, 41.6] | 58.3 [52.7, 63.5] | 90.6 [89.8, 91.3] | 88.1 [87.2, 88.9] | 64.2 [63.9, 64.5] | 46.0 [42.9, 49.1] | 83.5 [83.1, 83.9] | 54.6 [54.0, 55.1] | 50.4 [50.0, 50.8] | 57.2 [56.8, 57.7] | 92.5 [92.3, 92.7] |
| uni | 67.7 [59.6, 74.9] | 47.9 [44.0, 51.5] | 78.0 [73.1, 82.2] | 92.0 [91.3, 92.7] | 91.5 [90.7, 92.2] | 68.4 [68.1, 68.7] | 61.0 [57.9, 64.1] | 91.4 [91.1, 91.7] | 60.9 [60.3, 61.5] | 78.8 [78.4, 79.2] | 58.2 [57.8, 58.7] | 96.6 [96.5, 96.7] |
| uni2h | 78.4 [71.0, 85.0] | 49.1 [45.5, 52.5] | 69.4 [64.0, 74.1] | 90.8 [90.0, 91.5] | 95.0 [94.4, 95.6] | 73.9 [73.6, 74.2] | 55.9 [52.8, 59.1] | 92.4 [92.1, 92.7] | 60.6 [60.0, 61.1] | 44.2 [43.8, 44.6] | 65.7 [65.2, 66.1] | 95.9 [95.8, 96.1] |
| virchow | 48.0 [39.8, 55.5] | 35.9 [32.3, 39.5] | 51.3 [45.7, 56.5] | 85.5 [84.6, 86.4] | 89.7 [88.8, 90.6] | 72.2 [71.8, 72.5] | 57.4 [54.2, 60.4] | 88.1 [87.8, 88.4] | 59.3 [58.7, 59.9] | 58.6 [58.2, 59.0] | 47.3 [46.8, 47.8] | 94.7 [94.5, 94.8] |
| virchow2 | 77.3 [69.9, 83.7] | 49.1 [45.8, 52.5] | 57.1 [51.6, 62.0] | 83.5 [82.6, 84.5] | 90.3 [89.6, 91.1] | 69.0 [68.7, 69.4] | 54.1 [51.0, 57.3] | 79.0 [78.6, 79.5] | 59.7 [59.1, 60.3] | 79.5 [79.1, 79.9] | 56.0 [55.5, 56.4] | 95.4 [95.3, 95.6] |
| conch | 80.0 [73.2, 86.1] | 47.1 [43.4, 50.6] | 64.0 [58.0, 69.4] | 88.6 [87.8, 89.4] | 91.8 [91.0, 92.5] | 68.8 [68.4, 69.1] | 56.3 [53.1, 59.5] | 84.0 [83.6, 84.4] | 62.7 [62.1, 63.4] | 72.2 [71.8, 72.7] | 46.9 [46.5, 47.3] | 95.7 [95.5, 95.8] |
| titan | 78.4 [71.6, 85.0] | 51.9 [48.0, 55.4] | 67.4 [61.4, 72.6] | 87.4 [86.6, 88.2] | 90.9 [90.1, 91.6] | 71.4 [71.1, 71.7] | 59.1 [56.0, 62.2] | 87.3 [86.9, 87.6] | 63.7 [63.1, 64.3] | 67.8 [67.4, 68.2] | 51.2 [50.7, 51.6] | 95.0 [94.8, 95.1] |
| keep | 83.2 [76.0, 89.4] | 45.6 [42.5, 48.7] | 64.5 [58.7, 69.7] | 93.3 [92.7, 93.9] | 91.4 [90.6, 92.2] | 72.8 [72.5, 73.2] | 58.8 [55.8, 62.0] | 88.3 [88.0, 88.7] | 61.5 [60.8, 62.1] | 80.5 [80.1, 80.8] | 54.4 [53.9, 54.8] | 96.1 [96.0, 96.2] |
| musk | 62.9 [54.6, 70.3] | 45.4 [42.0, 48.6] | 71.9 [67.4, 76.1] | 81.4 [80.4, 82.4] | 89.1 [88.2, 89.9] | 64.0 [63.7, 64.3] | 50.1 [47.0, 53.3] | 84.5 [84.1, 84.9] | 59.5 [58.9, 60.1] | 72.5 [72.1, 72.9] | 39.6 [39.2, 40.0] | 88.5 [88.3, 88.7] |
| plip | 53.6 [44.7, 61.5] | 43.2 [39.4, 46.6] | 52.7 [48.1, 57.3] | 77.8 [76.8, 78.9] | 85.4 [84.3, 86.3] | 51.5 [51.2, 51.7] | 55.4 [52.3, 58.5] | 82.3 [81.9, 82.7] | 55.4 [54.8, 55.9] | 70.2 [69.8, 70.6] | 30.8 [30.4, 31.1] | 81.6 [81.3, 81.9] |
| quilt | 51.2 [42.6, 58.8] | 44.4 [40.8, 47.9] | 71.4 [66.0, 76.2] | 77.9 [76.8, 78.9] | 88.4 [87.5, 89.2] | 51.6 [51.3, 51.8] | 56.4 [53.2, 59.6] | 83.4 [83.0, 83.8] | 56.6 [56.1, 57.2] | 72.7 [72.3, 73.1] | 30.2 [29.8, 30.6] | 85.5 [85.2, 85.7] |
| dinob | 55.4 [47.0, 63.4] | 34.9 [31.5, 38.2] | 65.0 [59.4, 69.8] | 78.1 [77.0, 79.1] | 76.5 [75.4, 77.5] | 47.9 [47.7, 48.2] | 68.4 [65.4, 71.2] | 62.5 [62.0, 63.0] | 52.1 [51.5, 52.6] | 44.6 [44.2, 45.0] | 23.8 [23.4, 24.1] | 74.0 [73.7, 74.3] |
| dinol | 57.6 [49.1, 65.3] | 38.8 [35.2, 42.0] | 58.2 [52.4, 63.4] | 69.2 [67.9, 70.4] | 74.1 [73.0, 75.1] | 48.5 [48.2, 48.7] | 71.7 [68.8, 74.5] | 56.4 [55.9, 57.0] | 55.6 [55.0, 56.1] | 41.9 [41.5, 42.3] | 24.6 [24.2, 25.0] | 74.8 [74.5, 75.1] |
| vitb | 53.9 [45.3, 61.8] | 39.3 [35.9, 42.8] | 54.6 [49.0, 59.8] | 56.8 [55.5, 58.1] | 73.8 [72.7, 74.9] | 48.2 [47.9, 48.5] | 57.2 [54.1, 60.3] | 72.1 [71.7, 72.6] | 52.3 [51.8, 52.8] | 60.0 [59.6, 60.5] | 20.3 [19.9, 20.6] | 73.1 [72.8, 73.4] |
| vitl | 46.6 [38.4, 53.9] | 39.8 [36.3, 43.2] | 39.0 [33.9, 44.0] | 68.5 [67.3, 69.7] | 76.6 [75.6, 77.6] | 51.4 [51.1, 51.7] | 60.8 [57.7, 63.8] | 69.9 [69.4, 70.4] | 51.2 [50.7, 51.7] | 51.1 [50.7, 51.5] | 20.9 [20.5, 21.2] | 70.9 [70.6, 71.2] |
| clipb | 43.0 [35.4, 50.2] | 33.6 [30.2, 37.0] | 53.8 [48.4, 59.0] | 64.8 [63.4, 66.1] | 67.7 [66.6, 68.8] | 36.8 [36.6, 37.0] | 53.4 [50.2, 56.7] | 67.3 [66.7, 67.8] | 53.8 [53.3, 54.4] | 56.8 [56.4, 57.2] | 21.6 [21.2, 21.9] | 45.5 [45.1, 45.8] |
| clipl | 46.9 [38.3, 54.8] | 37.2 [33.8, 40.6] | 46.0 [40.7, 51.3] | 62.3 [61.0, 63.5] | 75.1 [74.0, 76.1] | 50.6 [50.4, 50.9] | 55.2 [52.1, 58.4] | 72.9 [72.4, 73.3] | 49.4 [48.8, 49.9] | 52.1 [51.7, 52.5] | 23.5 [23.2, 23.9] | 80.9 [80.6, 81.1] |

Table S44: Quantitative performance (Balanced accuracy) on 4-shot classification.

| Model | bach | bracs | break-h | ccrcc | crc | esca | mhist | pcam | tcga-crc | tcga-tils | tcga-unif | wilds |
|---|---|---|---|---|---|---|---|---|---|---|---|---|
| hiboub | 76.1 [69.6, 82.0] | 55.9 [52.7, 58.9] | 75.7 [70.4, 80.7] | 83.4 [82.5, 84.3] | 91.7 [90.9, 92.4] | 84.3 [83.9, 84.6] | 68.5 [66.4, 70.7] | 92.2 [91.9, 92.5] | 64.6 [63.9, 65.3] | 86.9 [86.5, 87.2] | 59.7 [59.2, 60.2] | 93.7 [93.5, 93.9] |
| hiboul | 74.7 [68.5, 80.7] | 53.3 [49.9, 56.6] | 71.0 [65.6, 76.2] | 83.7 [82.8, 84.7] | 81.8 [80.9, 82.7] | 78.6 [78.2, 79.0] | 65.7 [63.5, 67.9] | 91.4 [91.1, 91.7] | 66.3 [65.6, 67.1] | 84.4 [84.1, 84.8] | 63.2 [62.8, 63.7] | 71.0 [70.8, 71.2] |
| hopt0 | 60.4 [52.5, 68.4] | 45.8 [42.6, 49.0] | 71.9 [66.2, 77.7] | 86.4 [85.6, 87.3] | 89.4 [88.6, 90.3] | 85.1 [84.8, 85.4] | 69.2 [67.0, 71.5] | 86.1 [85.7, 86.4] | 69.7 [69.0, 70.3] | 88.2 [87.9, 88.6] | 73.4 [73.0, 73.9] | 95.6 [95.4, 95.7] |
| hopt1 | 67.6 [59.8, 74.8] | 48.4 [44.8, 51.8] | 74.7 [69.8, 79.6] | 87.4 [86.6, 88.2] | 92.0 [91.2, 92.7] | 86.8 [86.5, 87.1] | 68.9 [66.7, 71.1] | 88.2 [87.9, 88.6] | 71.2 [70.5, 71.9] | 84.6 [84.3, 84.9] | 76.2 [75.8, 76.6] | 74.9 [74.7, 75.2] |
| midnight | 85.5 [80.3, 90.6] | 49.5 [46.2, 52.7] | 40.6 [35.5, 45.7] | 78.8 [77.8, 79.8] | 94.8 [94.2, 95.4] | 83.5 [83.1, 83.9] | 65.6 [62.4, 68.6] | 82.4 [82.0, 82.8] | 56.6 [55.8, 57.3] | 81.0 [80.6, 81.4] | 66.8 [66.3, 67.3] | 89.7 [89.6, 89.9] |
| phikon | 55.5 [46.7, 63.6] | 45.4 [42.2, 48.6] | 71.8 [66.3, 77.1] | 93.0 [92.4, 93.6] | 92.4 [91.7, 93.1] | 81.7 [81.3, 82.0] | 66.0 [64.1, 68.1] | 87.5 [87.2, 87.8] | 67.2 [66.6, 67.9] | 89.8 [89.5, 90.1] | 62.1 [61.6, 62.6] | 94.7 [94.6, 94.9] |
| phikon2 | 63.9 [55.5, 71.4] | 45.4 [42.1, 48.6] | 61.9 [56.1, 67.4] | 91.1 [90.4, 91.8] | 88.5 [87.8, 89.2] | 83.1 [82.8, 83.5] | 62.5 [60.6, 64.3] | 83.2 [82.8, 83.6] | 62.8 [62.1, 63.6] | 85.9 [85.6, 86.2] | 67.4 [67.0, 67.9] | 92.1 [91.9, 92.2] |
| uni | 71.0 [63.5, 78.0] | 53.6 [50.1, 57.0] | 78.1 [73.3, 83.0] | 91.1 [90.3, 91.8] | 92.9 [92.2, 93.6] | 85.5 [85.2, 85.9] | 72.2 [70.1, 74.4] | 91.5 [91.2, 91.8] | 66.8 [66.0, 67.4] | 89.5 [89.2, 89.9] | 68.2 [67.8, 68.7] | 96.7 [96.5, 96.8] |
| uni2h | 85.4 [80.5, 90.2] | 56.3 [53.6, 59.2] | 68.8 [63.1, 74.3] | 90.3 [89.6, 91.1] | 95.6 [95.1, 96.2] | 87.6 [87.3, 88.0] | 69.4 [67.2, 71.6] | 92.6 [92.3, 92.9] | 69.1 [68.4, 69.8] | 81.4 [81.2, 81.7] | 73.9 [73.5, 74.4] | 96.1 [96.0, 96.2] |
| virchow | 53.3 [45.4, 61.0] | 42.6 [39.3, 45.9] | 52.2 [46.0, 58.5] | 89.1 [88.4, 89.8] | 89.6 [88.8, 90.4] | 85.4 [85.0, 85.8] | 66.0 [63.6, 68.5] | 88.0 [87.7, 88.3] | 63.6 [62.8, 64.3] | 80.2 [79.9, 80.6] | 56.8 [56.3, 57.3] | 96.4 [96.3, 96.6] |
| virchow2 | 83.3 [77.4, 88.5] | 54.9 [51.9, 58.0] | 57.9 [52.2, 63.4] | 80.0 [79.0, 81.0] | 90.4 [89.6, 91.2] | 83.0 [82.6, 83.4] | 72.0 [69.8, 74.3] | 80.5 [80.1, 80.9] | 62.8 [62.1, 63.5] | 86.1 [85.8, 86.5] | 64.8 [64.3, 65.2] | 95.5 [95.4, 95.7] |
| conch | 83.0 [77.1, 88.8] | 55.5 [52.8, 58.2] | 62.8 [57.7, 67.9] | 87.5 [86.6, 88.4] | 92.2 [91.5, 93.0] | 84.6 [84.2, 85.0] | 66.1 [63.8, 68.5] | 84.2 [83.8, 84.6] | 65.9 [65.2, 66.7] | 82.2 [81.8, 82.6] | 57.7 [57.2, 58.2] | 95.7 [95.6, 95.9] |
| titan | 84.7 [79.9, 89.5] | 58.5 [55.7, 61.3] | 68.6 [63.0, 74.1] | 86.6 [85.7, 87.4] | 91.4 [90.6, 92.1] | 85.0 [84.6, 85.5] | 68.4 [65.9, 70.9] | 86.9 [86.5, 87.2] | 66.5 [65.8, 67.2] | 84.0 [83.7, 84.3] | 60.5 [60.0, 61.0] | 94.6 [94.5, 94.8] |
| keep | 86.5 [80.2, 91.9] | 55.5 [53.0, 58.0] | 66.6 [61.2, 71.9] | 93.2 [92.5, 93.8] | 91.4 [90.6, 92.2] | 87.2 [86.8, 87.6] | 65.1 [62.3, 68.1] | 87.9 [87.5, 88.2] | 64.5 [63.7, 64.5] | 87.7 [87.3, 88.0] | 64.1 [63.6, 64.5] | 95.9 [95.7, 96.0] |
| musk | 65.3 [58.1, 72.7] | 54.5 [51.8, 57.2] | 73.0 [68.3, 77.6] | 82.4 [81.4, 83.3] | 90.5 [89.7, 91.2] | 82.0 [81.6, 82.4] | 65.4 [63.4, 67.4] | 84.7 [84.3, 85.1] | 64.0 [63.3, 64.7] | 84.8 [84.4, 85.1] | 51.0 [50.5, 51.6] | 92.0 [91.8, 92.1] |
| plip | 54.0 [46.2, 61.7] | 49.2 [46.1, 52.3] | 57.5 [52.0, 62.8] | 79.4 [78.4, 80.4] | 85.5 [84.6, 86.4] | 67.4 [66.9, 67.9] | 69.1 [67.1, 71.3] | 83.1 [82.7, 83.5] | 60.1 [59.4, 60.8] | 79.3 [78.9, 79.7] | 40.9 [40.4, 41.5] | 85.7 [85.4, 85.9] |
| quilt | 53.4 [45.8, 60.8] | 52.1 [49.0, 55.1] | 68.7 [63.1, 74.3] | 80.6 [79.7, 81.6] | 89.6 [88.8, 90.3] | 68.0 [67.5, 68.4] | 68.4 [66.3, 70.6] | 83.6 [83.2, 84.0] | 61.3 [60.6, 62.0] | 81.9 [81.5, 82.3] | 40.3 [39.8, 40.8] | 89.5 [89.3, 89.7] |
| dinob | 59.2 [51.3, 66.9] | 41.3 [38.1, 44.5] | 68.0 [62.3, 73.5] | 75.1 [74.0, 76.3] | 79.5 [78.5, 80.5] | 67.2 [66.7, 67.8] | 74.6 [72.2, 77.1] | 70.2 [69.7, 70.6] | 57.5 [56.8, 58.3] | 70.2 [69.8, 70.5] | 33.0 [32.4, 33.5] | 81.8 [81.6, 82.1] |
| dinol | 60.2 [52.3, 67.8] | 45.2 [42.0, 48.3] | 62.7 [56.9, 68.1] | 68.2 [67.0, 69.3] | 76.3 [75.3, 77.4] | 67.7 [67.2, 68.2] | 76.0 [73.4, 78.6] | 65.3 [64.9, 65.6] | 59.9 [59.2, 60.7] | 64.9 [64.6, 65.3] | 33.2 [32.7, 33.7] | 84.3 [84.0, 84.5] |
| vitb | 59.3 [51.4, 66.6] | 45.6 [42.4, 48.7] | 59.0 [53.3, 64.7] | 60.0 [58.9, 61.2] | 76.0 [75.0, 76.9] | 64.3 [63.7, 64.8] | 69.7 [67.4, 72.1] | 73.5 [73.0, 74.0] | 56.1 [55.4, 56.9] | 76.1 [75.7, 76.5] | 28.1 [27.6, 28.6] | 78.4 [78.1, 78.7] |
| vitl | 53.1 [45.6, 60.3] | 48.1 [45.1, 51.1] | 43.2 [37.5, 48.7] | 68.7 [67.5, 69.9] | 79.1 [78.1, 80.0] | 65.1 [64.6, 65.7] | 69.7 [67.0, 72.1] | 70.6 [70.1, 71.1] | 57.4 [56.6, 58.1] | 72.0 [71.6, 72.4] | 30.3 [29.7, 30.8] | 74.0 [73.7, 74.2] |
| clipb | 47.3 [40.0, 54.9] | 40.8 [37.9, 43.8] | 53.5 [47.9, 58.9] | 64.1 [63.0, 65.2] | 72.0 [71.0, 72.9] | 52.9 [52.4, 53.4] | 68.2 [66.2, 70.3] | 71.2 [70.8, 71.7] | 56.9 [56.1, 57.6] | 72.2 [71.8, 72.6] | 30.0 [29.5, 30.6] | 62.9 [62.6, 63.1] |
| clipl | 51.5 [43.8, 59.3] | 44.1 [41.1, 47.2] | 51.3 [45.9, 56.9] | 64.6 [63.4, 65.7] | 78.1 [77.2, 79.1] | 66.0 [65.5, 66.5] | 68.4 [66.3, 70.5] | 76.5 [76.1, 77.0] | 57.0 [56.3, 57.7] | 71.3 [70.9, 71.7] | 33.0 [32.4, 33.6] | 89.8 [89.5, 89.9] |

Table S45: Quantitative performance (F1-score) on 4-shot classification.

| Model | bach | bracs | break-h | ccrcc | crc | esca | mhist | pcam | tcga-crc | tcga-tils | tcga-unif | wilds |
|---|---|---|---|---|---|---|---|---|---|---|---|---|
| hiboub | 70.4 | 51.0 | 75.4 | 82.7 | 91.8 | 67.9 | 60.4 | 92.2 | 58.7 | 79.6 | 50.7 | 93.7 |
|  | [62.2, | [47.1, | [70.5, | [81.7, | [91.0, | [67.5, | [57.4, | [91.9, | [58.1, | [79.3, | [50.2, | [93.5, |
|  | 77.8] | 54.6] | 79.6] | 83.6] | 92.6] | 68.3] | 63.6] | 92.5] | 59.3] | 80.0] | 51.1] | 93.9] |
| hiboul | 69.2 | 50.1 | 71.6 | 82.5 | 82.7 | 66.0 | 57.2 | 91.4 | 59.4 | 79.2 | 54.3 | 68.6 |
|  | [61.4, | [46.4, | [66.7, | [81.5, | [81.8, | [65.7, | [54.2, | [91.1, | [58.8, | [78.9, | [53.8, | [68.3, |
|  | 76.4] | 53.7] | 76.0] | 83.5] | 83.5] | 66.3] | 60.3] | 91.7] | 59.9] | 79.6] | 54.8] | 68.9] |
| hopt0 | 58.5 | 41.2 | 69.6 | 87.5 | 90.2 | 72.8 | 62.4 | 86.1 | 62.5 | 86.9 | 65.0 | 95.6 |
|  | [49.8, | [37.5, | [64.2, | [86.7, | [89.3, | [72.6, | [59.3, | [85.7, | [61.9, | [86.6, | [64.5, | [95.4, |
|  | 66.4] | 45.0] | 74.6] | 88.3] | 91.0] | 73.1] | 65.4] | 86.4] | 63.0] | 87.3] | 65.4] | 95.7] |
| hopt1 | 64.2 | 45.5 | 67.7 | 88.4 | 92.5 | 75.4 | 61.3 | 88.2 | 63.4 | 74.0 | 68.7 | 73.5 |
|  | [55.2, | [41.6, | [62.4, | [87.7, | [91.8, | [75.2, | [58.3, | [87.9, | [62.8, | [73.6, | [68.2, | [73.2, |
|  | 71.9] | 49.3] | 72.6] | 89.2] | 93.3] | 75.7] | 64.4] | 88.6] | 64.0] | 74.4] | 69.1] | 73.8] |
| midnight | 82.4 | 46.3 | 41.6 | 80.7 | 94.7 | 74.2 | 63.9 | 82.3 | 54.6 | 75.5 | 61.8 | 89.6 |
|  | [75.3, | [42.6, | [36.5, | [79.6, | [94.0, | [73.9, | [60.7, | [81.8, | [54.0, | [75.1, | [61.3, | [89.4, |
|  | 88.6] | 49.9] | 46.7] | 81.7] | 95.3] | 74.5] | 66.8] | 82.7] | 55.2] | 75.9] | 62.3] | 89.8] |
| phikon | 54.4 | 41.0 | 69.6 | 92.5 | 92.7 | 64.8 | 56.7 | 87.4 | 58.5 | 86.2 | 53.4 | 94.7 |
|  | [45.9, | [37.1, | [64.3, | [91.8, | [92.0, | [64.5, | [53.5, | [87.1, | [57.9, | [85.8, | [53.0, | [94.5, |
|  | 62.0] | 44.7] | 74.3] | 93.1] | 93.4] | 65.2] | 59.9] | 87.8] | 59.1] | 86.5] | 53.8] | 94.8] |
| phikon2 | 59.5 | 40.9 | 59.8 | 90.4 | 88.9 | 65.9 | 51.2 | 83.1 | 56.1 | 76.5 | 58.2 | 92.0 |
|  | [50.7, | [37.0, | [54.2, | [89.7, | [88.1, | [65.6, | [48.1, | [82.7, | [55.6, | [76.1, | [57.7, | [91.9, |
|  | 67.2] | 44.6] | 64.9] | 91.1] | 89.8] | 66.3] | 54.3] | 83.5] | 56.7] | 76.9] | 58.6] | 92.2] |
| uni | 67.7 | 50.0 | 76.9 | 91.4 | 92.8 | 70.6 | 65.6 | 91.5 | 59.2 | 86.0 | 59.1 | 96.7 |
|  | [59.3, | [46.0, | [72.3, | [90.7, | [92.0, | [70.3, | [62.6, | [91.2, | [58.7, | [85.7, | [58.6, | [96.5, |
|  | 74.8] | 53.8] | 81.1] | 92.1] | 93.5] | 70.9] | 68.5] | 91.8] | 59.8] | 86.4] | 59.5] | 96.8] |
| uni2h | 81.3 | 50.3 | 68.6 | 90.9 | 95.3 | 75.5 | 62.0 | 92.6 | 59.8 | 67.8 | 66.9 | 96.1 |
|  | [74.3, | [46.6, | [63.2, | [90.2, | [94.7, | [75.2, | [58.9, | [92.3, | [59.2, | [67.4, | [66.4, | [96.0, |
|  | 87.6] | 53.7] | 73.4] | 91.6] | 95.9] | 75.8] | 65.0] | 92.9] | 60.3] | 68.2] | 67.3] | 96.2] |
| virchow | 50.3 | 37.6 | 49.8 | 87.7 | 89.8 | 73.3 | 59.5 | 88.0 | 57.5 | 68.6 | 49.1 | 96.4 |
|  | [42.1, | [33.9, | [43.9, | [86.9, | [88.9, | [72.9, | [56.3, | [87.6, | [56.9, | [68.2, | [48.7, | [96.3, |
|  | 57.7] | 41.3] | 55.3] | 88.6] | 90.7] | 73.6] | 62.5] | 88.3] | 58.1] | 69.0] | 49.6] | 96.6] |
| virchow2 | 79.9 | 50.3 | 58.8 | 81.8 | 90.8 | 69.8 | 65.5 | 80.4 | 57.6 | 79.7 | 56.4 | 95.5 |
|  | [72.7, | [47.0, | [53.4, | [80.8, | [90.0, | [69.4, | [62.5, | [80.0, | [57.1, | [79.3, | [55.9, | [95.4, |
|  | 86.0] | 53.7] | 63.5] | 82.8] | 91.5] | 70.1] | 68.5] | 80.9] | 58.2] | 80.1] | 56.8] | 95.7] |
| conch | 79.9 | 48.9 | 63.5 | 88.3 | 91.7 | 69.4 | 58.6 | 84.2 | 61.8 | 75.3 | 48.2 | 95.7 |
|  | [72.7, | [45.2, | [57.7, | [87.5, | [91.0, | [69.1, | [55.5, | [83.8, | [61.2, | [74.9, | [47.8, | [95.6, |
|  | 86.4] | 52.7] | 68.6] | 89.2] | 92.5] | 69.7] | 61.7] | 84.6] | 62.4] | 75.7] | 48.7] | 95.9] |
| titan | 80.1 | 53.7 | 67.5 | 86.9 | 91.2 | 72.0 | 62.8 | 86.9 | 61.6 | 74.1 | 52.9 | 94.6 |
|  | [73.3, | [49.8, | [61.5, | [86.1, | [90.4, | [71.7, | [59.7, | [86.5, | [61.0, | [73.7, | [52.4, | [94.4, |
|  | 86.3] | 57.4] | 72.6] | 87.8] | 91.9] | 72.2] | 65.8] | 87.2] | 62.2] | 74.5] | 53.3] | 94.8] |
| keep | 85.3 | 47.3 | 68.0 | 93.2 | 91.8 | 73.8 | 61.0 | 87.9 | 60.0 | 82.7 | 55.9 | 95.9 |
|  | [78.4, | [43.8, | [62.3, | [92.5, | [91.0, | [73.4, | [58.0, | [87.5, | [59.4, | [82.3, | [55.4, | [95.7, |
|  | 91.1] | 50.7] | 73.0] | 93.8] | 92.5] | 74.1] | 64.1] | 88.2] | 60.6] | 83.0] | 56.3] | 96.0] |
| musk | 61.4 | 48.4 | 72.4 | 81.9 | 89.3 | 64.8 | 55.5 | 84.7 | 58.4 | 76.7 | 41.2 | 92.0 |
|  | [53.1, | [44.7, | [67.8, | [80.9, | [88.4, | [64.5, | [52.4, | [84.3, | [57.8, | [76.4, | [40.8, | [91.8, |
|  | 68.9] | 51.9] | 76.7] | 82.9] | 90.1] | 65.1] | 58.6] | 85.1] | 59.0] | 77.1] | 41.6] | 92.1] |
| plip | 51.9 | 44.2 | 56.1 | 78.3 | 85.9 | 51.9 | 61.2 | 83.1 | 53.9 | 71.2 | 31.8 | 85.6 |
|  | [43.0, | [40.4, | [50.9, | [77.3, | [84.9, | [51.6, | [58.2, | [82.7, | [53.3, | [70.8, | [31.4, | [85.4, |
|  | 59.7] | 47.7] | 61.2] | 79.4] | 86.8] | 52.1] | 64.3] | 83.5] | 54.4] | 71.7] | 32.2] | 85.8] |
| quilt | 49.1 | 46.7 | 70.8 | 79.0 | 88.8 | 52.2 | 60.9 | 83.6 | 54.8 | 74.5 | 31.1 | 89.5 |
|  | [40.6, | [42.8, | [65.4, | [78.0, | [88.0, | [51.9, | [57.9, | [83.2, | [54.2, | [74.1, | [30.7, | [89.3, |
|  | 56.8] | 50.4] | 75.7] | 80.0] | 89.6] | 52.5] | 64.0] | 84.0] | 55.3] | 74.9] | 31.4] | 89.7] |
| dinob | 53.9 | 35.6 | 65.5 | 76.4 | 78.5 | 49.5 | 70.7 | 69.1 | 51.5 | 53.5 | 24.7 | 81.8 |
|  | [45.5, | [32.1, | [59.9, | [75.2, | [77.5, | [49.3, | [67.8, | [68.6, | [51.0, | [53.1, | [24.3, | [81.5, |
|  | 61.9] | 39.0] | 70.5] | 77.5] | 79.5] | 49.8] | 73.4] | 69.6] | 52.0] | 53.9] | 25.0] | 82.0] |
| dinol | 55.2 | 40.4 | 58.1 | 68.7 | 76.1 | 49.7 | 73.0 | 61.5 | 54.3 | 46.7 | 25.2 | 84.1 |
|  | [46.8, | [37.0, | [52.5, | [67.5, | [75.0, | [49.4, | [70.1, | [60.9, | [53.7, | [46.3, | [24.9, | [83.8, |
|  | 62.9] | 43.9] | 63.3] | 69.9] | 77.1] | 49.9] | 75.8] | 62.0] | 54.8] | 47.1] | 25.6] | 84.3] |
| vitb | 54.0 | 39.7 | 55.6 | 55.9 | 75.1 | 48.9 | 63.3 | 73.5 | 50.6 | 64.7 | 21.2 | 78.3 |
|  | [45.4, | [36.2, | [50.0, | [54.6, | [74.0, | [48.6, | [60.4, | [73.0, | [50.1, | [64.3, | [20.8, | [78.0, |
|  | 61.8] | 43.2] | 60.7] | 57.1] | 76.2] | 49.1] | 66.3] | 73.9] | 51.2] | 65.2] | 21.5] | 78.6] |
| vitl | 45.5 | 42.0 | 36.5 | 68.7 | 78.3 | 52.0 | 64.5 | 70.6 | 49.5 | 57.9 | 22.2 | 73.4 |
|  | [37.2, | [38.4, | [31.4, | [67.5, | [77.2, | [51.7, | [61.4, | [70.1, | [48.9, | [57.5, | [21.9, | [73.1, |
|  | 52.9] | 45.4] | 41.4] | 69.9] | 79.3] | 52.3] | 67.3] | 71.1] | 50.0] | 58.3] | 22.6] | 73.7] |
| clipb | 42.3 | 34.6 | 52.1 | 65.2 | 68.9 | 37.0 | 59.7 | 69.8 | 52.8 | 58.4 | 22.4 | 57.8 |
|  | [34.6, | [31.2, | [46.8, | [63.9, | [67.9, | [36.8, | [56.7, | [69.3, | [52.2, | [58.0, | [22.1, | [57.4, |
|  | 49.3] | 38.0] | 57.1] | 66.5] | 69.9] | 37.2] | 62.8] | 70.3] | 53.3] | 58.8] | 22.8] | 58.1] |
| clipl | 47.5 | 38.1 | 48.1 | 61.6 | 75.9 | 50.8 | 60.2 | 76.2 | 49.2 | 57.3 | 24.2 | 89.7 |
|  | [39.1, | [34.6, | [42.6, | [60.3, | [74.8, | [50.5, | [57.1, | [75.7, | [48.6, | [56.9, | [23.9, | [89.5, |
|  | 55.2] | 41.7] | 53.4] | 62.8] | 77.0] | 51.1] | 63.3] | 76.7] | 49.7] | 57.7] | 24.6] | 89.9] |

Table S46: Quantitative performance (Balanced accuracy) on 8-shot classification.

| Model | bach | bracs | break-h | ccrcc | crc | esca | mhist | pcam | tcga-crc | tcga-tils | tcga-unif | wilds |
|---|---|---|---|---|---|---|---|---|---|---|---|---|
| | 75.5 | 57.3 | 77.1 | 82.8 | 92.2 | 84.3 | 69.4 | 92.1 | 65.9 | 87.1 | 60.5 | 94.9 |
| hiboub | [68.8, | [54.1, | [71.8, | [81.9, | [91.4, | [83.9, | [67.3, | [91.8, | [65.2, | [86.7, | [60.0, | [94.7, |
| | 82.0] | 60.5] | 82.1] | 83.7] | 92.9] | 84.6] | 71.6] | 92.4] | 66.6] | 87.5] | 61.0] | 95.0] |
| | 74.1 | 52.6 | 71.2 | 82.1 | 82.3 | 78.3 | 65.9 | 91.2 | 67.1 | 84.1 | 64.0 | 72.4 |
| hiboul | [67.3, | [49.2, | [65.8, | [81.2, | [81.3, | [77.9, | [63.6, | [90.9, | [66.4, | [83.7, | [63.5, | [72.2, |
| | 80.5] | 56.0] | 76.2] | 83.1] | 83.1] | 78.7] | 68.2] | 91.5] | 67.8] | 84.5] | 64.5] | 72.7] |
| | 58.6 | 46.3 | 72.0 | 83.3 | 90.2 | 85.4 | 70.2 | 85.9 | 70.6 | 87.6 | 73.8 | 94.2 |
| hopt0 | [50.7, | [43.0, | [66.3, | [82.4, | [89.4, | [85.0, | [68.0, | [85.5, | [69.9, | [87.2, | [73.4, | [94.1, |
| | 66.3] | 49.8] | 77.8] | 84.2] | 91.0] | 85.7] | 72.7] | 86.3] | 71.3] | 88.0] | 74.2] | 94.4] |
| | 68.4 | 49.5 | 74.3 | 86.7 | 92.5 | 87.0 | 71.3 | 88.9 | 72.0 | 88.1 | 76.6 | 82.3 |
| hopt1 | [60.5, | [45.8, | [69.2, | [85.8, | [91.7, | [86.7, | [69.1, | [88.5, | [71.3, | [87.8, | [76.2, | [82.1, |
| | 75.4] | 53.1] | 79.3] | 87.5] | 93.2] | 87.3] | 73.7] | 89.2] | 72.7] | 88.5] | 77.0] | 82.5] |
| | 85.0 | 48.6 | 38.5 | 76.5 | 95.0 | 83.5 | 64.5 | 82.5 | 58.5 | 80.9 | 67.4 | 89.9 |
| midnight | [79.5, | [45.1, | [33.2, | [75.5, | [94.5, | [83.1, | [61.3, | [82.1, | [57.8, | [80.5, | [66.9, | [89.7, |
| | 90.1] | 52.0] | 43.8] | 77.5] | 95.6] | 83.9] | 67.5] | 82.9] | 59.2] | 81.4] | 67.9] | 90.1] |
| | 55.2 | 45.9 | 74.7 | 92.3 | 92.3 | 82.0 | 68.0 | 88.2 | 67.8 | 89.0 | 62.7 | 94.6 |
| phikon | [46.4, | [42.6, | [69.5, | [91.7, | [91.6, | [81.6, | [66.0, | [87.9, | [67.1, | [88.7, | [62.2, | [94.5, |
| | 63.6] | 49.2] | 79.6] | 92.9] | 93.0] | 82.3] | 70.1] | 88.5] | 68.5] | 89.4] | 63.2] | 94.8] |
| | 63.9 | 46.4 | 63.9 | 91.1 | 89.2 | 82.8 | 64.5 | 83.8 | 62.3 | 88.2 | 67.6 | 92.3 |
| phikon2 | [55.8, | [43.1, | [58.1, | [90.5, | [88.5, | [82.4, | [62.4, | [83.4, | [61.6, | [87.8, | [67.1, | [92.1, |
| | 71.5] | 49.8] | 69.5] | 91.8] | 89.9] | 83.1] | 66.6] | 84.2] | 63.1] | 88.5] | 68.0] | 92.5] |
| | 68.6 | 54.7 | 79.1 | 89.4 | 93.3 | 85.5 | 74.3 | 91.6 | 67.4 | 89.0 | 68.7 | 96.3 |
| uni | [60.8, | [51.1, | [73.8, | [88.6, | [92.6, | [85.1, | [72.1, | [91.4, | [66.7, | [88.6, | [68.2, | [96.2, |
| | 76.1] | 58.2] | 84.2] | 90.2] | 94.0] | 85.8] | 76.6] | 91.9] | 68.1] | 89.4] | 69.2] | 96.4] |
| | 87.0 | 57.6 | 69.5 | 89.8 | 95.7 | 87.8 | 71.8 | 92.7 | 69.7 | 86.1 | 74.4 | 96.0 |
| uni2h | [82.2, | [54.7, | [63.7, | [89.0, | [95.2, | [87.4, | [69.2, | [92.5, | [69.1, | [85.8, | [74.0, | [95.8, |
| | 91.7] | 60.6] | 75.1] | 90.6] | 96.3] | 88.1] | 74.2] | 93.0] | 70.4] | 86.4] | 74.9] | 96.1] |
| | 52.5 | 43.3 | 53.0 | 89.2 | 90.0 | 85.1 | 64.8 | 88.2 | 64.7 | 82.9 | 57.2 | 96.9 |
| virchow | [44.2, | [40.0, | [46.7, | [88.5, | [89.2, | [84.7, | [62.1, | [87.9, | [64.0, | [82.5, | [56.7, | [96.8, |
| | 60.3] | 46.7] | 59.4] | 89.9] | 90.8] | 85.5] | 67.5] | 88.6] | 65.4] | 83.3] | 57.8] | 97.0] |
| | 83.8 | 54.8 | 58.2 | 78.5 | 90.8 | 82.8 | 72.3 | 81.4 | 63.9 | 84.8 | 64.7 | 95.1 |
| virchow2 | [78.1, | [51.8, | [52.3, | [77.5, | [90.0, | [82.3, | [69.8, | [81.0, | [63.1, | [84.5, | [64.2, | [95.0, |
| | 88.9] | 57.9] | 63.9] | 79.5] | 91.6] | 83.2] | 74.8] | 81.8] | 64.5] | 85.2] | 65.2] | 95.2] |
| | 83.6 | 56.9 | 64.0 | 87.2 | 92.4 | 84.2 | 65.7 | 84.4 | 66.7 | 82.4 | 58.3 | 95.9 |
| conch | [77.5, | [53.9, | [59.1, | [86.3, | [91.7, | [83.8, | [63.1, | [84.0, | [66.0, | [82.0, | [57.8, | [95.7, |
| | 89.3] | 59.9] | 69.3] | 88.1] | 93.1] | 84.6] | 68.4] | 84.8] | 67.4] | 82.8] | 58.9] | 96.0] |
| | 83.9 | 60.2 | 68.7 | 86.1 | 91.5 | 84.9 | 67.4 | 86.7 | 67.1 | 85.7 | 61.4 | 94.4 |
| titan | [78.4, | [57.2, | [63.1, | [85.2, | [90.8, | [84.5, | [64.5, | [86.3, | [66.4, | [85.3, | [60.9, | [94.2, |
| | 89.0] | 63.2] | 74.2] | 86.9] | 92.3] | 85.3] | 70.1] | 87.0] | 67.8] | 86.0] | 61.9] | 94.5] |
| | 86.0 | 57.1 | 65.3 | 92.9 | 91.6 | 87.2 | 65.4 | 87.7 | 64.7 | 87.6 | 64.6 | 95.9 |
| keep | [79.8, | [54.5, | [59.7, | [92.2, | [90.8, | [86.8, | [62.5, | [87.3, | [63.9, | [87.2, | [64.1, | [95.8, |
| | 91.5] | 59.8] | 71.1] | 93.5] | 92.4] | 87.6] | 68.4] | 88.0] | 65.4] | 88.0] | 65.1] | 96.0] |
| | 65.8 | 54.3 | 73.6 | 82.1 | 90.7 | 81.7 | 66.6 | 84.6 | 64.4 | 85.2 | 52.0 | 92.3 |
| musk | [58.1, | [51.4, | [69.0, | [81.1, | [90.0, | [81.3, | [64.5, | [84.2, | [63.7, | [84.8, | [51.4, | [92.1, |
| | 73.4] | 57.3] | 78.2] | 83.0] | 91.5] | 82.1] | 68.8] | 85.0] | 65.1] | 85.5] | 52.5] | 92.4] |
| | 55.3 | 49.9 | 55.3 | 78.8 | 85.7 | 67.2 | 70.2 | 82.9 | 59.8 | 79.6 | 41.3 | 87.0 |
| plip | [47.7, | [46.8, | [49.8, | [77.7, | [84.8, | [66.7, | [68.0, | [82.5, | [59.0, | [79.2, | [40.8, | [86.8, |
| | 63.2] | 53.0] | 60.8] | 79.8] | 86.6] | 67.7] | 72.4] | 83.3] | 60.5] | 80.0] | 41.9] | 87.2] |
| | 54.7 | 51.6 | 67.9 | 81.2 | 89.7 | 68.0 | 68.7 | 83.4 | 60.9 | 82.2 | 40.5 | 90.9 |
| quilt | [46.8, | [48.6, | [62.0, | [80.2, | [89.0, | [67.5, | [66.4, | [83.0, | [60.2, | [81.8, | [40.0, | [90.7, |
| | 62.3] | 54.6] | 73.7] | 82.1] | 90.5] | 68.4] | 71.0] | 83.8] | 61.6] | 82.6] | 41.1] | 91.1] |
| | 62.7 | 42.2 | 65.0 | 73.9 | 80.2 | 67.6 | 75.8 | 71.3 | 58.5 | 74.1 | 33.1 | 83.1 |
| dinob | [54.9, | [39.0, | [58.9, | [72.7, | [79.2, | [67.1, | [73.3, | [70.8, | [57.8, | [73.8, | [32.5, | [82.8, |
| | 70.4] | 45.5] | 70.6] | 75.0] | 81.2] | 68.1] | 78.3] | 71.8] | 59.3] | 74.5] | 33.6] | 83.3] |
| | 58.8 | 44.3 | 63.4 | 68.8 | 76.8 | 67.9 | 76.2 | 66.8 | 59.7 | 66.7 | 33.5 | 86.7 |
| dinol | [50.4, | [41.0, | [57.8, | [67.7, | [75.8, | [67.4, | [73.5, | [66.4, | [59.0, | [66.3, | [33.0, | [86.5, |
| | 66.6] | 47.9] | 68.9] | 69.9] | 77.9] | 68.4] | 78.7] | 67.2] | 60.5] | 67.1] | 34.0] | 86.9] |
| | 57.8 | 45.5 | 60.9 | 59.7 | 76.9 | 64.5 | 72.1 | 73.2 | 56.3 | 77.2 | 28.5 | 79.9 |
| vitb | [50.0, | [42.1, | [55.1, | [58.5, | [76.0, | [63.9, | [69.7, | [72.7, | [55.6, | [76.8, | [28.0, | [79.7, |
| | 65.2] | 48.7] | 66.6] | 60.8] | 77.9] | 65.0] | 74.5] | 73.7] | 57.1] | 77.6] | 29.1] | 80.2] |
| | 52.0 | 48.3 | 44.6 | 67.7 | 80.2 | 64.8 | 70.5 | 70.4 | 57.8 | 75.5 | 30.8 | 75.5 |
| vitl | [44.3, | [45.2, | [38.7, | [66.5, | [79.2, | [64.2, | [67.8, | [69.9, | [57.1, | [75.0, | [30.3, | [75.2, |
| | 59.2] | 51.3] | 50.3] | 68.9] | 81.1] | 65.4] | 73.1] | 70.9] | 58.5] | 75.9] | 31.4] | 75.8] |
| | 45.7 | 42.1 | 51.9 | 63.8 | 72.4 | 52.5 | 68.9 | 72.5 | 56.8 | 73.1 | 30.2 | 67.8 |
| clipb | [38.4, | [39.0, | [45.9, | [62.7, | [71.4, | [52.0, | [66.8, | [72.1, | [56.1, | [72.7, | [29.6, | [67.6, |
| | 53.5] | 45.2] | 57.7] | 65.0] | 73.3] | 53.0] | 71.0] | 72.9] | 57.5] | 73.5] | 30.7] | 68.1] |
| | 52.4 | 45.5 | 51.0 | 63.8 | 78.3 | 65.9 | 69.1 | 77.3 | 57.0 | 74.0 | 33.2 | 90.1 |
| clipl | [44.6, | [42.3, | [45.5, | [62.6, | [77.4, | [65.4, | [67.0, | [76.8, | [56.2, | [73.6, | [32.7, | [90.0, |
| | 60.3] | 48.8] | 56.9] | 64.9] | 79.4] | 66.4] | 71.3] | 77.7] | 57.7] | 74.5] | 33.8] | 90.3] |

Table S47: Quantitative performance (F1-score) on 8-shot classification.

| Model | bach | bracs | break-h | ccrcc | crc | esca | mhist | pcam | tcga-crc | tcga-tils | tcga-unif | wilds |
|---|---|---|---|---|---|---|---|---|---|---|---|---|
| hiboub | 70.5 [62.5, 77.9] | 53.2 [49.4, 56.7] | 76.4 [71.6, 80.7] | 82.5 [81.5, 83.5] | 92.2 [91.5, 93.0] | 69.4 [69.0, 69.7] | 61.9 [58.9, 65.1] | 92.1 [91.8, 92.3] | 59.4 [58.8, 59.9] | 82.4 [82.0, 82.8] | 52.1 [51.6, 52.5] | 94.9 [94.7, 95.0] |
| hiboul | 70.0 [62.2, 77.2] | 50.0 [46.3, 53.6] | 72.1 [67.3, 76.2] | 82.0 [81.0, 82.9] | 83.1 [82.2, 83.9] | 66.3 [66.0, 66.6] | 58.1 [55.1, 61.2] | 91.2 [91.0, 91.5] | 59.6 [59.0, 60.2] | 80.0 [79.6, 80.4] | 55.7 [55.2, 56.1] | 70.4 [70.1, 70.7] |
| hopt0 | 56.0 [47.1, 64.1] | 42.5 [38.6, 46.2] | 69.9 [64.5, 75.0] | 84.8 [83.9, 85.6] | 90.9 [90.0, 91.7] | 73.8 [73.5, 74.0] | 64.3 [61.4, 67.3] | 85.9 [85.5, 86.3] | 63.1 [62.5, 63.7] | 88.1 [87.8, 88.5] | 65.9 [65.4, 66.3] | 94.2 [94.1, 94.4] |
| hopt1 | 65.2 [56.2, 72.8] | 47.3 [43.3, 51.1] | 67.8 [62.5, 72.7] | 87.8 [87.1, 88.6] | 93.0 [92.2, 93.7] | 76.3 [76.0, 76.5] | 65.1 [62.1, 68.1] | 88.9 [88.5, 89.2] | 64.4 [63.7, 64.9] | 82.3 [81.9, 82.6] | 69.6 [69.2, 70.1] | 81.9 [81.6, 82.1] |
| midnight | 81.7 [74.4, 87.9] | 46.2 [42.5, 49.7] | 39.8 [34.4, 44.8] | 78.2 [77.1, 79.4] | 94.8 [94.2, 95.4] | 74.8 [74.5, 75.1] | 63.2 [60.0, 66.2] | 82.3 [81.9, 82.8] | 55.8 [55.2, 56.4] | 75.6 [75.2, 76.0] | 62.7 [62.2, 63.2] | 89.8 [89.6, 90.0] |
| phikon | 54.5 [45.9, 62.5] | 42.1 [38.3, 45.9] | 71.7 [66.5, 76.5] | 91.8 [91.2, 92.5] | 92.5 [91.8, 93.2] | 66.0 [65.6, 66.3] | 59.6 [56.4, 62.8] | 88.2 [87.8, 88.5] | 59.5 [59.0, 60.1] | 86.9 [86.6, 87.3] | 54.6 [54.2, 55.0] | 94.6 [94.4, 94.8] |
| phikon2 | 59.6 [51.0, 67.4] | 42.6 [38.7, 46.3] | 61.1 [55.5, 66.5] | 90.6 [89.9, 91.3] | 89.7 [88.8, 90.5] | 66.4 [66.1, 66.8] | 55.1 [52.0, 58.1] | 83.7 [83.3, 84.1] | 56.3 [55.7, 56.9] | 83.2 [82.8, 83.5] | 59.2 [58.8, 59.7] | 92.3 [92.1, 92.4] |
| uni | 66.1 [57.8, 73.5] | 51.8 [47.8, 55.7] | 77.6 [72.9, 81.9] | 90.0 [89.2, 90.7] | 93.1 [92.4, 93.8] | 71.8 [71.5, 72.1] | 68.7 [65.8, 71.6] | 91.6 [91.3, 91.9] | 59.8 [59.3, 60.4] | 87.5 [87.1, 87.8] | 60.4 [60.0, 60.9] | 96.3 [96.2, 96.4] |
| uni2h | 83.5 [77.0, 89.4] | 53.0 [49.2, 56.6] | 69.7 [64.5, 74.5] | 90.5 [89.7, 91.2] | 95.4 [94.8, 96.0] | 76.4 [76.1, 76.6] | 66.8 [63.9, 69.6] | 92.7 [92.5, 93.0] | 60.8 [60.3, 61.4] | 77.3 [76.9, 77.7] | 68.1 [67.7, 68.5] | 95.9 [95.8, 96.1] |
| virchow | 49.6 [41.1, 57.2] | 38.8 [35.0, 42.5] | 51.0 [45.1, 56.6] | 88.2 [87.4, 89.0] | 90.3 [89.4, 91.1] | 73.6 [73.2, 73.9] | 59.9 [56.8, 62.9] | 88.2 [87.8, 88.5] | 58.2 [57.6, 58.8] | 73.7 [73.3, 74.1] | 50.0 [49.5, 50.5] | 96.9 [96.8, 97.0] |
| virchow2 | 80.7 [73.6, 86.8] | 50.8 [47.4, 54.2] | 58.6 [53.1, 63.5] | 80.5 [79.5, 81.5] | 91.1 [90.3, 91.8] | 69.9 [69.6, 70.2] | 67.5 [64.4, 70.3] | 81.3 [80.9, 81.7] | 57.6 [57.0, 58.1] | 79.1 [78.8, 79.6] | 57.1 [56.6, 57.5] | 95.1 [94.9, 95.2] |
| conch | 80.6 [73.5, 87.1] | 51.9 [48.0, 55.6] | 64.8 [59.4, 70.0] | 88.2 [87.4, 89.0] | 91.9 [91.1, 92.6] | 69.8 [69.4, 70.1] | 59.8 [56.8, 62.9] | 84.4 [84.0, 84.8] | 61.5 [60.9, 62.1] | 76.7 [76.3, 77.1] | 49.5 [49.1, 49.9] | 95.9 [95.7, 96.0] |
| titan | 80.0 [73.0, 86.5] | 56.6 [52.8, 60.1] | 67.7 [61.9, 72.8] | 86.6 [85.7, 87.4] | 91.3 [90.5, 92.0] | 72.4 [72.0, 72.7] | 63.8 [60.8, 66.7] | 86.6 [86.3, 87.0] | 61.5 [60.9, 62.1] | 78.1 [77.7, 78.5] | 54.5 [54.0, 54.9] | 94.4 [94.2, 94.5] |
| keep | 84.6 [77.6, 90.4] | 50.5 [46.8, 54.1] | 66.8 [61.0, 72.0] | 92.9 [92.3, 93.5] | 91.9 [91.1, 92.7] | 74.2 [73.9, 74.6] | 62.4 [59.4, 65.5] | 87.6 [87.3, 88.0] | 59.7 [59.1, 60.3] | 83.9 [83.5, 84.3] | 56.9 [56.5, 57.4] | 95.9 [95.7, 96.0] |
| musk | 62.6 [54.2, 70.4] | 49.2 [45.5, 52.8] | 73.3 [68.9, 77.3] | 81.9 [80.9, 82.8] | 89.5 [88.6, 90.3] | 65.1 [64.7, 65.4] | 57.8 [54.7, 61.0] | 84.6 [84.2, 85.0] | 57.8 [57.2, 58.4] | 79.7 [79.3, 80.1] | 42.7 [42.3, 43.1] | 92.3 [92.1, 92.4] |
| plip | 54.0 [45.3, 62.1] | 45.1 [41.4, 48.6] | 54.5 [49.1, 59.5] | 78.1 [77.1, 79.2] | 86.0 [85.1, 87.0] | 51.7 [51.4, 51.9] | 63.0 [60.1, 66.2] | 82.9 [82.4, 83.3] | 53.2 [52.6, 53.7] | 73.1 [72.7, 73.6] | 33.0 [32.6, 33.4] | 87.0 [86.8, 87.2] |
| quilt | 50.7 [42.1, 58.6] | 46.2 [42.5, 49.8] | 70.1 [64.5, 75.1] | 80.0 [79.0, 81.0] | 89.0 [88.2, 89.8] | 52.4 [52.1, 52.7] | 62.1 [59.0, 65.0] | 83.4 [83.0, 83.8] | 53.8 [53.3, 54.4] | 76.2 [75.8, 76.6] | 32.1 [31.7, 32.5] | 90.9 [90.7, 91.1] |
| dinob | 57.0 [48.6, 65.1] | 37.1 [33.6, 40.5] | 62.9 [57.3, 67.8] | 75.3 [74.1, 76.4] | 79.3 [78.3, 80.3] | 50.5 [50.2, 50.7] | 72.2 [69.4, 75.0] | 70.9 [70.4, 71.3] | 51.3 [50.7, 51.8] | 61.2 [60.7, 61.6] | 25.3 [24.9, 25.6] | 83.0 [82.8, 83.3] |
| dinol | 55.1 [46.3, 62.9] | 41.1 [37.3, 44.7] | 58.6 [53.0, 63.7] | 69.6 [68.3, 70.8] | 76.7 [75.7, 77.7] | 50.6 [50.4, 50.9] | 73.4 [70.5, 76.1] | 64.2 [63.7, 64.7] | 53.6 [53.0, 54.1] | 51.3 [50.9, 51.8] | 26.0 [25.6, 26.4] | 86.6 [86.4, 86.9] |
| vitb | 52.4 [43.9, 60.2] | 40.4 [36.7, 44.0] | 57.4 [51.9, 62.5] | 55.6 [54.3, 56.8] | 76.1 [75.0, 77.2] | 49.4 [49.2, 49.7] | 67.0 [64.2, 69.9] | 73.2 [72.7, 73.7] | 50.5 [50.0, 51.0] | 67.8 [67.4, 68.2] | 21.9 [21.6, 22.3] | 79.9 [79.7, 80.2] |
| vitl | 45.4 [37.1, 52.7] | 42.6 [39.0, 46.2] | 37.5 [32.3, 42.6] | 67.9 [66.7, 69.1] | 79.4 [78.3, 80.5] | 52.1 [51.8, 52.4] | 66.3 [63.3, 69.0] | 70.3 [69.8, 70.8] | 50.0 [49.5, 50.6] | 65.0 [64.6, 65.4] | 23.2 [22.9, 23.6] | 75.2 [74.9, 75.5] |
| clipb | 40.8 [33.4, 47.9] | 36.7 [33.2, 40.3] | 51.3 [45.8, 56.5] | 64.9 [63.6, 66.3] | 69.5 [68.5, 70.6] | 37.1 [36.9, 37.3] | 61.0 [58.1, 64.1] | 71.7 [71.2, 72.2] | 52.1 [51.5, 52.6] | 61.2 [60.8, 61.6] | 23.2 [22.8, 23.5] | 65.0 [64.6, 65.3] |
| clipl | 48.8 [40.6, 56.3] | 40.4 [36.7, 43.9] | 47.8 [42.3, 53.0] | 60.9 [59.6, 62.1] | 76.2 [75.2, 77.3] | 51.1 [50.9, 51.4] | 61.5 [58.4, 64.5] | 77.1 [76.6, 77.6] | 49.3 [48.8, 49.9] | 63.2 [62.8, 63.6] | 25.3 [24.9, 25.6] | 90.1 [90.0, 90.3] |

Table S48: Quantitative performance (Balanced accuracy) on 16-shot classification.

| Model | bach | bracs | break-h | ccrcc | crc | esca | mhist | pcam | tcga-crc | tcga-tils | tcga-unif | wilds |
|---|---|---|---|---|---|---|---|---|---|---|---|---|
| | 76.0 | 57.5 | 75.1 | 82.3 | 92.4 | 84.4 | 69.9 | 91.9 | 65.7 | 87.2 | 61.1 | 94.9 |
| hiboub | [69.4, | [54.2, | [69.7, | [81.4, | [91.7, | [84.0, | [67.8, | [91.6, | [65.0, | [86.8, | [60.6, | [94.8, |
| | 82.4] | 60.9] | 80.2] | 83.2] | 93.1] | 84.7] | 72.2] | 92.2] | 66.4] | 87.6] | 61.6] | 95.1] |
| | 75.9 | 53.1 | 70.3 | 81.8 | 82.4 | 78.8 | 66.6 | 91.1 | 66.5 | 84.7 | 64.7 | 72.9 |
| hiboul | [69.4, | [49.6, | [64.3, | [80.8, | [81.5, | [78.4, | [64.3, | [90.8, | [65.8, | [84.3, | [64.2, | [72.6, |
| | 82.2] | 56.6] | 75.7] | 82.7] | 83.3] | 79.2] | 68.9] | 91.4] | 67.2] | 85.1] | 65.3] | 73.1] |
| | 58.0 | 46.0 | 72.9 | 82.3 | 90.6 | 85.4 | 70.9 | 85.9 | 69.7 | 87.3 | 74.1 | 93.7 |
| hopt0 | [50.2, | [42.5, | [67.3, | [81.4, | [89.8, | [85.1, | [68.5, | [85.5, | [69.0, | [86.9, | [73.7, | [93.5, |
| | 65.9] | 49.6] | 78.4] | 83.2] | 91.5] | 85.8] | 73.4] | 86.3] | 70.3] | 87.7] | 74.6] | 93.8] |
| | 66.5 | 49.7 | 74.8 | 86.2 | 92.7 | 87.1 | 69.9 | 89.7 | 71.1 | 89.3 | 76.8 | 86.7 |
| hopt1 | [58.5, | [46.0, | [69.7, | [85.3, | [92.0, | [86.8, | [67.4, | [89.3, | [70.4, | [88.9, | [76.4, | [86.5, |
| | 74.0] | 53.4] | 79.7] | 87.0] | 93.5] | 87.4] | 72.5] | 90.0] | 71.7] | 89.6] | 77.2] | 86.9] |
| | 85.5 | 50.6 | 37.1 | 75.8 | 94.9 | 83.6 | 63.5 | 83.0 | 58.2 | 81.2 | 67.8 | 90.0 |
| midnight | [80.4, | [47.0, | [32.0, | [74.8, | [94.4, | [83.2, | [60.3, | [82.6, | [57.5, | [80.7, | [67.3, | [89.8, |
| | 90.7] | 54.1] | 42.4] | 76.8] | 95.5] | 84.1] | 66.5] | 83.4] | 58.9] | 81.6] | 68.3] | 90.2] |
| | 55.2 | 47.6 | 74.6 | 92.1 | 92.3 | 82.1 | 68.4 | 88.3 | 66.4 | 88.9 | 62.9 | 94.7 |
| phikon | [46.4, | [44.2, | [69.4, | [91.4, | [91.6, | [81.7, | [66.3, | [88.0, | [65.7, | [88.6, | [62.4, | [94.5, |
| | 63.6] | 51.1] | 79.5] | 92.7] | 93.0] | 82.4] | 70.5] | 88.7] | 67.1] | 89.3] | 63.4] | 94.8] |
| | 61.2 | 46.6 | 64.0 | 90.8 | 89.2 | 82.7 | 64.5 | 83.6 | 62.2 | 87.8 | 68.0 | 92.2 |
| phikon2 | [52.7, | [43.2, | [58.3, | [90.1, | [88.5, | [82.3, | [62.3, | [83.2, | [61.5, | [87.5, | [67.5, | [92.0, |
| | 69.1] | 50.2] | 69.5] | 91.5] | 89.9] | 83.1] | 66.7] | 84.0] | 63.0] | 88.2] | 68.4] | 92.4] |
| | 67.7 | 54.7 | 78.6 | 88.7 | 93.9 | 85.5 | 74.7 | 91.8 | 66.9 | 88.8 | 69.0 | 96.1 |
| uni | [59.8, | [51.1, | [73.3, | [87.9, | [93.2, | [85.2, | [72.5, | [91.5, | [66.2, | [88.4, | [68.5, | [95.9, |
| | 75.4] | 58.4] | 83.7] | 89.5] | 94.5] | 85.9] | 77.0] | 92.1] | 67.6] | 89.2] | 69.6] | 96.2] |
| | 87.4 | 58.4 | 69.5 | 89.9 | 95.9 | 87.8 | 72.5 | 92.9 | 69.3 | 88.0 | 74.9 | 95.9 |
| uni2h | [82.3, | [55.3, | [63.7, | [89.1, | [95.3, | [87.5, | [70.0, | [92.7, | [68.7, | [87.7, | [74.5, | [95.8, |
| | 92.3] | 61.5] | 75.1] | 90.7] | 96.4] | 88.2] | 74.9] | 93.2] | 70.0] | 88.4] | 75.4] | 96.1] |
| | 52.7 | 43.8 | 52.3 | 89.2 | 90.1 | 85.2 | 64.9 | 88.2 | 63.9 | 84.4 | 57.6 | 96.8 |
| virchow | [44.7, | [40.3, | [46.0, | [88.5, | [89.3, | [84.8, | [62.3, | [87.9, | [63.1, | [84.1, | [57.1, | [96.7, |
| | 60.2] | 47.4] | 58.7] | 89.9] | 90.9] | 85.5] | 67.7] | 88.6] | 64.6] | 84.8] | 58.2] | 96.9] |
| | 83.5 | 54.5 | 58.5 | 77.8 | 91.1 | 82.8 | 73.0 | 80.8 | 63.2 | 86.1 | 64.6 | 94.2 |
| virchow2 | [77.5, | [51.4, | [52.5, | [76.8, | [90.3, | [82.3, | [70.4, | [80.4, | [62.5, | [85.7, | [64.1, | [94.0, |
| | 89.0] | 57.8] | 64.5] | 78.9] | 91.8] | 83.2] | 75.4] | 81.2] | 63.9] | 86.4] | 65.1] | 94.3] |
| | 83.6 | 56.7 | 62.8 | 87.1 | 92.6 | 84.1 | 66.8 | 84.4 | 66.6 | 82.6 | 58.7 | 95.9 |
| conch | [77.5, | [53.4, | [57.7, | [86.3, | [91.9, | [83.7, | [64.2, | [84.0, | [65.8, | [82.2, | [58.2, | [95.8, |
| | 89.3] | 59.8] | 68.1] | 88.0] | 93.3] | 84.5] | 69.5] | 84.8] | 67.3] | 83.0] | 59.3] | 96.1] |
| | 83.9 | 60.7 | 68.4 | 85.8 | 91.7 | 84.9 | 67.7 | 86.1 | 66.8 | 86.0 | 61.9 | 94.4 |
| titan | [78.4, | [57.6, | [62.9, | [84.9, | [90.9, | [84.5, | [64.9, | [85.7, | [66.1, | [85.7, | [61.4, | [94.2, |
| | 89.0] | 63.8] | 73.9] | 86.6] | 92.5] | 85.3] | 70.6] | 86.4] | 67.5] | 86.4] | 62.4] | 94.5] |
| | 86.0 | 57.7 | 64.7 | 92.7 | 91.7 | 87.2 | 65.2 | 87.5 | 64.6 | 87.5 | 65.1 | 95.7 |
| keep | [79.8, | [54.9, | [59.1, | [92.1, | [90.9, | [86.8, | [62.2, | [87.1, | [63.9, | [87.1, | [64.6, | [95.6, |
| | 91.5] | 60.5] | 70.4] | 93.4] | 92.5] | 87.6] | 68.2] | 87.8] | 65.4] | 87.9] | 65.6] | 95.9] |
| | 65.5 | 55.1 | 73.8 | 82.1 | 90.8 | 81.5 | 66.5 | 84.5 | 63.9 | 84.9 | 52.5 | 92.2 |
| musk | [57.9, | [52.0, | [69.1, | [81.2, | [90.1, | [81.1, | [64.4, | [84.2, | [63.1, | [84.5, | [52.0, | [92.0, |
| | 73.2] | 58.2] | 78.3] | 83.1] | 91.5] | 81.9] | 68.6] | 84.9] | 64.6] | 85.3] | 53.1] | 92.4] |
| | 54.5 | 49.6 | 55.6 | 78.4 | 85.6 | 66.8 | 70.6 | 82.5 | 59.5 | 79.5 | 41.3 | 86.8 |
| plip | [47.0, | [46.4, | [50.2, | [77.4, | [84.7, | [66.3, | [68.5, | [82.1, | [58.7, | [79.1, | [40.8, | [86.6, |
| | 62.2] | 52.8] | 61.2] | 79.5] | 86.4] | 67.3] | 72.9] | 82.9] | 60.2] | 79.9] | 41.9] | 87.0] |
| | 53.9 | 51.1 | 68.5 | 81.2 | 89.9 | 67.7 | 69.6 | 82.9 | 60.6 | 81.9 | 40.4 | 90.3 |
| quilt | [46.1, | [48.0, | [62.8, | [80.3, | [89.1, | [67.3, | [67.2, | [82.5, | [59.9, | [81.5, | [39.9, | [90.1, |
| | 61.3] | 54.4] | 74.3] | 82.2] | 90.6] | 68.2] | 71.9] | 83.3] | 61.3] | 82.4] | 41.0] | 90.5] |
| | 61.5 | 42.8 | 65.9 | 73.2 | 81.0 | 67.7 | 75.5 | 71.1 | 58.3 | 75.8 | 33.3 | 83.3 |
| dinob | [53.4, | [39.5, | [59.9, | [72.0, | [80.0, | [67.1, | [73.0, | [70.6, | [57.5, | [75.4, | [32.7, | [83.0, |
| | 69.3] | 46.1] | 71.6] | 74.3] | 81.9] | 68.2] | 78.0] | 71.5] | 59.0] | 76.2] | 33.8] | 83.5] |
| | 58.5 | 43.6 | 62.0 | 68.1 | 77.4 | 67.5 | 76.0 | 67.3 | 59.6 | 67.5 | 33.6 | 87.9 |
| dinol | [49.9, | [40.2, | [56.2, | [66.9, | [76.4, | [67.0, | [73.4, | [66.9, | [58.9, | [67.1, | [33.0, | [87.6, |
| | 66.2] | 47.2] | 67.5] | 69.2] | 78.4] | 68.1] | 78.6] | 67.7] | 60.3] | 67.9] | 34.1] | 88.1] |
| | 55.3 | 45.3 | 60.9 | 59.6 | 77.5 | 64.3 | 72.4 | 73.1 | 56.4 | 77.2 | 28.8 | 79.7 |
| vitb | [47.4, | [41.9, | [55.3, | [58.4, | [76.5, | [63.7, | [69.9, | [72.7, | [55.6, | [76.8, | [28.3, | [79.4, |
| | 63.1] | 48.7] | 66.7] | 60.7] | 78.4] | 64.8] | 74.9] | 73.6] | 57.1] | 77.6] | 29.3] | 80.0] |
| | 49.8 | 48.8 | 42.8 | 67.7 | 80.8 | 64.7 | 71.0 | 70.1 | 58.1 | 76.3 | 31.3 | 75.3 |
| vitl | [41.5, | [45.7, | [37.0, | [66.5, | [79.8, | [64.1, | [68.3, | [69.6, | [57.4, | [75.9, | [30.8, | [75.0, |
| | 57.7] | 51.9] | 48.5] | 68.9] | 81.8] | 65.3] | 73.6] | 70.6] | 58.8] | 76.8] | 31.9] | 75.6] |
| | 46.2 | 41.8 | 51.6 | 63.9 | 72.5 | 52.5 | 69.8 | 72.4 | 56.3 | 73.8 | 30.3 | 67.8 |
| clipb | [38.9, | [38.6, | [45.7, | [62.7, | [71.6, | [51.9, | [67.7, | [71.9, | [55.6, | [73.4, | [29.8, | [67.5, |
| | 54.0] | 45.2] | 57.5] | 65.1] | 73.5] | 53.0] | 71.9] | 72.8] | 57.0] | 74.3] | 30.8] | 68.0] |
| | 52.4 | 46.0 | 53.3 | 64.1 | 78.5 | 65.6 | 69.9 | 77.3 | 56.8 | 74.6 | 33.7 | 90.2 |
| clipl | [44.6, | [42.7, | [47.7, | [63.0, | [77.5, | [65.1, | [67.8, | [76.9, | [56.1, | [74.2, | [33.2, | [90.1, |
| | 60.3] | 49.3] | 59.5] | 65.2] | 79.5] | 66.1] | 72.1] | 77.8] | 57.5] | 75.1] | 34.3] | 90.4] |

Table S49: Quantitative performance (F1-score) on 16-shot classification.

| Model | bach | bracs | break-h | ccrcc | crc | esca | mhist | pcam | tcga-crc | tcga-tils | tcga-unif | wilds |
|---|---|---|---|---|---|---|---|---|---|---|---|---|
| | 71.2 | 54.3 | 75.0 | 82.0 | 92.4 | 70.3 | 62.8 | 91.9 | 58.7 | 83.6 | 53.2 | 94.9 |
| hiboub | [63.3, | [50.3, | [69.9, | [81.0, | [91.7, | [69.9, | [59.8, | [91.6, | [58.2, | [83.2, | [52.7, | [94.8, |
| | 78.7] | 57.9] | 79.3] | 83.0] | 93.1] | 70.6] | 66.0] | 92.2] | 59.3] | 84.0] | 53.6] | 95.1] |
| | 72.2 | 51.3 | 71.4 | 81.4 | 83.2 | 67.2 | 59.1 | 91.0 | 58.8 | 81.7 | 57.0 | 71.0 |
| hiboul | [64.5, | [47.4, | [66.5, | [80.4, | [82.3, | [66.9, | [56.1, | [90.7, | [58.2, | [81.3, | [56.6, | [70.7, |
| | 79.2] | 54.9] | 75.7] | 82.3] | 84.0] | 67.5] | 62.2] | 91.3] | 59.4] | 82.1] | 57.5] | 71.3] |
| | 55.2 | 43.1 | 70.6 | 83.9 | 91.3 | 74.2 | 65.5 | 85.9 | 61.4 | 88.5 | 66.9 | 93.7 |
| hopt0 | [46.3, | [39.1, | [65.2, | [83.0, | [90.5, | [74.0, | [62.6, | [85.5, | [60.8, | [88.1, | [66.5, | [93.5, |
| | 63.4] | 47.0] | 75.5] | 84.8] | 92.0] | 74.5] | 68.6] | 86.2] | 62.0] | 88.8] | 67.4] | 93.8] |
| | 63.4 | 48.4 | 68.5 | 87.4 | 93.2 | 76.6 | 64.9 | 89.6 | 62.8 | 86.3 | 70.5 | 86.6 |
| hopt1 | [54.5, | [44.4, | [63.2, | [86.6, | [92.5, | [76.3, | [61.8, | [89.3, | [62.2, | [86.0, | [70.0, | [86.3, |
| | 71.4] | 52.4] | 73.3] | 88.2] | 93.9] | 76.8] | 67.9] | 90.0] | 63.4] | 86.7] | 70.9] | 86.8] |
| | 82.5 | 49.4 | 38.5 | 77.5 | 94.7 | 75.0 | 62.4 | 82.8 | 55.1 | 76.2 | 63.4 | 89.9 |
| midnight | [75.5, | [45.5, | [33.1, | [76.4, | [94.1, | [74.7, | [59.3, | [82.4, | [54.5, | [75.8, | [62.9, | [89.7, |
| | 88.5] | 53.0] | 43.7] | 78.7] | 95.3] | 75.4] | 65.4] | 83.2] | 55.7] | 76.6] | 63.9] | 90.1] |
| | 54.5 | 44.6 | 71.5 | 91.6 | 92.5 | 66.5 | 60.2 | 88.3 | 58.4 | 87.7 | 55.4 | 94.7 |
| phikon | [45.9, | [40.6, | [66.2, | [90.9, | [91.8, | [66.2, | [57.0, | [88.0, | [57.9, | [87.4, | [55.0, | [94.5, |
| | 62.5] | 48.3] | 76.3] | 92.3] | 93.2] | 66.9] | 63.4] | 88.7] | 59.0] | 88.1] | 55.9] | 94.8] |
| | 57.3 | 43.2 | 60.9 | 90.1 | 89.6 | 67.0 | 55.5 | 83.5 | 56.2 | 85.6 | 60.2 | 92.2 |
| phikon2 | [48.6, | [39.4, | [55.4, | [89.4, | [88.8, | [66.7, | [52.5, | [83.1, | [55.6, | [85.2, | [59.8, | [92.0, |
| | 65.2] | 47.0] | 66.2] | 90.8] | 90.4] | 67.3] | 58.7] | 83.9] | 56.8] | 85.9] | 60.7] | 92.4] |
| | 65.2 | 52.5 | 77.4 | 89.3 | 93.6 | 72.4 | 69.4 | 91.8 | 58.9 | 88.3 | 61.5 | 96.1 |
| uni | [56.6, | [48.5, | [72.8, | [88.6, | [92.9, | [72.1, | [66.5, | [91.5, | [58.3, | [87.9, | [61.1, | [95.9, |
| | 72.7] | 56.3] | 81.7] | 90.1] | 94.3] | 72.7] | 72.4] | 92.1] | 59.4] | 88.6] | 62.0] | 96.2] |
| | 84.7 | 54.7 | 69.9 | 90.6 | 95.5 | 76.7 | 68.1 | 92.9 | 60.1 | 82.5 | 69.2 | 95.9 |
| uni2h | [78.0, | [51.0, | [64.6, | [89.9, | [94.9, | [76.4, | [65.3, | [92.7, | [59.5, | [82.1, | [68.7, | [95.8, |
| | 90.6] | 58.3] | 74.6] | 91.3] | 96.1] | 77.0] | 70.9] | 93.2] | 60.6] | 82.9] | 69.6] | 96.1] |
| | 49.6 | 40.1 | 50.5 | 88.2 | 90.3 | 73.7 | 60.2 | 88.2 | 57.0 | 76.9 | 50.8 | 96.8 |
| virchow | [41.1, | [36.1, | [44.5, | [87.4, | [89.4, | [73.4, | [57.1, | [87.8, | [56.4, | [76.5, | [50.3, | [96.7, |
| | 57.1] | 43.9] | 55.9] | 89.0] | 91.1] | 74.1] | 63.2] | 88.5] | 57.5] | 77.3] | 51.2] | 96.9] |
| | 80.7 | 51.4 | 58.5 | 79.9 | 91.3 | 70.0 | 68.5 | 80.7 | 56.4 | 81.6 | 57.4 | 94.2 |
| virchow2 | [73.7, | [48.0, | [52.9, | [78.9, | [90.6, | [69.6, | [65.3, | [80.3, | [55.9, | [81.2, | [57.0, | [94.0, |
| | 86.8] | 54.9] | 63.7] | 80.9] | 92.1] | 70.3] | 71.3] | 81.1] | 57.0] | 82.0] | 57.9] | 94.3] |
| | 80.6 | 53.0 | 63.8 | 88.1 | 92.1 | 69.9 | 61.4 | 84.4 | 60.7 | 77.5 | 50.3 | 95.9 |
| conch | [73.5, | [49.0, | [58.4, | [87.3, | [91.3, | [69.6, | [58.3, | [84.0, | [60.1, | [77.1, | [49.8, | [95.8, |
| | 87.1] | 56.7] | 68.8] | 89.0] | 92.8] | 70.3] | 64.4] | 84.7] | 61.3] | 77.9] | 50.7] | 96.1] |
| | 80.0 | 57.7 | 67.2 | 86.3 | 91.5 | 72.5 | 64.4 | 86.1 | 60.4 | 79.7 | 55.5 | 94.4 |
| titan | [73.0, | [53.8, | [61.4, | [85.5, | [90.7, | [72.2, | [61.3, | [85.7, | [59.8, | [79.3, | [55.0, | [94.2, |
| | 86.5] | 61.2] | 72.3] | 87.2] | 92.2] | 72.8] | 67.4] | 86.4] | 61.0] | 80.1] | 55.9] | 94.5] |
| | 84.6 | 52.1 | 66.2 | 92.8 | 92.0 | 74.3 | 62.7 | 87.4 | 59.0 | 84.5 | 57.8 | 95.7 |
| keep | [77.6, | [48.3, | [60.5, | [92.2, | [91.2, | [74.0, | [59.6, | [87.1, | [58.4, | [84.1, | [57.4, | [95.6, |
| | 90.4] | 55.8] | 71.4] | 93.4] | 92.7] | 74.7] | 65.8] | 87.8] | 59.6] | 84.8] | 58.3] | 95.9] |
| | 62.5 | 50.9 | 73.3 | 82.0 | 89.5 | 65.1 | 58.0 | 84.5 | 57.0 | 80.4 | 44.0 | 92.2 |
| musk | [54.2, | [47.0, | [69.0, | [81.0, | [88.7, | [64.7, | [54.9, | [84.1, | [56.4, | [80.0, | [43.6, | [92.0, |
| | 70.3] | 54.6] | 77.4] | 82.9] | 90.3] | 65.4] | 61.2] | 84.9] | 57.5] | 80.8] | 44.5] | 92.4] |
| | 52.8 | 45.3 | 55.0 | 78.0 | 85.9 | 51.4 | 63.7 | 82.5 | 52.5 | 73.5 | 33.8 | 86.8 |
| plip | [44.2, | [41.5, | [49.7, | [76.9, | [84.9, | [51.2, | [60.7, | [82.1, | [51.9, | [73.1, | [33.4, | [86.6, |
| | 60.8] | 48.8] | 60.2] | 79.1] | 86.8] | 51.7] | 66.8] | 82.9] | 53.0] | 74.0] | 34.2] | 87.0] |
| | 49.7 | 46.8 | 70.7 | 80.1 | 89.1 | 52.5 | 63.5 | 82.9 | 53.1 | 76.6 | 32.8 | 90.3 |
| quilt | [41.2, | [43.0, | [65.4, | [79.1, | [88.3, | [52.2, | [60.5, | [82.5, | [52.5, | [76.2, | [32.4, | [90.1, |
| | 57.4] | 50.6] | 75.5] | 81.1] | 89.9] | 52.8] | 66.5] | 83.3] | 53.7] | 77.0] | 33.1] | 90.5] |
| | 56.6 | 38.5 | 63.8 | 74.6 | 80.2 | 51.1 | 71.8 | 70.7 | 51.2 | 64.7 | 26.0 | 83.3 |
| dinob | [48.3, | [34.8, | [58.3, | [73.4, | [79.2, | [50.9, | [68.9, | [70.2, | [50.7, | [64.3, | [25.7, | [83.0, |
| | 64.5] | 41.9] | 68.6] | 75.8] | 81.1] | 51.4] | 74.6] | 71.2] | 51.8] | 65.1] | 26.4] | 83.5] |
| | 55.1 | 41.1 | 57.6 | 68.9 | 77.3 | 51.0 | 73.1 | 65.2 | 53.3 | 53.8 | 26.7 | 87.8 |
| dinol | [46.2, | [37.4, | [52.1, | [67.6, | [76.3, | [50.7, | [70.2, | [64.6, | [52.8, | [53.4, | [26.3, | [87.6, |
| | 62.8] | 44.8] | 62.6] | 70.1] | 78.3] | 51.2] | 75.9] | 65.7] | 53.8] | 54.2] | 27.0] | 88.1] |
| | 51.8 | 40.9 | 57.5 | 55.6 | 76.7 | 49.2 | 67.8 | 73.1 | 50.5 | 68.7 | 22.5 | 79.7 |
| vitb | [43.3, | [37.2, | [52.3, | [54.3, | [75.6, | [48.9, | [65.0, | [72.7, | [50.0, | [68.3, | [22.2, | [79.4, |
| | 59.3] | 44.4] | 62.7] | 56.8] | 77.8] | 49.5] | 70.6] | 73.6] | 51.1] | 69.1] | 22.9] | 80.0] |
| | 44.3 | 43.7 | 35.8 | 68.1 | 80.0 | 52.0 | 67.3 | 69.9 | 50.4 | 67.3 | 24.3 | 75.1 |
| vitl | [36.0, | [40.0, | [30.5, | [66.9, | [79.0, | [51.7, | [64.3, | [69.4, | [49.9, | [66.9, | [23.9, | [74.8, |
| | 51.8] | 47.3] | 40.8] | 69.3] | 81.1] | 52.3] | 70.1] | 70.4] | 50.9] | 67.7] | 24.7] | 75.4] |
| | 41.7 | 37.4 | 51.2 | 64.9 | 69.8 | 37.0 | 62.3 | 71.7 | 51.0 | 63.2 | 23.9 | 65.1 |
| clipb | [34.1, | [33.8, | [45.6, | [63.6, | [68.8, | [36.8, | [59.3, | [71.2, | [50.5, | [62.8, | [23.6, | [64.8, |
| | 48.9] | 40.9] | 56.3] | 66.3] | 70.8] | 37.3] | 65.3] | 72.1] | 51.6] | 63.7] | 24.3] | 65.4] |
| | 48.8 | 41.6 | 49.7 | 60.9 | 76.4 | 50.8 | 62.6 | 77.2 | 48.9 | 65.3 | 26.5 | 90.2 |
| clipl | [40.6, | [37.8, | [44.1, | [59.6, | [75.4, | [50.5, | [59.5, | [76.7, | [48.4, | [64.9, | [26.1, | [90.0, |
| | 56.3] | 45.2] | 55.1] | 62.1] | 77.4] | 51.0] | 65.6] | 77.7] | 49.4] | 65.8] | 26.8] | 90.4] |

Table S50: Quantitative performance (Dice Score) on semantic segmentation.

| Model | ocelot | pannuke | segp-ep | segp-ly |
|---|---|---|---|---|
| | 80.2 | 60.9 | 67.8 | 62.3 |
| hiboub | [79.3, | [60.3, | [67.5, | [62.0, |
| | 81.1] | 61.6] | 68.0] | 62.7] |
| | 79.7 | 62.5 | 69.5 | 62.6 |
| hiboul | [78.7, | [61.9, | [69.3, | [62.3, |
| | 80.5] | 63.2] | 69.8] | 63.0] |
| | 78.4 | 55.4 | 68.0 | 59.1 |
| hopt0 | [77.5, | [54.8, | [67.8, | [58.7, |
| | 79.4] | 56.0] | 68.3] | 59.4] |
| | 77.4 | 53.3 | 68.3 | 59.0 |
| hopt1 | [76.5, | [52.7, | [68.1, | [58.6, |
| | 78.3] | 54.0] | 68.6] | 59.3] |
| | 79.4 | 62.0 | 71.2 | 62.7 |
| midnight | [78.5, | [61.4, | [71.0, | [62.4, |
| | 80.4] | 62.6] | 71.5] | 63.1] |
| | 80.2 | 60.7 | 69.2 | 61.9 |
| phikon | [79.3, | [60.1, | [68.9, | [61.5, |
| | 81.0] | 61.4] | 69.4] | 62.2] |
| | 78.7 | 61.0 | 69.1 | 60.9 |
| phikon2 | [77.7, | [60.4, | [68.8, | [60.6, |
| | 79.7] | 61.6] | 69.3] | 61.3] |
| | 79.1 | 60.6 | 69.7 | 61.7 |
| uni | [78.1, | [60.0, | [69.5, | [61.3, |
| | 80.0] | 61.2] | 70.0] | 62.1] |
| | 81.1 | 61.7 | 71.0 | 62.1 |
| uni2h | [80.1, | [61.0, | [70.7, | [61.8, |
| | 82.0] | 62.3] | 71.2] | 62.5] |
| | 81.1 | 62.6 | 70.2 | 62.8 |
| virchow | [80.2, | [62.0, | [70.0, | [62.5, |
| | 82.0] | 63.2] | 70.5] | 63.2] |
| | 80.8 | 62.2 | 71.3 | 63.0 |
| virchow2 | [79.8, | [61.6, | [71.1, | [62.6, |
| | 81.6] | 62.8] | 71.6] | 63.4] |
| | 78.7 | 63.7 | 67.4 | 63.5 |
| conch | [77.7, | [63.1, | [67.2, | [63.1, |
| | 79.7] | 64.4] | 67.7] | 63.9] |
| | 79.7 | 63.2 | 68.8 | 63.4 |
| titan | [78.7, | [62.5, | [68.6, | [63.0, |
| | 80.6] | 63.8] | 69.1] | 63.8] |
| | 80.7 | 60.2 | 69.9 | 61.4 |
| keep | [79.8, | [59.6, | [69.6, | [61.0, |
| | 81.6] | 60.9] | 70.1] | 61.8] |
| | 74.1 | 59.7 | 64.7 | 61.8 |
| musk | [73.1, | [59.0, | [64.4, | [61.5, |
| | 75.0] | 60.3] | 64.9] | 62.2] |
| | 68.6 | 46.5 | 62.2 | 56.7 |
| plip | [67.6, | [45.9, | [62.0, | [56.4, |
| | 69.6] | 47.1] | 62.5] | 57.0] |
| | 68.9 | 46.4 | 62.8 | 57.5 |
| quilt | [67.9, | [45.8, | [62.6, | [57.2, |
| | 69.9] | 47.1] | 63.0] | 57.8] |
| | 72.5 | 46.6 | 63.1 | 57.1 |
| dinob | [71.5, | [46.0, | [62.8, | [56.8, |
| | 73.5] | 47.2] | 63.3] | 57.4] |
| | 72.4 | 46.3 | 62.9 | 57.0 |
| dinol | [71.3, | [45.6, | [62.7, | [56.7, |
| | 73.3] | 46.9] | 63.2] | 57.3] |
| | 69.5 | 54.3 | 61.0 | 59.1 |
| vitb | [68.5, | [53.6, | [60.8, | [58.8, |
| | 70.5] | 54.9] | 61.3] | 59.5] |
| | 72.2 | 55.7 | 64.3 | 60.2 |
| vitl | [71.2, | [55.0, | [64.1, | [59.8, |
| | 73.2] | 56.4] | 64.6] | 60.5] |
| | 63.3 | 43.3 | 60.1 | 57.1 |
| clipb | [62.3, | [42.6, | [59.8, | [56.9, |
| | 64.3] | 43.9] | 60.3] | 57.4] |
| | 70.0 | 52.0 | 61.2 | 59.8 |
| clipl | [69.1, | [51.3, | [61.0, | [59.5, |
| | 71.0] | 52.6] | 61.4] | 60.2] |

Table S51: Quantitative performance (Jaccard Index) on semantic segmentation.

| Model | ocelot | pannuke | segp-ep | segp-ly |
|---|---|---|---|---|
| | 74.4 | 52.5 | 60.5 | 58.0 |
| hiboub | [73.4, | [51.8, | [60.2, | [57.7, |
| | 75.4] | 53.1] | 60.7] | 58.4] |
| | 73.6 | 53.9 | 62.0 | 58.7 |
| hiboul | [72.7, | [53.3, | [61.8, | [58.3, |
| | 74.5] | 54.5] | 62.2] | 59.0] |
| | 72.3 | 46.8 | 60.8 | 55.5 |
| hopt0 | [71.3, | [46.2, | [60.5, | [55.2, |
| | 73.3] | 47.4] | 61.0] | 55.8] |
| | 71.2 | 45.5 | 60.8 | 55.4 |
| hopt1 | [70.2, | [44.9, | [60.6, | [55.1, |
| | 72.2] | 46.1] | 61.0] | 55.7] |
| | 73.7 | 53.3 | 63.8 | 58.5 |
| midnight | [72.7, | [52.7, | [63.6, | [58.2, |
| | 74.7] | 53.9] | 64.0] | 58.8] |
| | 74.1 | 52.0 | 61.7 | 57.7 |
| phikon | [73.1, | [51.4, | [61.5, | [57.4, |
| | 75.0] | 52.6] | 62.0] | 58.0] |
| | 73.1 | 52.3 | 61.6 | 57.3 |
| phikon2 | [72.1, | [51.7, | [61.4, | [57.0, |
| | 74.1] | 52.9] | 61.8] | 57.6] |
| | 73.2 | 51.9 | 62.2 | 57.7 |
| uni | [72.2, | [51.3, | [62.0, | [57.4, |
| | 74.2] | 52.6] | 62.4] | 58.0] |
| | 75.6 | 53.0 | 63.5 | 57.8 |
| uni2h | [74.5, | [52.4, | [63.3, | [57.5, |
| | 76.5] | 53.7] | 63.7] | 58.1] |
| | 75.4 | 54.0 | 62.6 | 58.6 |
| virchow | [74.4, | [53.4, | [62.4, | [58.3, |
| | 76.3] | 54.6] | 62.9] | 58.9] |
| | 75.1 | 53.5 | 63.8 | 58.9 |
| virchow2 | [74.1, | [52.9, | [63.6, | [58.5, |
| | 76.1] | 54.1] | 64.0] | 59.2] |
| | 72.7 | 55.7 | 59.8 | 59.1 |
| conch | [71.7, | [55.1, | [59.6, | [58.8, |
| | 73.8] | 56.4] | 60.0] | 59.4] |
| | 74.1 | 55.3 | 61.2 | 59.1 |
| titan | [73.0, | [54.6, | [61.0, | [58.8, |
| | 75.1] | 55.9] | 61.4] | 59.5] |
| | 74.6 | 51.5 | 62.4 | 57.6 |
| keep | [73.6, | [50.9, | [62.2, | [57.3, |
| | 75.5] | 52.1] | 62.7] | 57.9] |
| | 67.5 | 51.5 | 57.0 | 57.6 |
| musk | [66.5, | [50.8, | [56.8, | [57.4, |
| | 68.5] | 52.1] | 57.2] | 58.0] |
| | 61.7 | 38.1 | 54.5 | 53.9 |
| plip | [60.6, | [37.6, | [54.3, | [53.6, |
| | 62.7] | 38.7] | 54.8] | 54.3] |
| | 61.9 | 38.1 | 55.0 | 55.0 |
| quilt | [60.8, | [37.5, | [54.8, | [54.7, |
| | 62.9] | 38.8] | 55.2] | 55.3] |
| | 65.4 | 38.9 | 56.0 | 54.5 |
| dinob | [64.4, | [38.3, | [55.7, | [54.2, |
| | 66.4] | 39.5] | 56.2] | 54.8] |
| | 65.5 | 38.9 | 55.5 | 54.3 |
| dinol | [64.4, | [38.3, | [55.3, | [54.0, |
| | 66.4] | 39.5] | 55.8] | 54.6] |
| | 62.4 | 46.1 | 53.5 | 55.9 |
| vitb | [61.4, | [45.5, | [53.3, | [55.6, |
| | 63.4] | 46.7] | 53.7] | 56.2] |
| | 65.2 | 47.4 | 56.5 | 56.4 |
| vitl | [64.2, | [46.8, | [56.3, | [56.1, |
| | 66.2] | 48.0] | 56.8] | 56.7] |
| | 56.2 | 36.1 | 52.6 | 54.9 |
| clipb | [55.1, | [35.5, | [52.4, | [54.6, |
| | 57.2] | 36.8] | 52.8] | 55.2] |
| | 62.8 | 44.0 | 53.6 | 56.5 |
| clipl | [61.8, | [43.4, | [53.4, | [56.2, |
| | 63.8] | 44.6] | 53.8] | 56.8] |

Table S52: Quantitative performance (ECE) on linear probing.

| Model | bach | bracs | break-h | ccrcc | crc | esca | mhist | pcam | tcga-crc | tcga-tils | tcga-unif | wilds |
|---|---|---|---|---|---|---|---|---|---|---|---|---|
|  | 16.1 | 3.6 | 3.8 | 3.7 | 2.8 | 1.5 | 4.5 | 1.8 | 0.6 | 0.6 | 3.9 | 1.0 |
| hiboub | [9.6, | [1.7, | [2.7, | [3.0, | [2.4, | [1.3, | [2.7, | [1.6, | [0.4, | [0.4, | [3.6, | [0.9, |
|  | 23.7] | 7.6] | 9.5] | 4.5] | 3.2] | 1.6] | 6.8] | 2.1] | 1.1] | 0.7] | 4.2] | 1.1] |
|  | 8.4 | 18.0 | 6.2 | 11.9 | 4.2 | 3.0 | 4.9 | 2.8 | 3.2 | 0.7 | 2.1 | 0.1 |
| hiboul | [4.2, | [14.3, | [3.6, | [11.2, | [3.7, | [2.9, | [3.0, | [2.5, | [2.8, | [0.5, | [1.8, | [0.0, |
|  | 15.8] | 21.7] | 10.9] | 12.6] | 4.6] | 3.1] | 7.1] | 3.0] | 3.6] | 0.8] | 2.4] | 0.2] |
|  | 13.3 | 14.5 | 2.9 | 6.2 | 4.0 | 2.9 | 1.6 | 2.9 | 3.1 | 0.5 | 3.1 | 1.9 |
| hopt0 | [7.6, | [11.0, | [1.4, | [5.6, | [3.6, | [2.8, | [0.6, | [2.7, | [2.7, | [0.3, | [2.8, | [1.9, |
|  | 21.2] | 18.2] | 7.4] | 6.9] | 4.5] | 3.0] | 3.7] | 3.2] | 3.6] | 0.6] | 3.3] | 2.0] |
|  | 5.1 | 15.3 | 3.6 | 4.5 | 3.3 | 0.7 | 9.7 | 2.8 | 1.7 | 0.2 | 1.9 | 0.6 |
| hopt1 | [3.0, | [11.8, | [1.7, | [3.8, | [2.9, | [0.7, | [8.1, | [2.6, | [1.4, | [0.1, | [1.6, | [0.5, |
|  | 13.4] | 18.9] | 7.5] | 5.1] | 3.8] | 0.8] | 12.1] | 3.1] | 2.2] | 0.4] | 2.1] | 0.7] |
|  | 6.0 | 12.3 | 2.1 | 3.5 | 2.7 | 0.9 | 1.7 | 0.7 | 2.5 | 0.8 | 4.9 | 0.2 |
| midnight | [2.6, | [9.2, | [1.5, | [2.9, | [2.3, | [0.8, | [0.8, | [0.4, | [2.1, | [0.7, | [4.7, | [0.1, |
|  | 11.5] | 16.1] | 7.2] | 4.2] | 3.1] | 1.0] | 4.2] | 0.9] | 2.9] | 1.0] | 5.1] | 0.3] |
|  | 16.8 | 12.8 | 17.9 | 5.7 | 5.2 | 2.3 | 6.4 | 2.9 | 1.2 | 0.9 | 4.1 | 1.0 |
| phikon | [10.2, | [9.3, | [14.0, | [5.1, | [4.7, | [2.2, | [4.7, | [2.6, | [0.9, | [0.8, | [3.8, | [1.0, |
|  | 24.8] | 16.3] | 23.1] | 6.5] | 5.7] | 2.4] | 9.0] | 3.1] | 1.7] | 1.1] | 4.4] | 1.1] |
|  | 5.7 | 10.1 | 17.5 | 9.3 | 4.8 | 0.8 | 2.6 | 0.6 | 0.9 | 0.9 | 0.8 | 1.6 |
| phikon2 | [2.6, | [7.0, | [13.7, | [8.5, | [4.3, | [0.7, | [1.4, | [0.4, | [0.4, | [0.7, | [0.6, | [1.5, |
|  | 14.2] | 13.6] | 22.8] | 10.2] | 5.3] | 0.9] | 5.1] | 0.9] | 1.3] | 1.0] | 1.0] | 1.7] |
|  | 5.5 | 15.3 | 9.5 | 3.8 | 4.2 | 1.9 | 3.1 | 2.5 | 1.1 | 1.2 | 2.4 | 1.0 |
| uni | [3.0, | [11.6, | [5.7, | [3.2, | [3.8, | [1.8, | [1.6, | [2.3, | [0.8, | [1.0, | [2.1, | [0.9, |
|  | 13.3] | 18.9] | 14.0] | 4.4] | 4.7] | 2.0] | 5.4] | 2.8] | 1.5] | 1.4] | 2.6] | 1.1] |
|  | 8.4 | 6.2 | 7.1 | 8.1 | 3.9 | 1.2 | 6.9 | 1.2 | 5.0 | 1.1 | 3.6 | 1.0 |
| uni2h | [4.0, | [4.1, | [4.3, | [7.4, | [3.5, | [1.1, | [4.8, | [1.0, | [4.6, | [1.0, | [3.3, | [1.0, |
|  | 14.1] | 10.1] | 10.7] | 8.8] | 4.4] | 1.3] | 9.3] | 1.4] | 5.4] | 1.3] | 3.8] | 1.1] |
|  | 20.3 | 11.7 | 12.8 | 2.1 | 3.8 | 0.5 | 3.5 | 1.5 | 2.6 | 0.8 | 5.8 | 0.6 |
| virchow | [13.5, | [8.2, | [8.6, | [1.6, | [3.3, | [0.4, | [1.8, | [1.3, | [2.1, | [0.6, | [5.5, | [0.5, |
|  | 28.0] | 15.5] | 17.1] | 2.7] | 4.3] | 0.6] | 5.8] | 1.7] | 3.0] | 1.0] | 6.1] | 0.7] |
|  | 9.6 | 6.3 | 5.7 | 4.3 | 4.1 | 0.5 | 0.6 | 4.3 | 5.5 | 0.3 | 7.2 | 7.0 |
| virchow2 | [7.2, | [3.7, | [2.8, | [3.9, | [3.7, | [0.4, | [0.5, | [4.0, | [5.1, | [0.2, | [7.0, | [6.8, |
|  | 16.8] | 10.0] | 9.5] | 4.9] | 4.6] | 0.6] | 3.1] | 4.5] | 5.9] | 0.5] | 7.5] | 7.1] |
|  | 7.6 | 15.8 | 7.0 | 2.2 | 3.2 | 3.2 | 3.2 | 0.6 | 4.4 | 1.1 | 2.2 | 0.6 |
| conch | [3.2, | [12.5, | [3.4, | [1.6, | [2.7, | [3.1, | [1.6, | [0.3, | [4.0, | [0.9, | [1.9, | [0.5, |
|  | 13.9] | 19.8] | 11.5] | 3.0] | 3.6] | 3.3] | 5.5] | 0.9] | 4.8] | 1.3] | 2.5] | 0.7] |
|  | 13.2 | 11.6 | 11.0 | 3.3 | 3.0 | 2.8 | 0.7 | 0.3 | 6.9 | 0.3 | 5.2 | 0.4 |
| titan | [8.1, | [8.3, | [7.4, | [2.7, | [2.5, | [2.7, | [0.5, | [0.1, | [6.4, | [0.1, | [4.9, | [0.3, |
|  | 19.5] | 15.2] | 15.1] | 4.1] | 3.4] | 2.9] | 3.2] | 0.6] | 7.4] | 0.4] | 5.5] | 0.5] |
|  | 25.7 | 6.9 | 6.6 | 3.8 | 2.5 | 1.3 | 0.8 | 1.5 | 2.4 | 0.2 | 4.4 | 0.7 |
| keep | [21.1, | [3.9, | [3.3, | [3.3, | [2.1, | [1.2, | [0.5, | [1.3, | [2.0, | [0.1, | [4.2, | [0.6, |
|  | 32.4] | 10.8] | 11.1] | 4.4] | 2.9] | 1.4] | 3.6] | 1.8] | 2.9] | 0.4] | 4.7] | 0.8] |
|  | 13.6 | 10.3 | 1.7 | 3.1 | 4.1 | 3.3 | 7.0 | 4.0 | 2.4 | 0.3 | 2.7 | 1.7 |
| musk | [8.6, | [7.2, | [1.0, | [2.6, | [3.6, | [3.2, | [5.2, | [3.7, | [2.0, | [0.2, | [2.4, | [1.6, |
|  | 21.1] | 14.1] | 5.7] | 3.9] | 4.6] | 3.4] | 9.3] | 4.3] | 2.9] | 0.5] | 3.0] | 1.9] |
|  | 7.8 | 11.6 | 7.3 | 5.9 | 6.1 | 4.6 | 1.7 | 1.3 | 2.7 | 0.1 | 6.3 | 3.5 |
| plip | [3.6, | [8.1, | [3.8, | [4.9, | [5.5, | [4.4, | [0.7, | [1.0, | [2.3, | [0.0, | [6.0, | [3.3, |
|  | 15.0] | 15.6] | 12.2] | 6.9] | 6.8] | 4.8] | 4.4] | 1.7] | 3.2] | 0.3] | 6.7] | 3.7] |
|  | 15.2 | 11.7 | 7.3 | 20.5 | 3.0 | 7.3 | 1.8 | 1.4 | 6.0 | 0.3 | 7.9 | 1.2 |
| quilt | [8.6, | [8.4, | [4.8, | [19.3, | [2.5, | [7.1, | [0.7, | [1.1, | [5.6, | [0.1, | [7.5, | [1.1, |
|  | 22.9] | 15.6] | 12.5] | 21.6] | 3.5] | 7.4] | 4.3] | 1.8] | 6.5] | 0.5] | 8.2] | 1.3] |
|  | 16.1 | 20.5 | 6.8 | 8.3 | 3.2 | 1.1 | 0.6 | 3.4 | 2.0 | 0.9 | 2.2 | 1.2 |
| dinob | [10.5, | [16.6, | [3.6, | [7.4, | [2.8, | [1.0, | [0.5, | [3.1, | [1.7, | [0.8, | [1.9, | [1.0, |
|  | 24.3] | 24.4] | 10.2] | 9.2] | 3.8] | 1.2] | 3.1] | 3.8] | 2.5] | 1.1] | 2.5] | 1.4] |
|  | 3.7 | 10.2 | 5.6 | 5.1 | 5.2 | 6.9 | 5.5 | 2.6 | 2.6 | 1.2 | 10.0 | 4.5 |
| dinol | [2.3, | [6.7, | [2.6, | [4.3, | [4.7, | [6.8, | [3.7, | [2.2, | [2.2, | [1.0, | [9.7, | [4.4, |
|  | 12.1] | 14.0] | 9.2] | 5.9] | 5.8] | 7.1] | 7.9] | 2.9] | 3.1] | 1.4] | 10.4] | 4.7] |
|  | 1.8 | 8.7 | 8.1 | 1.9 | 3.5 | 7.5 | 2.2 | 3.8 | 2.6 | 0.8 | 5.1 | 0.7 |
| vitb | [2.1, | [5.8, | [4.6, | [1.2, | [3.0, | [7.3, | [1.0, | [3.5, | [2.1, | [0.6, | [4.8, | [0.6, |
|  | 12.2] | 12.7] | 12.4] | 2.8] | 4.0] | 7.6] | 4.8] | 4.1] | 3.1] | 1.0] | 5.5] | 0.9] |
|  | 15.0 | 11.8 | 8.4 | 7.7 | 2.6 | 2.2 | 1.4 | 4.1 | 3.1 | 0.2 | 2.4 | 1.2 |
| vitl | [9.3, | [8.3, | [5.1, | [6.7, | [2.2, | [2.0, | [0.6, | [3.8, | [2.6, | [0.1, | [2.1, | [1.1, |
|  | 23.6] | 15.5] | 12.7] | 8.6] | 3.0] | 2.3] | 4.5] | 3.9] | 3.6] | 0.4] | 2.7] | 1.4] |
|  | 12.3 | 8.2 | 6.2 | 3.3 | 5.2 | 7.9 | 1.0 | 2.9 | 8.4 | 0.2 | 2.5 | 7.5 |
| clipb | [5.8, | [5.2, | [2.9, | [2.3, | [4.6, | [7.8, | [0.5, | [2.5, | [7.9, | [0.1, | [2.2, | [7.3, |
|  | 20.6] | 11.9] | 11.0] | 4.3] | 5.8] | 8.1] | 3.8] | 3.3] | 9.0] | 0.4] | 2.9] | 7.7] |
|  | 3.7 | 7.9 | 5.5 | 7.9 | 5.2 | 5.0 | 1.0 | 0.4 | 5.7 | 0.1 | 6.8 | 1.1 |
| clipl | [2.3, | [4.5, | [3.1, | [6.9, | [4.6, | [4.8, | [0.5, | [0.2, | [5.2, | [0.0, | [6.5, | [1.0, |
|  | 12.5] | 11.8] | 10.9] | 8.8] | 5.8] | 5.1] | 3.5] | 0.8] | 6.2] | 0.3] | 7.2] | 1.3] |

Table S53: Quantitative performance (MCE) on linear probing.

| Model | bach | bracs | break-h | ccrcc | crc | esca | mhist | pcam | tcga-crc | tcga-tils | tcga-unif | wilds |
|---|---|---|---|---|---|---|---|---|---|---|---|---|
| | 22.1 | 5.1 | 7.8 | 5.2 | 36.3 | 19.9 | 5.1 | 1.9 | 0.9 | 2.2 | 7.2 | 4.1 |
| hiboub | [16.6, | [3.2, | [5.1, | [3.7, | [15.9, | [2.7, | [3.6, | [1.7, | [0.6, | [1.1, | [6.4, | [2.5, |
| | 32.4] | 14.3] | 33.4] | 16.5] | 37.0] | 19.9] | 14.4] | 4.0] | 1.8] | 3.9] | 10.3] | 6.5] |
| | 11.7 | 22.9 | 35.0 | 20.6 | 19.8 | 18.5 | 8.9 | 4.8 | 4.8 | 1.2 | 6.0 | 0.7 |
| hiboul | [8.5, | [17.4, | [8.2, | [19.3, | [13.1, | [6.4, | [4.7, | [2.9, | [4.0, | [0.6, | [3.7, | [0.3, |
| | 62.4] | 37.2] | 35.4] | 22.0] | 32.8] | 18.5] | 20.5] | 8.9] | 5.5] | 3.5] | 13.2] | 4.7] |
| | 15.7 | 19.9 | 5.4 | 17.6 | 38.3 | 5.4 | 3.5 | 4.2 | 3.6 | 1.6 | 11.6 | 15.7 |
| hopt0 | [13.1, | [13.9, | [3.8, | [9.3, | [19.6, | [4.5, | [1.3, | [2.9, | [3.2, | [0.5, | [6.3, | [14.4, |
| | 62.5] | 28.0] | 33.3] | 51.2] | 40.3] | 6.3] | 12.7] | 8.1] | 4.1] | 4.2] | 17.4] | 17.0] |
| | 12.1 | 17.8 | 43.5 | 50.0 | 15.8 | 19.4 | 23.5 | 11.4 | 2.4 | 2.5 | 4.9 | 9.3 |
| hopt1 | [6.4, | [14.4, | [10.3, | [21.8, | [6.3, | [9.0, | [11.7, | [8.9, | [1.7, | [1.3, | [3.9, | [7.7, |
| | 62.2] | 28.5] | 64.5] | 64.0] | 37.4] | 19.4] | 34.7] | 13.8] | 3.2] | 4.2] | 16.0] | 10.9] |
| | 34.1 | 17.9 | 16.5 | 3.8 | 38.3 | 1.1 | 1.9 | 2.2 | 3.9 | 2.6 | 18.9 | 4.5 |
| midnight | [11.7, | [13.4, | [4.1, | [3.2, | [5.3, | [1.0, | [1.4, | [1.1, | [2.6, | [1.4, | [14.4, | [1.4, |
| | 47.0] | 31.8] | 44.0] | 7.4] | 38.3] | 9.6] | 11.6] | 5.3] | 5.4] | 4.2] | 19.1] | 9.0] |
| | 38.7 | 18.3 | 38.7 | 11.4 | 36.4 | 6.1 | 7.2 | 3.0 | 1.5 | 3.2 | 8.6 | 8.0 |
| phikon | [17.2, | [14.2, | [19.1, | [9.9, | [30.4, | [3.5, | [5.6, | [2.8, | [1.1, | [1.0, | [7.6, | [6.5, |
| | 39.0] | 28.2] | 38.7] | 21.5] | 42.4] | 81.9] | 14.6] | 4.1] | 2.3] | 5.7] | 19.0] | 9.5] |
| | 9.4 | 22.9 | 33.9 | 17.8 | 22.3 | 16.6 | 3.1 | 2.1 | 1.5 | 1.0 | 3.4 | 5.3 |
| phikon2 | [5.4, | [15.0, | [25.2, | [16.2, | [12.1, | [4.4, | [2.1, | [0.9, | [0.7, | [0.8, | [1.5, | [3.5, |
| | 38.9] | 31.7] | 44.6] | 19.4] | 32.9] | 33.1] | 11.6] | 3.7] | 2.7] | 3.1] | 9.6] | 7.1] |
| | 10.4 | 18.9 | 16.3 | 23.6 | 20.8 | 12.1 | 3.7 | 3.1 | 1.7 | 4.1 | 5.2 | 5.2 |
| uni | [6.0, | [14.8, | [10.8, | [6.9, | [9.5, | [10.5, | [2.4, | [2.4, | [1.2, | [2.8, | [4.3, | [3.2, |
| | 34.7] | 31.0] | 62.1] | 45.4] | 34.3] | 23.2] | 15.1] | 5.5] | 2.2] | 6.0] | 11.0] | 7.4] |
| | 37.0 | 19.3 | 29.4 | 16.4 | 37.8 | 12.1 | 8.5 | 1.2 | 7.7 | 4.2 | 7.6 | 9.0 |
| uni2h | [9.6, | [8.6, | [7.5, | [15.0, | [15.6, | [8.5, | [5.8, | [1.1, | [7.3, | [2.7, | [6.8, | [6.9, |
| | 37.0] | 32.5] | 32.9] | 18.6] | 37.8] | 15.7] | 19.2] | 5.4] | 8.2] | 5.7] | 22.4] | 11.0] |
| | 64.5 | 14.0 | 44.1 | 9.7 | 35.3 | 81.4 | 6.2 | 2.5 | 4.0 | 2.1 | 11.1 | 1.6 |
| virchow | [26.8, | [11.1, | [20.4, | [3.0, | [9.1, | [2.2, | [2.8, | [1.4, | [3.3, | [1.2, | [10.2, | [0.6, |
| | 65.8] | 21.1] | 66.1] | 33.3] | 35.3] | 81.4] | 14.9] | 6.0] | 4.7] | 4.8] | 18.7] | 4.4] |
| | 15.0 | 10.4 | 37.8 | 37.5 | 35.4 | 19.2 | 2.5 | 8.8 | 6.5 | 2.5 | 17.9 | 18.9 |
| virchow2 | [13.2, | [5.8, | [8.3, | [9.9, | [11.8, | [8.4, | [1.1, | [7.0, | [6.0, | [1.4, | [15.9, | [17.3, |
| | 64.0] | 18.9] | 38.1] | 61.6] | 54.3] | 19.2] | 11.6] | 13.1] | 7.2] | 5.2] | 18.3] | 20.6] |
| | 22.3 | 22.0 | 11.9 | 8.6 | 36.2 | 7.6 | 6.2 | 1.8 | 6.6 | 4.7 | 3.9 | 3.8 |
| conch | [7.6, | [16.1, | [7.1, | [2.8, | [12.3, | [6.8, | [2.4, | [0.7, | [6.0, | [3.4, | [3.5, | [2.8, |
| | 41.5] | 31.5] | 61.9] | 34.8] | 36.5] | 8.7] | 15.0] | 3.7] | 7.2] | 6.3] | 9.3] | 6.4] |
| | 36.7 | 17.7 | 32.7 | 8.2 | 19.5 | 8.5 | 0.7 | 1.0 | 7.9 | 1.2 | 9.8 | 1.1 |
| titan | [14.6, | [11.6, | [12.2, | [5.4, | [10.5, | [7.5, | [0.9, | [0.3, | [7.4, | [0.4, | [8.8, | [0.4, |
| | 38.3] | 26.3] | 35.6] | 30.1] | 38.4] | 9.6] | 10.0] | 3.6] | 8.5] | 3.0] | 12.2] | 4.1] |
| | 33.6 | 7.5 | 8.3 | 18.1 | 38.0 | 16.9 | 2.1 | 1.6 | 3.7 | 2.8 | 9.7 | 4.7 |
| keep | [32.2, | [6.0, | [6.6, | [7.2, | [14.0, | [7.2, | [0.9, | [1.4, | [3.1, | [0.8, | [7.6, | [2.7, |
| | 41.3] | 18.6] | 38.1] | 38.5] | 38.9] | 16.9] | 9.5] | 3.8] | 4.3] | 5.2] | 18.6] | 6.9] |
| | 35.7 | 12.3 | 8.1 | 7.1 | 11.5 | 7.0 | 8.0 | 6.9 | 3.8 | 1.4 | 5.3 | 7.8 |
| musk | [13.3, | [9.8, | [4.5, | [3.6, | [7.7, | [6.3, | [6.3, | [5.4, | [3.2, | [0.5, | [4.5, | [5.9, |
| | 39.2] | 22.0] | 61.2] | 19.2] | 60.9] | 7.7] | 16.5] | 8.8] | 4.5] | 3.1] | 8.7] | 9.8] |
| | 34.7 | 15.2 | 13.0 | 21.8 | 14.4 | 18.8 | 2.7 | 2.1 | 9.5 | 0.9 | 10.9 | 6.0 |
| plip | [32.5, | [11.5, | [7.6, | [10.2, | [11.0, | [18.3, | [1.3, | [1.3, | [8.4, | [0.2, | [9.9, | [5.0, |
| | 36.8] | 24.4] | 31.6] | 34.7] | 39.5] | 19.2] | 9.4] | 4.0] | 10.6] | 2.9] | 11.9] | 7.0] |
| | 21.2 | 16.0 | 11.0 | 29.6 | 24.0 | 81.3 | 2.2 | 2.8 | 7.2 | 2.2 | 12.6 | 3.2 |
| quilt | [14.9, | [11.6, | [8.0, | [21.0, | [9.7, | [15.6, | [1.2, | [1.7, | [6.6, | [0.9, | [11.6, | [2.3, |
| | 39.9] | 23.7] | 43.0] | 64.3] | 39.5] | 81.3] | 8.5] | 4.1] | 7.8] | 4.3] | 13.6] | 4.2] |
| | 35.6 | 26.0 | 64.6 | 13.1 | 20.7 | 19.4 | 3.9 | 3.9 | 7.0 | 1.5 | 3.8 | 1.7 |
| dinob | [23.7, | [22.0, | [14.3, | [10.4, | [8.6, | [1.2, | [1.1, | [3.3, | [6.1, | [0.9, | [3.2, | [1.2, |
| | 39.1] | 32.6] | 64.6] | 30.3] | 39.3] | 19.5] | 14.3] | 5.2] | 8.0] | 3.4] | 4.9] | 3.4] |
| | 4.1 | 17.9 | 61.2 | 11.9 | 39.2 | 19.7 | 8.4 | 2.6 | 6.4 | 2.2 | 16.2 | 7.8 |
| dinol | [5.5, | [11.4, | [6.9, | [10.1, | [7.8, | [12.3, | [4.6, | [2.4, | [5.6, | [1.2, | [15.2, | [6.6, |
| | 40.2] | 25.6] | 61.2] | 22.9] | 39.8] | 19.7] | 16.6] | 4.0] | 7.3] | 4.4] | 17.2] | 9.0] |
| | 3.5 | 15.8 | 35.5 | 3.2 | 11.0 | 12.0 | 5.3 | 5.4 | 5.9 | 3.4 | 7.5 | 1.0 |
| vitb | [4.4, | [9.8, | [16.4, | [2.0, | [7.2, | [11.4, | [2.1, | [4.1, | [4.9, | [1.2, | [6.6, | [0.6, |
| | 25.0] | 24.7] | 38.2] | 16.3] | 38.3] | 12.5] | 13.3] | 6.6] | 6.8] | 4.7] | 8.5] | 2.0] |
| | 39.0 | 18.2 | 15.9 | 14.8 | 16.8 | 19.1 | 3.6 | 6.4 | 4.2 | 0.8 | 4.2 | 1.6 |
| vitl | [19.0, | [11.8, | [8.4, | [12.6, | [5.4, | [5.3, | [1.2, | [5.0, | [3.6, | [0.3, | [3.7, | [1.1, |
| | 39.8] | 26.1] | 38.8] | 35.5] | 37.4] | 19.9] | 11.9] | 7.7] | 4.9] | 3.0] | 6.5] | 2.7] |
| | 17.1 | 11.9 | 6.8 | 12.2 | 15.0 | 47.8 | 2.7 | 4.1 | 31.0 | 0.7 | 3.9 | 12.5 |
| clipb | [10.4, | [8.3, | [5.3, | [4.6, | [11.5, | [12.6, | [1.1, | [3.1, | [28.5, | [0.2, | [3.3, | [11.5, |
| | 36.3] | 19.9] | 19.8] | 24.6] | 20.1] | 80.7] | 9.9] | 5.2] | 33.5] | 2.3] | 4.9] | 13.4] |
| | 10.9 | 9.8 | 13.2 | 16.9 | 27.1 | 9.9 | 3.5 | 1.0 | 9.8 | 0.5 | 11.2 | 2.8 |
| clipl | [4.8, | [7.2, | [6.9, | [13.2, | [13.8, | [9.2, | [1.0, | [0.4, | [8.5, | [0.2, | [10.2, | [1.6, |
| | 38.0] | 17.8] | 32.9] | 32.0] | 36.8] | 10.6] | 12.5] | 2.8] | 11.1] | 2.4] | 12.1] | 3.9] |

Table S54: Quantitative performance (ACE) on linear probing.

| Model | bach | bracs | break-h | ccrcc | crc | esca | mhist | pcam | tcga-crc | tcga-tils | tcga-unif | wilds |
|---|---|---|---|---|---|---|---|---|---|---|---|---|
| hiboub | 16.0 [11.2, 24.5] | 4.0 [2.6, 8.7] | 5.5 [3.9, 11.1] | 3.7 [2.9, 4.5] | 2.8 [2.3, 3.2] | 1.4 [1.3, 1.6] | 4.5 [2.7, 6.9] | 1.8 [1.6, 2.0] | 0.8 [0.5, 1.2] | 0.5 [0.3, 0.7] | 3.9 [3.6, 4.2] | 1.0 [0.9, 1.1] |
| hiboul | 7.9 [4.3, 15.6] | 18.0 [14.3, 21.7] | 5.3 [3.1, 10.1] | 11.9 [11.2, 12.6] | 4.2 [3.7, 4.6] | 3.0 [2.9, 3.1] | 4.9 [3.0, 7.1] | 2.8 [2.5, 3.0] | 3.2 [2.8, 3.6] | 0.7 [0.5, 0.9] | 2.1 [1.8, 2.4] | 0.1 [0.1, 0.2] |
| hopt0 | 13.4 [7.5, 21.1] | 14.5 [10.8, 18.1] | 2.7 [2.5, 8.3] | 6.2 [5.6, 6.9] | 4.0 [3.5, 4.5] | 2.9 [2.7, 3.0] | 0.7 [0.9, 3.6] | 2.9 [2.7, 3.2] | 3.1 [2.7, 3.6] | 0.3 [0.2, 0.5] | 3.1 [2.8, 3.3] | 2.0 [1.9, 2.0] |
| hopt1 | 4.8 [3.9, 14.5] | 15.1 [11.5, 18.7] | 1.8 [1.5, 7.3] | 4.5 [3.8, 5.1] | 3.3 [2.9, 3.8] | 1.3 [1.2, 1.4] | 9.2 [6.9, 11.3] | 2.8 [2.6, 3.1] | 1.7 [1.3, 2.2] | 0.3 [0.2, 0.5] | 1.8 [1.6, 2.1] | 0.8 [0.7, 0.9] |
| midnight | 5.9 [3.4, 13.0] | 12.3 [9.0, 16.0] | 3.7 [2.5, 9.6] | 3.5 [2.8, 4.2] | 2.6 [2.2, 3.0] | 0.8 [0.7, 0.9] | 1.6 [1.0, 4.3] | 0.9 [0.7, 1.1] | 3.4 [3.0, 3.8] | 0.5 [0.4, 0.7] | 4.9 [4.7, 5.1] | 0.1 [0.1, 0.2] |
| phikon | 13.9 [9.2, 23.5] | 12.8 [9.3, 16.4] | 18.1 [13.9, 23.0] | 5.7 [5.0, 6.4] | 5.2 [4.7, 5.7] | 2.2 [2.1, 2.4] | 6.4 [4.5, 8.9] | 2.8 [2.5, 3.1] | 1.3 [0.9, 1.8] | 0.9 [0.8, 1.1] | 4.1 [3.8, 4.4] | 1.4 [1.3, 1.5] |
| phikon2 | 5.4 [3.9, 16.1] | 10.1 [7.0, 13.6] | 17.9 [13.9, 23.0] | 9.1 [8.3, 10.0] | 4.8 [4.3, 5.3] | 1.3 [1.2, 1.4] | 2.2 [1.4, 4.9] | 0.6 [0.3, 0.9] | 0.9 [0.5, 1.4] | 0.8 [0.7, 1.0] | 0.8 [0.6, 1.0] | 1.6 [1.5, 1.7] |
| uni | 5.6 [4.5, 14.2] | 15.3 [11.6, 18.9] | 10.1 [6.0, 14.3] | 3.8 [3.2, 4.5] | 4.2 [3.8, 4.7] | 2.5 [2.4, 2.6] | 2.5 [1.6, 5.1] | 2.5 [2.3, 2.8] | 1.0 [0.8, 1.5] | 1.2 [1.0, 1.4] | 2.4 [2.1, 2.6] | 1.0 [0.9, 1.1] |
| uni2h | 8.4 [4.8, 14.8] | 6.7 [4.0, 10.4] | 7.3 [4.1, 10.8] | 8.0 [7.3, 8.8] | 3.9 [3.5, 4.4] | 2.1 [2.0, 2.2] | 7.0 [4.8, 9.3] | 1.2 [1.0, 1.4] | 5.0 [4.6, 5.4] | 1.1 [1.0, 1.3] | 3.6 [3.3, 3.8] | 1.0 [1.0, 1.1] |
| virchow | 18.4 [11.6, 26.0] | 11.7 [8.3, 15.5] | 11.6 [7.4, 16.0] | 2.1 [1.5, 2.7] | 3.8 [3.3, 4.3] | 0.8 [0.7, 0.9] | 3.5 [1.9, 5.7] | 1.5 [1.2, 1.7] | 2.6 [2.1, 3.0] | 0.8 [0.6, 1.0] | 5.8 [5.5, 6.1] | 0.6 [0.5, 0.7] |
| virchow2 | 11.8 [8.9, 18.4] | 6.3 [4.2, 10.4] | 6.0 [3.3, 10.2] | 4.3 [3.8, 4.9] | 4.1 [3.6, 4.6] | 1.2 [1.1, 1.3] | 1.2 [0.8, 3.6] | 4.3 [4.0, 4.5] | 5.5 [5.1, 5.9] | 0.5 [0.3, 0.7] | 7.2 [7.0, 7.5] | 7.0 [6.8, 7.1] |
| conch | 5.9 [3.2, 13.7] | 15.7 [12.2, 19.5] | 5.8 [3.9, 11.1] | 2.1 [1.4, 2.8] | 3.2 [2.7, 3.6] | 3.2 [3.1, 3.3] | 2.7 [1.7, 5.5] | 0.5 [0.3, 0.8] | 4.4 [4.0, 4.8] | 1.1 [1.0, 1.3] | 2.2 [1.9, 2.5] | 0.7 [0.6, 0.8] |
| titan | 12.2 [9.0, 19.5] | 11.6 [8.3, 15.1] | 11.1 [7.2, 15.2] | 3.3 [2.6, 4.0] | 3.0 [2.5, 3.4] | 2.8 [2.7, 2.9] | 2.5 [1.5, 4.9] | 0.3 [0.2, 0.6] | 6.9 [6.4, 7.4] | 0.3 [0.1, 0.4] | 5.2 [4.9, 5.5] | 0.4 [0.3, 0.5] |
| keep | 26.2 [22.2, 32.9] | 6.5 [3.9, 10.3] | 6.6 [3.6, 11.2] | 3.8 [3.2, 4.4] | 2.4 [2.0, 2.8] | 1.8 [1.7, 1.9] | 1.5 [0.9, 4.0] | 1.5 [1.2, 1.7] | 2.4 [2.0, 2.9] | 0.2 [0.1, 0.3] | 4.4 [4.2, 4.7] | 0.7 [0.6, 0.8] |
| musk | 14.0 [8.4, 22.2] | 10.3 [7.3, 14.0] | 2.6 [1.5, 6.3] | 3.2 [2.5, 4.0] | 4.1 [3.5, 4.6] | 3.3 [3.1, 3.4] | 6.3 [4.7, 8.7] | 4.0 [3.7, 4.3] | 2.6 [2.2, 3.1] | 0.3 [0.1, 0.5] | 2.7 [2.4, 3.0] | 1.7 [1.6, 1.8] |
| plip | 7.9 [4.3, 16.0] | 11.6 [8.2, 15.6] | 8.4 [4.9, 13.3] | 5.9 [4.9, 6.9] | 6.1 [5.5, 6.8] | 4.6 [4.4, 4.8] | 1.2 [0.9, 4.1] | 1.1 [0.8, 1.4] | 3.0 [2.6, 3.5] | 0.1 [0.1, 0.3] | 6.3 [6.0, 6.7] | 3.5 [3.3, 3.7] |
| quilt | 16.3 [9.9, 23.4] | 11.7 [8.4, 15.5] | 7.5 [5.2, 13.4] | 20.4 [19.3, 21.5] | 3.0 [2.5, 3.5] | 7.3 [7.1, 7.4] | 2.0 [1.3, 5.1] | 1.4 [1.1, 1.7] | 6.0 [5.6, 6.5] | 0.1 [0.1, 0.3] | 7.9 [7.5, 8.2] | 1.2 [1.1, 1.3] |
| dinob | 14.3 [9.8, 23.5] | 20.5 [16.7, 24.5] | 6.5 [3.5, 10.0] | 8.3 [7.3, 9.2] | 3.2 [2.7, 3.8] | 1.0 [0.9, 1.2] | 1.9 [1.1, 4.2] | 3.4 [3.1, 3.8] | 2.1 [1.8, 2.7] | 0.9 [0.7, 1.1] | 2.2 [1.9, 2.5] | 1.2 [1.0, 1.4] |
| dinol | 2.7 [3.0, 14.3] | 10.2 [6.7, 14.0] | 5.4 [2.6, 9.0] | 5.1 [4.3, 5.9] | 5.2 [4.6, 5.8] | 6.9 [6.8, 7.1] | 5.3 [3.5, 7.6] | 2.3 [2.0, 2.7] | 2.6 [2.2, 3.1] | 1.1 [0.9, 1.3] | 10.0 [9.7, 10.4] | 4.5 [4.4, 4.7] |
| vitb | 1.8 [3.2, 13.5] | 7.8 [4.9, 11.7] | 8.2 [4.7, 12.6] | 1.8 [1.2, 2.7] | 3.5 [2.9, 4.0] | 7.5 [7.3, 7.6] | 3.1 [1.5, 5.3] | 3.8 [3.4, 4.1] | 2.8 [2.4, 3.3] | 0.8 [0.6, 1.0] | 5.1 [4.8, 5.5] | 0.7 [0.6, 0.9] |
| vitl | 15.0 [9.4, 23.6] | 11.8 [8.2, 15.5] | 8.4 [5.0, 12.7] | 7.5 [6.6, 8.5] | 2.6 [2.1, 3.0] | 2.2 [2.0, 2.3] | 1.9 [1.2, 4.5] | 4.1 [3.8, 4.5] | 3.1 [2.6, 3.6] | 0.2 [0.1, 0.4] | 2.4 [2.1, 2.7] | 1.2 [1.0, 1.4] |
| clipb | 12.3 [6.9, 20.9] | 8.2 [5.2, 12.0] | 7.4 [4.5, 12.1] | 3.5 [2.5, 4.5] | 5.2 [4.5, 5.8] | 7.9 [7.8, 8.1] | 1.8 [1.2, 4.7] | 2.9 [2.5, 3.3] | 8.4 [7.9, 9.0] | 0.1 [0.1, 0.4] | 2.5 [2.2, 2.9] | 7.5 [7.3, 7.7] |
| clipl | 4.1 [3.4, 14.4] | 7.9 [5.0, 11.8] | 7.1 [3.9, 12.5] | 7.6 [6.6, 8.6] | 5.2 [4.6, 5.8] | 5.0 [4.8, 5.1] | 1.2 [0.8, 4.1] | 0.3 [0.3, 0.8] | 5.7 [5.2, 6.2] | 0.2 [0.1, 0.5] | 6.8 [6.5, 7.2] | 1.1 [0.9, 1.2] |

Table S55: Quantitative performance (TACE) on linear probing.

| Model | bach | bracs | break-h | ccrcc | crc | esca | mhist | pcam | tcga-crc | tcga-tils | tcga-unif | wilds |
|---|---|---|---|---|---|---|---|---|---|---|---|---|
|  | 16.0 | 4.0 | 5.5 | 3.7 | 2.8 | 1.4 | 4.5 | 1.8 | 0.8 | 0.5 | 3.9 | 1.0 |
| hiboub | [11.2, | [2.6, | [3.9, | [2.9, | [2.3, | [1.3, | [2.7, | [1.6, | [0.5, | [0.3, | [3.6, | [0.9, |
|  | 24.5] | 8.7] | 11.1] | 4.5] | 3.2] | 1.6] | 6.9] | 2.0] | 1.2] | 0.7] | 4.2] | 1.1] |
|  | 7.9 | 18.0 | 5.3 | 11.9 | 4.2 | 3.0 | 4.9 | 2.8 | 3.2 | 0.7 | 2.1 | 0.1 |
| hiboul | [4.3, | [14.3, | [3.1, | [11.2, | [3.7, | [2.9, | [3.0, | [2.5, |  | [0.5, | [1.8, | [0.1, |
|  | 15.6] | 21.7] | 10.1] | 12.6] | 4.6] | 3.1] | 7.1] | 3.0] | 3.6] | 0.9] | 2.4] | 0.2] |
|  | 13.4 | 14.5 | 2.7 | 6.2 | 4.0 | 2.9 | 0.7 | 2.9 | 3.1 | 0.3 | 3.1 | 2.0 |
| hopt0 | [7.5, | [10.8, | [2.5, | [5.6, | [3.5, | [2.7, | [0.9, | [2.7, | [2.7, | [0.2, | [2.8, | [1.9, |
|  | 21.1] | 18.1] | 8.3] | 6.9] | 4.5] | 3.0] | 3.6] | 3.2] | 3.6] | 0.5] | 3.3] | 2.0] |
|  | 4.8 | 15.1 | 1.8 | 4.5 | 3.3 | 1.3 | 9.2 | 2.8 | 1.7 | 0.3 | 1.8 | 0.8 |
| hopt1 | [3.9, | [11.5, | [1.5, | [3.8, | [2.9, | [1.2, | [6.9, | [2.6, | [1.3, | [0.2, | [1.6, | [0.7, |
|  | 14.5] | 18.7] | 7.3] | 5.1] | 3.8] | 1.4] | 11.3] | 3.1] | 2.2] | 0.5] | 2.1] | 0.9] |
|  | 5.9 | 12.3 | 3.7 | 3.5 | 2.6 | 0.8 | 1.6 | 0.9 | 3.4 | 0.5 | 4.9 | 0.1 |
| midnight | [3.4, | [9.0, | [2.5, | [2.8, | [2.2, | [0.7, | [1.0, | [0.7, | [3.0, | [0.4, | [4.7, | [0.1, |
|  | 13.0] | 16.0] | 9.6] | 4.2] | 3.0] | 0.9] | 4.3] | 1.1] | 3.8] | 0.7] | 5.1] | 0.2] |
|  | 13.9 | 12.8 | 18.1 | 5.7 | 5.2 | 2.2 | 6.4 | 2.8 | 1.3 | 0.9 | 4.1 | 1.4 |
| phikon | [9.2, | [9.3, | [13.9, | [5.0, | [4.7, | [2.1, | [4.5, | [2.5, | [0.9, | [0.8, | [3.8, | [1.3, |
|  | 23.5] | 16.4] | 23.0] | 6.4] | 5.7] | 2.4] | 8.9] | 3.1] | 1.8] | 1.1] | 4.4] | 1.5] |
|  | 5.4 | 10.1 | 17.9 | 9.1 | 4.8 | 1.3 | 2.2 | 0.6 | 0.9 | 0.8 | 0.8 | 1.6 |
| phikon2 | [3.9, | [7.0, | [13.9, | [8.3, | [4.3, | [1.2, | [1.4, | [0.3, | [0.5, | [0.7, | [0.6, | [1.5, |
|  | 16.1] | 13.6] | 23.0] | 10.0] | 5.3] | 1.4] | 4.9] | 0.9] | 1.4] | 1.0] | 1.0] | 1.7] |
|  | 5.6 | 15.3 | 10.1 | 3.8 | 4.2 | 2.5 | 2.5 | 2.5 | 1.0 | 1.2 | 2.4 | 1.0 |
| uni | [4.5, | [11.6, | [6.0, | [3.2, | [3.8, | [2.4, | [1.6, | [2.3, | [0.8, | [1.0, | [2.1, | [0.9, |
|  | 14.2] | 18.9] | 14.3] | 4.5] | 4.7] | 2.6] | 5.1] | 2.8] | 1.5] | 1.4] | 2.6] | 1.1] |
|  | 8.4 | 6.7 | 7.3 | 8.0 | 3.9 | 2.1 | 7.0 | 1.2 | 5.0 | 1.1 | 3.6 | 1.0 |
| uni2h | [4.8, | [4.0, | [4.1, | [7.3, | [3.5, | [2.0, | [4.8, | [1.0, | [4.6, | [1.0, | [3.3, | [1.0, |
|  | 14.8] | 10.4] | 10.8] | 8.8] | 4.4] | 2.2] | 9.3] | 1.4] | 5.4] | 1.3] | 3.8] | 1.1] |
|  | 18.4 | 11.7 | 11.6 | 2.1 | 3.8 | 0.8 | 3.5 | 1.5 | 2.6 | 0.8 | 5.8 | 0.6 |
| virchow | [11.6, | [8.3, | [7.4, | [1.5, | [3.3, | [0.7, | [1.9, | [1.2, | [2.1, | [0.6, | [5.5, | [0.5, |
|  | 26.0] | 15.5] | 16.0] | 2.7] | 4.3] | 0.9] | 5.7] | 1.7] | 3.0] | 1.0] | 6.1] | 0.7] |
|  | 11.8 | 6.3 | 6.0 | 4.3 | 4.1 | 1.2 | 1.2 | 4.3 | 5.5 | 0.5 | 7.2 | 7.0 |
| virchow2 | [8.9, | [4.2, | [3.3, | [3.8, | [3.6, | [1.1, | [0.8, | [4.0, | [5.1, | [0.3, | [7.0, | [6.8, |
|  | 18.4] | 10.4] | 10.2] | 4.9] | 4.6] | 1.3] | 3.6] | 4.5] | 5.9] | 0.7] | 7.5] | 7.1] |
|  | 5.9 | 15.7 | 5.8 | 2.1 | 3.2 | 3.2 | 2.7 | 0.5 | 4.4 | 1.1 | 2.2 | 0.7 |
| conch | [3.2, | [12.2, | [3.9, | [1.4, | [2.7, | [3.1, | [1.7, | [0.3, | [4.0, | [1.0, | [1.9, | [0.6, |
|  | 13.7] | 19.5] | 11.1] | 2.8] | 3.6] | 3.3] | 5.5] | 0.8] | 4.8] | 1.3] | 2.5] | 0.8] |
|  | 12.2 | 11.6 | 11.1 | 3.3 | 3.0 | 2.8 | 2.5 | 0.3 | 6.9 | 0.3 | 5.2 | 0.4 |
| titan | [9.0, | [8.3, | [7.2, | [2.6, | [2.5, | [2.7, | [1.5, | [0.2, | [6.4, | [0.1, | [4.9, | [0.3, |
|  | 19.5] | 15.1] | 15.2] | 4.0] | 3.4] | 2.9] | 4.9] | 0.6] | 7.4] | 0.4] | 5.5] | 0.5] |
|  | 26.2 | 6.5 | 6.6 | 3.8 | 2.4 | 1.8 | 1.5 | 1.5 | 2.4 | 0.2 | 4.4 | 0.7 |
| keep | [22.2, | [3.9, | [3.6, | [3.2, | [2.0, | [1.7, | [0.9, | [1.2, | [2.0, | [0.1, | [4.2, | [0.6, |
|  | 32.9] | 10.3] | 11.2] | 4.4] | 2.8] | 1.9] | 4.0] | 1.7] | 2.9] | 0.3] | 4.7] | 0.8] |
|  | 14.0 | 10.3 | 2.6 | 3.2 | 4.1 | 3.3 | 6.3 | 4.0 | 2.6 | 0.3 | 2.7 | 1.7 |
| musk | [8.4, | [7.3, | [1.5, | [2.5, | [3.5, | [3.1, | [4.7, | [3.7, | [2.2, | [0.1, | [2.4, | [1.6, |
|  | 22.2] | 14.0] | 6.3] | 4.0] | 4.6] | 3.4] | 8.7] | 4.3] | 3.1] | 0.5] | 3.0] | 1.8] |
|  | 7.9 | 11.6 | 8.4 | 5.9 | 6.1 | 4.6 | 1.2 | 1.1 | 3.0 | 0.1 | 6.3 | 3.5 |
| plip | [4.3, | [8.2, | [4.9, | [4.9, | [5.5, | [4.4, | [0.9, | [0.8, | [2.6, | [0.1, | [6.0, | [3.3, |
|  | 16.0] | 15.6] | 13.3] | 6.9] | 6.8] | 4.8] | 4.1] | 1.4] | 3.5] | 0.3] | 6.7] | 3.7] |
|  | 16.3 | 11.7 | 7.5 | 20.4 | 3.0 | 7.3 | 2.0 | 1.4 | 6.0 | 0.1 | 7.9 | 1.2 |
| quilt | [9.9, | [8.4, | [5.2, | [19.3, | [2.5, | [7.1, | [1.3, | [1.1, | [5.6, | [0.1, | [7.5, | [1.1, |
|  | 23.4] | 15.5] | 13.4] | 21.5] | 3.5] | 7.4] | 5.1] | 1.7] | 6.5] | 0.3] | 8.2] | 1.3] |
|  | 14.3 | 20.5 | 6.5 | 8.3 | 3.2 | 1.0 | 1.9 | 3.4 | 2.1 | 0.9 | 2.2 | 1.2 |
| dinob | [9.8, | [16.7, | [3.5, | [7.3, | [2.7, | [0.9, | [1.1, | [3.1, | [1.8, | [0.7, | [1.9, | [1.0, |
|  | 23.5] | 24.5] | 10.0] | 9.2] | 3.8] | 1.2] | 4.2] | 3.8] | 2.7] | 1.1] | 2.5] | 1.4] |
|  | 2.7 | 10.2 | 5.4 | 5.1 | 5.2 | 6.9 | 5.3 | 2.3 | 2.6 | 1.1 | 10.0 | 4.5 |
| dinol | [3.0, | [6.7, | [2.6, | [4.3, | [4.6, | [6.8, | [3.5, | [2.0, | [2.2, | [0.9, | [9.7, | [4.4, |
|  | 14.3] | 14.0] | 9.0] | 5.9] | 5.8] | 7.1] | 7.6] | 2.7] | 3.1] | 1.3] | 10.4] | 4.7] |
|  | 1.8 | 7.8 | 8.2 | 1.8 | 3.5 | 7.5 | 3.1 | 3.8 | 2.8 | 0.8 | 5.1 | 0.7 |
| vitb | [3.2, | [4.9, | [4.7, | [1.2, | [2.9, | [7.3, | [1.5, | [3.4, | [2.4, | [0.6, | [4.8, | [0.6, |
|  | 13.5] | 11.7] | 12.6] | 2.7] | 4.0] | 7.6] | 5.3] | 4.1] | 3.3] | 1.0] | 5.5] | 0.9] |
|  | 15.0 | 11.8 | 8.4 | 7.5 | 2.6 | 2.2 | 1.9 | 4.1 | 3.1 | 0.2 | 2.4 | 1.2 |
| vitl | [9.4, | [8.2, | [5.0, | [6.6, | [2.1, | [2.0, | [1.2, | [3.8, | [2.6, | [0.1, | [2.1, | [1.0, |
|  | 23.6] | 15.5] | 12.7] | 8.5] | 3.0] | 2.3] | 4.5] | 4.5] | 3.6] | 0.4] | 2.7] | 1.4] |
|  | 12.3 | 8.2 | 7.4 | 3.5 | 5.2 | 7.9 | 1.8 | 2.9 | 8.4 | 0.1 | 2.5 | 7.5 |
| clipb | [6.9, | [5.2, | [4.5, | [2.5, | [4.5, | [7.8, | [1.2, | [2.5, | [7.9, | [0.1, | [2.2, | [7.3, |
|  | 20.9] | 12.0] | 12.1] | 4.5] | 5.8] | 8.1] | 4.7] | 3.3] | 9.0] | 0.4] | 2.9] | 7.7] |
|  | 4.1 | 7.9 | 7.1 | 7.6 | 5.2 | 5.0 | 1.2 | 0.3 | 5.7 | 0.2 | 6.8 | 1.1 |
| clipl | [3.4, | [5.0, | [3.9, | [6.6, | [4.6, | [4.8, | [0.8, | [0.3, | [5.2, | [0.1, | [6.5, | [0.9, |
|  | 14.4] | 11.8] | 12.5] | 8.6] | 5.8] | 5.1] | 4.1] | 0.8] | 6.2] | 0.5] | 7.2] | 1.2] |

Table S56: Quantitative performance (SCE) on linear probing.

| Model | bach | bracs | break-h | ccrcc | crc | esca | mhist | pcam | tcga-crc | tcga-tils | tcga-unif | wilds |
|---|---|---|---|---|---|---|---|---|---|---|---|---|
| | 12.9 | 2.4 | 3.2 | 2.4 | 11.5 | 5.1 | 2.6 | 0.6 | 0.3 | 0.8 | 4.3 | 1.5 |
| hiboub | [8.0, | [1.4, | [2.3, | [1.9, | [4.7, | [1.0, | [1.2, | [0.5, | [0.2, | [0.4, | [4.1, | [0.9, |
| | 17.3] | 6.1] | 10.3] | 5.6] | 14.1] | 5.5] | 4.8] | 1.5] | 0.7] | 1.3] | 6.3] | 2.3] |
| | 6.7 | 14.6 | 10.4 | 8.7 | 8.3 | 6.3 | 4.1 | 2.3 | 1.8 | 0.5 | 3.6 | 0.2 |
| hiboul | [3.5, | [9.7, | [2.9, | [8.1, | [5.1, | [2.4, | [2.0, | [1.4, | [1.5, | [0.2, | [2.4, | [0.1, |
| | 21.8] | 19.2] | 14.2] | 10.7] | 11.6] | 6.7] | 7.0] | 3.3] | 2.1] | 1.0] | 5.1] | 1.2] |
| | 8.7 | 9.8 | 3.1 | 8.4 | 15.2 | 2.0 | 1.4 | 2.2 | 1.7 | 0.6 | 6.4 | 5.2 |
| hopt0 | [6.0, | [7.5, | [1.7, | [4.9, | [5.6, | [1.7, | [0.4, | [1.3, | [1.4, | [0.2, | [4.3, | [4.4, |
| | 23.1] | 14.5] | 9.9] | 15.1] | 20.0] | 2.8] | 3.7] | 3.1] | 2.0] | 1.2] | 7.9] | 6.1] |
| | 4.7 | 9.0 | 10.9 | 14.1 | 6.3 | 8.9 | 7.0 | 3.6 | 0.9 | 0.7 | 2.8 | 2.8 |
| hopt1 | [3.0, | [6.7, | [3.8, | [8.2, | [2.4, | [4.3, | [4.8, | [2.8, | [0.7, | [0.4, | [1.8, | [2.1, |
| | 17.0] | 14.7] | 17.7] | 17.3] | 12.1] | 9.6] | 9.9] | 4.6] | 1.2] | 1.3] | 5.2] | 3.5] |
| | 8.9 | 10.3 | 4.2 | 2.1 | 11.9 | 0.6 | 0.9 | 1.0 | 1.7 | 0.8 | 11.7 | 1.3 |
| midnight | [3.8, | [6.7, | [1.7, | [1.3, | [2.0, | [0.5, | [0.5, | [0.4, | [1.4, | [0.5, | [7.4, | [0.4, |
| | 13.5] | 14.2] | 11.1] | 3.1] | 16.4] | 2.5] | 3.5] | 1.7] | 2.0] | 1.4] | 12.6] | 2.3] |
| | 16.5 | 11.3 | 15.0 | 5.2 | 19.3 | 2.9 | 2.9 | 1.2 | 0.6 | 1.0 | 4.6 | 2.4 |
| phikon | [6.8, | [7.9, | [5.7, | [4.8, | [10.9, | [1.7, | [2.0, | [0.8, | [0.5, | [0.5, | [4.1, | [1.7, |
| | 22.8] | 14.6] | 18.7] | 9.4] | 22.0] | 17.8] | 5.7] | 1.9] | 0.9] | 1.6] | 8.3] | 3.1] |
| | 5.1 | 10.0 | 14.2 | 6.5 | 5.4 | 5.5 | 1.3 | 0.5 | 0.6 | 0.5 | 1.6 | 1.4 |
| phikon2 | [2.4, | [7.2, | [10.0, | [6.0, | [4.1, | [2.5, | [0.8, | [0.3, | [0.3, | [0.3, | [0.8, | [1.1, |
| | 13.3] | 13.0] | 21.0] | 7.7] | 10.4] | 8.8] | 3.7] | 1.2] | 0.9] | 1.0] | 2.9] | 2.2] |
| | 3.6 | 12.9 | 8.8 | 7.7 | 5.9 | 7.5 | 1.9 | 1.1 | 0.5 | 1.6 | 2.9 | 1.4 |
| uni | [2.7, | [9.2, | [5.2, | [3.7, | [3.4, | [6.1, | [0.9, | [0.8, | [0.4, | [1.0, | [2.2, | [1.0, |
| | 12.1] | 16.5] | 20.6] | 12.2] | 11.0] | 10.8] | 4.7] | 2.2] | 0.9] | 2.2] | 4.6] | 2.3] |
| | 13.3 | 6.9 | 11.5 | 7.4 | 15.0 | 5.5 | 4.6 | 0.5 | 2.8 | 1.1 | 5.4 | 2.5 |
| uni2h | [3.4, | [4.0, | [3.1, | [6.7, | [5.0, | [4.7, | [2.4, | [0.3, | [2.5, | [0.8, | [4.3, | [1.8, |
| | 17.5] | 10.8] | 16.3] | 9.8] | 18.7] | 6.2] | 7.3] | 1.6] | 3.1] | 1.7] | 8.9] | 3.5] |
| | 25.4 | 8.0 | 18.1 | 3.3 | 13.1 | 17.9 | 2.3 | 1.0 | 1.3 | 1.0 | 7.7 | 0.5 |
| virchow | [10.8, | [5.5, | [9.9, | [1.4, | [3.5, | [1.0, | [1.1, | [0.5, | [1.1, | [0.5, | [6.4, | [0.3, |
| | 31.4] | 11.7] | 24.9] | 8.4] | 16.6] | 18.6] | 5.0] | 1.8] | 1.6] | 1.6] | 10.2] | 1.4] |
| | 8.6 | 4.5 | 12.2 | 10.4 | 8.3 | 8.9 | 0.6 | 4.3 | 2.8 | 1.0 | 11.5 | 5.7 |
| virchow2 | [6.2, | [2.8, | [3.3, | [3.4, | [4.0, | [4.3, | [0.4, | [3.2, | [2.5, | [0.4, | [7.6, | [5.1, |
| | 19.8] | 8.4] | 16.7] | 16.1] | 14.4] | 9.8] | 3.3] | 5.4] | 3.1] | 1.6] | 12.2] | 6.3] |
| | 6.7 | 10.5 | 6.7 | 3.1 | 12.7 | 3.8 | 2.2 | 0.7 | 2.2 | 1.6 | 3.1 | 1.5 |
| conch | [2.8, | [8.8, | [3.4, | [1.4, | [4.3, | [3.1, | [0.9, | [0.3, | [1.9, | [1.1, | [2.3, | [0.9, |
| | 11.9] | 16.7] | 17.4] | 8.8] | 15.1] | 4.8] | 4.5] | 1.2] | 2.5] | 2.2] | 4.3] | 2.2] |
| | 15.0 | 8.5 | 14.5 | 3.0 | 9.4 | 3.1 | 0.4 | 0.4 | 3.4 | 0.3 | 6.1 | 0.3 |
| titan | [6.5, | [5.8, | [5.2, | [2.4, | [4.6, | [2.8, | [0.3, | [0.1, | [3.1, | [0.1, | [5.1, | [0.2, |
| | 18.1] | 12.7] | 19.1] | 8.8] | 14.9] | 4.7] | 2.8] | 1.0] | 3.8] | 1.0] | 7.7] | 1.1] |
| | 14.2 | 4.9 | 5.2 | 7.0 | 11.2 | 7.0 | 0.7 | 0.5 | 1.3 | 0.8 | 6.9 | 1.3 |
| keep | [12.3, | [3.0, | [3.1, | [3.4, | [4.2, | [3.1, | [0.3, | [0.4, | [1.1, | [0.3, | [5.1, | [0.8, |
| | 18.1] | 8.9] | 13.1] | 11.4] | 15.1] | 7.8] | 2.7] | 1.3] | 1.6] | 1.3] | 8.9] | 2.2] |
| | 14.3 | 7.3 | 2.5 | 2.9 | 5.8 | 3.1 | 3.2 | 3.2 | 1.3 | 0.4 | 3.5 | 2.0 |
| musk | [4.4, | [4.6, | [1.9, | [1.4, | [3.0, | [2.8, | [1.9, | [2.5, | [1.0, | [0.2, | [2.7, | [1.6, |
| | 20.0] | 11.3] | 15.3] | 5.5] | 17.3] | 3.5] | 6.0] | 3.8] | 1.6] | 1.0] | 4.5] | 2.8] |
| | 11.4 | 9.0 | 6.9 | 8.1 | 6.1 | 7.3 | 0.9 | 1.0 | 2.2 | 0.2 | 6.9 | 2.3 |
| plip | [8.4, | [5.9, | [3.3, | [4.9, | [5.3, | [7.0, | [0.5, | [0.6, | [2.0, | [0.1, | [6.2, | [1.9, |
| | 16.3] | 12.3] | 11.8] | 11.0] | 14.9] | 7.7] | 3.1] | 1.5] | 2.6] | 0.7] | 7.6] | 2.7] |
| | 12.5 | 8.1 | 4.3 | 15.8 | 8.9 | 24.1 | 1.1 | 1.1 | 2.7 | 0.7 | 8.2 | 1.1 |
| quilt | [7.1, | [6.0, | [3.4, | [10.1, | [4.3, | [7.6, | [0.5, | [0.7, | [2.4, | [0.3, | [7.4, | [0.8, |
| | 20.9] | 11.8] | 14.2] | 23.1] | 13.4] | 24.4] | 2.8] | 1.6] | 3.0] | 1.1] | 9.0] | 1.5] |
| | 14.6 | 14.1 | 21.4 | 7.2 | 7.3 | 4.6 | 0.9 | 1.8 | 1.6 | 0.7 | 2.5 | 0.9 |
| dinob | [8.5, | [10.7, | [5.1, | [5.2, | [3.5, | [0.6, | [0.4, | [1.4, | [1.4, | [0.4, | [1.9, | [0.6, |
| | 20.4] | 18.1] | 25.4] | 11.5] | 12.2] | 5.0] | 3.6] | 2.3] | 1.9] | 1.2] | 3.0] | 1.3] |
| | 2.8 | 7.8 | 17.7 | 7.2 | 12.2 | 9.4 | 3.1 | 1.3 | 1.7 | 0.9 | 10.3 | 2.9 |
| dinol | [2.5, | [5.1, | [2.8, | [5.0, | [3.4, | [5.1, | [1.8, | [0.9, | [1.4, | [0.6, | [9.2, | [2.5, |
| | 13.7] | 10.7] | 22.1] | 9.9] | 14.5] | 10.0] | 5.9] | 1.9] | 2.0] | 1.5] | 11.2] | 3.4] |
| | 1.5 | 6.5 | 12.4 | 1.8 | 3.5 | 6.4 | 2.0 | 1.9 | 1.9 | 0.9 | 4.9 | 0.4 |
| vitb | [1.9, | [4.4, | [5.8, | [1.0, | [3.0, | [6.0, | [1.7, | [1.6, | [1.6, | [0.6, | [4.3, | [0.2, |
| | 10.2] | 10.0] | 16.1] | 4.7] | 11.9] | 6.7] | 4.2] | 2.4] | 2.2] | 1.4] | 5.6] | 0.8] |
| | 16.0 | 8.7 | 7.8 | 8.3 | 6.0 | 6.7 | 1.3 | 2.3 | 1.8 | 0.3 | 2.9 | 0.6 |
| vitl | [7.1, | [6.0, | [4.0, | [5.3, | [2.6, | [2.9, | [0.4, | [1.9, | [1.4, | [0.1, | [2.3, | [0.4, |
| | 21.2] | 11.9] | 15.0] | 12.6] | 11.9] | 7.1] | 3.4] | 2.8] | 2.2] | 0.8] | 3.5] | 1.0] |
| | 10.0 | 6.4 | 4.8 | 4.7 | 6.2 | 16.3 | 0.7 | 1.7 | 9.0 | 0.2 | 2.7 | 4.3 |
| clipb | [4.9, | [4.1, | [2.5, | [2.3, | [5.2, | [6.8, | [0.4, | [1.3, | [8.5, | [0.1, | [2.2, | [4.0, |
| | 16.8] | 9.4] | 8.9] | 7.3] | 10.1] | 23.1] | 3.0] | 2.2] | 9.6] | 0.7] | 3.1] | 4.7] |
| | 3.9 | 6.0 | 5.4 | 8.9 | 10.9 | 4.7 | 0.9 | 0.3 | 3.8 | 0.2 | 6.8 | 0.8 |
| clipl | [2.1, | [3.5, | [2.8, | [6.0, | [6.9, | [4.2, | [0.4, | [0.1, | [3.5, | [0.1, | [6.1, | [0.6, |
| | 11.9] | 9.3] | 11.1] | 12.0] | 13.9] | 5.2] | 3.3] | 0.8] | 4.2] | 0.6] | 7.5] | 1.3] |

Table S57: Quantitative performance (Drop in Balanced accuracy) on adversarial attack ($\epsilon = 0.25 \cdot 10^{-3}$).

| Model | bach | bracs | break-h | ccrcc | crc | esca | mhist | pcam | tcga-crc | tcga-tils | tcga-unif | wilds |
|---|---|---|---|---|---|---|---|---|---|---|---|---|
| hiboub | 8.1 [3.7, 12.9] | 7.4 [5.3, 9.7] | 7.7 [4.5, 10.9] | 4.9 [4.4, 5.5] | 4.2 [3.7, 4.7] | 4.9 [3.6, 6.4] | 12.8 [10.5, 15.0] | 4.0 [0.0, 10.0] | 10.5 [9.6, 11.3] | 6.5 [5.8, 7.2] | 12.7 [11.6, 13.8] | 2.0 [1.7, 2.3] |
| hiboul | 3.8 [0.8, 8.0] | 11.8 [9.3, 14.6] | 4.0 [1.7, 6.8] | 3.0 [2.6, 3.4] | 4.1 [3.7, 4.5] | 3.6 [2.6, 4.7] | 11.3 [9.2, 13.4] | 6.0 [0.0, 13.0] | 5.5 [4.9, 6.2] | 4.0 [3.4, 4.6] | 8.3 [7.4, 9.2] | 1.0 [0.8, 1.2] |
| hopt0 | 4.3 [1.2, 8.2] | 7.2 [5.1, 9.4] | 5.3 [2.9, 8.1] | 3.4 [2.9, 3.9] | 2.2 [1.8, 2.6] | 3.8 [2.7, 5.2] | 8.5 [6.9, 10.3] | 10.0 [2.4, 18.2] | 6.3 [5.7, 7.0] | 5.4 [4.7, 6.1] | 7.0 [6.3, 7.8] | 1.0 [0.8, 1.1] |
| hopt1 | 9.0 [4.3, 14.1] | 12.5 [9.8, 15.3] | 12.2 [8.2, 16.4] | 2.9 [2.5, 3.3] | 3.2 [2.7, 3.8] | 3.1 [2.1, 4.3] | 27.0 [24.0, 29.9] | 18.3 [6.2, 32.8] | 9.3 [8.5, 10.1] | 6.3 [5.6, 6.9] | 10.0 [9.1, 11.0] | 3.6 [3.3, 4.0] |
| midnight | 4.6 [0.9, 9.0] | 12.8 [10.1, 15.6] | 9.4 [6.0, 13.2] | 3.4 [3.0, 3.9] | 2.8 [2.4, 3.3] | 3.6 [2.4, 4.9] | 15.0 [12.7, 17.4] | 4.2 [0.0, 14.3] | 4.5 [3.9, 5.1] | 2.3 [1.9, 2.7] | 5.5 [4.8, 6.2] | 0.8 [0.6, 1.0] |
| phikon | 7.8 [3.5, 13.1] | 4.2 [2.7, 5.9] | 7.2 [3.8, 10.9] | 2.9 [2.5, 3.4] | 1.6 [1.3, 2.0] | 3.1 [2.0, 4.3] | 10.9 [8.9, 13.1] | 4.0 [0.0, 10.0] | 7.3 [6.6, 8.0] | 4.0 [3.5, 4.6] | 6.3 [5.5, 7.1] | 0.8 [0.6, 1.0] |
| phikon2 | 7.0 [2.8, 12.1] | 6.9 [4.9, 9.0] | 7.4 [4.0, 11.2] | 3.7 [3.2, 4.1] | 3.0 [2.6, 3.5] | 4.2 [3.1, 5.4] | 17.3 [14.8, 19.8] | 2.0 [0.0, 6.5] | 7.3 [6.6, 8.0] | 4.0 [3.4, 4.5] | 7.3 [6.6, 8.1] | 4.4 [4.0, 4.8] |
| uni | 5.2 [1.7, 9.1] | 3.9 [2.3, 5.6] | 8.0 [4.6, 11.9] | 3.1 [2.7, 3.6] | 2.1 [1.7, 2.5] | 3.0 [1.9, 4.3] | 10.5 [8.5, 12.6] | 4.0 [0.0, 10.0] | 7.0 [6.3, 7.7] | 5.8 [5.1, 6.5] | 7.6 [6.8, 8.5] | 2.1 [1.8, 2.4] |
| uni2h | 1.6 [0.0, 3.5] | 6.2 [4.3, 8.3] | 5.4 [2.6, 8.8] | 2.6 [2.2, 3.0] | 2.7 [2.3, 3.1] | 2.3 [1.4, 3.4] | 13.2 [10.9, 15.5] | 2.0 [0.0, 6.8] | 5.1 [4.5, 5.7] | 4.7 [4.1, 5.4] | 5.5 [4.8, 6.3] | 0.7 [0.5, 0.8] |
| virchow | 5.9 [2.1, 10.3] | 7.2 [5.0, 9.4] | 7.0 [4.0, 10.4] | 2.9 [2.5, 3.4] | 2.3 [1.9, 2.7] | 2.9 [1.8, 4.0] | 12.7 [10.5, 15.0] | 2.0 [0.0, 6.8] | 7.1 [6.4, 7.8] | 3.6 [3.1, 4.2] | 8.0 [7.3, 8.8] | 1.7 [1.5, 2.0] |
| virchow2 | 4.9 [1.3, 9.2] | 6.2 [4.3, 8.2] | 1.4 [0.1, 3.4] | 1.7 [1.4, 2.0] | 1.8 [1.4, 2.3] | 3.1 [1.9, 4.5] | 5.3 [3.9, 6.8] | 2.0 [0.0, 6.8] | 4.0 [3.5, 4.6] | 4.5 [3.9, 5.1] | 6.5 [5.8, 7.2] | 5.5 [5.0, 5.9] |
| conch | 8.1 [3.7, 13.3] | 9.2 [7.0, 11.6] | 6.7 [3.8, 9.8] | 6.0 [5.4, 6.6] | 5.2 [4.6, 5.8] | 6.0 [4.5, 7.6] | 16.0 [13.6, 18.5] | 14.2 [4.2, 26.4] | 8.7 [7.9, 9.4] | 5.9 [5.2, 6.5] | 13.2 [12.2, 14.2] | 2.4 [2.1, 2.7] |
| titan | 5.2 [1.7, 9.1] | 13.4 [10.7, 16.2] | 10.0 [6.5, 13.6] | 9.5 [8.7, 10.2] | 6.8 [6.2, 7.5] | 10.2 [8.2, 12.4] | 27.7 [24.8, 30.7] | 10.2 [1.9, 21.8] | 16.2 [15.2, 17.1] | 10.9 [10.0, 11.8] | 22.4 [21.1, 23.6] | 6.2 [5.7, 6.7] |
| keep | 3.4 [1.1, 6.3] | 6.9 [4.9, 9.0] | 3.9 [1.8, 6.3] | 3.0 [2.6, 3.5] | 1.6 [1.2, 1.9] | 2.0 [1.2, 3.1] | 13.1 [10.9, 15.5] | 4.2 [0.0, 14.3] | 7.3 [6.6, 8.0] | 5.7 [5.1, 6.4] | 9.7 [8.8, 10.6] | 1.8 [1.6, 2.1] |
| musk | 4.9 [1.4, 9.1] | 11.8 [9.2, 14.4] | 14.1 [9.7, 18.6] | 8.3 [7.5, 9.0] | 13.4 [12.6, 14.2] | 7.7 [6.2, 9.4] | 24.8 [22.0, 27.7] | 20.5 [6.9, 35.6] | 14.9 [14.0, 15.9] | 13.0 [12.0, 13.9] | 27.3 [26.0, 28.6] | 6.2 [5.7, 6.7] |
| plip | 9.4 [4.5, 14.9] | 14.1 [11.3, 17.0] | 6.8 [3.9, 10.2] | 8.9 [8.2, 9.7] | 12.6 [11.8, 13.4] | 10.8 [9.1, 12.5] | 26.9 [24.0, 29.8] | 22.5 [8.7, 38.8] | 21.4 [20.3, 22.6] | 11.7 [10.7, 12.6] | 22.9 [21.7, 24.1] | 13.7 [13.0, 14.3] |
| quilt | 6.2 [2.3, 10.8] | 13.4 [10.6, 16.2] | 10.5 [6.9, 14.3] | 7.1 [6.4, 7.8] | 10.6 [9.9, 11.4] | 11.5 [9.8, 13.3] | 25.0 [22.1, 28.0] | 20.2 [9.1, 32.9] | 12.0 [11.1, 13.0] | 12.0 [11.1, 12.8] | 24.7 [23.5, 25.8] | 7.9 [7.4, 8.5] |
| dinob | 11.8 [6.2, 18.0] | 15.1 [12.3, 18.2] | 8.8 [5.2, 12.6] | 13.1 [12.2, 14.0] | 9.7 [8.9, 10.4] | 6.6 [5.1, 8.1] | 12.3 [10.0, 14.6] | 10.3 [0.0, 23.6] | 15.1 [14.1, 16.1] | 11.2 [10.3, 12.1] | 15.7 [14.6, 16.8] | 11.4 [10.8, 12.1] |
| dinol | 14.5 [8.8, 21.0] | 9.5 [7.2, 11.8] | 6.7 [3.6, 10.2] | 9.2 [8.4, 10.0] | 10.8 [10.1, 11.6] | 5.7 [4.6, 6.9] | 16.6 [14.2, 19.1] | 10.2 [1.8, 21.6] | 15.0 [14.0, 16.0] | 9.6 [8.7, 10.4] | 15.8 [14.7, 16.9] | 8.5 [8.0, 9.1] |
| vitb | 6.5 [2.5, 11.5] | 8.8 [6.6, 11.2] | 6.4 [3.7, 9.4] | 5.3 [4.7, 5.9] | 5.8 [5.2, 6.4] | 6.8 [5.6, 8.2] | 10.7 [8.6, 12.9] | 4.0 [0.0, 10.4] | 8.7 [7.9, 9.4] | 5.2 [4.5, 5.9] | 10.1 [9.3, 11.0] | 4.5 [4.1, 4.8] |
| vitl | 6.9 [3.2, 11.2] | 5.6 [3.8, 7.6] | 12.4 [8.3, 16.8] | 5.3 [4.7, 5.9] | 5.6 [5.0, 6.2] | 4.8 [3.7, 6.0] | 12.3 [10.1, 14.7] | 2.0 [0.0, 6.5] | 6.3 [5.6, 6.9] | 4.6 [4.0, 5.3] | 9.4 [8.6, 10.3] | 4.4 [4.0, 4.8] |
| clipb | 18.3 [11.8, 25.8] | 20.6 [17.3, 23.8] | 7.7 [4.7, 11.0] | 18.0 [17.0, 19.1] | 16.4 [15.5, 17.4] | 24.8 [22.9, 26.8] | 32.6 [29.6, 35.5] | 24.3 [11.4, 40.1] | 31.8 [30.6, 33.0] | 28.0 [26.8, 29.1] | 28.5 [27.3, 29.7] | 22.3 [21.5, 23.1] |
| clipl | 28.1 [20.4, 36.0] | 24.8 [21.3, 28.4] | 21.7 [16.7, 26.8] | 17.6 [16.7, 18.5] | 23.0 [22.0, 23.9] | 22.1 [20.1, 24.2] | 29.2 [26.2, 32.1] | 30.5 [15.8, 46.8] | 31.4 [30.1, 32.6] | 23.3 [22.2, 24.4] | 32.3 [30.9, 33.7] | 17.2 [16.4, 17.9] |

Table S58: Quantitative performance (Drop in F1-score) on adversarial attack ($\epsilon = 0.25 \cdot 10^{-3}$).

| Model | bach | bracs | break-h | ccrcc | crc | esca | mhist | pcam | tcga-crc | tcga-tils | tcga-unif | wilds |
|---|---|---|---|---|---|---|---|---|---|---|---|---|
| hiboub | 8.6 [4.2, 13.6] | 7.4 [5.3, 9.8] | 9.2 [5.4, 13.0] | 5.4 [4.8, 6.0] | 4.5 [4.0, 5.0] | 5.9 [4.5, 7.4] | 13.2 [10.9, 15.5] | 5.9 [0.0, 15.1] | 10.2 [9.5, 10.8] | 7.3 [6.6, 8.0] | 13.2 [12.1, 14.3] | 2.0 [1.7, 2.3] |
| hiboul | 3.2 [0.5, 6.8] | 11.4 [8.9, 14.2] | 4.0 [1.8, 6.8] | 3.5 [3.0, 3.9] | 4.8 [4.4, 5.3] | 4.1 [3.3, 5.1] | 11.1 [9.1, 13.2] | 8.7 [0.0, 19.8] | 5.7 [5.2, 6.2] | 4.2 [3.7, 4.8] | 8.4 [7.5, 9.3] | 1.0 [0.8, 1.2] |
| hopt0 | 4.8 [1.4, 9.1] | 6.9 [5.0, 9.1] | 5.7 [3.1, 8.6] | 3.4 [3.0, 3.9] | 2.1 [1.8, 2.5] | 4.6 [3.6, 6.0] | 8.8 [7.1, 10.6] | 14.2 [3.2, 27.4] | 6.4 [5.9, 6.9] | 5.4 [4.8, 6.0] | 7.2 [6.5, 8.0] | 1.0 [0.8, 1.2] |
| hopt1 | 10.0 [4.9, 15.6] | 12.2 [9.6, 15.0] | 13.1 [9.0, 17.5] | 3.3 [2.8, 3.7] | 3.0 [2.5, 3.5] | 4.5 [3.5, 5.6] | 27.5 [24.5, 30.5] | 20.4 [8.3, 36.1] | 9.0 [8.4, 9.7] | 6.5 [5.9, 7.2] | 10.1 [9.2, 11.0] | 3.6 [3.2, 4.0] |
| midnight | 3.8 [0.8, 7.5] | 12.6 [10.0, 15.3] | 10.2 [6.2, 14.6] | 3.8 [3.3, 4.3] | 3.0 [2.6, 3.5] | 4.7 [3.5, 6.0] | 15.2 [12.9, 17.6] | 3.2 [0.0, 11.5] | 4.4 [3.9, 4.9] | 2.6 [2.2, 3.0] | 5.5 [4.9, 6.2] | 0.8 [0.6, 1.0] |
| phikon | 7.5 [3.5, 12.4] | 4.1 [2.6, 5.7] | 7.0 [3.8, 10.5] | 2.9 [2.5, 3.3] | 1.8 [1.5, 2.2] | 3.2 [2.4, 4.2] | 10.9 [8.9, 12.9] | 5.5 [0.0, 14.3] | 6.8 [6.2, 7.3] | 4.3 [3.8, 4.9] | 6.6 [5.9, 7.4] | 0.8 [0.6, 1.0] |
| phikon2 | 6.8 [2.8, 11.3] | 7.1 [5.1, 9.2] | 6.5 [3.6, 9.9] | 4.8 [4.2, 5.4] | 3.2 [2.8, 3.7] | 4.6 [3.8, 5.5] | 17.2 [14.8, 19.7] | 2.7 [0.0, 8.9] | 7.2 [6.6, 7.7] | 4.3 [3.8, 4.8] | 7.6 [6.9, 8.3] | 4.4 [4.0, 4.8] |
| uni | 5.6 [1.9, 9.9] | 3.8 [2.3, 5.5] | 7.3 [4.3, 11.0] | 3.4 [3.0, 3.9] | 2.3 [1.9, 2.7] | 3.8 [2.9, 4.9] | 10.2 [8.3, 12.3] | 5.9 [0.0, 15.1] | 6.9 [6.4, 7.5] | 6.3 [5.7, 6.9] | 7.6 [6.8, 8.5] | 2.1 [1.8, 2.4] |
| uni2h | 2.1 [0.0, 4.8] | 6.4 [4.5, 8.6] | 5.3 [2.7, 8.5] | 3.1 [2.6, 3.5] | 3.0 [2.6, 3.5] | 3.1 [2.3, 4.1] | 13.5 [11.2, 15.9] | 3.0 [0.0, 10.8] | 5.3 [4.8, 5.8] | 5.0 [4.5, 5.6] | 5.6 [4.9, 6.3] | 0.7 [0.5, 0.8] |
| virchow | 5.7 [2.2, 9.8] | 6.9 [4.8, 9.0] | 6.8 [3.8, 10.1] | 3.2 [2.7, 3.6] | 2.2 [1.9, 2.6] | 3.4 [2.5, 4.2] | 12.4 [10.3, 14.6] | 2.8 [0.0, 9.9] | 6.8 [6.3, 7.4] | 3.8 [3.3, 4.3] | 8.4 [7.7, 9.2] | 1.7 [1.5, 2.0] |
| virchow2 | 4.5 [1.4, 8.5] | 6.0 [4.2, 8.1] | 1.5 [0.2, 3.4] | 1.7 [1.4, 2.0] | 1.7 [1.4, 2.1] | 3.2 [2.2, 4.4] | 5.3 [3.9, 6.8] | 2.8 [0.0, 9.7] | 4.4 [4.0, 4.9] | 4.6 [4.1, 5.2] | 6.7 [6.0, 7.4] | 5.7 [5.2, 6.1] |
| conch | 7.8 [3.7, 12.9] | 9.1 [7.0, 11.5] | 7.2 [4.2, 10.5] | 6.3 [5.7, 6.9] | 5.5 [5.0, 6.1] | 6.5 [5.3, 7.9] | 15.9 [13.6, 18.4] | 16.2 [5.4, 29.2] | 8.6 [8.0, 9.2] | 6.3 [5.6, 7.0] | 13.6 [12.7, 14.7] | 2.4 [2.1, 2.7] |
| titan | 5.4 [1.7, 9.6] | 14.0 [11.2, 16.9] | 10.9 [7.4, 14.6] | 10.1 [9.4, 10.9] | 7.1 [6.4, 7.7] | 11.1 [9.4, 12.9] | 27.7 [24.8, 30.8] | 11.3 [2.7, 23.3] | 15.1 [14.4, 15.8] | 11.5 [10.7, 12.4] | 23.1 [22.0, 24.3] | 6.2 [5.7, 6.7] |
| keep | 4.7 [1.6, 8.4] | 6.8 [4.8, 8.9] | 3.6 [1.7, 5.8] | 3.0 [2.6, 3.4] | 1.7 [1.3, 2.0] | 2.6 [1.9, 3.5] | 13.4 [11.2, 15.8] | 3.1 [0.0, 11.1] | 6.9 [6.4, 7.5] | 6.0 [5.4, 6.6] | 10.1 [9.2, 11.1] | 1.8 [1.6, 2.1] |
| musk | 4.2 [1.3, 7.8] | 11.7 [9.1, 14.3] | 14.7 [10.0, 19.7] | 8.1 [7.4, 8.8] | 14.5 [13.6, 15.3] | 8.7 [7.4, 10.1] | 25.6 [22.8, 28.6] | 19.3 [7.9, 33.1] | 14.4 [13.7, 15.1] | 14.4 [13.5, 15.3] | 27.5 [26.2, 28.8] | 6.2 [5.7, 6.7] |
| plip | 9.4 [4.8, 14.6] | 13.8 [11.1, 16.7] | 9.9 [5.5, 14.6] | 9.0 [8.3, 9.8] | 13.7 [12.9, 14.5] | 12.2 [10.4, 14.2] | 26.5 [23.7, 29.3] | 21.9 [10.2, 37.3] | 17.8 [17.0, 18.6] | 13.0 [12.1, 13.9] | 24.5 [23.2, 25.6] | 14.7 [14.0, 15.4] |
| quilt | 6.7 [2.6, 11.4] | 13.5 [10.7, 16.3] | 12.2 [8.1, 16.6] | 8.3 [7.5, 9.2] | 11.7 [11.0, 12.5] | 13.9 [11.8, 15.9] | 25.2 [22.3, 28.2] | 23.4 [10.6, 37.5] | 10.5 [9.8, 11.2] | 15.2 [14.4, 16.1] | 26.1 [24.8, 27.2] | 7.9 [7.4, 8.5] |
| dinob | 11.8 [6.2, 17.8] | 14.4 [11.7, 17.3] | 9.0 [5.6, 12.9] | 13.1 [12.2, 14.0] | 9.9 [9.2, 10.7] | 7.2 [5.9, 8.6] | 12.6 [10.3, 14.8] | 9.4 [0.0, 21.5] | 13.2 [12.5, 13.8] | 12.7 [11.8, 13.6] | 16.4 [15.3, 17.5] | 11.5 [10.8, 12.1] |
| dinol | 14.4 [8.9, 20.6] | 9.3 [7.0, 11.7] | 7.0 [4.0, 10.6] | 9.0 [8.3, 9.8] | 11.4 [10.7, 12.2] | 6.1 [5.0, 7.2] | 16.4 [14.1, 18.8] | 11.7 [2.7, 24.3] | 12.9 [12.2, 13.6] | 11.3 [10.4, 12.1] | 16.6 [15.6, 17.6] | 8.8 [8.2, 9.3] |
| vitb | 6.2 [2.4, 10.7] | 8.7 [6.5, 11.1] | 6.7 [4.0, 9.7] | 5.3 [4.7, 5.8] | 6.2 [5.6, 6.8] | 6.6 [5.5, 7.7] | 10.8 [8.7, 12.9] | 5.1 [0.0, 13.2] | 7.9 [7.4, 8.5] | 5.8 [5.2, 6.4] | 10.6 [9.8, 11.5] | 4.5 [4.1, 4.9] |
| vitl | 7.7 [3.6, 12.3] | 5.6 [3.8, 7.5] | 12.2 [8.2, 16.5] | 5.2 [4.6, 5.9] | 6.1 [5.5, 6.7] | 5.0 [4.1, 6.1] | 12.3 [10.2, 14.8] | 2.6 [0.0, 8.3] | 5.6 [5.2, 6.1] | 5.1 [4.5, 5.7] | 10.0 [9.1, 10.9] | 4.4 [4.0, 4.8] |
| clipb | 18.6 [12.2, 25.6] | 20.1 [16.9, 23.3] | 10.2 [5.9, 14.7] | 18.9 [17.9, 20.0] | 17.5 [16.6, 18.5] | 24.5 [22.7, 26.4] | 33.4 [30.3, 36.6] | 26.5 [13.5, 42.2] | 29.1 [28.3, 30.0] | 32.4 [31.4, 33.4] | 29.8 [28.4, 31.1] | 23.8 [23.0, 24.7] |
| clipl | 27.1 [19.8, 34.5] | 24.4 [20.9, 27.9] | 24.7 [19.2, 30.1] | 19.1 [18.1, 20.1] | 24.1 [23.2, 25.1] | 22.7 [20.5, 24.9] | 29.4 [26.5, 32.3] | 31.8 [17.8, 47.5] | 26.9 [26.0, 27.7] | 26.0 [25.0, 27.0] | 33.8 [32.4, 35.1] | 17.2 [16.5, 18.0] |

Table S59: Quantitative performance (Drop in Balanced accuracy) on adversarial attack ($\epsilon = 1.5 \cdot 10^{-3}$).

| Model | bach | bracs | break-h | ccrcc | crc | esca | mhist | pcam | tcga-crc | tcga-tils | tcga-unif | wilds |
|---|---|---|---|---|---|---|---|---|---|---|---|---|
| | 53.5 | 42.9 | 36.9 | 42.5 | 54.9 | 32.5 | 61.6 | 92.0 | 57.6 | 54.8 | 61.0 | 46.3 |
| hiboub | [45.4, | [39.1, | [31.3, | [41.2, | [53.7, | [30.1, | [58.5, | [84.5, | [56.3, | [53.5, | [59.6, | [45.2, |
| | 62.0] | 47.0] | 42.7] | 43.8] | 56.1] | 34.8] | 64.7] | 98.1] | 59.0] | 56.1] | 62.4] | 47.3] |
| | 40.0 | 53.3 | 35.2 | 28.7 | 34.0 | 26.8 | 62.6 | 55.2 | 40.9 | 29.4 | 55.4 | 17.2 |
| hiboul | [31.9, | [49.4, | [29.8, | [27.6, | [33.0, | [24.5, | [59.5, | [37.6, | [39.5, | [28.2, | [53.9, | [16.5, |
| | 48.6] | 57.5] | 40.7] | 29.8] | 35.0] | 29.0] | 65.5] | 71.8] | 42.2] | 30.6] | 56.8] | 18.0] |
| | 41.0 | 45.1 | 25.3 | 32.7 | 29.6 | 28.4 | 61.4 | 100.0 | 45.1 | 41.4 | 51.9 | 30.4 |
| hopt0 | [32.5, | [41.2, | [20.3, | [31.5, | [28.5, | [26.0, | [58.4, | [100.0, | [43.7, | [40.1, | [50.5, | [29.5, |
| | 49.6] | 49.1] | 30.2] | 33.9] | 30.8] | 30.8] | 64.4] | 100.0] | 46.4] | 42.6] | 53.4] | 31.2] |
| | 52.4 | 59.6 | 60.5 | 35.3 | 42.1 | 31.6 | 77.1 | 100.0 | 57.7 | 46.5 | 63.2 | 74.5 |
| hopt1 | [43.4, | [55.7, | [54.5, | [34.1, | [41.0, | [29.3, | [74.3, | [100.0, | [56.4, | [45.1, | [61.9, | [73.7, |
| | 60.9] | 63.4] | 66.3] | 36.5] | 43.2] | 34.0] | 80.0] | 100.0] | 59.0] | 47.7] | 64.7] | 75.3] |
| | 37.7 | 49.7 | 34.5 | 29.5 | 27.9 | 26.0 | 61.0 | 57.5 | 28.4 | 16.3 | 36.5 | 23.0 |
| midnight | [29.3, | [45.6, | [29.3, | [28.3, | [26.9, | [23.9, | [57.8, | [41.2, | [27.2, | [15.3, | [35.1, | [22.2, |
| | 45.8] | 53.9] | 40.1] | 30.6] | 28.9] | 28.1] | 63.9] | 73.1] | 29.6] | 17.3] | 37.8] | 23.8] |
| | 37.6 | 33.5 | 34.8 | 25.8 | 19.4 | 17.7 | 59.3 | 53.3 | 43.2 | 33.5 | 45.1 | 16.1 |
| phikon | [29.2, | [29.7, | [29.0, | [24.7, | [18.5, | [15.7, | [56.1, | [35.5, | [41.9, | [32.3, | [43.6, | [15.4, |
| | 46.0] | 37.4] | 40.8] | 27.0] | 20.4] | 19.9] | 62.3] | 69.9] | 44.5] | 34.7] | 46.6] | 16.8] |
| | 45.9 | 44.5 | 25.5 | 34.8 | 43.9 | 30.0 | 66.5 | 71.5 | 45.0 | 31.5 | 52.9 | 60.0 |
| phikon2 | [36.7, | [40.5, | [19.9, | [33.5, | [42.7, | [27.5, | [63.5, | [54.6, | [43.7, | [30.2, | [51.4, | [59.0, |
| | 54.6] | 48.7] | 31.3] | 36.0] | 45.1] | 32.4] | 69.4] | 86.2] | 46.4] | 32.7] | 54.2] | 61.0] |
| | 36.6 | 35.9 | 32.3 | 25.6 | 27.4 | 23.2 | 61.6 | 69.7 | 47.2 | 53.1 | 49.1 | 52.8 |
| uni | [28.2, | [32.1, | [26.7, | [24.5, | [26.4, | [21.0, | [58.6, | [54.1, | [45.9, | [51.8, | [47.6, | [51.9, |
| | 44.8] | 39.9] | 37.6] | 26.8] | 28.5] | 25.5] | 64.6] | 83.3] | 48.6] | 54.3] | 50.4] | 53.7] |
| | 26.3 | 42.7 | 28.1 | 21.4 | 29.3 | 16.9 | 62.1 | 40.8 | 34.6 | 37.3 | 40.9 | 23.7 |
| uni2h | [18.7, | [38.8, | [22.7, | [20.4, | [28.3, | [15.0, | [58.8, | [24.3, | [33.3, | [36.0, | [39.5, | [22.9, |
| | 33.9] | 46.8] | 33.5] | 22.3] | 30.4] | 18.9] | 65.2] | 59.3] | 35.9] | 38.6] | 42.3] | 24.5] |
| | 39.9 | 42.1 | 40.0 | 27.2 | 27.2 | 21.4 | 65.3 | 67.7 | 44.0 | 32.0 | 52.6 | 38.3 |
| virchow | [31.5, | [38.3, | [34.2, | [26.1, | [26.2, | [19.2, | [62.3, | [52.6, | [42.7, | [30.7, | [51.1, | [37.4, |
| | 48.4] | 46.3] | 46.0] | 28.4] | 28.3] | 23.6] | 68.2] | 81.2] | 45.4] | 33.2] | 54.0] | 39.2] |
| | 19.8 | 33.7 | 24.8 | 19.8 | 21.0 | 15.1 | 43.5 | 59.3 | 32.8 | 40.7 | 43.2 | 44.2 |
| virchow2 | [13.0, | [29.8, | [19.6, | [18.7, | [20.0, | [13.0, | [40.5, | [ , | [31.6, | [39.4, | [41.7, | [43.4, |
| | 26.5] | 37.6] | 30.1] | 20.8] | 22.0] | 17.3] | 46.7] | 75.5] | 34.1] | 41.9] | 44.5] | 45.1] |
| | 36.7 | 50.0 | 50.7 | 60.8 | 63.3 | 45.4 | 70.6 | 77.7 | 54.0 | 54.2 | 60.3 | 46.8 |
| conch | [28.6, | [46.0, | [45.0, | [59.5, | [62.1, | [43.0, | [67.7, | [63.3, | [52.7, | [53.0, | [59.1, | [45.8, |
| | 44.8] | 54.1] | 56.5] | 62.0] | 64.5] | 47.8] | 73.5] | 90.5] | 55.3] | 55.4] | 61.6] | 47.8] |
| | 64.9 | 58.8 | 73.4 | 82.4 | 90.9 | 71.6 | 78.2 | 94.0 | 68.3 | 81.8 | 67.3 | 92.3 |
| titan | [58.1, | [55.1, | [68.1, | [81.4, | [90.2, | [69.4, | [75.5, | [86.8, | [67.0, | [80.8, | [66.0, | [91.8, |
| | 71.5] | 62.8] | 78.8] | 83.4] | 91.7] | 73.9] | 80.8] | 100.0] | 69.6] | 82.8] | 68.5] | 92.8] |
| | 33.2 | 45.4 | 36.6 | 34.2 | 37.9 | 23.3 | 54.9 | 65.2 | 47.0 | 49.3 | 54.8 | 49.6 |
| keep | [25.1, | [41.3, | [30.9, | [32.9, | [36.8, | [20.9, | [51.7, | [47.8, | [45.7, | [48.1, | [53.4, | [48.7, |
| | 41.5] | 49.5] | 42.6] | 35.4] | 39.1] | 25.7] | 57.9] | 81.3] | 48.3] | 50.6] | 56.2] | 50.6] |
| | 44.0 | 51.1 | 64.4 | 66.7 | 86.7 | 56.2 | 76.3 | 88.0 | 65.3 | 84.9 | 64.9 | 93.8 |
| musk | [35.0, | [47.1, | [59.1, | [65.5, | [85.8, | [53.7, | [73.5, | [78.8, | [64.0, | [83.9, | [63.6, | [93.3, |
| | 53.0] | 55.0] | 69.7] | 67.9] | 87.6] | 58.6] | 79.1] | 96.0] | 66.6] | 85.9] | 66.2] | 94.3] |
| | 47.8 | 50.8 | 25.8 | 45.6 | 73.2 | 53.0 | 71.9 | 79.8 | 59.7 | 70.4 | 52.0 | 49.3 |
| plip | [39.1, | [47.0, | [20.6, | [44.4, | [72.1, | [50.8, | [69.1, | [66.8, | [58.3, | [69.2, | [50.8, | [48.5, |
| | 56.7] | 54.6] | 31.1] | 46.9] | 74.3] | 55.4] | 74.6] | 91.1] | 61.0] | 71.6] | 53.2] | 50.1] |
| | 44.2 | 48.5 | 40.1 | 34.1 | 70.4 | 44.9 | 55.5 | 77.5 | 44.8 | 67.7 | 49.8 | 45.7 |
| quilt | [35.3, | [44.6, | [34.4, | [32.8, | [69.4, | [42.8, | [52.4, | [61.2, | [43.5, | [66.5, | [48.5, | [44.7, |
| | 53.4] | 52.6] | 45.9] | 35.3] | 71.4] | 46.9] | 58.6] | 91.1] | 46.1] | 68.9] | 51.0] | 46.6] |
| | 58.3 | 50.3 | 64.1 | 75.0 | 77.6 | 51.3 | 69.5 | 73.5 | 63.6 | 76.7 | 51.3 | 89.6 |
| dinob | [50.3, | [46.5, | [58.1, | [74.0, | [76.5, | [48.9, | [66.4, | [57.9, | [62.3, | [75.6, | [49.9, | [89.0, |
| | 66.1] | 54.1] | 69.6] | 76.2] | 78.5] | 53.6] | 72.6] | 87.7] | 64.9] | 77.8] | 52.6] | 90.2] |
| | 62.0 | 42.8 | 61.4 | 73.5 | 75.1 | 48.0 | 72.1 | 77.5 | 64.0 | 74.6 | 53.6 | 82.6 |
| dinol | [53.7, | [38.8, | [55.8, | [72.4, | [74.0, | [45.8, | [69.4, | [61.2, | [62.7, | [73.4, | [52.2, | [81.9, |
| | 70.1] | 46.6] | 66.9] | 74.6] | 76.1] | 50.4] | 74.9] | 91.1] | 65.3] | 75.7] | 55.0] | 83.3] |
| | 37.9 | 41.4 | 46.3 | 43.9 | 53.4 | 40.6 | 52.7 | 75.7 | 50.6 | 41.2 | 45.7 | 46.9 |
| vitb | [29.3, | [37.4, | [40.3, | [42.7, | [52.2, | [38.4, | [49.7, | [60.6, | [49.3, | [40.0, | [44.3, | [45.9, |
| | 46.5] | 45.4] | 52.1] | 45.2] | 54.6] | 42.8] | 55.8] | 89.1] | 52.0] | 42.5] | 47.0] | 47.9] |
| | 41.7 | 35.0 | 46.4 | 36.4 | 51.7 | 36.0 | 45.0 | 67.5 | 40.1 | 36.2 | 44.4 | 49.4 |
| vitl | [33.2, | [31.2, | [40.4, | [35.2, | [50.5, | [33.8, | [42.1, | [51.0, | [38.7, | [35.0, | [43.0, | [48.5, |
| | 50.5] | 38.8] | 52.5] | 37.7] | 52.9] | 38.4] | 47.9] | 82.2] | 41.4] | 37.4] | 45.6] | 50.4] |
| | 47.4 | 50.8 | 29.1 | 65.4 | 80.1 | 58.0 | 63.9 | 90.0 | 58.8 | 77.2 | 46.0 | 74.5 |
| clipb | [38.9, | [47.1, | [24.9, | [64.1, | [79.1, | [56.0, | [60.9, | [81.6, | [57.6, | [76.0, | [44.8, | [73.6, |
| | 55.7] | 54.6] | 33.9] | 66.6] | 81.1] | 60.0] | 66.8] | 97.4] | 60.1] | 78.4] | 47.3] | 75.3] |
| | 58.8 | 55.8 | 48.1 | 73.7 | 82.2 | 67.3 | 75.6 | 83.7 | 58.5 | 78.7 | 52.9 | 91.7 |
| clipl | [50.4, | [52.0, | [42.5, | [72.7, | [81.3, | [65.0, | [72.7, | [69.7, | [57.2, | [77.7, | [51.5, | [91.1, |
| | 67.2] | 59.6] | 54.3] | 74.9] | 83.1] | 69.5] | 78.4] | 95.5] | 59.9] | 79.9] | 54.2] | 92.2] |

Table S60: Quantitative performance (Drop in F1-score) on adversarial attack ($\epsilon = 1.5 \cdot 10^{-3}$).

| Model | bach | bracs | break-h | ccrcc | crc | esca | mhist | pcam | tcga-crc | tcga-tils | tcga-unif | wilds |
|---|---|---|---|---|---|---|---|---|---|---|---|---|
| | 50.9 | 41.0 | 43.8 | 43.2 | 55.5 | 34.7 | 61.4 | 90.2 | 48.9 | 55.4 | 62.0 | 46.3 |
| hiboub | [42.5, 59.0] | [37.0, 45.0] | [37.4, 49.7] | [42.0, 44.5] | [54.3, 56.7] | [32.5, 36.8] | [58.3, 64.4] | [82.2, 97.4] | [47.7, 50.1] | [54.4, 56.4] | [60.6, 63.4] | [45.3, 47.3] |
| | 39.4 | 51.1 | 35.1 | 31.1 | 35.7 | 27.6 | 62.1 | 56.1 | 35.5 | 32.9 | 55.8 | 17.2 |
| hiboul | [31.3, 47.7] | [46.9, 55.4] | [29.3, 41.2] | [30.0, 32.3] | [34.6, 36.8] | [25.5, 29.7] | [59.1, 65.0] | [41.0, 71.6] | [34.5, 36.5] | [31.8, 33.9] | [54.3, 57.1] | [16.5, 18.0] |
| | 40.8 | 43.7 | 28.0 | 33.3 | 28.9 | 28.6 | 59.9 | 100.0 | 40.5 | 43.4 | 51.8 | 31.0 |
| hopt0 | [32.4, 49.3] | [39.7, 47.6] | [22.4, 33.3] | [32.1, 34.5] | [27.8, 30.1] | [26.4, 30.8] | [56.9, 62.8] | [100.0, 100.0] | [39.4, 41.6] | [42.3, 44.4] | [50.4, 53.2] | [30.0, 31.9] |
| | 51.4 | 57.0 | 62.2 | 35.7 | 42.0 | 32.1 | 76.8 | 100.0 | 50.5 | 48.5 | 62.7 | 76.7 |
| hopt1 | [42.6, 59.7] | [52.7, 61.1] | [56.4, 67.5] | [34.5, 36.9] | [40.9, 43.2] | [29.8, 34.3] | [74.0, 79.6] | [100.0, 100.0] | [49.4, 51.6] | [47.5, 49.5] | [61.3, 64.1] | [76.0, 77.5] |
| | 37.6 | 47.0 | 38.1 | 31.1 | 29.7 | 28.9 | 61.0 | 57.4 | 26.2 | 19.6 | 36.6 | 23.0 |
| midnight | [29.7, 45.8] | [42.8, 51.1] | [31.8, 44.3] | [30.0, 32.3] | [28.6, 30.7] | [26.6, 31.1] | [58.0, 63.8] | [41.5, 72.5] | [25.3, 27.2] | [18.7, 20.5] | [35.4, 37.9] | [22.2, 23.8] |
| | 37.8 | 31.8 | 34.5 | 25.3 | 20.0 | 17.7 | 58.3 | 52.2 | 36.7 | 36.5 | 46.2 | 16.1 |
| phikon | [29.6, 45.9] | [27.9, 35.5] | [28.6, 40.4] | [24.2, 26.5] | [19.1, 21.0] | [16.0, 19.6] | [55.2, 61.2] | [35.8, 68.0] | [35.7, 37.8] | [35.5, 37.6] | [44.7, 47.6] | [15.4, 16.9] |
| | 43.7 | 42.2 | 27.8 | 37.6 | 44.7 | 29.3 | 65.2 | 69.9 | 38.9 | 35.6 | 52.5 | 60.3 |
| phikon2 | [35.2, 51.8] | [38.1, 46.5] | [21.9, 33.6] | [36.4, 38.8] | [43.5, 45.9] | [27.1, 31.6] | [62.1, 68.1] | [54.2, 84.1] | [37.7, 40.0] | [34.5, 36.6] | [51.1, 53.9] | [59.3, 61.2] |
| | 35.8 | 33.9 | 34.3 | 26.6 | 28.8 | 24.8 | 60.8 | 68.9 | 41.0 | 53.6 | 49.4 | 55.5 |
| uni | [27.7, 43.9] | [30.1, 37.8] | [28.8, 39.5] | [25.5, 27.8] | [27.8, 29.9] | [22.8, 26.9] | [57.9, 63.6] | [55.4, 82.2] | [39.9, 42.1] | [52.6, 54.6] | [48.0, 50.7] | [54.6, 56.5] |
| | 27.7 | 40.3 | 28.9 | 24.1 | 31.6 | 18.4 | 61.7 | 42.5 | 31.1 | 40.4 | 41.0 | 24.1 |
| uni2h | [20.3, 35.4] | [36.4, 44.2] | [23.6, 34.5] | [23.0, 25.1] | [30.5, 32.7] | [16.7, 20.1] | [58.5, 64.7] | [27.2, 59.9] | [30.2, 32.1] | [39.3, 41.4] | [39.6, 42.3] | [23.2, 24.9] |
| | 38.8 | 40.1 | 41.9 | 28.7 | 28.0 | 20.9 | 64.0 | 65.0 | 38.3 | 34.9 | 53.0 | 39.1 |
| virchow | [30.5, 47.0] | [36.2, 44.2] | [35.6, 48.1] | [27.5, 29.8] | [26.9, 29.1] | [18.9, 22.8] | [61.1, 66.9] | [49.8, 78.9] | [37.2, 39.3] | [33.7, 35.9] | [51.5, 54.3] | [38.2, 40.1] |
| | 20.5 | 31.5 | 25.8 | 19.4 | 21.0 | 17.2 | 42.3 | 58.2 | 29.8 | 43.0 | 43.5 | 51.6 |
| virchow2 | [13.8, 27.1] | [27.9, 35.4] | [20.6, 31.3] | [18.4, 20.4] | [20.0, 22.0] | [15.4, 19.1] | [39.3, 45.4] | [43.0, 73.6] | [28.8, 30.8] | [42.0, 44.1] | [42.1, 44.8] | [50.7, 52.6] |
| | 36.9 | 48.0 | 53.6 | 61.0 | 62.3 | 43.5 | 69.0 | 75.3 | 47.3 | 54.9 | 60.8 | 47.2 |
| conch | [29.0, 44.9] | [43.9, 52.1] | [47.8, 58.9] | [59.7, 62.2] | [61.2, 63.4] | [41.4, 45.5] | [66.2, 71.9] | [60.8, 88.8] | [46.2, 48.4] | [53.9, 55.9] | [59.5, 62.0] | [46.2, 48.2] |
| | 60.2 | 56.7 | 73.9 | 82.0 | 90.7 | 69.1 | 77.5 | 91.4 | 61.3 | 80.6 | 67.7 | 92.3 |
| titan | [52.1, 67.9] | [52.6, 60.9] | [69.0, 78.8] | [81.0, 83.0] | [89.2, 91.4] | [67.0, 71.0] | [74.8, 80.2] | [80.6, 100.0] | [60.2, 62.3] | [79.7, 81.6] | [66.5, 68.8] | [91.7, 92.8] |
| | 37.9 | 42.7 | 37.6 | 33.6 | 39.5 | 25.0 | 55.4 | 67.0 | 41.2 | 50.9 | 55.3 | 49.9 |
| keep | [30.1, 46.0] | [38.6, 46.7] | [31.8, 43.4] | [32.4, 34.8] | [38.4, 40.6] | [22.9, 27.1] | [52.4, 58.3] | [51.2, 81.3] | [40.2, 42.3] | [50.0, 52.0] | [53.9, 56.7] | [48.9, 50.9] |
| | 41.1 | 48.8 | 68.3 | 67.4 | 85.3 | 53.8 | 76.4 | 83.2 | 60.7 | 86.2 | 66.2 | 93.8 |
| musk | [32.6, 50.0] | [44.7, 52.7] | [62.4, 73.8] | [66.3, 68.6] | [84.4, 86.3] | [51.6, 55.9] | [73.6, 79.1] | [68.8, 94.5] | [59.6, 61.7] | [85.3, 87.1] | [64.9, 67.3] | [93.3, 94.3] |
| | 48.3 | 49.2 | 32.5 | 48.2 | 72.3 | 53.2 | 70.4 | 75.3 | 52.1 | 68.3 | 53.6 | 58.9 |
| plip | [39.6, 56.9] | [45.2, 53.2] | [25.5, 38.9] | [47.0, 49.4] | [71.2, 73.3] | [50.9, 55.4] | [67.3, 73.1] | [60.5, 88.7] | [51.1, 53.2] | [67.3, 69.3] | [52.2, 54.8] | [58.1, 59.6] |
| | 45.0 | 46.0 | 46.8 | 41.2 | 72.8 | 43.7 | 54.9 | 76.9 | 37.2 | 70.8 | 51.0 | 46.3 |
| quilt | [36.0, 53.5] | [41.9, 50.1] | [40.6, 52.3] | [39.9, 42.6] | [71.9, 73.7] | [41.4, 45.9] | [51.9, 57.9] | [60.8, 89.5] | [36.1, 38.2] | [69.8, 71.8] | [49.6, 52.2] | [45.3, 47.3] |
| | 55.4 | 49.2 | 65.4 | 75.2 | 74.6 | 53.4 | 69.1 | 72.6 | 56.3 | 77.0 | 52.0 | 89.5 |
| dinob | [46.6, 63.5] | [45.3, 52.9] | [59.4, 70.6] | [74.1, 76.3] | [73.6, 75.6] | [51.3, 55.4] | [66.0, 72.1] | [58.2, 86.5] | [55.3, 57.4] | [76.0, 78.1] | [50.6, 53.2] | [88.9, 90.1] |
| | 59.9 | 41.2 | 66.1 | 72.6 | 71.9 | 47.7 | 70.4 | 77.6 | 56.0 | 74.2 | 54.0 | 82.7 |
| dinol | [51.5, 68.1] | [37.1, 45.1] | [60.3, 71.1] | [71.4, 73.7] | [70.8, 72.9] | [45.5, 50.0] | [67.6, 73.2] | [62.8, 90.2] | [55.0, 57.1] | [73.1, 75.3] | [52.6, 55.1] | [82.0, 83.4] |
| | 33.8 | 39.8 | 48.7 | 43.2 | 51.3 | 38.2 | 53.9 | 72.1 | 44.0 | 43.7 | 46.6 | 46.8 |
| vitb | [25.7, 41.8] | [35.6, 43.8] | [42.6, 54.3] | [42.0, 44.5] | [50.2, 52.4] | [36.2, 40.3] | [51.0, 56.8] | [56.4, 86.2] | [43.0, 45.1] | [42.6, 44.8] | [45.1, 47.9] | [45.8, 47.8] |
| | 41.1 | 33.4 | 49.0 | 36.6 | 52.4 | 35.1 | 49.7 | 64.4 | 32.5 | 39.2 | 45.6 | 49.5 |
| vitl | [32.6, 49.9] | [29.5, 37.1] | [43.0, 54.8] | [35.4, 37.8] | [51.3, 53.6] | [32.9, 37.3] | [46.9, 52.3] | [48.5, 79.2] | [31.5, 33.5] | [38.2, 40.2] | [44.2, 46.9] | [48.6, 50.5] |
| | 45.4 | 49.9 | 30.7 | 66.3 | 79.0 | 55.2 | 62.6 | 85.8 | 48.1 | 80.1 | 47.0 | 74.9 |
| clipb | [36.9, 53.5] | [46.0, 53.5] | [24.7, 36.6] | [65.1, 67.5] | [77.9, 80.0] | [53.3, 57.1] | [59.6, 65.5] | [72.8, 96.7] | [47.2, 49.1] | [79.1, 81.2] | [45.6, 48.3] | [74.1, 75.8] |
| | 55.9 | 54.5 | 51.4 | 71.8 | 79.9 | 66.9 | 75.0 | 82.2 | 51.1 | 78.7 | 54.1 | 91.6 |
| clipl | [47.1, 64.1] | [50.5, 58.4] | [44.9, 57.5] | [70.7, 72.9] | [79.0, 80.9] | [64.5, 68.9] | [72.1, 77.8] | [67.9, 94.1] | [50.1, 52.1] | [77.7, 79.8] | [52.6, 55.3] | [91.1, 92.2] |

Table S61: Quantitative performance (Drop in Balanced accuracy) on adversarial attack ($\epsilon = 35 \cdot 10^{-3}$).

| Model | bach | bracs | break-h | ccrcc | crc | esca | mhist | pcam | tcga-crc | tcga-tils | tcga-unif | wilds |
|---|---|---|---|---|---|---|---|---|---|---|---|---|
| hiboub | 70.7 | 62.0 | 54.8 | 86.1 | 94.0 | 77.9 | 75.1 | 100.0 | 70.6 | 87.4 | 69.8 | 97.2 |
| | [64.2, | [58.4, | [48.9, | [85.3, | [93.4, | [75.7, | [72.2, | [100.0, | [69.4, | [86.4, | [68.6, | [96.9, |
| | 77.1] | 65.6] | 60.9] | 86.9] | 94.6] | 80.2] | 77.9] | 100.0] | 71.9] | 88.3] | 71.2] | 97.5] |
| hiboul | 74.5 | 62.8 | 64.1 | 77.8 | 80.2 | 63.1 | 79.2 | 100.0 | 73.7 | 78.2 | 74.0 | 85.4 |
| | [67.0, | [59.1, | [58.7, | [76.7, | [79.2, | [60.7, | [76.4, | [100.0, | [72.5, | [77.1, | [72.8, | [84.7, |
| | 81.7] | 66.6] | 69.9] | 78.9] | 81.1] | 65.5] | 81.7] | 100.0] | 74.9] | 79.2] | 75.3] | 86.0] |
| hopt0 | 70.2 | 60.3 | 59.2 | 85.8 | 85.8 | 72.2 | 81.5 | 100.0 | 73.1 | 85.1 | 76.8 | 97.7 |
| | [62.2, | [56.3, | [53.2, | [85.0, | [84.9, | [70.0, | [79.0, | [100.0, | [71.9, | [84.1, | [75.6, | [97.4, |
| | 77.8] | 64.1] | 65.2] | 86.7] | 86.6] | 74.4] | 83.9] | 100.0] | 74.4] | 86.1] | 78.1] | 98.0] |
| hopt1 | 75.2 | 65.2 | 74.6 | 87.2 | 88.5 | 70.9 | 81.2 | 100.0 | 74.7 | 87.7 | 79.2 | 98.3 |
| | [68.5, | [61.6, | [68.9, | [86.4, | [87.7, | [68.7, | [78.5, | [100.0, | [73.5, | [86.7, | [78.0, | [98.1, |
| | 81.8] | 68.8] | 80.0] | 88.0] | 89.3] | 73.2] | 83.9] | 100.0] | 75.9] | 88.6] | 80.3] | 98.6] |
| midnight | 70.9 | 63.6 | 48.8 | 58.1 | 75.7 | 62.0 | 76.1 | 94.0 | 57.5 | 50.9 | 68.2 | 85.4 |
| | [63.5, | [60.0, | [43.4, | [56.8, | [74.7, | [59.6, | [73.3, | [87.0, | [56.2, | [49.7, | [67.0, | [84.6, |
| | 77.7] | 67.2] | 54.3] | 59.4] | 76.7] | 64.3] | 78.7] | 100.0] | 58.8] | 52.2] | 69.4] | 86.0] |
| phikon | 69.4 | 57.2 | 54.5 | 77.3 | 77.8 | 63.0 | 77.8 | 91.8 | 68.4 | 86.6 | 74.8 | 84.7 |
| | [61.3, | [53.5, | [48.6, | [76.3, | [76.7, | [60.8, | [75.0, | [80.6, | [67.1, | [85.6, | [73.5, | [84.0, |
| | 77.3] | 61.0] | 60.3] | 78.3] | 78.8] | 65.1] | 80.4] | 100.0] | 69.7] | 87.6] | 76.1] | 85.4] |
| phikon2 | 67.3 | 59.1 | 49.3 | 77.9 | 87.4 | 70.2 | 78.9 | 94.0 | 64.2 | 86.0 | 76.6 | 95.6 |
| | [59.5, | [55.3, | [43.6, | [77.1, | [86.5, | [67.8, | [76.3, | [87.0, | [63.0, | [85.1, | [75.5, | [95.2, |
| | 74.9] | 62.6] | 55.5] | 78.8] | 88.2] | 72.5] | 81.6] | 100.0] | 65.5] | 87.0] | 77.8] | 96.0] |
| uni | 73.1 | 59.2 | 63.0 | 85.4 | 82.7 | 67.4 | 81.3 | 100.0 | 71.0 | 87.3 | 73.2 | 97.3 |
| | [65.4, | [55.5, | [57.1, | [84.6, | [81.8, | [65.0, | [78.7, | [100.0, | [69.8, | [86.4, | [71.8, | [96.9, |
| | 80.4] | 62.8] | 68.9] | 86.3] | 83.6] | 69.7] | 83.7] | 100.0] | 72.3] | 88.3] | 74.4] | 97.6] |
| uni2h | 77.5 | 63.2 | 72.5 | 68.4 | 80.3 | 60.7 | 77.2 | 98.0 | 73.2 | 88.2 | 74.2 | 96.6 |
| | [70.3, | [59.5, | [67.1, | [67.3, | [79.4, | [58.4, | [74.6, | [93.5, | [71.9, | [87.2, | [73.1, | [96.3, |
| | 84.3] | 67.1] | 78.0] | 69.6] | 81.2] | 63.0] | 80.0] | 100.0] | 74.4] | 89.2] | 75.4] | 97.0] |
| virchow | 65.4 | 59.2 | 60.0 | 77.1 | 77.8 | 65.7 | 81.5 | 96.0 | 70.4 | 80.2 | 73.2 | 88.9 |
| | [57.6, | [55.5, | [54.1, | [76.0, | [76.9, | [63.6, | [79.0, | [90.0, | [69.1, | [79.2, | [71.9, | [88.3, |
| | 73.4] | 63.0] | 65.8] | 78.1] | 78.8] | 67.7] | 84.0] | 100.0] | 71.6] | 81.3] | 74.5] | 89.5] |
| virchow2 | 72.7 | 60.7 | 65.2 | 70.2 | 75.9 | 52.1 | 76.7 | 91.8 | 72.8 | 88.5 | 70.6 | 88.5 |
| | [65.1, | [57.1, | [59.9, | [69.2, | [74.8, | [49.8, | [74.1, | [80.9, | [71.6, | [87.6, | [69.4, | [87.9, |
| | 80.0] | 64.4] | 70.7] | 71.3] | 77.0] | 54.3] | 79.2] | 100.0] | 74.1] | 89.4] | 71.9] | 89.2] |
| conch | 87.3 | 59.9 | 65.8 | 89.6 | 93.5 | 80.9 | 79.8 | 94.0 | 68.6 | 83.6 | 66.6 | 97.2 |
| | [82.1, | [56.3, | [60.4, | [88.9, | [92.8, | [78.8, | [77.1, | [87.0, | [67.4, | [82.5, | [65.4, | [96.9, |
| | 92.0] | 63.7] | 71.4] | 90.3] | 94.1] | 83.1] | 82.4] | 100.0] | 69.9] | 84.6] | 67.9] | 97.5] |
| titan | 83.2 | 63.2 | 74.6 | 88.3 | 94.3 | 83.0 | 81.2 | 96.0 | 68.6 | 87.6 | 67.8 | 96.7 |
| | [79.2, | [59.8, | [69.4, | [87.6, | [93.7, | [80.8, | [78.7, | [89.6, | [67.3, | [86.6, | [66.5, | [96.4, |
| | 87.3] | 66.8] | 80.0] | 89.1] | 94.9] | 85.2] | 83.8] | 100.0] | 69.8] | 88.5] | 69.1] | 97.1] |
| keep | 75.6 | 63.1 | 62.7 | 83.5 | 93.6 | 71.7 | 74.2 | 96.0 | 71.3 | 86.8 | 71.0 | 97.4 |
| | [69.1, | [59.5, | [56.9, | [82.7, | [93.0, | [69.7, | [71.3, | [90.0, | [70.0, | [85.9, | [69.7, | [97.1, |
| | 82.0] | 66.7] | 68.5] | 84.4] | 94.3] | 73.8] | 77.0] | 100.0] | 72.5] | 87.8] | 72.2] | 97.8] |
| musk | 70.7 | 63.9 | 73.2 | 84.7 | 90.9 | 77.8 | 79.1 | 88.0 | 68.1 | 86.5 | 65.2 | 96.1 |
| | [63.5, | [60.2, | [68.3, | [83.8, | [90.1, | [75.5, | [76.3, | [78.8, | [66.8, | [85.5, | [63.9, | [95.7, |
| | 77.7] | 67.3] | 78.3] | 85.6] | 91.7] | 80.0] | 81.7] | 96.0] | 69.4] | 87.6] | 66.5] | 96.4] |
| plip | 64.3 | 57.3 | 40.1 | 74.8 | 86.9 | 68.5 | 78.4 | 90.0 | 62.7 | 85.2 | 52.6 | 88.0 |
| | [56.1, | [53.6, | [35.1, | [73.7, | [86.0, | [66.4, | [75.8, | [81.6, | [61.4, | [84.2, | [51.3, | [87.4, |
| | 72.4] | 61.0] | 45.4] | 75.9] | 87.8] | 70.7] | 81.1] | 97.6] | 64.0] | 86.2] | 53.8] | 88.6] |
| quilt | 59.3 | 56.9 | 52.9 | 62.0 | 88.7 | 63.6 | 69.2 | 85.7 | 63.9 | 84.4 | 52.5 | 87.5 |
| | [50.9, | [53.3, | [47.1, | [60.8, | [87.9, | [61.6, | [66.2, | [71.6, | [62.7, | [83.4, | [51.2, | [86.8, |
| | 67.7] | 60.5] | 58.6] | 63.2] | 89.5] | 65.4] | 72.3] | 96.7] | 65.2] | 85.5] | 53.7] | 88.1] |
| dinob | 66.9 | 53.3 | 76.4 | 79.5 | 89.6 | 75.0 | 81.4 | 91.8 | 66.3 | 82.5 | 53.1 | 91.0 |
| | [59.1, | [49.5, | [70.7, | [78.5, | [88.8, | [72.8, | [78.8, | [81.2, | [65.1, | [81.4, | [51.7, | [90.4, |
| | 74.3] | 56.9] | 81.7] | 80.6] | 90.4] | 77.0] | 84.0] | 100.0] | 67.6] | 83.6] | 54.4] | 91.5] |
| dinol | 71.1 | 51.7 | 74.7 | 82.6 | 90.1 | 72.2 | 82.2 | 89.7 | 67.2 | 82.1 | 55.4 | 90.7 |
| | [63.3, | [47.9, | [69.4, | [81.7, | [89.4, | [70.1, | [79.8, | [76.7, | [65.9, | [80.9, | [54.1, | [90.1, |
| | 78.6] | 55.5] | 80.1] | 83.6] | 90.9] | 74.3] | 84.6] | 100.0] | 68.4] | 83.2] | 56.8] | 91.3] |
| vitb | 57.1 | 56.9 | 60.0 | 78.8 | 87.9 | 65.8 | 74.3 | 83.8 | 64.8 | 81.0 | 52.6 | 89.6 |
| | [48.9, | [53.2, | [54.2, | [77.8, | [87.1, | [63.7, | [71.4, | [70.8, | [63.5, | [79.9, | [51.3, | [89.0, |
| | 65.5] | 60.8] | 66.0] | 79.8] | 88.8] | 67.8] | 77.2] | 94.2] | 66.1] | 82.1] | 53.9] | 90.2] |
| vitl | 56.8 | 55.3 | 61.0 | 75.4 | 87.6 | 70.0 | 73.8 | 90.0 | 61.9 | 82.7 | 54.0 | 91.2 |
| | [48.7, | [51.5, | [55.1, | [74.3, | [86.8, | [67.7, | [71.0, | [81.5, | [60.6, | [81.7, | [52.7, | [90.6, |
| | 65.0] | 58.9] | 67.0] | 76.5] | 88.4] | 72.3] | 76.7] | 97.5] | 63.2] | 83.8] | 55.3] | 91.7] |
| clipb | 53.0 | 52.7 | 38.8 | 72.8 | 87.8 | 62.2 | 76.3 | 90.0 | 60.2 | 80.4 | 46.4 | 83.8 |
| | [44.9, | [49.1, | [34.9, | [71.7, | [87.1, | [60.3, | [73.6, | [81.6, | [58.9, | [79.4, | [45.2, | [83.1, |
| | 61.1] | 56.5] | 43.3] | 74.0] | 88.6] | 64.2] | 79.0] | 97.4] | 61.5] | 81.6] | 47.6] | 84.5] |
| clipl | 66.0 | 57.2 | 51.3 | 79.1 | 87.7 | 68.8 | 79.1 | 83.7 | 59.0 | 82.6 | 53.1 | 93.4 |
| | [58.5, | [53.4, | [45.6, | [78.2, | [86.9, | [66.6, | [76.3, | [69.7, | [57.6, | [81.5, | [51.7, | [93.0, |
| | 73.4] | 61.0] | 57.2] | 80.0] | 88.5] | 71.0] | 81.8] | 95.5] | 60.3] | 83.6] | 54.4] | 93.9] |

Table S62: Quantitative performance (Drop in F1-score) on adversarial attack ($\epsilon = 35 \cdot 10^{-3}$).

| Model | bach | bracs | break-h | ccrcc | crc | esca | mhist | pcam | tcga-crc | tcga-tils | tcga-unif | wilds |
|---|---|---|---|---|---|---|---|---|---|---|---|---|
| | 65.6 | 61.7 | 58.5 | 84.0 | 93.5 | 76.7 | 75.7 | 100.0 | 65.7 | 88.8 | 71.5 | 97.2 |
| hiboub | [57.1, | [57.9, | [51.8, | [83.1, | [92.8, | [74.5, | [72.8, | [100.0, | [64.6, | [87.9, | [70.2, | [96.8, |
| | 73.2] | 65.4] | 64.8] | 85.0] | 94.1] | 78.6] | 78.5] | 100.0] | 66.9] | 89.6] | 72.8] | 97.5] |
| | 71.1 | 62.4 | 65.5 | 80.1 | 78.7 | 58.4 | 78.8 | 100.0 | 66.3 | 77.0 | 75.3 | 86.0 |
| hiboul | [62.9, | [58.6, | [59.9, | [79.1, | [77.7, | [56.4, | [76.1, | [100.0, | [65.3, | [76.0, | [74.1, | [85.4, |
| | 78.8] | 66.2] | 71.0] | 81.1] | 79.7] | 60.4] | 81.4] | 100.0] | 67.4] | 77.9] | 76.5] | 86.7] |
| | 68.7 | 60.6 | 62.2 | 85.4 | 85.5 | 71.4 | 80.1 | 100.0 | 67.1 | 84.8 | 77.6 | 97.7 |
| hopt0 | [60.2, | [56.5, | [55.7, | [84.5, | [84.6, | [69.3, | [77.6, | [100.0, | [66.1, | [83.9, | [76.3, | [97.4, |
| | 76.5] | 64.1] | 68.0] | 86.3] | 86.4] | 73.3] | 82.5] | 100.0] | 68.2] | 85.6] | 78.8] | 98.0] |
| | 71.7 | 64.3 | 76.0 | 86.5 | 88.7 | 70.4 | 81.6 | 100.0 | 67.6 | 88.4 | 80.1 | 98.3 |
| hopt1 | [63.6, | [60.5, | [70.4, | [85.7, | [87.9, | [68.2, | [79.0, | [100.0, | [66.5, | [87.5, | [78.9, | [98.1, |
| | 79.3] | 68.0] | 81.0] | 87.4] | 89.5] | 72.5] | 84.2] | 100.0] | 68.7] | 89.2] | 81.1] | 98.6] |
| | 70.2 | 62.9 | 51.0 | 59.4 | 75.5 | 63.6 | 76.0 | 92.5 | 54.6 | 53.2 | 67.9 | 85.4 |
| midnight | [63.2, | [59.1, | [44.2, | [58.2, | [74.5, | [61.4, | [73.1, | [84.1, | [53.5, | [52.3, | [66.7, | [84.7, |
| | 77.2] | 66.6] | 57.3] | 60.7] | 76.4] | 65.6] | 78.6] | 100.0] | 55.7] | 54.2] | 69.0] | 86.1] |
| | 68.4 | 56.5 | 51.9 | 79.5 | 78.0 | 61.4 | 77.4 | 90.9 | 62.7 | 86.2 | 76.1 | 85.3 |
| phikon | [60.0, | [52.5, | [45.9, | [78.6, | [77.0, | [59.8, | [74.6, | [79.4, | [61.6, | [85.4, | [74.8, | [84.7, |
| | 75.8] | 60.1] | 57.7] | 80.5] | 79.0] | 63.0] | 80.0] | 100.0] | 63.8] | 87.1] | 77.3] | 86.0] |
| | 63.9 | 58.2 | 48.7 | 73.9 | 86.8 | 67.4 | 78.6 | 91.3 | 60.6 | 86.6 | 77.6 | 95.6 |
| phikon2 | [55.5, | [54.3, | [42.4, | [72.7, | [85.9, | [65.5, | [75.9, | [80.6, | [59.5, | [85.7, | [76.3, | [95.2, |
| | 71.6] | 61.8] | 54.9] | 75.0] | 87.7] | 69.1] | 81.3] | 100.0] | 61.7] | 87.4] | 78.7] | 96.0] |
| | 71.1 | 59.0 | 64.2 | 84.4 | 80.9 | 66.9 | 80.9 | 100.0 | 65.7 | 88.8 | 74.4 | 97.2 |
| uni | [63.4, | [55.1, | [58.6, | [83.5, | [80.0, | [64.9, | [78.3, | [100.0, | [64.6, | [87.9, | [73.0, | [96.9, |
| | 78.6] | 62.5] | 69.4] | 85.3] | 81.9] | 68.7] | 83.3] | 100.0] | 66.8] | 89.6] | 75.5] | 97.6] |
| | 75.0 | 61.2 | 74.5 | 68.6 | 81.3 | 59.2 | 77.7 | 97.4 | 66.0 | 88.4 | 74.2 | 96.7 |
| uni2h | [67.0, | [57.2, | [69.0, | [67.5, | [80.4, | [57.0, | [75.0, | [92.5, | [64.9, | [87.6, | [73.0, | [96.3, |
| | 82.4] | 65.1] | 79.8] | 69.7] | 82.2] | 61.2] | 80.4] | 100.0] | 67.0] | 89.3] | 75.4] | 97.0] |
| | 64.3 | 58.5 | 62.0 | 76.1 | 78.5 | 64.1 | 80.8 | 94.1 | 64.9 | 78.6 | 74.1 | 89.4 |
| virchow | [55.9, | [54.5, | [55.9, | [75.0, | [77.5, | [62.3, | [78.2, | [84.9, | [63.8, | [77.7, | [72.8, | [88.9, |
| | 72.1] | 62.3] | 67.7] | 77.1] | 79.4] | 65.8] | 83.3] | 100.0] | 65.9] | 79.5] | 75.3] | 90.0] |
| | 71.4 | 57.8 | 66.5 | 70.2 | 76.2 | 50.8 | 74.4 | 90.9 | 67.5 | 88.9 | 70.2 | 88.5 |
| virchow2 | [63.7, | [53.8, | [60.6, | [69.2, | [75.1, | [48.5, | [71.7, | [79.4, | [66.4, | [88.1, | [68.9, | [87.9, |
| | 78.8] | 61.7] | 72.1] | 71.3] | 77.3] | 53.1] | 77.0] | 100.0] | 68.7] | 89.7] | 71.4] | 89.2] |
| | 84.4 | 59.8 | 68.6 | 88.1 | 92.4 | 79.0 | 79.4 | 91.3 | 63.9 | 86.4 | 67.6 | 97.2 |
| conch | [78.0, | [56.0, | [63.0, | [87.3, | [91.7, | [77.2, | [76.8, | [80.2, | [62.8, | [85.5, | [66.3, | [96.9, |
| | 90.0] | 63.4] | 74.0] | 89.0] | 93.1] | 80.7] | 82.0] | 100.0] | 64.9] | 87.3] | 68.7] | 97.5] |
| | 76.8 | 62.4 | 75.3 | 87.4 | 93.8 | 81.9 | 81.2 | 94.1 | 61.6 | 88.9 | 68.3 | 96.7 |
| titan | [70.0, | [58.6, | [70.4, | [86.5, | [93.1, | [79.9, | [78.7, | [84.1, | [60.5, | [88.1, | [67.0, | [96.4, |
| | 83.1] | 66.1] | 80.2] | 88.2] | 94.4] | 83.6] | 83.7] | 100.0] | 62.6] | 89.7] | 69.4] | 97.1] |
| | 72.1 | 62.2 | 65.4 | 82.9 | 92.7 | 71.0 | 74.3 | 94.1 | 64.3 | 88.1 | 72.1 | 97.4 |
| keep | [64.6, | [58.4, | [59.9, | [82.0, | [92.0, | [69.2, | [71.4, | [84.9, | [63.2, | [87.3, | [70.8, | [97.1, |
| | 79.3] | 65.9] | 70.4] | 83.8] | 93.4] | 72.7] | 77.1] | 100.0] | 65.4] | 89.0] | 73.3] | 97.7] |
| | 66.2 | 62.8 | 77.3 | 85.3 | 91.0 | 74.6 | 79.8 | 83.2 | 62.8 | 88.7 | 66.5 | 96.0 |
| musk | [57.9, | [59.0, | [71.7, | [84.4, | [90.2, | [72.7, | [77.0, | [68.8, | [61.7, | [87.9, | [65.2, | [95.6, |
| | 74.2] | 66.3] | 82.3] | 86.1] | 91.7] | 76.2] | 82.3] | 94.5] | 63.9] | 89.6] | 67.7] | 96.4] |
| | 63.9 | 56.5 | 43.4 | 75.7 | 86.3 | 66.1 | 77.8 | 55.9 | 58.5 | 85.8 | 54.1 | 88.0 |
| plip | [55.2, | [52.7, | [36.4, | [74.6, | [85.4, | [63.8, | [75.1, | [72.8, | [54.9, | [84.9, | [52.7, | [87.3, |
| | 71.7] | 60.1] | 49.9] | 76.7] | 87.2] | 68.1] | 80.4] | 96.8] | 57.0] | 86.7] | 55.3] | 88.6] |
| | 56.9 | 55.9 | 57.2 | 63.4 | 87.3 | 61.8 | 68.3 | 84.9 | 58.2 | 85.9 | 53.4 | 87.8 |
| quilt | [48.2, | [52.1, | [50.8, | [62.0, | [86.4, | [59.5, | [65.4, | [70.4, | [57.2, | [85.0, | [52.0, | [87.2, |
| | 65.1] | 59.5] | 62.8] | 64.7] | 88.1] | 64.0] | 71.5] | 96.7] | 59.3] | 86.8] | 54.6] | 88.4] |
| | 64.4 | 52.9 | 77.8 | 79.7 | 88.9 | 74.0 | 82.1 | 90.9 | 58.1 | 84.7 | 54.0 | 90.9 |
| dinob | [55.9, | [49.1, | [72.1, | [78.7, | [88.1, | [72.1, | [79.4, | [79.6, | [57.1, | [83.7, | [52.6, | [90.3, |
| | 72.3] | 56.4] | 82.7] | 80.8] | 89.8] | 75.7] | 84.6] | 100.0] | 59.2] | 85.7] | 55.2] | 91.5] |
| | 68.9 | 50.6 | 78.4 | 81.4 | 88.7 | 71.4 | 81.4 | 90.5 | 58.9 | 84.8 | 56.0 | 90.6 |
| dinol | [60.3, | [46.6, | [72.7, | [80.4, | [87.9, | [69.4, | [79.0, | [77.9, | [57.9, | [83.8, | [54.6, | [90.0, |
| | 76.6] | 54.2] | 83.4] | 82.4] | 89.5] | 73.2] | 83.8] | 100.0] | 59.9] | 85.7] | 57.2] | 91.1] |
| | 53.1 | 56.6 | 60.2 | 77.9 | 85.6 | 62.2 | 74.1 | 80.2 | 58.4 | 83.2 | 53.7 | 89.6 |
| vitb | [44.4, | [52.9, | [54.2, | [76.9, | [84.7, | [60.3, | [71.1, | [64.8, | [57.4, | [82.2, | [52.3, | [89.0, |
| | 61.6] | 60.3] | 66.1] | 79.0] | 86.6] | 63.9] | 76.9] | 91.8] | 59.5] | 84.1] | 54.9] | 90.2] |
| | 55.3 | 54.3 | 61.1 | 75.1 | 85.6 | 68.0 | 73.5 | 85.8 | 55.1 | 83.6 | 55.7 | 91.1 |
| vitl | [46.8, | [50.2, | [54.9, | [74.0, | [84.7, | [65.7, | [70.6, | [72.6, | [54.1, | [82.6, | [54.3, | [90.6, |
| | 63.6] | 57.9] | 66.8] | 76.2] | 86.5] | 70.1] | 76.3] | 96.5] | 56.2] | 84.5] | 56.9] | 91.7] |
| | 51.1 | 52.4 | 39.7 | 73.7 | 86.7 | 60.0 | 75.7 | 85.8 | 48.7 | 82.8 | 47.3 | 83.6 |
| clipb | [43.0, | [48.6, | [33.7, | [72.5, | [85.8, | [58.1, | [72.8, | [72.8, | [47.8, | [81.8, | [45.9, | [82.9, |
| | 58.4] | 56.0] | 45.7] | 74.8] | 87.5] | 61.9] | 78.3] | 96.7] | 49.7] | 83.8] | 48.6] | 84.4] |
| | 63.5 | 56.4 | 54.5 | 75.8 | 86.9 | 68.2 | 79.1 | 82.2 | 51.5 | 84.3 | 54.2 | 93.4 |
| clipl | [55.2, | [52.5, | [48.0, | [74.8, | [86.0, | [65.9, | [76.4, | [67.9, | [50.5, | [83.3, | [52.7, | [92.9, |
| | 71.0] | 60.1] | 60.6] | 76.9] | 87.8] | 70.2] | 81.9] | 94.1] | 52.5] | 85.2] | 55.5] | 93.9] |

