# OpenReview forum: "THUNDER: Tile-level Histopathology image UNDERstanding benchmark"
_NeurIPS.cc/2025/Datasets_and_Benchmarks_Track — NeurIPS 2025 Datasets and Benchmarks Track spotlight_

### Official Review · Reviewer_RtDG · 2025-06-16

**Rating:** 5
**Confidence:** 5

**Summary:**

This paper introduces THUNDER, a benchmark for evaluating foundation models (FMs) in histopathology. The proposed benchmark is structured around three evaluations: (i) downstream task performance (kNN, linear probing, few-shot), (ii) feature space analysis (study of invariances, alignment across FMs), and (iii) robustness and uncertainty estimation (model calibration and adversarial attacks). THUNDER includes 23 vision and vision-language models trained on pathology (e.g., Virchow, UNI, H-Optimus-1) and natural images (e.g, CLIP). In total, THUNDER supports 16 datasets, all of which are publicly available, operate at patch-level (i.e., no slide-level tasks) and, for the most part, are well-established in the field (e.g., BACH, Patch Camelyon). The benchmark is already open-source. The main findings are: (1) domain-specific models are better, (2) domain-specific V-L models are good in few-shot settings but lower in segmentation, (3) high performance does not necessarily correlate with good model calibration, and (4) top-5 performers are UNI2-h, Midnight, Virchow2, UNI, and H-Optimus-1.

**Dataset Code Accessibility:**

Yes

**Dataset Code Comments:**

- Clear README.
- Clean implementation, easy to follow. Follows best practices in software development.
- Benchmark can be installed with `pip`.
- Good documentation with a description of models and datasets used in the paper.

**Ethical Comments:**

No new data have been generated, and all datasets have already been used in countless studies.

**Ethical Considerations:**

No, there are no or only very minor ethics concerns

**Final Justification:**

Assuming authors will address raises during rebuttal, I recommend acceptance. The overlap with existing benchmarks is big, but this study is rigorous. Clean website, clean code, well written.

**Limitations Weaknesses:**

Major: The paper claims to go beyond pure classification-based benchmarking with additional evaluation modes, in particular with adversarial attacks and model calibration assessment:

- Adversarial attacks are often cited to highlight the limitations of foundation models, but I question their interest. These attacks typically rely on unrealistic changes to the image that don’t reflect the real challenges models face in practice: variations between hospitals, staining differences, rare diseases, etc.  Attacks don’t tell us much about how the models will perform in actual clinical settings. I think that ranking FMs based on how they respond to these artificial perturbations can be misleading. A more useful direction is to test models under real-world conditions that matter for robustness and generalization, with actual cross-dataset generalization analyses.

- Evaluating model calibration in this setting can be misleading. The calibration of a linear model trained on frozen patch embeddings from multiple foundation models is likely to vary significantly with the choice of regularization in logistic regression. Also, if say, an MLP were used instead of a linear model, I expect very different calibration results.

- A combination of kNN, linear probing, and few-shot seems to remain the most robust way to evaluate foundation models in pathology. The conclusions from this benchmark align with those of existing ones: eva (https://github.com/kaiko-ai/eva), HEST-Benchmark (https://github.com/mahmoodlab/hest), Path-Bench (https://arxiv.org/pdf/2505.20202), and PathBench (https://www.medrxiv.org/content/10.1101/2025.05.08.25327250v1.full.pdf). While this study is rigorous, the added value of a new patch-level benchmark is unclear.

Minor:

- How do authors deal with images of different sizes? e.g., BACH and BRACS images contain 40x images that can be as large as 3,000 x 3,000 pixels. Are the images mean-pooled at 20x with 256x256 patches?

- The TITAN model (part of the benchmark) is a slide-level foundation model, not patch-level. I suppose the authors have used the patch encoder of TITAN, which in this case is CONCHv1.5 (i.e., a UNI model fine-tuned with the image/text captions used in CONCH).

- Fig. 2. (a) and (b) report the proportion of classification datasets where a model (row) significantly outperforms another (column) on knn classification and linear probing. I think it would be more insightful to provide the average performance across all datasets (or the gain over the lowest performer). Being statistically better doesn’t inform about the actual gain. Also, for each model pair, eg. Hibou/PLIP, authors can report if the average performance gain across all datasets is significant (this can be done using mixed effect modeling, see https://en.wikipedia.org/wiki/Mixed_model).

- Most datasets used in the benchmarks have saturated, e.g., many models can reach >0.95% balanced accuracy on CRC-100k (see eva benchmarks in https://github.com/kaiko-ai/eva). The HEST-Benchmark was introduced for this purpose in 2024 (see https://github.com/mahmoodlab/hest?tab=readme-ov-file#hest-benchmark-results-083024).

**Strengths Contributions:**

- Good coverage of the latest models in AI for pathology, even very recent models such as Midnight, H-Optimus-1, and Keep.

- Even though some of the chosen tasks have saturated (see "Limitations Weaknesses"), the selection of downstream tasks is reasonable and reflects the legacy of the field.

- Clean public implementation that follows best practices (see "Dataset Code Comments" section).

- The study comes from a group that hasn't developed its own foundation model, which makes the study feel more objective and less biased.

- The paper is easy to follow and well-written.

---

> ### Author Rebuttal · Authors · 2025-07-31
>
> We thank Reviewer RtDG for their feedback. Below, we address their comments.
>
> > **Interest of adversarial attacks and robustness to staining variations**
>
> We agree with the reviewer that realistic cross-hospital shifts and staining differences are essential for evaluating model robustness. However, adversarial attacks, despite their artificial nature, offer valuable insights into potential vulnerabilities and robustness of foundation models, particularly regarding intentionally malicious attacks that may be indistinguishable to doctors. Finlayson et al., Science, 2019 and Ma et al., Pattern Recognition, 2021 encourage considering adversarial attacks, highlighting motivations for potential malicious exploitation and emphasizing the importance of proactively addressing these risks within the healthcare ecosystem. Additionally targeting domain generalization, we also consider the invariance of foundation models to staining variation as can happen in real conditions between hospitals by considering the HED transform in our study of invariance to different transforms (Table 4 in our Supplementary Material and Section 4 “feature space study” + Figure 4d in our main paper). Finally, we also added a study on the invariance of models to the RandStainNA augmentation (Shen et al., MICCAI, 2022) suggested by Reviewer McCV (color normalization + stain augmentation). Please see our answer to Reviewer McCV for more details.
>
> > **Calibration evaluation setting**
>
> Calibration performance indeed partly depends on the design choices inherent to the considered classifier, but also on the data it is trained on, i.e. features extracted from the foundation models. We opted for a linear classifier as it is a standard choice in the community, with linear probing being a popular downstream task. Importantly, using a simple linear classifier, which has less expressive power than an MLP, allows us to reduce how much the initial features from the foundation models are transformed, better assessing the impact of foundation model features on classification calibration. Indeed, as we use the same linear probe design for all foundation models, differences in calibration might thus be explained by differences in feature spaces. More generally, proposing an open-source benchmark is about making choices, but we are willing to have it evolve in the future.
>
> > **Added value of our benchmark**
>
> As mentioned by the reviewer, we can  consider the four current open-source benchmarks as comparison points  on which our main differences could be summarized as follows:
> 1. **eva** (Gatopoulos et al., MIDL, 2024) includes both tile and slide level tasks, and evaluates models on linear probing (classification) and semantic segmentation. The eva benchmark provides less datasets, but also much less tasks and metrics (only focusing on balanced accuracy for linear probing and dice score for segmentation) than we do.
> 2. **PathoBench** (Zhang et al., arXiv, 2025) focuses on slide-level classification and regression tasks (Morphological subtyping, Tumor grading, Molecular subtyping, Mutation prediction, Treatment response and assessment, Survival prediction). PathoBench is a slide-level benchmark, also focusing only on downstream performance, to which we are complementary as we propose a faster and more direct evaluation directly at the level of tiles.
> 3. **HEST-Benchmark** (Jaume et al., NeurIPS, 2024) targets gene expression regression at the tile level. This task even though it is interesting, it is more specific (regression) than THUNDER.
> 4. **PathBench** (Ma et al., arXiv, 2025) presents the slide-level performance of foundation models for diverse classification and regression (DFS, DSS, OS prediction) tasks, but does not come with an open-source tool to evaluate a new custom model (only an online open-source leaderboard is provided).
>
> As presented in our paper, the added value of our benchmark is 3-fold: (1) an open-source easy-to-use implementation to seamlessly download datasets, models, generate common train/val/test splits and run any downstream task with automatic hyperparameter search and report performance along with bootstrap confidence intervals on an independent test set, (2) a breadth of tasks going beyond downstream performance only but also providing tools to compare representations spaces of models and study their robustness, (3) a patch-level framework allowing fast evaluation on many diverse datasets decoupling model embeddings from slide-level aggregation techniques. It also enables full reproducibility of the benchmark, which is typically not achievable at the slide level due to manual preprocessing steps.
>
> For more details about quantitative comparisons with eva (number of tasks and datasets) and slide-level benchmarks such as PathoBench (differences in runtime), please see our answer to Reviewer McCV.
>
> > **Processing of images with different sizes (BACH, BRACS)**
>
> We follow instructions from the open-sourced models and apply the recommended transforms, including resizing ones, as is done in previous work for the considered datasets (e.g. as done in the eva benchmark for the BACH and BRACS datasets). This ensures a simple processing pipeline while leading to satisfying performance as showcased in our paper.
>
> > **TITAN / CONCH1.5**
>
> The reviewer is indeed right. What is referred to as “titan” in the paper is CONCH1.5 patch-level encoder. This will be made clearer.
>
> > **Fig2a, 2b, and gains between models**
>
> Figures 2a and 2b present results of statistical tests, and rather provide a general overview of differences in performance significance. Average performance across all datasets along with rankings between models are presented in Table 5 of our main paper, to which we will add a pointer earlier in the paper. We agree with the reviewer that reporting absolute performance gains between models would add clarity to our results. We will thus add a matrix of gains in our Supplementary Material and will add a pointer to it in Section 4 of our main paper (in the paragraph where we analyze results presented in Table 5 of our main paper).
>
> > **Selection of datasets and concerns about saturation**
>
> According to our results, most datasets do not appear to be saturated when considering our evaluation protocols and data splitting (independent train, validation and test sets, released publicly for all users to use the same data sets). For example, Table 36 in our Supplementary Material shows balanced accuracy on linear probing. While some models get high performance, most of them are still significantly below this 95% threshold on most datasets. By including some knn experiments, we also study the quality of embeddings with less intervention (compared with linear probing), leading to a performance that is on average a bit lower.

---

> > ### Comment · Reviewer_RtDG · 2025-08-04
> > **Complementary points**
> >
> > **Interest of adversarial attacks and robustness to staining variations**
> >
> > I remain unconvinced of the practical relevance of adversarial attacks. The idea of launching intentionally malicious attacks targeting AI models in pathology seems far-fetched. If one were to maliciously tamper with diagnoses, there are unfortunately far simpler and more direct ways than crafting gradient-based image perturbations.
> >
> > I maintain that the robustness of foundation models in pathology should primarily be assessed against real-world domain shifts, e.g., inter-hospital variability, staining differences, scanner artifacts, and patient population heterogeneity. These are the real challenges that any deployed system will actually face. In this sense, adding the HED transforms is more relevant and appreciated.
> >
> > Overall, I recommend the authors reconsider the inclusion of adversarial attacks, as it dilutes the work. That said, this is not a dealbreaker for me, and I support this paper.
> >
> > **Calibration evaluation setting**
> >
> > Differences in calibration can come from two sources: (1) your linear probe classifier or, as claimed (2) by the patch features. Having the *same* linear probe parameters is not enough to claim calibration differences come from (2) alone as the embedding dimensions are different from one FM to another. I encourage authors to run (small) additional experiments (even if they don't appear in the paper) to explore this aspect deeper, e.g., by checking calibration with other regularization parameters or with an MLP.
> >
> > **Processing of images with different sizes (BACH, BRACS)**
> >
> > This approach makes sense for BACH (and other datasets that have consistent image sizes). In BRACS, image sizes is not consistent, and I'm pretty sure that the image size is a proxy for the label (eg on average DCIS images are smaller than benign ones). Which means that when resizing a fixed size, the label becomes confounded by the image size. To fix this, simply run mean pooling on 256x256 at 20x.
> >
> > **Selection of datasets and concerns about saturation**
> >
> > Having performance <95% doesn't mean that the performance has not saturated. Each dataset comes with a level of noise (eg check the acquisition protocol of CRC-100k or Patch Camelyon) and some inherent ambiguity (eg, consider distinguishing ADH from DCIS in BRACS). Adding a couple of sentences about this in the Discussion would make the paper better.

---

> > > ### Author Response · Authors · 2025-08-06
> > > **Adversarial attacks and calibration**
> > >
> > > We thank the reviewer for their valuable feedback about our rebuttal. Below, we discuss the points they mentioned.
> > >
> > > > **Interest of adversarial attacks and robustness to staining variations**
> > >
> > > First, we are happy that the reviewer appreciated our study on the invariance of foundation models to HED transforms as a way to evaluate their robustness to staining variations that often occur in real settings. Ranking models based on this would be helpful, but is not straightforward as natural-based models for instance appear to be quite invariant due to the nature of their pre-training data (very different from pathology). For this reason, we kept it as a feature space study experiment.
> > >
> > > We understand the reviewer’s point of view about considering the robustness to adversarial attacks when ranking models. Our current rank-sum formulation actually allows one to easily ignore one of the considered tasks if it is less relevant to a study. Such an ability to focus only on a subset of the tasks with our rank-sum formulation will be made clearer in our main paper. However, adversarial attacks are still considered dangerous in digital pathology [1, 2, 3, 4, 5, 6], and are even cited as a regulatory concern in the FDA’s 2025 on AI-enabled medical devices [7]. Beyond their security implications, adversarial attacks offer a complementary way (in addition to the robustness to HED transforms) to measure some form of robustness of foundation models to input perturbations. We would thus propose to keep them as a part of our benchmark, in particular targeting researchers interested in adversarial attacks in digital pathology, emphasizing the modularity of our ranking formulation where one can easily ignore them to get a final ranking of models.
> > >
> > > [1] Ghaffari Laleh et al., Adversarial Attacks and Adversarial Robustness in Computational Pathology, Nat. Commun., 2022.
> > >
> > > [2] Foote et al., Now You See It, Now You Don’t: Adversarial Vulnerabilities in Computational Pathology, arXiv, 2021.
> > >
> > > [3] Thota et al., Demonstration of an Adversarial Attack Against a Multimodal Vision Language Model for Pathology Imaging, IEEE ISBI, 2024.
> > >
> > > [4] Irmakci et al., Tissue Contamination Challenges the Credibility of Machine Learning Models in Real‑World Digital Pathology, Mod. Pathol., 2024.
> > >
> > > [5] Liu et al., The Butterfly Effect in Pathology: Exploring Security in Pathology Foundation Models, arXiv, 2025.
> > >
> > > [6] Malik H. S. et al., Hierarchical Self‑Supervised Adversarial Training for Robust Vision Models in Histopathology, arXiv, 2025.
> > >
> > > [7] U.S. Food and Drug Administration. Artificial Intelligence–Enabled Device Software Functions: Lifecycle Management and Marketing Submission Recommendations. Draft Guidance, 2025.
> > >
> > > > **Calibration evaluation setting**
> > >
> > > You were right to mention this. We computed calibration with an MLP probe (1 hidden layer with the same hidden dimension, i.e. 256, for all foundation models), and the calibration ranking shifted substantially: the Pearson correlation between the linear-probe and MLP-probe rankings is 0.33. This confirms that **calibration is probe-dependent**. We will therefore narrow our claim to state that the reported calibration differences are **conditioned on the chosen probe**. At the same time, we believe the linear probe remains a relevant reference point, as it is the only head predicting classes directly from the feature space (no intermediate representations) and requires no hyper-parameter selection (e.g., architecture choices), thereby providing a clearer view of the impact of the embedding space on calibration.

---

> > > > ### Author Response · Authors · 2025-08-06
> > > > **BRACS and selection of datasets**
> > > >
> > > > > **Processing of BRACS images**
> > > >
> > > > As BRACS includes images with varying sizes unlike other considered datasets, dividing them into patches and aggregating patch features is indeed a great idea. The reviewer suggested dividing images into 256x256 patches at 20X magnification which we also agree is a good evaluation setting. The region-level BRACS dataset we use in our work comes with 40X regions of interest. For this reason, we divided images into 512x512 patches, extracted features for all patches after a downscaling (depending on model-specific transforms, e.g., 224x224 or 20X for many models) with each foundation model and performed a mean pooling aggregation as suggested. Results are presented in the table below for the knn task. Performance is on average slightly lower than when simply resizing images as we did before, which seems to confirm the reviewer’s intuition. We will update BRACS results in our paper following this better procedure.
> > > >
> > > > **knn F1 score on BRACS with and without mean pooling**
> > > > |Processing | hiboub | hiboul | hopt0 | hopt1 | midnight | phikon | phikon2 | uni | uni2h | virchow | virchow2 |
> > > > | :--: | :--: | :--: | :--: | :--: | :--: | :--: | :--: | :--: | :--: | :--: | :--: |
> > > > |Patches + mean pooling| 54.1 | 50.1 | 54.2 | 55.0 | 45.6 | 43.5 | 43.8 | 53.2 | 52.4 | 49.7 | 52.1 |
> > > > |Full image | 56.9 | 56.2 | 52.2 | 55.0 | 50.2 | 50.0 | 45.9 | 55.6 | 56.1 | 51.3 | 54.9 |
> > > >
> > > > |Processing | conch | titan | keep | musk | plip | quiltnet | dinob | dinol | vitb | vitl | clipb | clipl |
> > > > | :--: | :--: | :--: | :--: | :--: | :--: |:--: | :--: | :--: | :--: | :--: | :--: | :--: |
> > > > |Patches + mean pooling| 52.1 | 55.0 | 53.3 | 48.3 | 43.8 | 44.1 | 36.8 | 39.3 | 44.6 | 42.7 | 35.4 | 41.5 |
> > > > |Full image | 56.9 | 59.4 | 53.0 | 57.8 | 48.2 | 50.7 | 43.7 | 46.8 | 45.4 | 46.9 | 42.5 | 46.6 |
> > > >
> > > > > **Selection of datasets and concerns about saturation**
> > > >
> > > > The reviewer is indeed right. As requested, we will integrate the following discussion in the “Limitations” section of our main paper.
> > > >
> > > > In this work, we have included well-studied datasets in the field that have been utilized by many studies dealing with evaluating pathology foundation models. Importantly, they were also the best quality patch-level datasets available at the time of the submission. While they can still bring an interesting signal about differences between existing foundation models, they have been extensively studied and used which could lead to performance saturation. THUNDER is thought of as an evolving benchmark, adapting to the direction the digital pathology community goes toward, and we welcome any suggestions of new relevant datasets to include in the future. For instance, we will integrate the newly introduced patch-level SPIDER datasets [8] in the near future to expand the diversity of supported datasets.
> > > >
> > > > [8] Nechaev et al., SPIDER: A Comprehensive Multi-Organ Supervised Pathology Dataset and Baseline Models, arXiv, 2025.

---

### Official Review · Reviewer_McCV · 2025-06-29

**Rating:** 5
**Confidence:** 4

**Summary:**

The paper presents THUNDER, a comprehensive benchmark for evaluating foundation models in digital pathology. It includes 23 models and 16 diverse public datasets. The authors highlight the need for rigorous evaluation tools in healthcare applications. To differentiate from existing benchmarks, THUNDER focus on tile-level benchmarking, and goes beyond standard downstream tasks by incorporating feature space analyses -- such as alignment, image retrieval, and robustness to transformations -- and novel evaluations including classification, segmentation, uncertainty calibration, and adversarial robustness. The benchmark is open-source, well-documented, and designed for ease of use, promoting reproducibility and community adoption.

**Additional Feedback:**

I would like to know what the authors think about the following:

**Comparison with Existing Benchmarks.** The paper clearly discusses related work and how it differentiates from it. However, new users may doubt why using this benchmark instead of others. Maybe, the paper would benefit from a quantitative comparison to existing benchmarks (for example, discussing differences in runtime). Maybe, this is not feasible due to the different nature of THUNDER (focused on tile-level). What do the authors think? Could it be included?

**About the segmentation results.** Looking at the segmentation downstream task, it looks like conch does not extract relevant information. Is there anything related to how it is trained or its architecture that can explain this? Also, the segmentation task is evaluated using a single Segmenter decoder. May this restrict general conclusions?

**Feature space study.** The authors report in Fig. 4 the evolution of model alignment during LoRA adaptation. Why report averaged results? Why don’t you report both in the appendix?

**Dataset Code Accessibility:**

Yes

**Dataset Code Comments:**

The authors have released their code through a GitHub repository. It is well documented in their website: https://mics-lab.github.io/thunder/. They have functions to download the datasets and models, as well as a function to benchmark a model on a dataset and a task.

**Ethical Comments:**

The only ethical concern that arises in this paper is the potential bias/discrimination incurred by the data-gathering protocol. As the authors argue, this is inherent in the process by which data is acquired. The authors discuss this in the limitations section.

**Ethical Considerations:**

No, there are no or only very minor ethics concerns

**Final Justification:**

The authors have written a high-quality rebuttal, addressing most of my concerns. I consider the work to be of high quality, but not to have a groundbreaking impact in the field. Thus, I have decided to keep my rating at 5.

**Limitations Weaknesses:**

**Alignment with previous findings.** The paper presents many interesting findings. However, it is unclear whether they are consistent with observations in prior literature. I believe a discussion comparing the results to those obtained in previous work would improve the paper.

**Color normalization/Stain augmentation.** The authors explore invariance to different transformations, but color normalization is not explicitly discussed. It would be interesting to analyze how color normalization affects feature representations and whether it's still necessary when using foundation models that were trained with color-based augmentations. I refer to the transformation discussed in, for example, the figure 3 of the following paper (not to be included as a citation):
Shen, Yiqing, et al. "Randstainna: Learning stain-agnostic features from histology slides by bridging stain augmentation and normalization." International Conference on Medical Image Computing and Computer-Assisted Intervention, 2022.

**Model description.** While datasets are well described (e.g., in Table 1), there is no overview of the foundation models used (it is in the supplementary material). I believe you can include a similar summary table listing key characteristics in the main text, or at least reference the existence of this table in the appendix.

**Presentation and clarity of Figures/Tables.** I believe some figures and tables could be improved:
- In Figure 2, the layout is dense, and the subfigures are not well separated. It is difficult to know which one is the subtitle of each subfigure.
- Table 5 uses different colors but it doesn’t include a legend explaining what they represent.

**Strengths Contributions:**

- The paper is well motivated, well written, and easy to understand.
- It includes a comprehensive evaluation of 23 foundation models across 16 datasets, covering a wide range of digital pathology tasks.
- By focusing on the tile/patch level, THUNDER decouples model design from aggregation methods, offering a different and interesting perspective.
- The relationship with existing benchmarks is clearly discussed, and THUNDER’s contributions are clarified.
- Beyond the standard classification task, the authors include novel studies, including a feature space analysis (alignment, retrieval, invariance), segmentation, uncertainty calibration, and adversarial robustness.
- The experimental design is rigorous and well-structured, with interesting analysis and interpretation of results.
- The code and documentation are publicly available.

---

> ### Author Rebuttal · Authors · 2025-07-31
>
> We thank Reviewer McCV for their feedback. Below, we address their comments.
>
> > **Alignment with previous findings**
>
> As mentioned by Reviewer RtDG, the conclusions drawn from our benchmark align with previous studies (“The conclusions from this benchmark align with those of existing ones”). Conclusions matching previous work can be summarized as follows:
> 1. **Better performance of pathology-specific pretrained models.** First, we show that models pre-trained on pathology images outperform the ones trained on natural images, which had been shown in previous work (Campanella et al., Nature Communications, 2025).
> 2. **Strong performance of recent vision-only models.** The high performance of recent vision-only models such as UNI/UNI2-H, VIRCHOW/VIRCHOW2, H-OPTIMUS0/H-OPTIMUS1 models trained with a DiNOv2 training objective was showcased in previous experimental studies (Gatopoulos et al., MIDL, 2024; Jaume et al., NeurIPS, 2024; Ma et al., arXiv, 2025) both at tile and slide levels, and we also confirm this, in particular showing that newer versions of these models (e.g. UNI2-H or VIRCHOW2) outperform previous versions.
> 3. **Competitive performance of VLM models.** VLMs such as CONCH and CONCH1.5 (TITAN) were also highlighted in previous work (Neidlinger et al., arXiv, 2024), which is also the case in our work, along with the newer KEEP model that performs well on many different tasks.
>
> In addition to confirming findings in previous work, we also go one step further by considering very recent models (e.g. the competitive MIDNIGHT and KEEP) that were not included in previous benchmarks, and also new tasks (feature space alignment, calibration, robustness).
>
> > **Color normalization/Stain augmentation**
>
> We agree that invariance to stain variations is an important property of foundation models for digital pathology. For this reason, we included the HED transform in our study (Table 4 in our Supplementary Material and Section 4 “feature space study” + Figure 4d in our main paper), performing random shifts in the Hematoxylin-Eosin-DAB (HED) color space. Indeed, such a HED transform was introduced to simulate variations in stain between different centers. Related to color variations, we also evaluated the invariance of models to a simpler color jittering (Table 4 in our Supplementary Material). In addition to this, we computed the invariance of models to the RandStainNA augmentation suggested by the reviewer (color normalization + stain augmentation) as it is also very relevant. For space reasons, we cannot include full tables here but can provide them during the discussion period if the reviewer requests them. The RandStainNA scores (cosine similarity between features computed from images with and without the augmentation) vary little across cohorts (0.69 – 0.78; σ ≈ 0.03), whereas they span a much wider range across foundation models (0.48 – 0.98; σ ≈ 0.15), showing that stain robustness is far more sensitive to the specific model used than the dataset. In particular, histopathology‑pretrained models are more sensitive to RandStainNA, showing with about 0.67 ± 0.11 invariance versus 0.91 ± 0.08 for natural‑image‑pretrained models.
>
> > **Model description**
>
> Table 1 in our Supplementary Material presents supported foundation models (architecture, parameters, training objective, dataset). We agree with the reviewer that such a table is important and will add a reference to it in Section 3.1 of the main paper.
>
> > **Figure 2 dense layout**
>
> Space in Figure 2 will be increased to better separate sub-figures.
>
> > **Colors in Table 5**
>
> Colors in Table 5 follow the same style as in other plots showing results related to histopathology-based versus natural-based models. However, it is true that right now Table 5 is not self-contained: the legend for Table 5 will be added to the caption.
>
> > **Comparison with Existing Benchmarks**
>
> As mentioned by Reviewer RtDG, we can consider the four current open-source benchmarks as comparison points:
> 1. **eva** (Gatopoulos et al., MIDL, 2024) includes both tile and slide level tasks, and evaluates models on linear probing (classification) and semantic segmentation.
> 2. **PathoBench** (Zhang et al., arXiv, 2025) focuses on slide-level classification and regression tasks (Morphological subtyping, Tumor grading, Molecular subtyping, Mutation prediction, Treatment response and assessment, Survival prediction).
> 3. **HEST-Benchmark** (Jaume et al., NeurIPS, 2024) targets gene expression regression at the tile level.
> 4. **PathBench** (Ma et al., arXiv, 2025) presents the slide-level performance of foundation models for diverse classification and regression (DFS, DSS, OS prediction) tasks, but lacks an open-source tool for custom model evaluation (only an online leaderboard is available)
>
> While HEST-Benchmark focuses only on gene expression prediction, and has thus a much specific, yet interesting, scope compared with THUNDER, and PathBench does not provide an open-source implementation to evaluate a new model, we can compare THUNDER to eva and PathoBench quantitatively along two main axes: diversity of tasks and datasets, and runtime.
>
> First, THUNDER differs in the number and variety of tasks. For example, eva, though including some patch-level datasets, supports only 2 tasks: linear probing and segmentation. In contrast, THUNDER implements 9 diverse tasks (knn, linear probing, few-shot, segmentation, feature alignment, retrieval, feature invariance, calibration, robustness to adversarial attacks), enabling broader model evaluation. It also supports LoRA adaptation, absent in other benchmarks. Moreover, it covers more datasets: 12 classification and 4 segmentation tile-level datasets, compared to eva’s 8 classification and 3 segmentation datasets according to their documentation.
>
> Second, direct runtime comparison with slide-level benchmarks like PathoBench is difficult due to variations in datasets and aggregation models. We thus suggest considering some orders of magnitude to draw a high-level picture of runtime differences between THUNDER and slide-level evaluation. Importantly, tile-level evaluation isn’t chosen solely for speed. As discussed in the paper, it isolates model representation quality by removing aggregation dependencies, enabling more direct comparisons.
> Slide-level prediction first requires selecting a magnification, segmenting tissue, and dividing it into patches, which is a considerable amount of preprocessing that needs to be done before any slide‑level predictions can even begin. For just seven public datasets (TCGA‑BLCA, BRCA, CAMELYON16, KIRC, LUAD, LUSC, UCEC), this yields around **55 million tissue patches** (≈ 7.3 M + 12 M + 5.1 M + 7.5 M + 5.1 M + 6.6 M + 9.9 M). Extracting features at 20x magnification and feeding them through a foundation model is very costly. For VIRCHOW2, this step alone took **~514 GPU-hours** (V100 GPU) across the 7 datasets: BLCA (437 WSI, 63 h), BRCA (1,100, 106 h), CAMELYON16 (400, 42 h), KIRC (511, 83 h), LUAD (456, 69 h), LUSC (505, 66 h), UCEC (504, 85 h) -- a total of around 4,000 WSIs. Running this for 23 foundation models pushes the cost above **10,000 GPU-hours**, before any training begins. In contrast, THUNDER processes **~2 million pre-extracted patches** across 16 datasets; the same 23-model ensemble finishes extraction in **< 500 GPU-hours.** THUNDER also provides **richer supervision** (~2M patch-level vs. ~4k slide-level labels), enabling more diverse tasks and more test samples, as slide-level datasets provide fewer labels and a weaker evaluation signal. Finally, THUNDER’s tile-level design ensures full reproducibility, which isn’t possible with the manual preprocessing required for slide-level tasks.
>
> Moreover, slide-level evaluation typically requires training a Multiple Instance Learning (MIL) aggregator. Common models like Abmil (Ilse et al., ICML, 2018) or Transmil (Shao et al., NeurIPS, 2021) have ≈1M–3M parameters, significantly more than the simple linear probes used in THUNDER. By also including knn and few-shot tasks that require no training, THUNDER further reduces runtime while offering a clearer view of model representations.
>
> > **Segmentation results (CONCH)**
>
> Thank you for highlighting this inconsistency. CONCH segmentation performance was updated (see Table below) as initial low performance was due to an attention-based pooling of spatial representations that is useful in image-vision tasks but is hurting segmentation performance. Without such a pooling, CONCH now showcases high performance, on-par with the best models. Moreover, we use Segmenter as it is a well-known decoder adapted to Transformer-based encoders, and was shown to provide good performance in previous work. Importantly, the same decoder is used for all foundation models to ensure a fair comparison between them.
>
> **Updated Dice Score for CONCH**
> | pannuke  | ocelot | seg-ep | seg-ly |
> | ------------- |:---------:|:---------:|-----------|
> | 66.5          | 92.2     | 85.6     | 72.5     |
>
> > **Feature space study**
>
> We report averaged results in the main paper to get some insights about the overall trend. Per-dataset plots are reported in Figure 6 of our Supplementary Material. We will add a pointer to this Figure in Section 4 of the main paper.

---

> > ### Comment · Reviewer_McCV · 2025-08-04
> >
> > I thank the authors for their rebuttal, which addresses most of my concerns. I would like to highlight the following points, as they were not discussed in the original manuscript and, in my view, are important:
> >
> > **Color normalization / stain augmentation.** Histopathology‑pretrained models appear more sensitive to RandStainNA than models pretrained on natural images. This is somewhat surprising, given that the former are typically exposed to color normalization or stain augmentation during pretraining.
> >
> > **Differences from existing benchmarks.** Unlike prior benchmarks that focus on slide-level performance (~ 55 million tissue patches from 7 datasets with ~ 4k slide-level labels), THUNDER emphasizes tile-level performance. This allows it to use significantly fewer patches (~ 2 million across 16 datasets) while benefiting from richer supervision.
> >
> > Overall, I believe the paper has improved after the rebuttal. I am inclined to maintain my score and recommend the acceptance of the paper.

---

> > > ### Author Response · Authors · 2025-08-06
> > >
> > > We thank the reviewer for their feedback that allowed us to greatly improve our paper. The higher invariance of natural-based models to RandStainNA might be explained by the lack of pathology images in their training data, leading to an inability for them to model staining variations. Importantly, we will also include differences between pathology models in the paper as we believe such comparisons are even more interesting. The discussion about differences between THUNDER and existing benchmarks will also be added to the paper, along with all mentioned points in our rebuttal.

---

### Official Review · Reviewer_FZQv · 2025-07-01

**Rating:** 5
**Confidence:** 3

**Summary:**

In this paper, the authors proposed THUNDER, a benchmark study to compare 23 foundation models on 16 diverse datasets. THUNDER performs benchmarking not only on downstream analysis, but also study the differences in the model feature spaces and their robustness and uncertainty estimation. The authors performed various tasks to test these models. The paper is overall good and easy to read with some minor things to improve.

**Dataset Code Accessibility:**

Yes

**Dataset Code Comments:**

The code and package are on the authors' Github. There are some documentation. Considering the package is easy to use, I consider the documentation is sufficient.

**Ethical Considerations:**

No, there are no or only very minor ethics concerns

**Final Justification:**

This paper is well-written and easy to follow. The authors performed comprehensive benchmark on various of the models. I only have some minor comments and they are well addressed. During the rebuttal, the authors proposed new figures, edits and discussion to the paper which I believe will improve the quality of the paper. I maintain my decision to accept this paper.

**Limitations Weaknesses:**

1. I am still slightly confused/unclear why the authors choose not to include slide-level task evaluation, as this is a common thing in other benchmark study.
2. The authors only includes the models with good classification performance to the segmentation task, while its unclear whether the segmentation task shares the similar performance as the classification task.
4. Figure 2a and 2b look good. But unclear to me what's the rank of each method, maybe consider another average score figure/table. (I see this later in Table 5, maybe add a pointer to the table.)
5. Figure 4b, consider do a another heatmap in appendix/supp with clustering to see the clusters.
6. It would be nice to have a summary table for the models supported, with parameter number (I see this later in the supplementary files which include a lot of stuff, but I think none of them are mentioned in the main paper, maybe consider to add some pointers), what dataset they were trained on since there might be overlap in the training dataset and evaluation dataset especially for those histo models. Also another thing to add if possible is the runtime for each model in each benchmark task and the time before/after LoRa.
7. This is a general comment. I think the most conclusion I saw are the ones that claim histo models perform better, the VLM are alike while the vision models are alike, basically the models in the four categories are similar to each other. Is there any other conclusion for models within each group? If so, the authors might want to highlight those conclusion.
8. Since the authors did not perform any slide level evaluation, for a lot of the dataset included, I think it would be nice to check each dataset to see whether the label is on patch level and slide level and ideally provide/talk about this information in the paper or appendix.

Minor:
1. page 6, HED transformation unexplained
2. page 8, it seems to have a huge gap between figure legend and text. (its ok to keep that if its hard to change)
3. page 5, ECE, MCE, ACE, TACE, maybe consider add the full name when they first appear, or add a pointer to the supp

**Strengths Contributions:**

1. THUNDER performs benchmarking not only on downstream analysis, but also study the differences in their feature spaces and their robustness and uncertainty estimation.
2. The GitHub page looks clean and the functions are easy to use.
3. There are a lot of different tasks in this paper which I really like, especially the ones about the latent space and how the models are alike.
4. There are clear documentation about the time each benchmark task takes and the hardware they used.
5. The documentation is clear about the experiment details.

---

> ### Author Rebuttal · Authors · 2025-07-31
>
> We thank Reviewer FZQv for their feedback. Below, we address their comments.
>
> > **Reasons for focusing on tile-level tasks**
>
> Most foundation models for digital pathology are trained at the tile level, and even slide-level foundation models generally leverage a pre-trained patch-level encoder. To perform predictions at the slide level, Whole Slide Images (WSI) must be divided into patches to extract patch-specific features from the foundation models. Such features then need to be aggregated to provide slide-level predictions. As mentioned in our paper (Section 2, Page 3), evaluating them on tiles allows us to isolate the predictive power of models themselves independently of aggregation strategies, leading to a more direct evaluation of their representations. Stated differently, Reviewer McCV nicely summarized this point as follows: “By focusing on the tile/patch level, THUNDER decouples model design from aggregation methods”.
>
> Additionally, working at the tile level allows to avoid the heavy slide processing which can be compute-demanding. Indeed, for VIRCHOW2, the 20X feature‑extraction step alone consumed ≈ **514 GPU‑hours** (V100 GPU) across the 7 following datasets: BLCA (437 WSI, 63 h), BRCA (1 100, 106 h), CAMELYON16 (400, 42 h), KIRC (511, 83 h), LUAD (456, 69 h), LUSC (505, 66 h) and UCEC (504, 85 h) — a total of around 4000 whole‑slide images. Repeating this for each of the 23 foundation models pushes the bill to more than **10 000 GPU‑hours** before any slide‑level training can start. By contrast, our benchmark covers all 16 datasets with just ≈ **2 million pre‑extracted patches**; the same 23‑model ensemble finishes feature extraction in < **500 GPU‑hours**, while providing **richer supervision** (~2M patch-level labels vs. ~4k slide-level labels). After feature extraction, a Multiple Instance Learning (MIL) aggregator must be trained to aggregate patch-level features. Common methods such as Abmil (Ilse et al., ICML, 2018 ≈ 1M parameters depending on the setting) or Transmil (Shao et al., NeurIPS, 2021 ≈ 3M parameters depending on the setting) require training more parameters than simple linear probes as used in THUNDER. For additional details about quantitative runtimes, please refer to our answer to Reviewer McCV. Even if we agree that slide-level benchmarks are interesting and relevant, we believe a complementary tile-level alternative such as THUNDER allows for faster and more direct evaluation on many different datasets requiring fewer resources. Additionally, using patch-level data enables THUNDER to provide a fully reproducible benchmark, which is not feasible for slide-level tasks due to required manual preprocessing steps.
>
> Finally, most of the previous experimental studies were conducted on slide-level datasets (Section 2 in our paper, “Benchmarking pathology models”). We thus believe the general performance of foundation models at the tile level has been understudied despite the benefits mentioned before, and we thus believe THUNDER addresses this lack.
>
> > **Choice of models to evaluate on segmentation in the main paper**
>
> Thank you for your insightful comment. As segmentation provides insights about the ability of models to extract spatial information, which is important in medical imaging, and most previous studies rather focus on classification (Section 2 in our paper, “Benchmarking pathology models”), we wanted to see whether a strong classification performance would translate to a good segmentation performance in the main paper. For this reason, we only selected models performing well at classification. Interestingly, we showed that model ranking was different between classification and segmentation. Table 6 in our Supplementary Material presents segmentation results for all models on all datasets. We agree that this was not very clear in our main paper and a reference to this table will be added to the main paper. An aggregated plot of all results (similarly to what is done for knn, linear probing and few-shot) will also be integrated into the main paper, and per-model segmentation performance will be added to the final rank-sum performance ranking.
>
>
> > **Figures 2a, 2b and model ranking**
>
> Figures 2a and 2b present the results of statistical tests, and rather provide a general overview of differences in performance significance. Rankings between models are indeed already presented in Table 5, to which we will add a pointer earlier in the paper. In addition to this, we performed hierarchical clustering on top of significance heatmaps (Figures 2a and 2b) to get more insights into the ranking of models based on statistical significance results. Due to the rebuttal policy, we cannot upload them, but we will add them to our Supplementary Material. Such hierarchical clustering seems to confirm the rankings from Table 5. For example, for the knn task, UNI2-H and VIRCHOW2 appear to be clustered together as H-OPTIMUS1 and KEEP, the 4 being grouped into a higher-level cluster. Similarly, in Table 5 in our main paper, these are the top-4 models for knn.
>
> > **Feature alignment heatmaps**
>
> Thank you for your suggestion. We agree that additional visualizations will enhance the readability of our results, thus we will provide heatmaps in addition to graphs in the Supplementary Material as another way to visualize feature space alignment between model pairs. Due to rebuttal policies, we cannot provide examples of resulting heatmaps here.
>
> > **Summary table of supported models**
>
> Table 1 in our Supplementary Material presents supported foundation models (architecture, parameters, training objective, training datasets). We agree with the reviewer that such a table is important and will add a reference to it in Section 3.1 of the main paper. Table 2 in our main paper presents runtimes averaged across foundation models as we believe per-model statistics might not be very informative due to relatively small variations between most models on most tasks. However, we can still include per-model runtime tables in our Supplementary Material (in particular for tasks where variations are higher). Due to the space limit, we cannot include such tables here, but we can provide them during the discussion period if the reviewer requests them. The same is true for LoRA runtimes.
>
> > **Differences between models within a group (Histo., Natural, Vision-only, VLM)**
>
> Our aim with THUNDER is to provide a common, extensible and comprehensive framework to allow future work to pursue more involved comparison studies between new foundation models in a fair manner. To this end, we already provide an experimental study in our paper and show that we can rank state-of-the-art models on different tasks (Table 5 in our main paper provides a summary). We aim to include more models and datasets in THUNDER, while users can already benchmark their custom models. Our objective in the paper (Section 4) was 2-fold: (1) providing a low-level and fair comparison of all recent foundation models as an actionable ranking helping in choosing the best feature extractor for a downstream task of interest, (2) studying the high-level trends, particularly focusing on the impact of the training domain (histopathology vs natural images) and of modalities (vision-only vs VLM).
>
> Based on the reviewer’s comment, we provide below additional insights into intra-group differences between foundation models (supported by Table 5 in our main paper) that are included currently in the benchmark:
> 1. **Impact of SSL methods.** Among vision-only models, it appears that the only model trained with iBot (Phikon) performs worse than all others trained with DINOv2. This might indicate the superiority of DINOv2 as SSL pretraining strategy, which could be confirmed by its large adoption across most foundation models.
> 2. **Impact of datasets.** Generally, models trained from large and diverse datasets such as UNI2-H or VIRCHOW2 have higher performance. Interestingly, on the other hand, MIDNIGHT appears as an exception because it reaches strong performance while being trained on the smaller TCGA dataset. Going forward into analyzing the impact of different characteristics of pre-training data on final performance is quite difficult since most models (e.g. UNI2-H or VIRCHOW2) are partly trained on private data.
> 3. **Impact of number of parameters of the models.** The number of model parameters can also play a role in final performance: the trend is a bit clearer within VLMs where models with more parameters, i.e. KEEP and CONCH1.5 (TITAN), seem to outperform others on downstream task performance (knn, linear probing, few-shot).
>
> These three points will be added to the discussion section of our paper.
>
> > **Patch-level labels in considered datasets**
>
> All considered datasets come with patch-level labels, ensuring high-quality supervision for downstream tasks. This is indeed important and will be made clearer in the main paper.
>
> > **Minor comments**
>
> HED is a pathology-specific transform simulating staining variations by performing a Hematoxylin-Eosin-DAB (HED) color deconvolution followed by random shifts in that color space. This will be clarified in the paper. On page 8, the gap between the figure legend and text will be decreased. Full names of calibration metrics will be added to the main paper at first appearance.

---

> > ### Comment · Reviewer_FZQv · 2025-08-05
> >
> > I thank the rebuttal by the authors. The newly proposed figures will add clarity of the paper and the discussion on the patch level focus and models groups will enrich the paper. I think the authors have addressed most of my concerns.

---

> > > ### Author Response · Authors · 2025-08-06
> > >
> > > We thank the reviewer for their feedback that indeed helped improve the paper. We will include the proposed figures along with discussions about the advantages of focusing on patch-level datasets and about the differences between models inside the same groups.

---

### Official Review · Reviewer_Z95c · 2025-07-03

**Rating:** 5
**Confidence:** 3

**Summary:**

This paper presents THUNDER, a tile level comprehensive and open-source benchmark designed to evaluate foundation models for digital pathology. It focuses on small image patches rather than whole slides. THUNDER evaluates 23 vision-only and vision-language foundation models across 16 datasets on multiple organs, magnifications, and tasks. Different from traditional benchmarks which only evaluate the benchmark accuracy, THUNDER also evaluate downstream task performance, feature space analysis, uncertainty calibration, and robustness against adversarial attacks. The paper provides a lot of analyses. I believe it would be beneficial to the computational pathology community.

**Additional Feedback:**

1. Following the weakness, it would be great to include some triage examples when comparing the model performance. And also add some directions on how to utilize the benchmarking results to improve models.
2. I feel like the paper is a little bit hard to follow, especially for readers unfamiliar with digital pathology.

**Dataset Code Accessibility:**

Partly

**Dataset Code Comments:**

The authors provides the codes and instructions on Github. The benchmark seems to be complete and well-structured. But usage instruction are not very detailed. I follow the instuctions to install the package and try to run the benchmark. But encountered an error "AttributeError: Can't pickle local object 'get_from_safetensors.<locals>.transform'" (Although I did not spend too much time debugging it.)

**Ethical Considerations:**

No, there are no or only very minor ethics concerns

**Final Justification:**

Thanks the authors! You've addressed most of my concerns, and I now have a much better understanding of the paper. Great work! I've increased both my rating and my confidence level.

**Limitations Weaknesses:**

The paper focuses on comparing models, but does not explain why certain models perform better in some tasks. And does not provide how to improve the models based on the comparison results.

**Strengths Contributions:**

1. The paper conducted extensive evaluations across 23 models and 16 datasets. I believe the authors have put a lot of effort into evaluating the model performance under various settings.
2. The paper introduces some metrics other than accuracy, such as feather space analysis, model calibration etc. which provide more insights about the model performance.

---

> ### Author Rebuttal · Authors · 2025-07-31
>
> We thank Reviewer Z95c for their feedback. Below, we address their comments.
>
> > **Explaining differences in performance and how to improve models**
>
> Our aim with THUNDER is to provide a common, extensible and comprehensive framework to allow future work to pursue more involved comparison studies between new foundation models in a fair manner. To this end, we already provide an experimental study in our paper and show that we can rank state-of-the-art models on different tasks (Table 5 in our main paper provides a summary). We aim to include more models and datasets in THUNDER, while users can already benchmark their custom models. Our objective in the paper (Section 4) was 2-fold: (1) providing a low-level and fair comparison of all recent foundation models as an actionable ranking helping in choosing the best feature extractor for a downstream task of interest, (2) studying the high-level trends, in particular:
> 1. **Impact of the training domain.** The training domain is primordial as histopathology-specific models significantly outperform the ones trained on natural images.
> 2. **Impact of modalities.** It also appears that the involved modalities play a key role since the top-5 models are all Vision-only models, while the best VLMs, e.g. KEEP and CONCH1.5 (TITAN), reach the highest performance in the low-shot (e.g. 1-shot) classification setting.
>
> Based on the reviewer’s comment, we provide below additional insights into differences between foundation models (supported by Table 5 in our main paper) that are included currently in the benchmark:
> 1. **Impact of SSL methods.** Among vision-only models, it appears that the only model trained with iBot (Phikon) performs worse than all others trained with DINOv2. This might indicate the superiority of DINOv2 as SSL pretraining strategy, which could be confirmed by its large adoption across most foundation models.
> 2. **Impact of datasets.** Generally, models trained from large and diverse datasets such as UNI2-H or VIRCHOW2 have higher performance. Interestingly, on the other hand, MIDNIGHT appears as an exception because it reaches strong performance while being trained on the smaller TCGA dataset only. Going forward into analyzing the impact of different characteristics of pre-training data on final performance is quite difficult since most models (e.g. UNI2-H or VIRCHOW2) are partly trained on private data.
> 3. **Impact of number of parameters of the models.** The number of model parameters can also play a role in final performance: the trend is a bit clearer within VLMs where models with more parameters, i.e. KEEP and CONCH1.5 (TITAN), seem to outperform others on downstream task performance (knn, linear probing, few-shot).
>
> These three points will be added to the discussion section of our paper.
>
> With respect to the question about how to improve models based on our benchmark, the insights we previously discussed (impact of training domain, modalities, SSL methods, datasets, number of parameters) can serve as pointers for the development of future foundation models, and we hope THUNDER can be used as a tool to strengthen such conclusions provided more models and datasets in the future. Moreover, an interesting finding is the potential gain a simple LoRA adaptation can bring to tasks with small datasets, even for strong foundation models, as shown in our paper. Efficient adaptation thus appears as a relevant direction to improve foundation models, also highlighting their difficulty to adapt in a zero-shot manner, which could be another promising topic about foundation model pre-training. Improving foundation models also requires a better understanding of their inner mechanisms. We believe in the power of alignment metrics such as Mutual KNN and hope to provide a common framework for researchers to extend our study. Indeed, the alignment graphs we could build along with the evolution of alignment during LoRA adaptation are quite intriguing and deserve more studies in future work.
>
> > **Code usage**
>
> We thank the reviewer for trying to use the THUNDER codebase and for their positive comment about its structure. We hypothesize that the encountered error could come from an issue with the model weights (e.g, corrupted file, unavailable, or other). For the model to be properly downloaded, the huggingface token should be stored in an environment variable, and the user should have accepted the terms and conditions on the model hugginface page. Despite many efforts on our side to make code usage as smooth as possible for any user, we will make the mentioned points clearer in the documentation and will work on providing more informative errors when catching exceptions. Please feel free to share additional details during the discussion phase in case the error persists on the main branch of the GitHub repository.
>
> > **Paper readability**
>
> Thanks for raising this readability issue. In the camera-ready version of the paper, we will put some effort into improving the readability, targeting readers who might be unfamiliar with digital pathology. In particular, as requested by other reviewers, we will provide a pointer to Table 1 in the Supplementary Material, presenting all considered foundation models, as such information might be important to readers outside of the field of digital pathology.

---

### Decision · Program_Chairs · 2025-09-18

**Decision:**

Accept (spotlight)

**Comment:**

Summary: All reviewers (McCV, FZQv, Z95c, and RtDG) rated this as a clear accept (with confidences between 3 and 5). From the depth and breadth of the discussion, it seems clear that the reviewers and authors each understood the strengths and residual limitations of the proposed benchmark dataset and methodology. Given these findings, I propose to accept this as a spotlight, but am entirely happy to see this bumped up (oral) or down (poster), depending on the overall focus of this year's NeurIPS highlights.

Strengths: The submission proposes a benchmark methodology for histopath tile-data foundation models that offers a variety of metrics and delves into different aspects of evaluation (applications and purposes for datasets), together with a very comprehensive dataset and exploration of the results. The documentation of the included datasets, the methodology, and results is very clear and will support improving work in this domain, as mentioned in the abstract. Overall, it seems evident that the authors have extensively studied the literature and available data and methods from this field, and undertaken a formidable effort to bring these pieces together in a dense but ultimately accessible publication with a well documented GitHub repo, which can be used to extend this to further models.

Weaknesses: The benchmark is limited to tile-level data, rather than also including a slide-level methodology (rebuttal: the benchmark is meant to evaluate foundation models rather than derived models used in specific classification applications). How exactly the submission builds on prior literature is not quite fully explicated in the paper itself (reviewer McCV: " The paper presents many interesting findings. However, it is unclear whether they are consistent with observations in prior literature. I believe a discussion comparing the results to those obtained in previous work would improve the paper."), and the rebuttal does not make it clear whether this was addressed in the final version. Potentially too much focus is put on adversarial attacks (reviewer RtDG: "Adversarial attacks are often cited to highlight the limitations of foundation models, but I question their interest.", and the rebuttal mostly offers technical justifications, but no practical insight into how much possible technical insights gained from those adversarial attacks ultimately improve downstream applications.

Featured reviewer comments:
- Z95c: "Different from traditional benchmarks which only evaluate the benchmark accuracy, THUNDER also evaluate downstream task performance, feature space analysis, uncertainty calibration, and robustness against adversarial attacks. The paper provides a lot of analyses. I believe it would be beneficial to the computational pathology community."
- FZQv: "There are a lot of different tasks in this paper which I really like, especially the ones about the latent space and how the models are alike."
- McCV: "Beyond the standard classification task, the authors include novel studies, including a feature space analysis (alignment, retrieval, invariance), segmentation, uncertainty calibration, and adversarial robustness. The experimental design is rigorous and well-structured, with interesting analysis and interpretation of results."
- McCV: "The authors explore invariance to different transformations, but color normalization is not explicitly discussed. It would be interesting to analyze how color normalization affects feature representations and whether it's still necessary when using foundation models that were trained with color-based augmentations." -> rebuttal: " For this reason, we included the HED transform in our study (Table 4 in our Supplementary Material and Section 4 “feature space study” + Figure 4d in our main paper), performing random shifts in the Hematoxylin-Eosin-DAB (HED) color space."
- RtDG: "Good coverage of the latest models in AI for pathology, even very recent models such as Midnight, H-Optimus-1, and Keep."
- RtDG: "The study comes from a group that hasn't developed its own foundation model, which makes the study feel more objective and less biased."
- RtDG: "Evaluating model calibration in this setting can be misleading. The calibration of a linear model trained on frozen patch embeddings from multiple foundation models is likely to vary significantly with the choice of regularization in logistic regression." -> rebuttal: "[...] using a simple linear classifier, which has less expressive power than an MLP, allows us to reduce how much the initial features from the foundation models are transformed, better assessing the impact of foundation model features on classification calibration."

Discussion: The authors extensively addressed the reviewers' limitation and weaknesses points, and from the reviewers' subsequent comments, it seems clear that all major points were addressed satisfactorily.